# A Computation and Communication Efficient Method for Distributed Nonconvex Problems in the Partial Participation Setting

**Alexander Tyurin**
KAUST
Saudi Arabia
alexandertiurin@gmail.com

**Peter Richtárik**
KAUST
Saudi Arabia
richtarik@gmail.com

## Abstract

We present a new method that includes three key components of distributed optimization and federated learning: variance reduction of stochastic gradients, partial participation, and compressed communication. We prove that the new method has optimal oracle complexity and state-of-the-art communication complexity in the partial participation setting. Regardless of the communication compression feature, our method successfully combines variance reduction and partial participation: we get the optimal oracle complexity, never need the participation of all nodes, and do not require the bounded gradients (dissimilarity) assumption.

## 1 Introduction

Federated and distributed learning have become very popular in recent years (Konečný et al., 2016; McMahan et al., 2017). The current optimization tasks require much computational resources and machines. Such requirements emerge in machine learning, where massive datasets and computations are distributed between cluster nodes (Lin et al., 2017; Ramesh et al., 2021). In federated learning, nodes, represented by mobile phones, laptops, and desktops, do not send their data to a server due to privacy and their huge number (Ramaswamy et al., 2019), and the server remotely orchestrates the nodes and communicates with them to solve an optimization problem.

As in classical optimization tasks, one of the main current challenges is to find **computationally efficient** optimization algorithms. However, the nature of distributed problems induces many other (Kairouz et al., 2021), including i) **partial participation** of nodes in algorithm steps: due to stragglers (Li et al., 2020) or communication delays (Vogels et al., 2021), ii) **communication bottleneck**: even if a node participates, it can be costly to transmit information to a server or other nodes (Alistarh et al., 2017; Ramesh et al., 2021; Kairouz et al., 2021; Sapio et al., 2019; Narayanan et al., 2019). It is necessary to develop a method that considers these problems.

## 2 Optimization Problem

Let us consider the nonconvex distributed optimization problem

$$\min_{x \in \mathbb{R}^d} \left\{ f(x) := \frac{1}{n} \sum_{i=1}^{n} f_i(x) \right\}, \tag{1}$$

where $f_i : \mathbb{R}^d \to \mathbb{R}$ is a smooth nonconvex function for all $i \in [n] := \{1, \ldots, n\}$. The full information about function $f_i$ is stored on $i^{\text{th}}$ node. The communication between nodes is maintained in the parameters server fashion (Kairouz et al., 2021): we have a server that receives compressed

37th Conference on Neural Information Processing Systems (NeurIPS 2023).

information from nodes, updates a state, and broadcasts an updated model.[1] Since we work in the nonconvex world, our goal is to find an $\varepsilon$-solution ($\varepsilon$-stationary point) of (1): a (possibly random) point $\widehat{x} \in \mathbb{R}^d$, such that $\mathrm{E}\left[\|\nabla f(\widehat{x})\|^2\right] \leq \varepsilon$.

We consider three settings:

1. **Gradient Setting.** The $i^{\text{th}}$ node has only access to the gradient $\nabla f_i : \mathbb{R}^d \to \mathbb{R}^d$ of function $f_i$. Moreover, the following assumptions for the functions $f_i$ hold.

**Assumption 1.** *There exists $f^* \in \mathbb{R}$ such that $f(x) \geq f^*$ for all $x \in \mathbb{R}$.*

**Assumption 2.** *The function $f$ is $L$–smooth, i.e., $\|\nabla f(x) - \nabla f(y)\| \leq L\|x - y\|$ for all $x, y \in \mathbb{R}^d$.*

**Assumption 3.** *The functions $f_i$ are $L_i$–smooth for all $i \in [n]$. Let us define $\widehat{L}^2 := \frac{1}{n}\sum_{i=1}^n L_i^2$.*[2]

2. **Finite-Sum Setting.** The functions $\{f_i\}_{i=1}^n$ have the finite-sum form

$$f_i(x) = \frac{1}{m}\sum_{j=1}^m f_{ij}(x), \qquad \forall i \in [n], \tag{2}$$

where $f_{ij} : \mathbb{R}^d \to \mathbb{R}$ is a smooth nonconvex function for all $j \in [m]$.

We assume that Assumptions 1, 2 and 3 hold and the following assumption.

**Assumption 4.** *The function $f_{ij}$ is $L_{ij}$-smooth for all $i \in [n], j \in [m]$. Let $L_{\max} := \max_{i \in [n], j \in [m]} L_{ij}$.*

3. **Stochastic Setting.** The function $f_i$ is an expectation of a stochastic function,

$$f_i(x) = \mathrm{E}_\xi\left[f_i(x; \xi)\right], \qquad \forall i \in [n], \tag{3}$$

where $f_i : \mathbb{R}^d \times \Omega_\xi \to \mathbb{R}$. For a fixed $x \in \mathbb{R}$, $f_i(x; \xi)$ is a random variable over some distribution $\mathcal{D}_i$, and, for a fixed $\xi \in \Omega_\xi$, $f_i(x; \xi)$ is a smooth nonconvex function. The $i^{\text{th}}$ node has only access to a stochastic gradients $\nabla f_i(\cdot; \xi_{ij})$ of the function $f_i$ through the distribution $\mathcal{D}_i$, where $\xi_{ij}$ is a sample from $\mathcal{D}_i$. We assume that Assumptions 1, 2 and 3 hold and the following assumptions.

**Assumption 5.** *For all $i \in [n]$ and for all $x \in \mathbb{R}^d$, the stochastic gradient $\nabla f_i(x; \xi)$ is unbiased and has bounded variance, i.e., $\mathrm{E}_\xi\left[\nabla f_i(x; \xi)\right] = \nabla f_i(x)$, and $\mathrm{E}_\xi\left[\|\nabla f_i(x; \xi) - \nabla f_i(x)\|^2\right] \leq \sigma^2$, where $\sigma^2 \geq 0$.*

**Assumption 6.** *For all $i \in [n]$ and for all $x, y \in \mathbb{R}$, the stochastic gradient $\nabla f_i(x; \xi)$ satisfies the mean-squared smoothness property, i.e., $\mathrm{E}_\xi\left[\|\nabla f_i(x; \xi) - \nabla f_i(y; \xi)\|^2\right] \leq L_\sigma^2\|x - y\|^2$.*

We compare algorithms using *the oracle complexity*, i.e., the number of (stochastic) gradients that each node has to calculate to get $\varepsilon$-solution, and *the communication complexity*, i.e., the number of bits that each node has to send to the server to get $\varepsilon$-solution.

## 2.1 Unbiased Compressors

We use the concept of unbiased compressors to alleviate the communication bottleneck. The unbiased compressors quantize and/or sparsify vectors that the nodes send to the server.

**Definition 1.** A stochastic mapping $\mathcal{C} : \mathbb{R}^d \to \mathbb{R}^d$ is an *unbiased compressor* if there exists $\omega \in \mathbb{R}$

$$\text{such that} \quad \mathrm{E}\left[\mathcal{C}(x)\right] = x \quad \text{and} \quad \mathrm{E}\left[\|\mathcal{C}(x) - x\|^2\right] \leq \omega\|x\|^2 \quad \forall x \in \mathbb{R}^d. \tag{4}$$

We denote a set of stochastic mappings that satisfy Definition 1 as $\mathbb{U}(\omega)$. In our methods, the nodes make use of unbiased compressors $\{\mathcal{C}_i\}_{i=1}^n$. The community developed a large number of unbiased compressors, including Rand$K$ (see Definition 5) (Beznosikov et al., 2020; Stich et al., 2018), Adaptive sparsification (Wangni et al., 2018) and Natural compression and dithering (Horváth et al., 2019a). We are aware of correlated compressors by Szlendak et al. (2021) and quantizers by Suresh et al. (2022) that help in the homogeneous regimes, but in this work, we are mainly concentrated on generic heterogeneous regimes, though, for simplicity, assume the independence of the compressors.

**Assumption 7.** *$\mathcal{C}_i \in \mathbb{U}(\omega)$ for all $i \in [n]$, and the compressors are statistically independent.*

---

[1]Note that this strategy can be used in peer-to-peer communication, assuming that the server is an abstraction and all its algorithmic steps are performed on each node.

[2]Note that $L \leq \widehat{L}$, $\widehat{L} \leq L_{\max}$, and $\widehat{L} \leq L_\sigma$.

Table 1: Summary of methods that solve the problem (1) in the stochastic setting (3). Abbr.: *VR (Variance Reduction)* = Does a method have the optimal oracle complexity $\mathcal{O}\left(\frac{\sigma^2}{\varepsilon} + \frac{\sigma}{\varepsilon^{3/2}}\right)$? *PP (Partial Participation)* = Does a method support partial participation from Section 2.2? *CC* = Does a method have the communication complexity equals to $\mathcal{O}\left(\frac{\omega}{\sqrt{n}\varepsilon}\right)$?

| Method | VR | PP | CC | Limitations |
|---|---|---|---|---|
| SPIDER, SARAH, PAGE, STORM (Fang et al., 2018; Nguyen et al., 2017) (Li et al., 2021a; Cutkosky and Orabona, 2019) | ✓ | ✗ | ✗ | — |
| MARINA (Gorbunov et al., 2021) | ✓ | ✗[a] | ✓[b] | Suboptimal convergence rate (see (Tyurin and Richtárik, 2023)). |
| FedPAGE (Zhao et al., 2021b) | ✗ | ✗[a] | ✗ | Suboptimal oracle complexity $\mathcal{O}\left(\frac{\sigma^2}{\varepsilon^2}\right)$. |
| FRECON (Zhao et al., 2021a) | ✗ | ✓ | ✓ | — |
| FedAvg (McMahan et al., 2017; Karimireddy et al., 2020b) | ✗ | ✓ | ✗ | Bounded gradients (dissimilarity) assumption of $f_i$. |
| SCAFFOLD (Karimireddy et al., 2020b) | ✗ | ✓ | ✗ | Suboptimal convergence rate[e]. |
| MIME[c] (Karimireddy et al., 2020a) | ✗[d] | ✓ | ✗ | Calculates full gradient. Bounded gradients (dissimilarity) assumption of $f_i$. Suboptimal oracle compl. $\mathcal{O}\left(1/\varepsilon^{3/2}\right)$ in the setting (2). |
| CE-LSGD (for Partial Participation)[c] (Patel et al., 2022) (concurrent work) | ✓ | ✓ | ✗ | Bounded gradients (dissimilarity) assumption of $f_i$. Suboptimal oracle compl. $\mathcal{O}\left(1/\varepsilon^{3/2}\right)$ in the setting (2). |
| DASHA (Tyurin and Richtárik, 2023) | ✓ or ✗ | ✗ or ✓ | ✓ ✓ | — |
| DASHA-PP (new) | ✓ | ✓ | ✓ | — |

[a] MARINA and FedPAGE, with a small probability, require the participation of all nodes so that they can not support partial participation from Section 2.2. Moreover, these methods provide suboptimal oracle complexities.

[b] On average, MARINA provides the compressed communication mechanism with complexity $\mathcal{O}\left(\frac{\omega}{\sqrt{n}\varepsilon}\right)$. However, with a small probability, this method sends non-compressed vectors.

[c] Note that MIME and CE-LSGD can not be directly compared with DASHA-PP because MIME and CE-LSGD consider the online version of the problem (1), and require more strict assumptions.

[d] Although MIME obtains the convergence rate $\mathcal{O}\left(\frac{1}{\varepsilon^{3/2}}\right)$ of a variance reduced method, it requires the calculation of the full (exact) gradients.

[e] It can be seen when $\sigma^2 = 0$. Consider the $s$-nice sampling of the nodes, then SCAFFOLD requires $\mathcal{O}\left(n^{3/2}/\varepsilon s^{3/2}\right)$ communication rounds to get $\varepsilon$-solution, while DASHA-PP requires $\mathcal{O}\left(\sqrt{n}/\varepsilon s\right)$ communication rounds (see Theorem 4 with $\omega = 0$, $b = p_a/2 - p_a$, and $p_a = \frac{s}{n}$).

## 2.2 Nodes Partial Participation Assumptions

We now try to formalize the notion of partial participation. Let us assume that we have $n$ events $\{i^{\text{th}}$ node is *participating*$\}$ with the following properties.

**Assumption 8.** *The partial participation of nodes has the following distribution: exists constants $p_a \in (0, 1]$ and $p_{aa} \in [0, 1]$, such that*

1.
$$\mathbf{Prob}\left(i^{\text{th}} \text{ node is } participating\right) = p_a \quad \forall i \in [n],$$

2.
$$\mathbf{Prob}\left(i^{\text{th}} \text{ and } j^{\text{th}} \text{ nodes are } participating\right) = p_{aa} \quad \forall i \neq j \in [n].$$

3.
$$p_{aa} \leq p_a^2, \tag{5}$$

*and these events from different communication rounds are independent.*

We are not fighting for the full generality and believe that more complex sampling strategies can be considered in the analysis. For simplicity, we settle upon Assumption 8. Standard partial participation strategies, including $s$–nice sampling, where the server chooses uniformly $s$ nodes without replacement ($p_a = s/n$ and $p_{aa} = s(s-1)/n(n-1)$), and independent participation, where each

Table 2: Summary of methods that solve the problem (1) in the finite-sum setting (2). Abbr.: *VR* (Variance Reduction) = Does a method have the optimal oracle complexity $\mathcal{O}\left(m + \frac{\sqrt{m}}{\varepsilon}\right)$? *PP* and *CC* are defined in Table 1.

| Method | VR | PP | CC | Limitations |
|---|---|---|---|---|
| SPIDER, PAGE (Fang et al., 2018; Li et al., 2021a) | ✓ | ✗ | ✗ | — |
| MARINA (Gorbunov et al., 2021) | ✓ | ✗[(a)] | ✓[(b)] | Suboptimal convergence rate (see (Tyurin and Richtárik, 2023)). |
| ZeroSARAH (Li et al., 2021b) | ✓ | ✓ | ✗ | Only homogeneous regime, i.e., the functions $f_i$ are equal. |
| FedPAGE (Zhao et al., 2021b) | ✗ | ✗[(a)] | ✗ | Suboptimal oracle complexity $\mathcal{O}\left(\frac{m}{\varepsilon}\right)$. |
| DASHA (Tyurin and Richtárik, 2023) | ✓ | ✗ | ✓ | — |
| DASHA-PP (new) | ✓ | ✓ | ✓ | — |

[(a), (b)] : see Table 1.

node independently participates with probability $p_a$ (due to independence, we have $p_{aa} = p_a^2$), satisfy Assumption 8. In the literature, $s$–nice sampling is one of the most popular strategies (Zhao et al., 2021a; Richtárik et al., 2021; Reddi et al., 2020; Konečný et al., 2016).

## 3 Motivation and Related Work

The main goal of our paper is to develop a method for the nonconvex distributed optimization that will include three key features: variance reduction of stochastic gradients, compressed communication, and partial participation. We now provide an overview of the literature (see also Table 1 and Table 2).

**1. Variance reduction of stochastic gradients**
It is important to consider finite-sum (2) and stochastic (3) settings because, in machine learning tasks, either the number of local functions $m$ is huge or the functions $f_i$ is an expectation of a stochastic function due to the batch normalization (Ioffe and Szegedy, 2015) or random augmentation (Goodfellow et al., 2016), and it is infeasible to calculate the full gradients analytically. Let us recall the results from the nondistributed optimization. In the gradient setting, the optimal oracle complexity is $\mathcal{O}\left(1/\varepsilon\right)$, achieved by the vanilla gradient descent (GD) (Carmon et al., 2020; Nesterov, 2018). In the finite-sum setting and stochastic settings, the optimal oracle complexities are $\mathcal{O}\left(m + \frac{\sqrt{m}}{\varepsilon}\right)$ and $\mathcal{O}\left(\frac{\sigma^2}{\varepsilon} + \frac{\sigma}{\varepsilon^{3/2}}\right)$ (Fang et al., 2018; Li et al., 2021a; Arjevani et al., 2019), accordingly, achieved by methods SPIDER, SARAH, PAGE, and STORM from (Fang et al., 2018; Nguyen et al., 2017; Li et al., 2021a; Cutkosky and Orabona, 2019).

**2. Compressed communication**
In distributed optimization (Ramesh et al., 2021; Xu et al., 2021), lossy communication compression can be a powerful tool to increase the communication speed between the nodes and the server. Different types of compressors are considered in the literature, including unbiased compressors (Alistarh et al., 2017; Beznosikov et al., 2020; Szlendak et al., 2021), contractive (biased) compressors (Richtárik et al., 2021), 3PC compressors (Richtárik et al., 2022). We will focus on unbiased compressors because methods DASHA and MARINA (Tyurin and Richtárik, 2023; Szlendak et al., 2021; Gorbunov et al., 2021) that employ unbiased compressors provide the current theoretical state-of-the-art (SOTA) communication complexities.

Many methods analyzed optimization methods with the unbiased compressors (Alistarh et al., 2017; Mishchenko et al., 2019; Horváth et al., 2019b; Gorbunov et al., 2021; Tyurin and Richtárik, 2023). In the gradient setting, the methods MARINA and DASHA by Gorbunov et al. (2021) and Tyurin and Richtárik (2023) establish the current SOTA communication complexity, each method needs $\frac{1 + \omega/\sqrt{n}}{\varepsilon}$ communication rounds to get an $\varepsilon$–solution. In the finite-sum and stochastic settings, the current

SOTA communication complexity is attained by the DASHA method, while maintaining the optimal oracle complexities $\mathcal{O}\left(m + \frac{\sqrt{m}}{\varepsilon\sqrt{n}}\right)$ and $\mathcal{O}\left(\frac{\sigma^2}{\varepsilon n} + \frac{\sigma}{\varepsilon^{3/2}n}\right)$ per node.

## 3. Partial participation

From the beginning of federated learning era, the partial participation has been considered to be the essential feature of distributed optimization methods (McMahan et al., 2017; Konečný et al., 2016; Kairouz et al., 2021). However, previously proposed methods have limitations: i) methods MARINA and FedPAGE from (Gorbunov et al., 2021; Zhao et al., 2021b) still require synchronization of all nodes with a small probability. ii) in the stochastic settings, methods FedAvg, SCAFFOLD, and FRECON with the partial participation mechanism (McMahan et al., 2017; Karimireddy et al., 2020b; Zhao et al., 2021a) provide results without variance reduction techniques from (Fang et al., 2018; Li et al., 2021a; Cutkosky and Orabona, 2019) and, therefore, get suboptimal oracle complexities. Note that FRECON and DASHA reduce the variance *only from compressors* (in the partial participation and stochastic setting). iii) in the finite-sum setting, the ZeroSARAH method by Li et al. (2021b) focuses on the homogeneous regime only (the functions $f_i$ are equal). iv) The MIME method by Karimireddy et al. (2020a) and the CE-LSGD method (for Partial Participation) by the concurrent paper (Patel et al., 2022) consider the online version of the problem (1). Therefore, MIME and CE-LSGD (for Partial Participation) require stricter assumptions, including the bounded inter-client gradient variance assumption. In the finite-sum setting (2), MIME and CE-LSGD obtain a suboptimal oracle complexity $\mathcal{O}\left(1/\varepsilon^{3/2}\right)$ while, in the full participation setting, it is possible to get the complexity $\mathcal{O}\left(1/\varepsilon\right)$.

# 4 Contributions

We propose a new method DASHA-PP for the nonconvex distributed optimization.

● As far as we know, this is the first method that includes three key ingredients of federated learning methods: *variance reduction of stochastic gradients, compressed communication, and partial participation.*

● Moreover, this is the first method that combines *variance reduction of stochastic gradients and partial participation* flawlessly: i) it gets the optimal oracle complexity ii) does not require the participation of all nodes iii) does not require the bounded gradients assumption of the functions $f_i$.

● We prove convergence rates and show that this method has *the optimal oracle complexity and the state-of-the-art communication complexity in the partial participation setting.* Moreover, in our work, we observe a nontrivial side-effect from mixing the variance reduction of stochastic gradients and partial participation. It is a general problem not related to our methods or analysis that we discuss in Section C.

● In Section A, we present experiments where we validate our theory and compare our new methods to previous ones.

# 5 Algorithm Description and Main Challenges Towards Partial Participation

We now present DASHA-PP (see Algorithm 1), a family of methods to solve the optimization problem (1). When we started investigating the problem, we took DASHA as a baseline method for two reasons: the family of algorithms DASHA provides the current state-of-the-art communication complexities in the *non-partial participation* setting, and, unlike MARINA, it does not send non-compressed gradients and does not synchronize all nodes. Let us briefly discuss the main idea of DASHA, its problem in the *partial participation* setting, and why the refinement of DASHA is not an exercise.

In fact, the original DASHA method supports the partial participation of nodes *in the gradient setting*. Since the nodes only do the following steps (see full algorithm in Algorithm 6):

$$g_i^{t+1} = g_i^t + \mathcal{C}_i \left( \nabla f_i(x^{t+1}) - (1-a)\nabla f_i(x^t) - a g_i^t \right). \tag{6}$$

---

**Algorithm 1** DASHA-PP

---

1: **Input:** starting point $x^0 \in \mathbb{R}^d$, stepsize $\gamma > 0$, momentum $a \in (0,1]$, momentum $b \in (0,1]$, probability $p_{\text{page}} \in (0,1]$ (only in DASHA-PP-PAGE), batch size $B$ (only in DASHA-PP-PAGE, DASHA-PP-FINITE-MVR and DASHA-PP-MVR), probability $p_{\text{a}} \in (0,1]$ that a node is *participating*[a], number of iterations $T \geq 1$
2: Initialize $g_i^0 \in \mathbb{R}^d$, $h_i^0 \in \mathbb{R}^d$ on the nodes and $g^0 = \frac{1}{n} \sum_{i=1}^n g_i^0$ on the server
3: Initialize $h_{ij}^0 \in \mathbb{R}^d$ on the nodes and take $h_i^0 = \frac{1}{m} \sum_{j=1}^m h_{ij}^0$ (only in DASHA-PP-FINITE-MVR)
4: **for** $t = 0, 1, \ldots, T-1$ **do**
5:     $x^{t+1} = x^t - \gamma g^t$
6:     Broadcast $x^{t+1}, x^t$ to all *participating*[a] nodes
7:     **for** $i = 1, \ldots, n$ in parallel **do**
8:       **if** $i^{\text{th}}$ node is *participating*[a] **then**
9:         Calculate $k_i^{t+1}$ using Algorithm 2, 3, 4 or 5
10:         $h_i^{t+1} = h_i^t + \frac{1}{p_{\text{a}}} k_i^{t+1}$
11:         $m_i^{t+1} = \mathcal{C}_i \left( \frac{1}{p_{\text{a}}} k_i^{t+1} - \frac{a}{p_{\text{a}}} \left( g_i^t - h_i^t \right) \right)$
12:         $g_i^{t+1} = g_i^t + m_i^{t+1}$
13:         Send $m_i^{t+1}$ to the server
14:       **else**
15:         $h_{ij}^{t+1} = h_{ij}^t$ (only in DASHA-PP-FINITE-MVR)
16:         $h_i^{t+1} = h_i^t, \quad g_i^{t+1} = g_i^t, \quad m_i^{t+1} = 0$
17:       **end if**
18:     **end for**
19:     $g^{t+1} = g^t + \frac{1}{n} \sum_{i=1}^n m_i^{t+1}$
20: **end for**
21: **Output:** $\hat{x}^T$ chosen uniformly at random from $\{x^t\}_{k=0}^{T-1}$
    [a]: For the formal description see Section 2.2.

---

**Algorithm 2** Calculate $k_i^{t+1}$ for DASHA-PP in the gradient setting. See line 9 in Alg. 1

---

1: $k_i^{t+1} = \nabla f_i(x^{t+1}) - \nabla f_i(x^t) - b \left( h_i^t - \nabla f_i(x^t) \right)$

---

**Algorithm 3** Calculate $k_i^{t+1}$ for DASHA-PP-PAGE in the finite-sum setting. See line 9 in Alg. 1

---

1: Generate a random set $I_i^t$ of size $B$ from $[m]$ *with replacement*
2: $k_i^{t+1} = \begin{cases} \nabla f_i(x^{t+1}) - \nabla f_i(x^t) - \frac{b}{p_{\text{page}}} \left( h_i^t - \nabla f_i(x^t) \right), \\ \quad \text{with probability } p_{\text{page}} \text{ on all } \textit{participating} \text{ nodes}, \\ \frac{1}{B} \sum_{j \in I_i^t} \left( \nabla f_{ij}(x^{t+1}) - \nabla f_{ij}(x^t) \right), \\ \quad \text{with probability } 1 - p_{\text{page}} \text{ on all } \textit{participating} \text{ nodes} \end{cases}$

---

**Algorithm 4** Calc. $k_i^{t+1}$ for DASHA-PP-FINITE-MVR in the finite-sum setting. See line 9 in Alg. 1

---

1: Generate a random set $I_i^t$ of size $B$ from $[m]$ *without replacement*
2: $k_{ij}^{t+1} = \begin{cases} \frac{m}{B} \left( \nabla f_{ij}(x^{t+1}) - \nabla f_{ij}(x^t) - b \left( h_{ij}^t - \nabla f_{ij}(x^t) \right) \right), & j \in I_i^t, \\ 0, & j \notin I_i^t \end{cases}$
3: $h_{ij}^{t+1} = h_{ij}^t + \frac{1}{p_{\text{a}}} k_{ij}^{t+1}$
4: $k_i^{t+1} = \frac{1}{m} \sum_{j=1}^m k_{ij}^{t+1}$

---

**Algorithm 5** Calculate $k_i^{t+1}$ for DASHA-PP-MVR in the stochastic setting. See line 9 in Alg. 1

---

1: Generate i.i.d. samples $\{\xi_{ij}^{t+1}\}_{j=1}^B$ of size $B$ from $\mathcal{D}_i$.
2: $k_i^{t+1} = \frac{1}{B} \sum_{j=1}^B \nabla f_i(x^{t+1}; \xi_{ij}^{t+1}) - \frac{1}{B} \sum_{j=1}^B \nabla f_i(x^t; \xi_{ij}^{t+1}) - b \left( h_i^t - \frac{1}{B} \sum_{j=1}^B \nabla f_i(x^t; \xi_{ij}^{t+1}) \right)$

---

The partial participation mechanism (independent participation from Section 2.2) can be easily implemented here if we temporally redefine the compressor and use another one[3] instead:

$$\mathcal{C}_i^{p_a} := \begin{cases} \frac{1}{p_a}\mathcal{C}_i, \text{ w.p. } p_a, \\ 0, \quad \text{w.p. } 1 - p_a. \end{cases} \quad \overset{(6)}{\Rightarrow} g_i^{t+1} = \begin{cases} g_i^t + \frac{1}{p_a}\mathcal{C}_i\left(\nabla f_i(x^{t+1}) - (1-a)\nabla f_i(x^t) - ag_i^t\right), \text{ w.p. } p_a \\ g_i^t, \hspace{6.2cm} \text{w.p. } 1 - p_a. \end{cases}$$

With probability $1 - p_a$, a node does not update $g_i^t$ and does not send anything to the server. The main observation is that we can do this trick since $g_i^{t+1}$ depends only on the vectors $x^{t+1}$, $x^t$, and $g_i^t$. The points $x^{t+1}$ and $x^t$ are only available in a node only during its participation.

However, we focus our attention on partial participation *in the finite-sum and stochastic settings*. Consider the nodes' steps in DASHA-MVR (Tyurin and Richtárik, 2023) (see Algorithm 7) that is designed for the stochastic setting:

$$h_i^{t+1} = \nabla f_i(x^{t+1}; \xi_i^{t+1}) + (1-b)(h_i^t - \nabla f_i(x^t; \xi_i^{t+1})), \tag{7}$$

$$g_i^{t+1} = g_i^t + \mathcal{C}_i\left(h_i^{t+1} - h_i^t - a\left(g_i^t - h_i^t\right)\right). \tag{8}$$

Now we have two sequences $h_i^t$ and $g_i^t$. Even if we use the same trick for (8), we still have to update (7) in every iteration of the algorithm since $g_i^{t+1}$ additionally depends on $h_i^{t+1}$ and $h_i^t$. In other words, if a node does not update $g_i^t$ and does not send anything to the server, it still has to update $h_i^t$, what is impossible without the points $x^{t+1}$ and $x^t$. One of the main challenges was to "guess" how to generalize (7) and (8) to the partial participation setting. We now provide a solution (DASHA-PP-MVR with the batch size $B = 1$):

$$h_i^{t+1} = h_i^t + \frac{1}{p_a}k_i^{t+1}, \; k_i^{t+1} = \nabla f_i(x^{t+1}; \xi_i^{t+1}) - \nabla f_i(x^t; \xi_i^{t+1}) - b\left(h_i^t - \nabla f_i(x^t; \xi_i^{t+1})\right),$$

$$g_i^{t+1} = g_i^t + \mathcal{C}_i\left(\frac{1}{p_a}k_i^{t+1} - \frac{a}{p_a}\left(g_i^t - h_i^t\right)\right) \text{ with probability } p_a, \tag{9}$$

and $h_i^{t+1} = h_i^t$, $g_i^{t+1} = g_i^t$ with probability $1 - p_a$.

Now both control variables $g_i^t$ and $h_i^t$ do not change with the probability $1 - p_a$. When the $i^{th}$ node participates, the update rules of $g_i^{t+1}$ and $h_i^{t+1}$ in (9) were adapted to make the proof work. When $p_a = 1$ (no partial participation), the update rules from (9) reduce to (7) and (8).

The theoretical analysis of the new algorithm became more complicated: unlike (7) and (8), the control variables $h_i^{t+1}$ and $g_i^{t+1}$ in (9) (see also main Algorithm 1) are coupled by the randomness from the partial participation. Going deeper into details, for instance, one can compare Lemma I.2 from (Tyurin and Richtárik, 2023) and Lemma 5, which both bound $\left\|g_i^{t+1} - h_i^{t+1}\right\|^2$. The former lemma does not use the knowledge about the update rules of $h_i^{t+1}$, works with one expectation $\mathrm{E}_{\mathcal{C}}\left[\cdot\right]$, uses only (4), (15), and (16). The latter lemma additionally requires and uses the structure of the update rule of $h_i^{t+1}$ (the structure is very important in the lemma since the control variables $h_i^{t+1}$ and $g_i^{t+1}$ are coupled), surgically copes with the expectations $\mathrm{E}_{\mathcal{C}}\left[\cdot\right]$ and $\mathrm{E}_{p_a}\left[\cdot\right]$ (for instance, it is not trivial in each order one should apply the expectations), and uses the sampling lemma (Lemma 1). The same reasoning applies to other parts of the analysis and the finite-sum setting: the generalization of the previous algorithm and the additional randomness from the partial participation required us to rethink the previous proofs.

At the first reading of the proofs, we suggest the reader follow the proof of Theorem 2 in the gradient setting (DASHA-PP), which takes a small part of the paper. Although the appendix seems to be dense and large, the size is justified by the fact that we consider four different sub-algorithms, DASHA-PP, DASHA-PP-PAGE, DASHA-PP-FINITE-MVR, and DASHA-PP-MVR, and also PŁ-condition (The theory is designed so that the proofs do not repeat steps of each other and use one framework).

# 6 Theorems

We now present the convergence rates theorems of DASHA-PP in different settings. We will compare the theorems with the results of the current state-of-the-art methods, MARINA and DASHA, that work

---

[3]If $\mathcal{C}_i \in \mathbb{U}\left(\omega\right)$, then $\mathcal{C}_i^{p_a} \in \mathbb{U}\left(\omega + 1/p_a - 1\right)$.

in the full participation setting. Suppose that MARINA or DASHA converges to $\varepsilon$-solution after $T$ communication rounds. Then, ideally, we would expect the convergence of the new algorithms to $\varepsilon$-solution after up to $T/p_{\mathrm{a}}$ communication rounds due to the partial participation constraints[4]. The detailed analysis of the algorithms under Polyak-Łojasiewicz condition we provide in Section F. Let us define $\Delta_0 := f(x^0) - f^*$.

## 6.1 Gradient Setting

**Theorem 2.** *Suppose that Assumptions 1, 2, 3, 7 and 8 hold. Let us take $a = \frac{p_{\mathrm{a}}}{2\omega+1}$, $b = \frac{p_{\mathrm{a}}}{2-p_{\mathrm{a}}}$,*

$$\gamma \leq \left( L + \left[ \frac{48\omega(2\omega+1)}{np_{\mathrm{a}}^2} + \frac{16}{np_{\mathrm{a}}^2}\left(1 - \frac{p_{\mathrm{aa}}}{p_{\mathrm{a}}}\right) \right]^{1/2} \widehat{L} \right)^{-1},$$

*and $g_i^0 = h_i^0 = \nabla f_i(x^0)$ for all $i \in [n]$ in Algorithm 1 (DASHA-PP), then $\mathrm{E}\left[\left\|\nabla f(\widehat{x}^T)\right\|^2\right] \leq \frac{2\Delta_0}{\gamma T}$.*

Let us recall the convergence rate of MARINA or DASHA, the number of communication rounds to get $\varepsilon$-solution equals $\mathcal{O}\left(\frac{\Delta_0}{\varepsilon}\left[L + \frac{\omega}{\sqrt{n}}\widehat{L}\right]\right)$, while the rate of DASHA-PP equals $\mathcal{O}\left(\frac{\Delta_0}{\varepsilon}\left[L + \frac{\omega+1}{p_{\mathrm{a}}\sqrt{n}}\widehat{L}\right]\right)$. Up to Lipschitz constants factors, we get the degeneration up to $1/p_{\mathrm{a}}$ factor due to the partial participation. This is the expected result since each worker sends useful information only with the probability $p_{\mathrm{a}}$.

## 6.2 Finite-Sum Setting

**Theorem 3.** *Suppose that Assumptions 1, 2, 3, 4, 7, and 8 hold. Let us take $a = \frac{p_{\mathrm{a}}}{2\omega+1}$, $b = \frac{p_{page}p_{\mathrm{a}}}{2-p_{\mathrm{a}}}$, probability $p_{page} \in (0,1]$,*

$$\gamma \leq \left( L + \left[ \frac{48\omega(2\omega+1)}{np_{\mathrm{a}}^2}\left(\widehat{L}^2 + \frac{(1-p_{page})L_{\max}^2}{B}\right) + \frac{16}{np_{\mathrm{a}}^2 p_{page}}\left(\left(1 - \frac{p_{\mathrm{aa}}}{p_{\mathrm{a}}}\right)\widehat{L}^2 + \frac{(1-p_{page})L_{\max}^2}{B}\right) \right]^{1/2} \right)^{-1}$$

*and $g_i^0 = h_i^0 = \nabla f_i(x^0)$ for all $i \in [n]$ in Algorithm 1 (DASHA-PP-PAGE) then $\mathrm{E}\left[\left\|\nabla f(\widehat{x}^T)\right\|^2\right] \leq \frac{2\Delta_0}{\gamma T}$.*

We now choose $p_{page}$ to balance heavy full gradient and light mini-batch calculations. Let us define $\mathbb{1}_{p_{\mathrm{a}}} := \sqrt{1 - \frac{p_{\mathrm{aa}}}{p_{\mathrm{a}}}} \in [0,1]$. Note that if $p_{\mathrm{a}} = 1$ then $p_{\mathrm{aa}} = 1$ and $\mathbb{1}_{p_{\mathrm{a}}} = 0$.

**Corollary 1.** *Let the assumptions from Theorem 3 hold and $p_{page} = B/(m+B)$. Then DASHA-PP-PAGE needs*

$$T := \mathcal{O}\left(\frac{\Delta_0}{\varepsilon}\left[L + \frac{\omega}{p_{\mathrm{a}}\sqrt{n}}\left(\widehat{L} + \frac{L_{\max}}{\sqrt{B}}\right) + \frac{1}{p_{\mathrm{a}}}\sqrt{\frac{m}{n}}\left(\frac{\mathbb{1}_{p_{\mathrm{a}}}\widehat{L}}{\sqrt{B}} + \frac{L_{\max}}{B}\right)\right]\right) \tag{10}$$

*communication rounds to get an $\varepsilon$-solution and the expected number of gradient calculations per node equals $\mathcal{O}(m + BT)$.*

The convergence rate the rate of the current state-of-the-art method DASHA-PAGE without partial participation equals $\mathcal{O}\left(\frac{\Delta_0}{\varepsilon}\left[L + \frac{\omega}{\sqrt{n}}\left(\widehat{L} + \frac{L_{\max}}{\sqrt{B}}\right) + \sqrt{\frac{m}{n}}\frac{L_{\max}}{B}\right]\right)$. Let us closer compare it with (10). As expected, we see that the second term w.r.t. $\omega$ degenerates up to $1/p_{\mathrm{a}}$. Surprisingly, the third term w.r.t. $\sqrt{m/n}$ can degenerate up to $\sqrt{B}/p_{\mathrm{a}}$ when $\widehat{L} \approx L_{\max}$. Hence, in order to keep degeneration up to $1/p_{\mathrm{a}}$, one should take the batch size $B = \mathcal{O}\left(L_{\max}^2/\widehat{L}^2\right)$. This interesting effect we analyze separately in Section C. The fact that the degeneration is up to $1/p_{\mathrm{a}}$ we check numerically in Section A.

In the following corollary, we consider $\mathrm{Rand}K$ compressors[5] (see Definition 5) and show that with the particular choice of parameters, up to the Lipschitz constants factors, DASHA-PP-PAGE gets the

---

[4]We check this numerically in Section A.

[5]The choice of the compressor is driven by simplicity, and the following analysis can be used for other unbiased compressors.

optimal oracle complexity and SOTA communication complexity. Indeed, comparing the following result with (Tyurin and Richtárik, 2023, Corollary 6.6), one can see that we get the degeneration up to $1/p_a$ factor, which is expected in the partial participation setting. Note that the complexities improve with the number of workers $n$.

**Corollary 2.** *Suppose that assumptions of Corollary 1 hold, $B \leq \min\left\{\frac{1}{p_a}\sqrt{\frac{m}{n}}, \frac{L_{\max}^2}{\mathbb{1}_{p_a}^2 \widehat{L}^2}\right\}$[6], and we use the unbiased compressor RandK with $K = \Theta\left(Bd/\sqrt{m}\right)$. Then the communication complexity of Algorithm 1 is $\mathcal{O}\left(d + \frac{L_{\max}\Delta_0 d}{p_a \varepsilon \sqrt{n}}\right)$, and the expected number of gradient calculations per node equals $\mathcal{O}\left(m + \frac{L_{\max}\Delta_0\sqrt{m}}{p_a \varepsilon \sqrt{n}}\right)$.*

The convergence rate of DASHA-PP-FINITE-MVR is provided in Section E.5.

## 6.3 Stochastic Setting

We define $h^t := \frac{1}{n}\sum_{i=1}^n h_i^t$.

**Theorem 4.** *Suppose that Assumptions 1, 2, 3, 5, 6, 7 and 8 hold. Let us take $a = \frac{p_a}{2\omega+1}$, $b \in \left(0, \frac{p_a}{2-p_a}\right]$, $\gamma \leq \left(L + \left[\frac{48\omega(2\omega+1)}{np_a^2}\left(\widehat{L}^2 + \frac{(1-b)^2 L_\sigma^2}{B}\right) + \frac{12}{np_a b}\left(\left(1 - \frac{p_{aa}}{p_a}\right)\widehat{L}^2 + \frac{(1-b)^2 L_\sigma^2}{B}\right)\right]^{1/2}\right)^{-1}$, and $g_i^0 = h_i^0$ for all $i \in [n]$ in Algorithm 1 (DASHA-PP-MVR). Then*

$$\mathrm{E}\left[\left\|\nabla f(\widehat{x}^T)\right\|^2\right] \leq \frac{1}{T}\left[\frac{2\Delta_0}{\gamma} + \frac{2}{b}\left\|h^0 - \nabla f(x^0)\right\|^2 + \left(\frac{32b\omega(2\omega+1)}{np_a^2} + \frac{4\left(1 - \frac{p_{aa}}{p_a}\right)}{np_a}\right)\left(\frac{1}{n}\sum_{i=1}^n \left\|h_i^0 - \nabla f_i(x^0)\right\|^2\right)\right]$$

$$+ \left(\frac{48b^2\omega(2\omega+1)}{p_a^2} + \frac{12b}{p_a}\right)\frac{\sigma^2}{nB}.$$

In the next corollary, we choose momentum $b$ and initialize vectors $h_i^0$ to get $\varepsilon$-solution.

**Corollary 3.** *Suppose that assumptions from Theorem 4 hold, momentum $b = \Theta\left(\min\left\{\frac{p_a}{\omega}\sqrt{\frac{n\varepsilon B}{\sigma^2}}, \frac{p_a n\varepsilon B}{\sigma^2}\right\}\right)$, $\frac{\sigma^2}{n\varepsilon B} \geq 1$, and $h_i^0 = \frac{1}{B_{\mathrm{init}}}\sum_{k=1}^{B_{\mathrm{init}}}\nabla f_i(x^0; \xi_{ik}^0)$ for all $i \in [n]$, and batch size $B_{\mathrm{init}} = \Theta\left(\frac{\sqrt{p_a}B}{b}\right)$, then Algorithm 1 (DASHA-PP-MVR) needs*

$$T := \mathcal{O}\left(\frac{\Delta_0}{\varepsilon}\left[L + \frac{\omega}{p_a\sqrt{n}}\left(\widehat{L} + \frac{L_\sigma}{\sqrt{B}}\right) + \frac{\sigma}{p_a\sqrt{\varepsilon}n}\left(\frac{\mathbb{1}_{p_a}\widehat{L}}{\sqrt{B}} + \frac{L_\sigma}{B}\right)\right] + \frac{\sigma^2}{\sqrt{p_a}n\varepsilon B}\right)$$

*communication rounds to get an $\varepsilon$-solution and the number of stochastic gradient calculations per node equals $\mathcal{O}(B_{\mathrm{init}} + BT)$.*

The convergence rate of the DASHA-SYNC-MVR, the state-of-the-art method without partial participation, equals $\mathcal{O}\left(\frac{\Delta_0}{\varepsilon}\left[L + \frac{\omega}{\sqrt{n}}\left(\widehat{L} + \frac{L_\sigma}{\sqrt{B}}\right) + \frac{\sigma}{\sqrt{\varepsilon}n}\frac{L_\sigma}{B}\right] + \frac{\sigma^2}{n\varepsilon B}\right)$. Similar to Section 6.2, we see that in the regimes when $\widehat{L} \approx L_\sigma$ the third term w.r.t. $1/\varepsilon^{3/2}$ can degenerate up to $\sqrt{B}/p_a$. However, if we take $B = \mathcal{O}\left(L_\sigma^2/\widehat{L}^2\right)$, then the degeneration of the third term will be up to $1/p_a$. This effect we analyze in Section C. The fact that the degeneration is up to $1/p_a$ we check numerically in Section A.

In the following corollary, we consider RandK compressors (see Definition 5) and show that with the particular choice of parameters, up to the Lipschitz constants factors, DASHA-PP-MVR gets the optimal oracle complexity and SOTA communication complexity of DASHA-SYNC-MVR method. Indeed, comparing the following result with (Tyurin and Richtárik, 2023, Corollary 6.9), one can see that we get the degeneration up to $1/p_a$ factor, which is expected in the partial participation setting. Note that the complexities improve with the number of workers $n$.

**Corollary 4.** *Suppose that assumptions of Corollary 3 hold, batch size $B \leq \min\left\{\frac{\sigma}{p_a\sqrt{\varepsilon}n}, \frac{L_\sigma^2}{\mathbb{1}_{p_a}^2 \widehat{L}^2}\right\}$, we take RandK compressors with $K = \Theta\left(\frac{Bd\sqrt{\varepsilon n}}{\sigma}\right)$. Then the communication complexity equals*

---

[6]If $\mathbb{1}_{p_a} = 0$, then $\frac{L_\sigma^2}{\mathbb{1}_{p_a}^2 \widehat{L}^2} = +\infty$

$\mathcal{O}\left(\frac{d\sigma}{\sqrt{p_\mathrm{a}}\sqrt{n}\varepsilon} + \frac{L_\sigma\Delta_0 d}{p_\mathrm{a}\sqrt{n}\varepsilon}\right)$, *and the expected number of stochastic gradient calculations per node equals* $\mathcal{O}\left(\frac{\sigma^2}{\sqrt{p_\mathrm{a}}n\varepsilon} + \frac{L_\sigma\Delta_0\sigma}{p_\mathrm{a}\varepsilon^{3/2}n}\right)$.

We are aware that the initial batch size $B_\mathrm{init}$ can be suboptimal w.r.t. $\omega$ in DASHA-PP-MVR in some regimes (see also (Tyurin and Richtárik, 2023)). This is a side effect of mixing the variance reduction of stochastic gradients and compression. However, Corollary 4 reveals that we can escape these regimes by choosing the parameter $K$ of Rand$K$ compressors in a particular way. To get the complete picture, we analyze the same phenomenon under PŁ condition (see Section F) and provide a new method DASHA-PP-SYNC-MVR (see Section G).

### Acknowledgements

This work of P. Richtárik and A. Tyurin was supported by the KAUST Baseline Research Scheme (KAUST BRF) and the KAUST Extreme Computing Research Center (KAUST ECRC), and the work of P. Richtárik was supported by the SDAIA-KAUST Center of Excellence in Data Science and Artificial Intelligence (SDAIA-KAUST AI).

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

# Contents

## A  Numerical Verification of Theoretical Dependencies

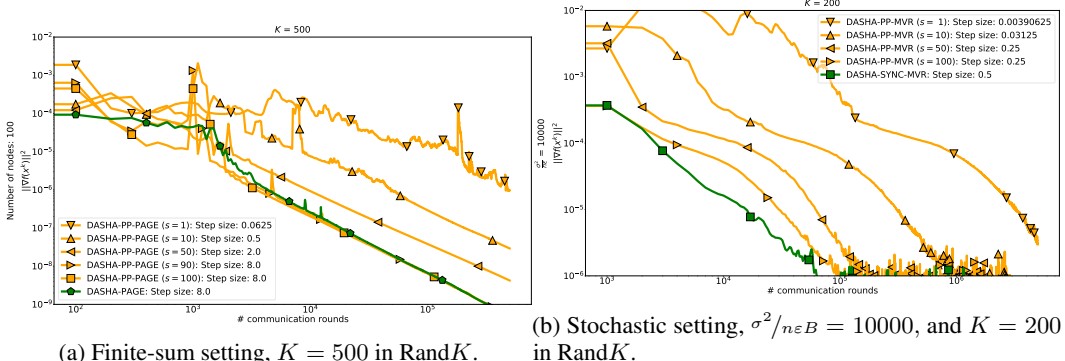

(a) Finite-sum setting, $K = 500$ in $\text{Rand}K$.

(b) Stochastic setting, $\sigma^2/n\varepsilon B = 10000$, and $K = 200$ in $\text{Rand}K$.

Figure 1: Classification task with the *real-sim* dataset.

Our main goal is to verify the dependeces from the theory. We compare DASHA-PP with DASHA. Clearly, DASHA-PP can not generally perform better than DASHA. In different settings, we verify that the bigger $p_a$, the closer DASHA-PP is to DASHA, i.e., DASHA-PP converges no slower than $1/p_a$ times.

In all experiments, we take the *real-sim* dataset with dimension $d = 20{,}958$ and the number of samples equals 72,309 from LIBSVM datasets (Chang and Lin, 2011) (under the 3-clause BSD license), and randomly split the dataset between $n = 100$ nodes equally, ignoring residual samples. In the finite-sum setting, we solve a classification problem with functions

$$f_i(x) := \frac{1}{m} \sum_{j=1}^{m} \left( 1 - \frac{1}{1 + \exp(y_{ij} a_{ij}^\top x)} \right)^2, \tag{11}$$

where $a_{ij} \in \mathbb{R}^d$ is the feature vector of a sample on the $i^{\text{th}}$ node, $y_{ij} \in \{-1, 1\}$ is the corresponding label, and $m$ is the number of samples on the $i^{\text{th}}$ node for all $i \in [n]$. In the stochastic setting, we consider functions

$$f_i(x_1, x_2) := \mathrm{E}_{j\sim[m]} \left[ -\log \left( \frac{\exp\left(a_{ij}^\top x_{y_{ij}}\right)}{\sum_{y\in\{1,2\}} \exp\left(a_{ij}^\top x_y\right)} \right) + \lambda \sum_{y\in\{1,2\}} \sum_{k=1}^{d} \frac{\{x_y\}_k^2}{1 + \{x_y\}_k^2} \right], \tag{12}$$

where $x_1, x_2 \in \mathbb{R}^d$, $\{\cdot\}_k$ is an indexing operation, $a_{ij} \in \mathbb{R}^d$ is a feature of a sample on the $i^{\text{th}}$ node, $y_{ij} \in \{1, 2\}$ is a corresponding label, $m$ is the number of samples located on the $i^{\text{th}}$ node, constant $\lambda = 0.001$ for all $i \in [n]$.

The code was written in Python 3.6.8 using PyTorch 1.9 (Paszke et al., 2019). A distributed environment was emulated on a machine with Intel(R) Xeon(R) Gold 6226R CPU @ 2.90GHz and 64 cores.

We use the standard setting in experiments[7] where all parameters except step sizes are taken as suggested in theory. Step sizes are finetuned from a set $\{2^i \mid i \in [-10, 10]\}$. We emulate the partial participation setting using $s$-nice sampling with the number of nodes $n = 100$. We consider the $\text{Rand}K$ compressor and take the batch size $B = 1$. We plot the relation between communication rounds and values of the norm of gradients at each communication round.

In the finite-sum (Figure 1a) and in the stochastic setting (Figure 1b), we see that the bigger probability $p_a = s/n$ to 1, the closer DASHA-PP to DASHA. Moreover, DASHA-PP with $s = 10$ and $s = 1$ converges approximately $\times 10$ ($= 1/p_a$) and $\times 100$ ($= 1/p_a$) times slower, accordingly. Our theory predicts such behavior.

---

[7]Code: https://github.com/mysteryresearcher/dasha-partial-participation

## A.1 Experiments in Partial Participation Setting

In this experiments, we compare our new algorithm DASHA-PP with previous baselines MARINA and FRECON in the partial participation setting. We consider MARINA and FRECON because they are the previous SOTA methods in the *partial participation setting with compression*. We investigate the same optimization problem and setup as in Section A of the paper. All methods use the Rand$K$ compressor in these experiments.

1. **Finite-Sum Setting.** We now consider the function from (11). In Figures 2 and 3, we compare all three methods in the finite-sum setting on two different datasets: *real-sim* and *MNIST*. The parameter $s$ is the number of clients participating in each round that are selected randomly using the $s$-nice sampling (server chooses uniformly $s$ nodes without replacement). We can see that DASHA-PP converges faster than MARINA. Since FRECON does not support variance reduction of stochastic gradients, it converges to less accurate solutions.

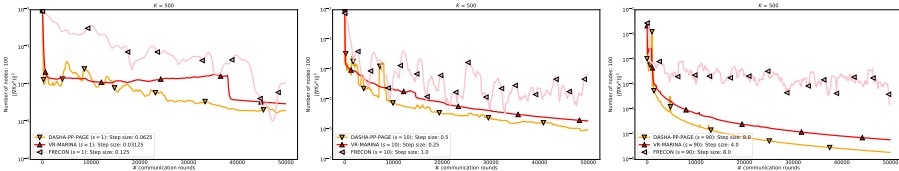

(a) 1 % of nodes participating   (b) 10 % of nodes participating   (c) 90 % of nodes participating

Figure 2: Classification task on *real-sim*

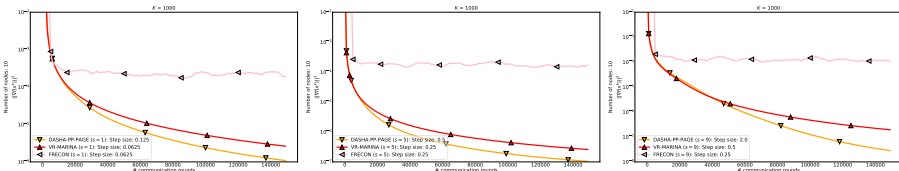

(a) 10 % of nodes participating   (b) 50 % of nodes participating   (c) 90 % of nodes participating

Figure 3: Classification task on *MNIST*

2. **Stochastic Setting.** In Figures 4 and 5, we consider the stochastic setting with the function from (11). We can see that DASHA-PP convergences to high accuracy solutions, unlike FRECON. Moreover, DASHA-PP improves the convergence rates of MARINA.

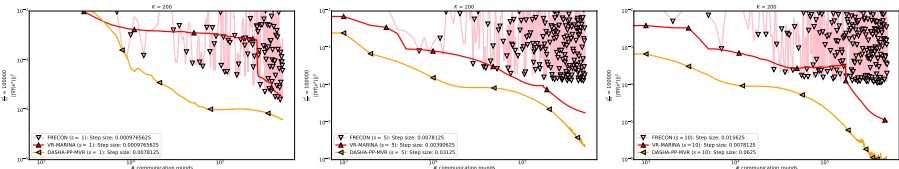

(a) 10 % of nodes participating   (b) 50 % of nodes participating   (c) 100 % of nodes participating

Figure 4: Classification task on *real-sim*

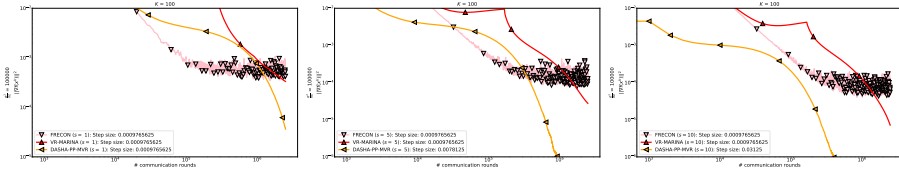

(a) 10 % of nodes participating   (b) 50 % of nodes participating   (c) 100 % of nodes participating

Figure 5: Classification task on *MNIST*

# B Original DASHA and DASHA-MVR Methods

To simplify the discussion and explanation from the main part, we present the algorithms from (Tyurin and Richtárik, 2023)

---

**Algorithm 6** DASHA

---

1: **Input:** starting point $x^0 \in \mathbb{R}^d$, stepsize $\gamma > 0$, momentum $a \in (0, 1]$, number of iterations $T \geq 1$

2: Initialize $g_i^0 \in \mathbb{R}^d$ on the nodes and $g^0 = \frac{1}{n} \sum_{i=1}^n g_i^0$ on the server
3: **for** $t = 0, 1, \ldots, T - 1$ **do**
4: $\quad x^{t+1} = x^t - \gamma g^t$
5: $\quad$ Broadcast $x^{t+1}$ and $x^t$
6: $\quad$ **for** $i = 1, \ldots, n$ in parallel **do**
7: $\quad\quad m_i^{t+1} = \mathcal{C}_i \left( \nabla f_i(x^{t+1}) - \nabla f_i(x^t) - a \left( g_i^t - \nabla f_i(x^t) \right) \right)$
8: $\quad\quad g_i^{t+1} = g_i^t + m_i^{t+1}$
9: $\quad\quad$ Send $m_i^{t+1}$ to the server
10: $\quad$ **end for**
11: $\quad g^{t+1} = g^t + \frac{1}{n} \sum_{i=1}^n m_i^{t+1}$
12: **end for**
13: **Output:** $\hat{x}^T$ chosen uniformly at random from $\{x^t\}_{k=0}^{T-1}$

---

---

**Algorithm 7** DASHA-MVR (with batch size $B = 1$)

---

1: **Input:** starting point $x^0 \in \mathbb{R}^d$, stepsize $\gamma > 0$, momentums $a, b \in (0, 1]$, number of iterations $T \geq 1$
2: Initialize $g_i^0 \in \mathbb{R}^d$ on the nodes and $g^0 = \frac{1}{n} \sum_{i=1}^n g_i^0$ on the server
3: **for** $t = 0, 1, \ldots, T - 1$ **do**
4: $\quad x^{t+1} = x^t - \gamma g^t$
5: $\quad$ Broadcast $x^{t+1}$ and $x^t$
6: $\quad$ **for** $i = 1, \ldots, n$ in parallel **do**
7: $\quad\quad h_i^{t+1} = \nabla f_i(x^{t+1}; \xi_i^{t+1}) + (1-b)(h_i^t - \nabla f_i(x^t; \xi_i^{t+1})), \quad \xi_i^{t+1} \sim \mathcal{D}_i$
8: $\quad\quad m_i^{t+1} = \mathcal{C}_i \left( h_i^{t+1} - h_i^t - a \left( g_i^t - h_i^t \right) \right)$
9: $\quad\quad g_i^{t+1} = g_i^t + m_i^{t+1}$
10: $\quad\quad$ Send $m_i^{t+1}$ to the server
11: $\quad$ **end for**
12: $\quad g^{t+1} = g^t + \frac{1}{n} \sum_{i=1}^n m_i^{t+1}$
13: **end for**
14: **Output:** $\hat{x}^T$ chosen uniformly at random from $\{x^t\}_{k=0}^{T-1}$

---

## C  Problem of Estimating the Mean in the Partial Participation Setting

We now provide the example to explain why the only choice of $B = \mathcal{O}\left(\min\left\{\frac{1}{p_\mathrm{a}}\sqrt{\frac{m}{n}}, \frac{L_{\max}^2}{\mathbb{1}_{p_\mathrm{a}}^2 \widehat{L}^2}\right\}\right)$ and
$B = \mathcal{O}\left(\min\left\{\frac{\sigma}{p_\mathrm{a}\sqrt{\varepsilon}n}, \frac{L_\sigma^2}{\mathbb{1}_{p_\mathrm{a}}^2 \widehat{L}^2}\right\}\right)$ in DASHA-PP-PAGE and DASHA-PP-MVR, accordingly, guarantees
the degeneration up to $1/p_\mathrm{a}$. This is surprising, because in methods with the variance reduction of
stochastic gradients (Li et al., 2021a; Tyurin and Richtárik, 2023) we can take the size of batch size
$B = \mathcal{O}\left(\sqrt{\frac{m}{n}}\right)$ and $B = \mathcal{O}\left(\frac{\sigma}{\sqrt{\varepsilon}n}\right)$ and guarantee the optimality. Note that the smaller the batch size
$B$, the more the server and the nodes have to communicate to get $\varepsilon$-solution.

Let us consider the task of estimating the mean of vectors in the distributed setting. Suppose that we
have $n$ nodes, and each of them contains $m$ vectors $\{x_{ij}\}_{j=1}^m$, where $x_{ij} \in \mathbb{R}^d$ for all $i \in [n], j \in [m]$.
First, let us consider that each node samples a mini-batch $I^i$ of size $B$ with replacement and sends it
to the server. Then the server calculates the mean of the mini-batches from nodes. One can easily
show that the variance of the estimator is

$$
\mathrm{E}\left[\left\|\frac{1}{nB}\sum_{i=1}^n\sum_{j\in I^i} x_{ij} - \frac{1}{nm}\sum_{i=1}^n\sum_{j=1}^m x_{ij}\right\|^2\right] \tag{13}
$$

$$
= \frac{1}{nB}\frac{1}{nm}\sum_{i=1}^n\sum_{j=1}^m\left\|x_{ij} - \frac{1}{m}\sum_{j=1}^m x_{ij}\right\|^2.
$$

Next, we consider the same task in the partial participation setting with $s$–nice sampling, i.e., we
sample a random set $S \subset [n]$ of $s \in [n]$ nodes without replacement and receive the mini-batches
only from the sampled nodes. Such sampling of nodes satisfy Assumption 8 with $p_\mathrm{a} = s/n$ and
$p_\mathrm{a} = s(s-1)/n(n-1)$. In this case, the variance of the estimator (See Lemma 1 with $r_i = 0$ and
$s_i = \sum_{j\in I^i} x_{ij}$) is

$$
\mathrm{E}\left[\left\|\frac{1}{sB}\sum_{i\in S}\sum_{j\in I^i} x_{ij} - \frac{1}{nm}\sum_{i=1}^n\sum_{j=1}^m x_{ij}\right\|^2\right] \tag{14}
$$

$$
= \frac{1}{sB}\frac{1}{nm}\underbrace{\sum_{i=1}^n\sum_{j=1}^m\left\|x_{ij} - \frac{1}{m}\sum_{j=1}^m x_{ij}\right\|^2}_{\mathcal{L}_{\max}^2}
$$

$$
+ \frac{n-s}{s(n-1)}\frac{1}{n}\underbrace{\sum_{i=1}^n\left\|\frac{1}{m}\sum_{j=1}^m x_{ij} - \frac{1}{nm}\sum_{i=1}^n\sum_{j=1}^m x_{ij}\right\|^2}_{\widehat{\mathcal{L}}^2}.
$$

Let us assume that $s \leq n/2$. Note that (13) scales with any $B \geq 1$, while (14) only scales when
$B = \mathcal{O}\left(\mathcal{L}_{\max}^2/\widehat{\mathcal{L}}^2\right)$. In other words, for large enough $B$, the variance in (14) does not significantly
improves with the growth of $B$ due to the term $\widehat{\mathcal{L}}^2$. In our proof, due to partial participation, the
variance from (14) naturally appears, and we get the same effect. As was mentioned in Sections 6.2
and 6.3, it can be seen in our convergence rate bounds.

# D   Auxiliary facts

We list auxiliary facts that we use in our proofs:

1. For all $x, y \in \mathbb{R}^d$, we have

$$\|x + y\|^2 \leq 2 \|x\|^2 + 2 \|y\|^2 \tag{15}$$

2. Let us take a *random vector* $\xi \in \mathbb{R}^d$, then

$$\mathrm{E}\left[\|\xi\|^2\right] = \mathrm{E}\left[\|\xi - \mathrm{E}[\xi]\|^2\right] + \|\mathrm{E}[\xi]\|^2. \tag{16}$$

## D.1   Sampling Lemma

This section provides a lemma that we regularly use in our proofs, and it is useful for samplings that satisfy Assumption 8.

**Lemma 1.** *Suppose that a set $S$ is a random subset of a set $[n]$ such that*

*1.* $$\mathbf{Prob}\,(i \in S) = p_{\mathrm{a}}, \quad \forall i \in [n],$$

*2.* $$\mathbf{Prob}\,(i \in S, j \in S) = p_{\mathrm{aa}}, \quad \forall i \neq j \in [n],$$

*3.* $$p_{\mathrm{aa}} \leq p_{\mathrm{a}}^2,$$

*where $p_{\mathrm{a}} \in (0, 1]$ and $p_{\mathrm{aa}} \in [0, 1]$. Let us take random independent vectors $s_i \in \mathbb{R}^d$ for all $i \in [n]$, nonrandom vector $r_i \in \mathbb{R}^d$ for all $i \in [n]$, and random vectors*

$$v_i = \begin{cases} r_i + \frac{1}{p_{\mathrm{a}}} s_i, i \in S, \\ r_i, i \notin S, \end{cases}$$

*then*

$$\mathrm{E}\left[\left\|\frac{1}{n}\sum_{i=1}^n v_i - \mathrm{E}\left[\frac{1}{n}\sum_{i=1}^n v_i\right]\right\|^2\right]$$

$$= \frac{1}{n^2 p_{\mathrm{a}}}\sum_{i=1}^n \mathrm{E}\left[\|s_i - \mathrm{E}[s_i]\|^2\right] + \frac{p_{\mathrm{a}} - p_{\mathrm{aa}}}{n^2 p_{\mathrm{a}}^2}\sum_{i=1}^n \|\mathrm{E}[s_i]\|^2 + \frac{p_{\mathrm{aa}} - p_{\mathrm{a}}^2}{p_{\mathrm{a}}^2}\left\|\frac{1}{n}\sum_{i=1}^n \mathrm{E}[s_i]\right\|$$

$$\leq \frac{1}{n^2 p_{\mathrm{a}}}\sum_{i=1}^n \mathrm{E}\left[\|s_i - \mathrm{E}[s_i]\|^2\right] + \frac{p_{\mathrm{a}} - p_{\mathrm{aa}}}{n^2 p_{\mathrm{a}}^2}\sum_{i=1}^n \|\mathrm{E}[s_i]\|^2.$$

*Proof.* Let us define additional constants $p_{\mathrm{an}}$ and $p_{\mathrm{nn}}$, such that

1. $$\mathbf{Prob}\,(i \in S, j \notin S) = p_{\mathrm{an}}, \quad \forall i \neq j \in [n],$$

2. $$\mathbf{Prob}\,(i \notin S, j \notin S) = p_{\mathrm{nn}}, \quad \forall i \neq j \in [n].$$

Note, that

$$p_{\mathrm{a}} = p_{\mathrm{aa}} + p_{\mathrm{an}} \tag{17}$$

and

$$p_{\mathrm{nn}} = 1 - p_{\mathrm{aa}} - 2p_{\mathrm{an}}. \tag{18}$$

Using the law of total expectation and

$$\mathrm{E}[v_i] = p_{\mathrm{a}}\left(r_i + \mathrm{E}\left[\frac{1}{p_{\mathrm{a}}}s_i\right]\right) + (1 - p_{\mathrm{a}})r_i = r_i + \mathrm{E}[s_i],$$

we have

$$
\mathrm{E}\left[\left\|\frac{1}{n}\sum_{i=1}^{n}v_i - \mathrm{E}\left[\frac{1}{n}\sum_{i=1}^{n}v_i\right]\right\|^2\right]
$$

$$
= \frac{1}{n^2}\sum_{i=1}^{n}\mathrm{E}\left[\|v_i - (r_i + \mathrm{E}[s_i])\|^2\right]
$$

$$
+ \frac{1}{n^2}\sum_{i\neq j}^{n}\mathrm{E}\left[\langle v_i - (r_i + \mathrm{E}[s_i]), v_j - (r_j + \mathrm{E}[s_j])\rangle\right]
$$

$$
= \frac{p_{\mathrm{a}}}{n^2}\sum_{i=1}^{n}\mathrm{E}\left[\left\|r_i + \frac{1}{p_{\mathrm{a}}}s_i - (r_i + \mathrm{E}[s_i])\right\|^2\right]
$$

$$
+ \frac{1-p_{\mathrm{a}}}{n^2}\sum_{i=1}^{n}\|r_i - (r_i + \mathrm{E}[s_i])\|^2
$$

$$
+ \frac{p_{\mathrm{aa}}}{n^2}\sum_{i\neq j}^{n}\mathrm{E}\left[\left\langle r_i + \frac{1}{p_{\mathrm{a}}}s_i - (r_i + \mathrm{E}[s_i]), r_j + \frac{1}{p_{\mathrm{a}}}s_j - (r_j + \mathrm{E}[s_j])\right\rangle\right]
$$

$$
+ \frac{2p_{\mathrm{an}}}{n^2}\sum_{i\neq j}^{n}\mathrm{E}\left[\left\langle r_i + \frac{1}{p_{\mathrm{a}}}s_i - (r_i + \mathrm{E}[s_i]), r_j - (r_j + \mathrm{E}[s_j])\right\rangle\right]
$$

$$
+ \frac{p_{\mathrm{nn}}}{n^2}\sum_{i\neq j}^{n}\langle r_i - (r_i + \mathrm{E}[s_i]), r_j - (r_j + \mathrm{E}[s_j])\rangle.
$$

From the independence of random vectors $s_i$, we obtain

$$
\mathrm{E}\left[\left\|\frac{1}{n}\sum_{i=1}^{n}v_i - \mathrm{E}\left[\frac{1}{n}\sum_{i=1}^{n}v_i\right]\right\|^2\right]
$$

$$
= \frac{p_{\mathrm{a}}}{n^2}\sum_{i=1}^{n}\mathrm{E}\left[\left\|\frac{1}{p_{\mathrm{a}}}s_i - \mathrm{E}[s_i]\right\|^2\right]
$$

$$
+ \frac{1-p_{\mathrm{a}}}{n^2}\sum_{i=1}^{n}\|\mathrm{E}[s_i]\|^2
$$

$$
+ \frac{p_{\mathrm{aa}}(1-p_{\mathrm{a}})^2}{n^2 p_{\mathrm{a}}^2}\sum_{i\neq j}^{n}\langle \mathrm{E}[s_i], \mathrm{E}[s_j]\rangle
$$

$$
+ \frac{2p_{\mathrm{an}}(p_{\mathrm{a}}-1)}{n^2 p_{\mathrm{a}}}\sum_{i\neq j}^{n}\langle \mathrm{E}[s_i], \mathrm{E}[s_j]\rangle
$$

$$
+ \frac{p_{\mathrm{nn}}}{n^2}\sum_{i\neq j}^{n}\langle \mathrm{E}[s_i], \mathrm{E}[s_j]\rangle.
$$

Using (17) and (18), we have

$$
\mathrm{E}\left[\left\|\frac{1}{n}\sum_{i=1}^{n}v_i - \mathrm{E}\left[\frac{1}{n}\sum_{i=1}^{n}v_i\right]\right\|^2\right]
$$

$$
= \frac{p_{\mathrm{a}}}{n^2}\sum_{i=1}^{n}\mathrm{E}\left[\left\|\frac{1}{p_{\mathrm{a}}}s_i - \mathrm{E}[s_i]\right\|^2\right]
$$

$$
+ \frac{1-p_{\mathrm{a}}}{n^2}\sum_{i=1}^{n}\|\mathrm{E}[s_i]\|^2
$$

$$+ \frac{p_{\mathrm{aa}} - p_{\mathrm{a}}^2}{n^2 p_{\mathrm{a}}^2} \sum_{i \neq j}^{n} \langle \mathrm{E}\left[s_i\right], \mathrm{E}\left[s_j\right] \rangle$$

$$\overset{(16)}{=} \frac{1}{n^2 p_{\mathrm{a}}} \sum_{i=1}^{n} \mathrm{E}\left[\|s_i - \mathrm{E}\left[s_i\right]\|^2\right]$$

$$+ \frac{1 - p_{\mathrm{a}}}{n^2 p_{\mathrm{a}}} \sum_{i=1}^{n} \|\mathrm{E}\left[s_i\right]\|^2$$

$$+ \frac{p_{\mathrm{aa}} - p_{\mathrm{a}}^2}{n^2 p_{\mathrm{a}}^2} \sum_{i \neq j}^{n} \langle \mathrm{E}\left[s_i\right], \mathrm{E}\left[s_j\right] \rangle$$

$$= \frac{1}{n^2 p_{\mathrm{a}}} \sum_{i=1}^{n} \mathrm{E}\left[\|s_i - \mathrm{E}\left[s_i\right]\|^2\right]$$

$$+ \frac{p_{\mathrm{a}} - p_{\mathrm{aa}}}{n^2 p_{\mathrm{a}}^2} \sum_{i=1}^{n} \|\mathrm{E}\left[s_i\right]\|^2$$

$$+ \frac{p_{\mathrm{aa}} - p_{\mathrm{a}}^2}{p_{\mathrm{a}}^2} \left\| \frac{1}{n} \sum_{i=1}^{n} \mathrm{E}\left[s_i\right] \right\|.$$

Finally, using that $p_{\mathrm{aa}} \leq p_{\mathrm{a}}^2$, we have

$$\mathrm{E}\left[\left\| \frac{1}{n} \sum_{i=1}^{n} v_i - \mathrm{E}\left[\frac{1}{n} \sum_{i=1}^{n} v_i\right] \right\|^2\right]$$

$$\leq \frac{1}{n^2 p_{\mathrm{a}}} \sum_{i=1}^{n} \mathrm{E}\left[\|s_i - \mathrm{E}\left[s_i\right]\|^2\right] + \frac{p_{\mathrm{a}} - p_{\mathrm{aa}}}{n^2 p_{\mathrm{a}}^2} \sum_{i=1}^{n} \|\mathrm{E}\left[s_i\right]\|^2.$$

$\square$

## D.2 Compressors Facts

We define the Rand$K$ compressor that chooses without replacement $K$ coordinates, scales them by a constant factor to preserve unbiasedness and zero-out other coordinates.

**Definition 5.** Let us take a random subset $S$ from $[d]$, $|S| = K$, $K \in [d]$. We say that a stochastic mapping $\mathcal{C} : \mathbb{R}^d \to \mathbb{R}^d$ is Rand$K$ if

$$\mathcal{C}(x) = \frac{d}{K} \sum_{j \in S} x_j e_j,$$

where $\{e_i\}_{i=1}^{d}$ is the standard unit basis.

**Theorem 6.** *If $\mathcal{C}$ is RandK, then $\mathcal{C} \in \mathbb{U}\left(\frac{d}{k} - 1\right)$.*

See the proof in (Beznosikov et al., 2020).

## E Proofs of Theorems

There are three different sources of randomness in Algorithm 1: the first one from vectors $\{k_i^{t+1}\}_{i=1}^{n}$, the second one from compressors $\{\mathcal{C}_i\}_{i=1}^{n}$, and the third one from availability of nodes. We define $\mathrm{E}_k\left[\cdot\right]$, $\mathrm{E}_{\mathcal{C}}\left[\cdot\right]$ and $\mathrm{E}_{p_{\mathrm{a}}}\left[\cdot\right]$ to be conditional expectations w.r.t. $\{k_i^{t+1}\}_{i=1}^{n}$, $\{\mathcal{C}_i\}_{i=1}^{n}$, and availability, accordingly, conditioned on all previous randomness. Moreover, we define $\mathrm{E}_{t+1}\left[\cdot\right]$ to be a conditional expectation w.r.t. all randomness in iteration $t + 1$ conditioned on all previous randomness. Note, that $\mathrm{E}_{t+1}\left[\cdot\right] = \mathrm{E}_k\left[\mathrm{E}_{\mathcal{C}}\left[\mathrm{E}_{p_{\mathrm{a}}}\left[\cdot\right]\right]\right]$.

In the case of DASHA-PP-PAGE, there are two different sources of randomness from $\{k_i^{t+1}\}_{i=1}^{n}$. We define $\mathrm{E}_{p_{\mathrm{page}}}\left[\cdot\right]$ and $\mathrm{E}_B\left[\cdot\right]$ to be conditional expectations w.r.t. the probabilistic switching and mini-batch indices $I_i^t$, accordingly, conditioned on all previous randomness. Note, that $\mathrm{E}_{t+1}\left[\cdot\right] = \mathrm{E}_B\left[\mathrm{E}_{\mathcal{C}}\left[\mathrm{E}_{p_{\mathrm{a}}}\left[\mathrm{E}_{p_{\mathrm{page}}}\left[\cdot\right]\right]\right]\right]$ and $\mathrm{E}_{t+1}\left[\cdot\right] = \mathrm{E}_B\left[\mathrm{E}_{p_{\mathrm{page}}}\left[\mathrm{E}_{\mathcal{C}}\left[\mathrm{E}_{p_{\mathrm{a}}}\left[\cdot\right]\right]\right]\right]$.

### E.1 Standard Lemmas in the Nonconvex Setting

We start the proof of theorems by providing standard lemmas from the nonconvex optimization.

**Lemma 2.** *Suppose that Assumption 2 holds and let $x^{t+1} = x^t - \gamma g^t$. Then for any $g^t \in \mathbb{R}^d$ and $\gamma > 0$, we have*

$$f(x^{t+1}) \leq f(x^t) - \frac{\gamma}{2} \left\| \nabla f(x^t) \right\|^2 - \left( \frac{1}{2\gamma} - \frac{L}{2} \right) \left\| x^{t+1} - x^t \right\|^2 + \frac{\gamma}{2} \left\| g^t - \nabla f(x^t) \right\|^2. \quad (19)$$

*Proof.* Using $L$−smoothness, we have

$$f(x^{t+1}) \leq f(x^t) + \left\langle \nabla f(x^t), x^{t+1} - x^t \right\rangle + \frac{L}{2} \left\| x^{t+1} - x^t \right\|^2$$
$$= f(x^t) - \gamma \left\langle \nabla f(x^t), g^t \right\rangle + \frac{L}{2} \left\| x^{t+1} - x^t \right\|^2.$$

Next, due to $- \langle x, y \rangle = \frac{1}{2} \|x - y\|^2 - \frac{1}{2} \|x\|^2 - \frac{1}{2} \|y\|^2$, we obtain

$$f(x^{t+1}) \leq f(x^t) - \frac{\gamma}{2} \left\| \nabla f(x^t) \right\|^2 - \left( \frac{1}{2\gamma} - \frac{L}{2} \right) \left\| x^{t+1} - x^t \right\|^2 + \frac{\gamma}{2} \left\| g^t - \nabla f(x^t) \right\|^2.$$

$\square$

**Lemma 3.** *Suppose that Assumption 1 holds and*

$$\mathrm{E}\left[ f(x^{t+1}) \right] + \gamma \Psi^{t+1} \leq \mathrm{E}\left[ f(x^t) \right] - \frac{\gamma}{2} \mathrm{E}\left[ \left\| \nabla f(x^t) \right\|^2 \right] + \gamma \Psi^t + \gamma C, \quad (20)$$

*where $\Psi^t$ is a sequence of numbers, $\Psi^t \geq 0$ for all $t \in [T]$, constant $C \geq 0$, and constant $\gamma > 0$. Then*

$$\mathrm{E}\left[ \left\| \nabla f(\widehat{x}^T) \right\|^2 \right] \leq \frac{2\Delta_0}{\gamma T} + \frac{2\Psi^0}{T} + 2C, \quad (21)$$

*where a point $\widehat{x}^T$ is chosen uniformly from a set of points $\{x^t\}_{t=0}^{T-1}$.*

*Proof.* By unrolling (20) for $t$ from 0 to $T - 1$, we obtain

$$\frac{\gamma}{2} \sum_{t=0}^{T-1} \mathrm{E}\left[ \left\| \nabla f(x^t) \right\|^2 \right] + \mathrm{E}\left[ f(x^T) \right] + \gamma \Psi^T \leq f(x^0) + \gamma \Psi^0 + \gamma T C.$$

We subtract $f^*$, divide inequality by $\frac{\gamma T}{2}$, and take into account that $f(x) \geq f^*$ for all $x \in \mathbb{R}$, and $\Psi^t \geq 0$ for all $t \in [T]$, to get the following inequality:

$$\frac{1}{T} \sum_{t=0}^{T-1} \mathrm{E}\left[ \left\| \nabla f(x^t) \right\|^2 \right] \leq \frac{2\Delta_0}{\gamma T} + \frac{2\Psi^0}{T} + 2C.$$

It is left to consider the choice of a point $\widehat{x}^T$ to complete the proof of the lemma. $\square$

**Lemma 4.** *If $0 < \gamma \leq (L + \sqrt{A})^{-1}$, $L > 0$, and $A \geq 0$, then*

$$\frac{1}{2\gamma} - \frac{L}{2} - \frac{\gamma A}{2} \geq 0.$$

The lemma can be easily checked with the direct calculation.

## E.2 Generic Lemmas

**Lemma 5.** *Suppose that Assumptions 7 and 8 hold and let us consider sequences $g_i^{t+1}$, $h_i^{t+1}$, and $k_i^{t+1}$ from Algorithm 1, then*

$$\mathrm{E}_{\mathcal{C}}\left[\mathrm{E}_{p_{\mathrm{a}}}\left[\left\|g^{t+1}-h^{t+1}\right\|^2\right]\right]$$

$$\leq \frac{2\omega}{n^2 p_{\mathrm{a}}}\sum_{i=1}^n \left\|k_i^{t+1}\right\|^2 + \frac{a^2((2\omega+1)\,p_{\mathrm{a}}-p_{\mathrm{aa}})}{n^2 p_{\mathrm{a}}^2}\sum_{i=1}^n \left\|g_i^t-h_i^t\right\|^2 + (1-a)^2\left\|g^t-h^t\right\|^2, \quad (22)$$

*and*

$$\mathrm{E}_{\mathcal{C}}\left[\mathrm{E}_{p_{\mathrm{a}}}\left[\left\|g_i^{t+1}-h_i^{t+1}\right\|^2\right]\right]$$

$$\leq \frac{2\omega}{p_{\mathrm{a}}}\left\|k_i^{t+1}\right\|^2 + \left(\frac{a^2(2\omega+1-p_{\mathrm{a}})}{p_{\mathrm{a}}}+(1-a)^2\right)\left\|g_i^t-h_i^t\right\|^2 \quad \forall i\in[n]. \quad (23)$$

*Proof.* First, we estimate $\mathrm{E}_{\mathcal{C}}\left[\mathrm{E}_{p_{\mathrm{a}}}\left[\left\|g^{t+1}-h^{t+1}\right\|^2\right]\right]$:

$$\mathrm{E}_{\mathcal{C}}\left[\mathrm{E}_{p_{\mathrm{a}}}\left[\left\|g^{t+1}-h^{t+1}\right\|^2\right]\right]$$

$$= \mathrm{E}_{\mathcal{C}}\left[\mathrm{E}_{p_{\mathrm{a}}}\left[\left\|g^{t+1}-h^{t+1}-\mathrm{E}_{\mathcal{C}}\left[\mathrm{E}_{p_{\mathrm{a}}}\left[g^{t+1}-h^{t+1}\right]\right]\right\|^2\right]\right] + \left\|\mathrm{E}_{\mathcal{C}}\left[\mathrm{E}_{p_{\mathrm{a}}}\left[g^{t+1}-h^{t+1}\right]\right]\right\|^2,$$

where we used (16). Due to Assumption 8, we have

$$\mathrm{E}_{\mathcal{C}}\left[\mathrm{E}_{p_{\mathrm{a}}}\left[g_i^{t+1}\right]\right]$$

$$= p_{\mathrm{a}}\mathrm{E}_{\mathcal{C}}\left[g_i^t+\mathcal{C}_i\left(\frac{1}{p_{\mathrm{a}}}k_i^{t+1}-\frac{a}{p_{\mathrm{a}}}\left(g_i^t-h_i^t\right)\right)\right] + (1-p_{\mathrm{a}})g_i^t$$

$$= g_i^t + p_{\mathrm{a}}\mathrm{E}_{\mathcal{C}}\left[\mathcal{C}_i\left(\frac{1}{p_{\mathrm{a}}}k_i^{t+1}-\frac{a}{p_{\mathrm{a}}}\left(g_i^t-h_i^t\right)\right)\right]$$

$$= g_i^t + k_i^{t+1} - a\left(g_i^t-h_i^t\right),$$

and

$$\mathrm{E}_{\mathcal{C}}\left[\mathrm{E}_{p_{\mathrm{a}}}\left[h_i^{t+1}\right]\right] = p_{\mathrm{a}}\mathrm{E}_{\mathcal{C}}\left[h_i^t+\frac{1}{p_{\mathrm{a}}}k_i^{t+1}\right] + (1-p_{\mathrm{a}})h_i^t = h_i^t + k_i^{t+1}.$$

Thus, we can get

$$\mathrm{E}_{\mathcal{C}}\left[\mathrm{E}_{p_{\mathrm{a}}}\left[\left\|g^{t+1}-h^{t+1}\right\|^2\right]\right]$$

$$= \mathrm{E}_{\mathcal{C}}\left[\mathrm{E}_{p_{\mathrm{a}}}\left[\left\|g^{t+1}-h^{t+1}-\mathrm{E}_{\mathcal{C}}\left[\mathrm{E}_{p_{\mathrm{a}}}\left[g^{t+1}-h^{t+1}\right]\right]\right\|^2\right]\right] + (1-a)^2\left\|g^t-h^t\right\|^2.$$

Due to the independence of compressors, we can use Lemma 1 with $r_i = g_i^t - h_i^t$ and $s_i = p_{\mathrm{a}}\mathcal{C}_i\left(\frac{1}{p_{\mathrm{a}}}k_i^{t+1}-\frac{a}{p_{\mathrm{a}}}\left(g_i^t-h_i^t\right)\right) - k_i^{t+1}$, and obtain

$$\mathrm{E}_{\mathcal{C}}\left[\mathrm{E}_{p_{\mathrm{a}}}\left[\left\|g^{t+1}-h^{t+1}\right\|^2\right]\right]$$

$$\leq \frac{1}{n^2 p_{\mathrm{a}}}\sum_{i=1}^n \mathrm{E}_{\mathcal{C}}\left[\left\|p_{\mathrm{a}}\mathcal{C}_i\left(\frac{1}{p_{\mathrm{a}}}k_i^{t+1}-\frac{a}{p_{\mathrm{a}}}\left(g_i^t-h_i^t\right)\right)-k_i^{t+1}-\mathrm{E}_{\mathcal{C}}\left[p_{\mathrm{a}}\mathcal{C}_i\left(\frac{1}{p_{\mathrm{a}}}k_i^{t+1}-\frac{a}{p_{\mathrm{a}}}\left(g_i^t-h_i^t\right)\right)-k_i^{t+1}\right]\right\|^2\right]$$

$$+ \frac{p_{\mathrm{a}}-p_{\mathrm{aa}}}{n^2 p_{\mathrm{a}}^2}\sum_{i=1}^n \left\|\mathrm{E}_{\mathcal{C}}\left[p_{\mathrm{a}}\mathcal{C}_i\left(\frac{1}{p_{\mathrm{a}}}k_i^{t+1}-\frac{a}{p_{\mathrm{a}}}\left(g_i^t-h_i^t\right)\right)-k_i^{t+1}\right]\right\|^2$$

$$+ (1-a)^2 \left\| g^t - h^t \right\|^2$$

$$= \frac{p_{\mathrm{a}}}{n^2} \sum_{i=1}^{n} \mathrm{E}_{\mathcal{C}} \left[ \left\| \mathcal{C}_i \left( \frac{1}{p_{\mathrm{a}}} k_i^{t+1} - \frac{a}{p_{\mathrm{a}}} \left( g_i^t - h_i^t \right) \right) - \left( \frac{1}{p_{\mathrm{a}}} k_i^{t+1} - \frac{a}{p_{\mathrm{a}}} \left( g_i^t - h_i^t \right) \right) \right\|^2 \right]$$

$$+ \frac{a^2 \left( p_{\mathrm{a}} - p_{\mathrm{aa}} \right)}{n^2 p_{\mathrm{a}}^2} \sum_{i=1}^{n} \left\| g_i^t - h_i^t \right\|^2 + (1-a)^2 \left\| g^t - h^t \right\|^2 .$$

From Assumption 7, we have

$$\mathrm{E}_{\mathcal{C}} \left[ \mathrm{E}_{p_{\mathrm{a}}} \left[ \left\| g^{t+1} - h^{t+1} \right\|^2 \right] \right]$$

$$\leq \frac{\omega p_{\mathrm{a}}}{n^2} \sum_{i=1}^{n} \left\| \frac{1}{p_{\mathrm{a}}} k_i^{t+1} - \frac{a}{p_{\mathrm{a}}} \left( g_i^t - h_i^t \right) \right\|^2 + \frac{a^2 \left( p_{\mathrm{a}} - p_{\mathrm{aa}} \right)}{n^2 p_{\mathrm{a}}^2} \sum_{i=1}^{n} \left\| g_i^t - h_i^t \right\|^2 + (1-a)^2 \left\| g^t - h^t \right\|^2$$

$$= \frac{\omega}{n^2 p_{\mathrm{a}}} \sum_{i=1}^{n} \left\| k_i^{t+1} - a \left( g_i^t - h_i^t \right) \right\|^2 + \frac{a^2 \left( p_{\mathrm{a}} - p_{\mathrm{aa}} \right)}{n^2 p_{\mathrm{a}}^2} \sum_{i=1}^{n} \left\| g_i^t - h_i^t \right\|^2 + (1-a)^2 \left\| g^t - h^t \right\|^2$$

$$\overset{(15)}{\leq} \frac{2\omega}{n^2 p_{\mathrm{a}}} \sum_{i=1}^{n} \left\| k_i^{t+1} \right\|^2 + \frac{a^2 \left( (2\omega+1) p_{\mathrm{a}} - p_{\mathrm{aa}} \right)}{n^2 p_{\mathrm{a}}^2} \sum_{i=1}^{n} \left\| g_i^t - h_i^t \right\|^2 + (1-a)^2 \left\| g^t - h^t \right\|^2 .$$

The second inequality can be proved almost in the same way:

$$\mathrm{E}_{\mathcal{C}} \left[ \mathrm{E}_{p_{\mathrm{a}}} \left[ \left\| g_i^{t+1} - h_i^{t+1} \right\|^2 \right] \right]$$

$$= \mathrm{E}_{\mathcal{C}} \left[ \mathrm{E}_{p_{\mathrm{a}}} \left[ \left\| g_i^{t+1} - h_i^{t+1} - \mathrm{E}_{\mathcal{C}} \left[ \mathrm{E}_{p_{\mathrm{a}}} \left[ g_i^{t+1} - h_i^{t+1} \right] \right] \right\|^2 \right] \right] + \left\| \mathrm{E}_{\mathcal{C}} \left[ \mathrm{E}_{p_{\mathrm{a}}} \left[ g_i^{t+1} - h_i^{t+1} \right] \right] \right\|^2$$

$$= \mathrm{E}_{\mathcal{C}} \left[ \mathrm{E}_{p_{\mathrm{a}}} \left[ \left\| g_i^{t+1} - h_i^{t+1} - g_i^t + a \left( g_i^t - h_i^t \right) + h_i^t \right\|^2 \right] \right] + (1-a)^2 \left\| g_i^t - h_i^t \right\|^2$$

$$= p_{\mathrm{a}} \mathrm{E}_{\mathcal{C}} \left[ \left\| \mathcal{C}_i \left( \frac{1}{p_{\mathrm{a}}} k_i^{t+1} - \frac{a}{p_{\mathrm{a}}} \left( g_i^t - h_i^t \right) \right) - \frac{1}{p_{\mathrm{a}}} k_i^{t+1} + a \left( g_i^t - h_i^t \right) \right\|^2 \right]$$

$$+ a^2 (1-p_{\mathrm{a}}) \left\| g_i^t - h_i^t \right\|^2 + (1-a)^2 \left\| g_i^t - h_i^t \right\|^2$$

$$\overset{(16)}{=} p_{\mathrm{a}} \mathrm{E}_{\mathcal{C}} \left[ \left\| \mathcal{C}_i \left( \frac{1}{p_{\mathrm{a}}} k_i^{t+1} - \frac{a}{p_{\mathrm{a}}} \left( g_i^t - h_i^t \right) \right) - \left( \frac{1}{p_{\mathrm{a}}} k_i^{t+1} - \frac{a}{p_{\mathrm{a}}} \left( g_i^t - h_i^t \right) \right) \right\|^2 \right]$$

$$+ a^2 \frac{(1-p_{\mathrm{a}})^2}{p_{\mathrm{a}}} \left\| g_i^t - h_i^t \right\|^2$$

$$+ a^2 (1-p_{\mathrm{a}}) \left\| g_i^t - h_i^t \right\|^2 + (1-a)^2 \left\| g_i^t - h_i^t \right\|^2$$

$$\leq \frac{\omega}{p_{\mathrm{a}}} \left\| k_i^{t+1} - a \left( g_i^t - h_i^t \right) \right\|^2$$

$$+ \frac{a^2 (1-p_{\mathrm{a}})}{p_{\mathrm{a}}} \left\| g_i^t - h_i^t \right\|^2 + (1-a)^2 \left\| g_i^t - h_i^t \right\|^2$$

$$\overset{(15)}{\leq} \frac{2\omega}{p_{\mathrm{a}}} \left\| k_i^{t+1} \right\|^2 + \frac{a^2 (2\omega + 1 - p_{\mathrm{a}})}{p_{\mathrm{a}}} \left\| g_i^t - h_i^t \right\|^2 + (1-a)^2 \left\| g_i^t - h_i^t \right\|^2 .$$

$$\square$$

**Lemma 6.** *Suppose that Assumptions 2, 7, and 8 hold and let us take $a = \frac{p_{\mathrm{a}}}{2\omega + 1}$, then*

$$\mathrm{E} \left[ f(x^{t+1}) \right] + \frac{\gamma(2\omega+1)}{p_{\mathrm{a}}} \mathrm{E} \left[ \left\| g^{t+1} - h^{t+1} \right\|^2 \right] + \frac{\gamma((2\omega+1) p_{\mathrm{a}} - p_{\mathrm{aa}})}{n p_{\mathrm{a}}^2} \mathrm{E} \left[ \frac{1}{n} \sum_{i=1}^{n} \left\| g_i^{t+1} - h_i^{t+1} \right\|^2 \right]$$

$$\leq \mathrm{E} \left[ f(x^t) - \frac{\gamma}{2} \left\| \nabla f(x^t) \right\|^2 - \left( \frac{1}{2\gamma} - \frac{L}{2} \right) \left\| x^{t+1} - x^t \right\|^2 + \gamma \left\| h^t - \nabla f(x^t) \right\|^2 \right]$$

$$+ \frac{\gamma(2\omega+1)}{p_{\mathrm{a}}} \mathrm{E} \left[ \left\| g^t - h^t \right\|^2 \right] + \frac{\gamma((2\omega+1) p_{\mathrm{a}} - p_{\mathrm{aa}})}{n p_{\mathrm{a}}^2} \mathrm{E} \left[ \frac{1}{n} \sum_{i=1}^{n} \left\| g_i^t - h_i^t \right\|^2 \right] + \frac{4\gamma\omega(2\omega+1)}{n p_{\mathrm{a}}^2} \mathrm{E} \left[ \frac{1}{n} \sum_{i=1}^{n} \left\| k_i^{t+1} \right\|^2 \right] .$$

*Proof.* Due to Lemma 2 and the update step from Line 5 in Algorithm 1, we have

$\mathrm{E}_{t+1}\left[f(x^{t+1})\right]$

$\leq \mathrm{E}_{t+1}\left[f(x^t) - \frac{\gamma}{2}\left\|\nabla f(x^t)\right\|^2 - \left(\frac{1}{2\gamma} - \frac{L}{2}\right)\left\|x^{t+1} - x^t\right\|^2 + \frac{\gamma}{2}\left\|g^t - \nabla f(x^t)\right\|^2\right]$

$= \mathrm{E}_{t+1}\left[f(x^t) - \frac{\gamma}{2}\left\|\nabla f(x^t)\right\|^2 - \left(\frac{1}{2\gamma} - \frac{L}{2}\right)\left\|x^{t+1} - x^t\right\|^2 + \frac{\gamma}{2}\left\|g^t - h^t + h^t - \nabla f(x^t)\right\|^2\right]$

$\overset{(16)}{\leq} \mathrm{E}_{t+1}\left[f(x^t) - \frac{\gamma}{2}\left\|\nabla f(x^t)\right\|^2 - \left(\frac{1}{2\gamma} - \frac{L}{2}\right)\left\|x^{t+1} - x^t\right\|^2 + \gamma\left(\left\|g^t - h^t\right\|^2 + \left\|h^t - \nabla f(x^t)\right\|^2\right)\right).$

Let us fix some constants $\kappa, \eta \in [0, \infty)$ that we will define later. Combining the last inequality, bounds (22), (23) and using the law of total expectation, we get

$\mathrm{E}\left[f(x^{t+1})\right]$

$\quad + \kappa\mathrm{E}\left[\left\|g^{t+1} - h^{t+1}\right\|^2\right] + \eta\mathrm{E}\left[\frac{1}{n}\sum_{i=1}^n\left\|g_i^{t+1} - h_i^{t+1}\right\|^2\right]$

$= \mathrm{E}\left[\mathrm{E}_{t+1}\left[f(x^{t+1})\right]\right]$

$\quad + \kappa\mathrm{E}\left[\mathrm{E}_{\mathcal{C}}\left[\mathrm{E}_{p_\mathrm{a}}\left[\left\|g^{t+1} - h^{t+1}\right\|^2\right]\right]\right] + \eta\mathrm{E}\left[\mathrm{E}_{\mathcal{C}}\left[\mathrm{E}_{p_\mathrm{a}}\left[\frac{1}{n}\sum_{i=1}^n\left\|g_i^{t+1} - h_i^{t+1}\right\|^2\right]\right]\right]$

$\leq \mathrm{E}\left[f(x^t) - \frac{\gamma}{2}\left\|\nabla f(x^t)\right\|^2 - \left(\frac{1}{2\gamma} - \frac{L}{2}\right)\left\|x^{t+1} - x^t\right\|^2 + \gamma\left(\left\|g^t - h^t\right\|^2 + \left\|h^t - \nabla f(x^t)\right\|^2\right)\right]$

$\quad + \kappa\mathrm{E}\left[\frac{2\omega}{n^2 p_\mathrm{a}}\sum_{i=1}^n\left\|k_i^{t+1}\right\|^2 + \frac{a^2((2\omega+1)p_\mathrm{a} - p_\mathrm{aa})}{n^2 p_\mathrm{a}^2}\sum_{i=1}^n\left\|g_i^t - h_i^t\right\|^2 + (1-a)^2\left\|g^t - h^t\right\|^2\right]$

$\quad + \eta\mathrm{E}\left[\frac{2\omega}{n p_\mathrm{a}}\sum_{i=1}^n\left\|k_i^{t+1}\right\|^2 + \left(\frac{a^2(2\omega+1 - p_\mathrm{a})}{p_\mathrm{a}} + (1-a)^2\right)\frac{1}{n}\sum_{i=1}^n\left\|g_i^t - h_i^t\right\|^2\right]$

$= \mathrm{E}\left[f(x^t) - \frac{\gamma}{2}\left\|\nabla f(x^t)\right\|^2 - \left(\frac{1}{2\gamma} - \frac{L}{2}\right)\left\|x^{t+1} - x^t\right\|^2 + \gamma\left\|h^t - \nabla f(x^t)\right\|^2\right]$

$\quad + \left(\gamma + \kappa(1-a)^2\right)\mathrm{E}\left[\left\|g^t - h^t\right\|^2\right]$

$\quad + \left(\frac{\kappa a^2((2\omega+1)p_\mathrm{a} - p_\mathrm{aa})}{n p_\mathrm{a}^2} + \eta\left(\frac{a^2(2\omega+1 - p_\mathrm{a})}{p_\mathrm{a}} + (1-a)^2\right)\right)\mathrm{E}\left[\frac{1}{n}\sum_{i=1}^n\left\|g_i^t - h_i^t\right\|^2\right]$

$\quad + \left(\frac{2\kappa\omega}{n p_\mathrm{a}} + \frac{2\eta\omega}{p_\mathrm{a}}\right)\mathrm{E}\left[\frac{1}{n}\sum_{i=1}^n\left\|k_i^{t+1}\right\|^2\right].$

Now, by taking $\kappa = \frac{\gamma}{a}$, we can see that $\gamma + \kappa(1-a)^2 \leq \kappa$, and thus

$\mathrm{E}\left[f(x^{t+1})\right]$

$\quad + \frac{\gamma}{a}\mathrm{E}\left[\left\|g^{t+1} - h^{t+1}\right\|^2\right] + \eta\mathrm{E}\left[\frac{1}{n}\sum_{i=1}^n\left\|g_i^{t+1} - h_i^{t+1}\right\|^2\right]$

$\leq \mathrm{E}\left[f(x^t) - \frac{\gamma}{2}\left\|\nabla f(x^t)\right\|^2 - \left(\frac{1}{2\gamma} - \frac{L}{2}\right)\left\|x^{t+1} - x^t\right\|^2 + \gamma\left\|h^t - \nabla f(x^t)\right\|^2\right]$

$\quad + \frac{\gamma}{a}\mathrm{E}\left[\left\|g^t - h^t\right\|^2\right]$

$\quad + \left(\frac{\gamma a((2\omega+1)p_\mathrm{a} - p_\mathrm{aa})}{n p_\mathrm{a}^2} + \eta\left(\frac{a^2(2\omega+1 - p_\mathrm{a})}{p_\mathrm{a}} + (1-a)^2\right)\right)\mathrm{E}\left[\frac{1}{n}\sum_{i=1}^n\left\|g_i^t - h_i^t\right\|^2\right]$

$\quad + \left(\frac{2\gamma\omega}{a n p_\mathrm{a}} + \frac{2\eta\omega}{p_\mathrm{a}}\right)\mathrm{E}\left[\frac{1}{n}\sum_{i=1}^n\left\|k_i^{t+1}\right\|^2\right].$

Next, by taking $\eta = \frac{\gamma((2\omega+1)p_a - p_{aa})}{np_a^2}$ and considering the choice of $a$, one can show that $\left(\frac{\gamma a((2\omega+1)p_a - p_{aa})}{np_a^2} + \eta\left(\frac{a^2(2\omega+1-p_a)}{p_a} + (1-a)^2\right)\right) \leq \eta$. Thus

$$
\mathrm{E}\left[f(x^{t+1})\right]
$$
$$
+ \frac{\gamma(2\omega+1)}{p_a}\mathrm{E}\left[\left\|g^{t+1} - h^{t+1}\right\|^2\right] + \frac{\gamma((2\omega+1)p_a - p_{aa})}{np_a^2}\mathrm{E}\left[\frac{1}{n}\sum_{i=1}^n \left\|g_i^{t+1} - h_i^{t+1}\right\|^2\right]
$$
$$
\leq \mathrm{E}\left[f(x^t) - \frac{\gamma}{2}\left\|\nabla f(x^t)\right\|^2 - \left(\frac{1}{2\gamma} - \frac{L}{2}\right)\left\|x^{t+1} - x^t\right\|^2 + \gamma\left\|h^t - \nabla f(x^t)\right\|^2\right]
$$
$$
+ \frac{\gamma(2\omega+1)}{p_a}\mathrm{E}\left[\left\|g^t - h^t\right\|^2\right] + \frac{\gamma((2\omega+1)p_a - p_{aa})}{np_a^2}\mathrm{E}\left[\frac{1}{n}\sum_{i=1}^n \left\|g_i^t - h_i^t\right\|^2\right]
$$
$$
+ \left(\frac{2\gamma(2\omega+1)\omega}{np_a^2} + \frac{2\gamma((2\omega+1)p_a - p_{aa})\omega}{np_a^3}\right)\mathrm{E}\left[\frac{1}{n}\sum_{i=1}^n \left\|k_i^{t+1}\right\|^2\right].
$$

Considering that $p_{aa} \geq 0$, we can simplify the last term and get

$$
\mathrm{E}\left[f(x^{t+1})\right]
$$
$$
+ \frac{\gamma(2\omega+1)}{p_a}\mathrm{E}\left[\left\|g^{t+1} - h^{t+1}\right\|^2\right] + \frac{\gamma((2\omega+1)p_a - p_{aa})}{np_a^2}\mathrm{E}\left[\frac{1}{n}\sum_{i=1}^n \left\|g_i^{t+1} - h_i^{t+1}\right\|^2\right]
$$
$$
\leq \mathrm{E}\left[f(x^t) - \frac{\gamma}{2}\left\|\nabla f(x^t)\right\|^2 - \left(\frac{1}{2\gamma} - \frac{L}{2}\right)\left\|x^{t+1} - x^t\right\|^2 + \gamma\left\|h^t - \nabla f(x^t)\right\|^2\right]
$$
$$
+ \frac{\gamma(2\omega+1)}{p_a}\mathrm{E}\left[\left\|g^t - h^t\right\|^2\right] + \frac{\gamma((2\omega+1)p_a - p_{aa})}{np_a^2}\mathrm{E}\left[\frac{1}{n}\sum_{i=1}^n \left\|g_i^t - h_i^t\right\|^2\right]
$$
$$
+ \frac{4\gamma(2\omega+1)\omega}{np_a^2}\mathrm{E}\left[\frac{1}{n}\sum_{i=1}^n \left\|k_i^{t+1}\right\|^2\right].
$$

$\square$

### E.3   Proof for DASHA-PP

**Lemma 7.** *Suppose that Assumptions 3 and 8 hold. For $h_i^{t+1}$ and $k_i^{t+1}$ from Algorithm 1 (DASHA-PP) we have*

1.
$$
\mathrm{E}_{p_a}\left[\left\|h^{t+1} - \nabla f(x^{t+1})\right\|^2\right]
$$
$$
\leq \frac{2(p_a - p_{aa})\widehat{L}^2}{np_a^2}\left\|x^{t+1} - x^t\right\|^2 + \frac{2b^2(p_a - p_{aa})}{n^2 p_a^2}\sum_{i=1}^n \left\|h_i^t - \nabla f_i(x^t)\right\|^2 + (1-b)^2\left\|h^t - \nabla f(x^t)\right\|^2.
$$

2.
$$
\mathrm{E}_{p_a}\left[\left\|h_i^{t+1} - \nabla f_i(x^{t+1})\right\|^2\right]
$$
$$
\leq \frac{2(1-p_a)}{p_a}L_i^2\left\|x^{t+1} - x^t\right\|^2 + \left(\frac{2b^2(1-p_a)}{p_a} + (1-b)^2\right)\left\|h_i^t - \nabla f_i(x^t)\right\|^2, \quad \forall i \in [n].
$$

3.
$$
\left\|k_i^{t+1}\right\|^2 \leq 2L_i^2\left\|x^{t+1} - x^t\right\|^2 + 2b^2\left\|h_i^t - \nabla f_i(x^t)\right\|^2, \quad \forall i \in [n].
$$

*Proof.* First, let us proof the bound for $\mathrm{E}_k\left[\mathrm{E}_{p_a}\left[\left\|h^{t+1} - \nabla f(x^{t+1})\right\|^2\right]\right]$:

$$
\mathrm{E}_{p_a}\left[\left\|h^{t+1} - \nabla f(x^{t+1})\right\|^2\right]
$$

$$= \mathrm{E}_{p_\mathrm{a}} \left[ \left\| h^{t+1} - \mathrm{E}_{p_\mathrm{a}} \left[ h^{t+1} \right] \right\|^2 \right] + \left\| \mathrm{E}_{p_\mathrm{a}} \left[ h^{t+1} \right] - \nabla f(x^{t+1}) \right\|^2.$$

Using

$$\mathrm{E}_{p_\mathrm{a}} \left[ h_i^{t+1} \right] = h_i^t + \nabla f_i(x^{t+1}) - \nabla f_i(x^t) - b(h_i^t - \nabla f_i(x^t))$$

and (16), we have

$$\mathrm{E}_{p_\mathrm{a}} \left[ \left\| h^{t+1} - \nabla f(x^{t+1}) \right\|^2 \right]$$
$$= \mathrm{E}_{p_\mathrm{a}} \left[ \left\| h^{t+1} - \mathrm{E}_{p_\mathrm{a}} \left[ h^{t+1} \right] \right\|^2 \right] + (1-b)^2 \left\| h^t - \nabla f(x^t) \right\|^2.$$

We can use Lemma 1 with $r_i = h_i^t$ and $s_i = k_i^{t+1}$ to obtain

$$\mathrm{E}_{p_\mathrm{a}} \left[ \left\| h^{t+1} - \nabla f(x^{t+1}) \right\|^2 \right]$$

$$\leq \frac{1}{n^2 p_\mathrm{a}} \sum_{i=1}^{n} \left\| k_i^{t+1} - k_i^{t+1} \right\|^2 + \frac{p_\mathrm{a} - p_\mathrm{aa}}{n^2 p_\mathrm{a}^2} \sum_{i=1}^{n} \left\| k_i^{t+1} \right\|^2 + (1-b)^2 \left\| h^t - \nabla f(x^t) \right\|^2$$

$$= \frac{p_\mathrm{a} - p_\mathrm{aa}}{n^2 p_\mathrm{a}^2} \sum_{i=1}^{n} \left\| \nabla f_i(x^{t+1}) - \nabla f_i(x^t) - b \left( h_i^t - \nabla f_i(x^t) \right) \right\|^2 + (1-b)^2 \left\| h^t - \nabla f(x^t) \right\|^2$$

$$\overset{(15)}{\leq} \frac{2 \left( p_\mathrm{a} - p_\mathrm{aa} \right)}{n^2 p_\mathrm{a}^2} \sum_{i=1}^{n} \left\| \nabla f_i(x^{t+1}) - \nabla f_i(x^t) \right\|^2 + \frac{2 b^2 \left( p_\mathrm{a} - p_\mathrm{aa} \right)}{n^2 p_\mathrm{a}^2} \sum_{i=1}^{n} \left\| h_i^t - \nabla f_i(x^t) \right\|^2 + (1-b)^2 \left\| h^t - \nabla f(x^t) \right\|^2$$

$$\leq \frac{2 \left( p_\mathrm{a} - p_\mathrm{aa} \right) \widehat{L}^2}{n p_\mathrm{a}^2} \left\| x^{t+1} - x^t \right\|^2 + \frac{2 b^2 \left( p_\mathrm{a} - p_\mathrm{aa} \right)}{n^2 p_\mathrm{a}^2} \sum_{i=1}^{n} \left\| h_i^t - \nabla f_i(x^t) \right\|^2 + (1-b)^2 \left\| h^t - \nabla f(x^t) \right\|^2.$$

In the last in inequality, we used Assumption 3. Now, we prove the second inequality:

$$\mathrm{E}_{p_\mathrm{a}} \left[ \left\| h_i^{t+1} - \nabla f_i(x^{t+1}) \right\|^2 \right]$$

$$= \mathrm{E}_{p_\mathrm{a}} \left[ \left\| h_i^{t+1} - \mathrm{E}_{p_\mathrm{a}} \left[ h_i^{t+1} \right] \right\|^2 \right] + \left\| \mathrm{E}_{p_\mathrm{a}} \left[ h_i^{t+1} \right] - \nabla f_i(x^{t+1}) \right\|^2$$

$$= \mathrm{E}_{p_\mathrm{a}} \left[ \left\| h_i^{t+1} - \left( h_i^t + \nabla f_i(x^{t+1}) - \nabla f_i(x^t) - b(h_i^t - \nabla f_i(x^t)) \right) \right\|^2 \right] + (1-b)^2 \left\| h_i^t - \nabla f_i(x^t) \right\|^2$$

$$= \frac{(1 - p_\mathrm{a})^2}{p_\mathrm{a}} \left\| \nabla f_i(x^{t+1}) - \nabla f_i(x^t) - b(h_i^t - \nabla f_i(x^t)) \right\|^2$$

$$\quad + (1 - p_\mathrm{a}) \left\| \nabla f_i(x^{t+1}) - \nabla f_i(x^t) - b(h_i^t - \nabla f_i(x^t)) \right\|^2 + (1-b)^2 \left\| h_i^t - \nabla f_i(x^t) \right\|^2$$

$$= \frac{(1 - p_\mathrm{a})}{p_\mathrm{a}} \left\| \nabla f_i(x^{t+1}) - \nabla f_i(x^t) - b(h_i^t - \nabla f_i(x^t)) \right\|^2 + (1-b)^2 \left\| h_i^t - \nabla f_i(x^t) \right\|^2$$

$$\leq \frac{2(1 - p_\mathrm{a})}{p_\mathrm{a}} L_i^2 \left\| x^{t+1} - x^t \right\|^2 + \left( \frac{2 b^2 (1 - p_\mathrm{a})}{p_\mathrm{a}} + (1-b)^2 \right) \left\| h_i^t - \nabla f_i(x^t) \right\|^2.$$

Finally, the third inequality of the theorem follows from (15) and Assumption 3. $\qquad \square$

**Theorem 2.** *Suppose that Assumptions 1, 2, 3, 7 and 8 hold. Let us take $a = \frac{p_\mathrm{a}}{2\omega + 1}$, $b = \frac{p_\mathrm{a}}{2 - p_\mathrm{a}}$,*

$$\gamma \leq \left( L + \left[ \frac{48\omega \left( 2\omega + 1 \right)}{n p_\mathrm{a}^2} + \frac{16}{n p_\mathrm{a}^2} \left( 1 - \frac{p_\mathrm{aa}}{p_\mathrm{a}} \right) \right]^{1/2} \widehat{L} \right)^{-1},$$

*and $g_i^0 = h_i^0 = \nabla f_i(x^0)$ for all $i \in [n]$ in Algorithm 1 (DASHA-PP), then $\mathrm{E} \left[ \left\| \nabla f(\widehat{x}^T) \right\|^2 \right] \leq \frac{2\Delta_0}{\gamma T}$.*

*Proof.* Let us fix constants $\nu, \rho \in [0, \infty)$ that we will define later. Considering Lemma 6, Lemma 7, and the law of total expectation, we obtain

$$\mathrm{E} \left[ f(x^{t+1}) \right] + \frac{\gamma(2\omega + 1)}{p_\mathrm{a}} \mathrm{E} \left[ \left\| g^{t+1} - h^{t+1} \right\|^2 \right] + \frac{\gamma((2\omega + 1) p_\mathrm{a} - p_\mathrm{aa})}{n p_\mathrm{a}^2} \mathrm{E} \left[ \frac{1}{n} \sum_{i=1}^{n} \left\| g_i^{t+1} - h_i^{t+1} \right\|^2 \right]$$

$$+ \nu \mathrm{E}\left[\left\|h^{t+1} - \nabla f(x^{t+1})\right\|^2\right] + \rho \mathrm{E}\left[\frac{1}{n}\sum_{i=1}^{n}\left\|h_i^{t+1} - \nabla f_i(x^{t+1})\right\|^2\right]$$

$$= \mathrm{E}\left[f(x^{t+1})\right] + \frac{\gamma(2\omega+1)}{p_{\mathrm{a}}}\mathrm{E}\left[\left\|g^{t+1} - h^{t+1}\right\|^2\right] + \frac{\gamma((2\omega+1)\,p_{\mathrm{a}} - p_{\mathrm{aa}})}{np_{\mathrm{a}}^2}\mathrm{E}\left[\frac{1}{n}\sum_{i=1}^{n}\left\|g_i^{t+1} - h_i^{t+1}\right\|^2\right]$$

$$+ \nu \mathrm{E}\left[\mathrm{E}_{p_{\mathrm{a}}}\left[\left\|h^{t+1} - \nabla f(x^{t+1})\right\|^2\right]\right] + \rho \mathrm{E}\left[\mathrm{E}_{p_{\mathrm{a}}}\left[\frac{1}{n}\sum_{i=1}^{n}\left\|h_i^{t+1} - \nabla f_i(x^{t+1})\right\|^2\right]\right]$$

$$\leq \mathrm{E}\left[f(x^t) - \frac{\gamma}{2}\left\|\nabla f(x^t)\right\|^2 - \left(\frac{1}{2\gamma} - \frac{L}{2}\right)\left\|x^{t+1} - x^t\right\|^2 + \gamma\left\|h^t - \nabla f(x^t)\right\|^2\right]$$

$$+ \frac{\gamma(2\omega+1)}{p_{\mathrm{a}}}\mathrm{E}\left[\left\|g^t - h^t\right\|^2\right] + \frac{\gamma((2\omega+1)\,p_{\mathrm{a}} - p_{\mathrm{aa}})}{np_{\mathrm{a}}^2}\mathrm{E}\left[\frac{1}{n}\sum_{i=1}^{n}\left\|g_i^t - h_i^t\right\|^2\right]$$

$$+ \frac{4\gamma\omega(2\omega+1)}{np_{\mathrm{a}}^2}\mathrm{E}\left[2\widehat{L}^2\left\|x^{t+1} - x^t\right\|^2 + 2b^2\frac{1}{n}\sum_{i=1}^{n}\left\|h_i^t - \nabla f_i(x^t)\right\|^2\right]$$

$$+ \nu\mathrm{E}\left[\frac{2\,(p_{\mathrm{a}} - p_{\mathrm{aa}})\,\widehat{L}^2}{np_{\mathrm{a}}^2}\left\|x^{t+1} - x^t\right\|^2 + \frac{2b^2\,(p_{\mathrm{a}} - p_{\mathrm{aa}})}{n^2p_{\mathrm{a}}^2}\sum_{i=1}^{n}\left\|h_i^t - \nabla f_i(x^t)\right\|^2 + (1-b)^2\left\|h^t - \nabla f(x^t)\right\|^2\right]$$

$$+ \rho\mathrm{E}\left[\frac{2(1-p_{\mathrm{a}})}{p_{\mathrm{a}}}\widehat{L}^2\left\|x^{t+1} - x^t\right\|^2 + \left(\frac{2b^2(1-p_{\mathrm{a}})}{p_{\mathrm{a}}} + (1-b)^2\right)\frac{1}{n}\sum_{i=1}^{n}\left\|h_i^t - \nabla f_i(x^t)\right\|^2\right].$$

After rearranging the terms, we get

$$\mathrm{E}\left[f(x^{t+1})\right] + \frac{\gamma(2\omega+1)}{p_{\mathrm{a}}}\mathrm{E}\left[\left\|g^{t+1} - h^{t+1}\right\|^2\right] + \frac{\gamma((2\omega+1)\,p_{\mathrm{a}} - p_{\mathrm{aa}})}{np_{\mathrm{a}}^2}\mathrm{E}\left[\frac{1}{n}\sum_{i=1}^{n}\left\|g_i^{t+1} - h_i^{t+1}\right\|^2\right]$$

$$+ \nu\mathrm{E}\left[\left\|h^{t+1} - \nabla f(x^{t+1})\right\|^2\right] + \rho\mathrm{E}\left[\frac{1}{n}\sum_{i=1}^{n}\left\|h_i^{t+1} - \nabla f_i(x^{t+1})\right\|^2\right]$$

$$\leq \mathrm{E}\left[f(x^t)\right] - \frac{\gamma}{2}\mathrm{E}\left[\left\|\nabla f(x^t)\right\|^2\right]$$

$$+ \frac{\gamma(2\omega+1)}{p_{\mathrm{a}}}\mathrm{E}\left[\left\|g^t - h^t\right\|^2\right] + \frac{\gamma((2\omega+1)\,p_{\mathrm{a}} - p_{\mathrm{aa}})}{np_{\mathrm{a}}^2}\mathrm{E}\left[\frac{1}{n}\sum_{i=1}^{n}\left\|g_i^t - h_i^t\right\|^2\right]$$

$$- \left(\frac{1}{2\gamma} - \frac{L}{2} - \frac{8\gamma\omega\,(2\omega+1)\,\widehat{L}^2}{np_{\mathrm{a}}^2} - \nu\frac{2\,(p_{\mathrm{a}} - p_{\mathrm{aa}})\,\widehat{L}^2}{np_{\mathrm{a}}^2} - \rho\frac{2(1-p_{\mathrm{a}})\widehat{L}^2}{p_{\mathrm{a}}}\right)\mathrm{E}\left[\left\|x^{t+1} - x^t\right\|^2\right]$$

$$+ \left(\gamma + \nu(1-b)^2\right)\mathrm{E}\left[\left\|h^t - \nabla f(x^t)\right\|^2\right]$$

$$+ \left(\frac{8b^2\gamma\omega(2\omega+1)}{np_{\mathrm{a}}^2} + \nu\frac{2b^2\,(p_{\mathrm{a}} - p_{\mathrm{aa}})}{np_{\mathrm{a}}^2} + \rho\left(\frac{2b^2(1-p_{\mathrm{a}})}{p_{\mathrm{a}}} + (1-b)^2\right)\right)\mathrm{E}\left[\frac{1}{n}\sum_{i=1}^{n}\left\|h_i^t - \nabla f_i(x^t)\right\|^2\right].$$

By taking $\nu = \frac{\gamma}{b}$, one can show that $\left(\gamma + \nu(1-b)^2\right) \leq \nu$, and

$$\mathrm{E}\left[f(x^{t+1})\right] + \frac{\gamma(2\omega+1)}{p_{\mathrm{a}}}\mathrm{E}\left[\left\|g^{t+1} - h^{t+1}\right\|^2\right] + \frac{\gamma((2\omega+1)\,p_{\mathrm{a}} - p_{\mathrm{aa}})}{np_{\mathrm{a}}^2}\mathrm{E}\left[\frac{1}{n}\sum_{i=1}^{n}\left\|g_i^{t+1} - h_i^{t+1}\right\|^2\right]$$

$$+ \frac{\gamma}{b}\mathrm{E}\left[\left\|h^{t+1} - \nabla f(x^{t+1})\right\|^2\right] + \rho\mathrm{E}\left[\frac{1}{n}\sum_{i=1}^{n}\left\|h_i^{t+1} - \nabla f_i(x^{t+1})\right\|^2\right]$$

$$\leq \mathrm{E}\left[f(x^t)\right] - \frac{\gamma}{2}\mathrm{E}\left[\left\|\nabla f(x^t)\right\|^2\right]$$

$$+ \frac{\gamma(2\omega+1)}{p_{\mathrm{a}}}\mathrm{E}\left[\left\|g^t - h^t\right\|^2\right] + \frac{\gamma((2\omega+1)\,p_{\mathrm{a}} - p_{\mathrm{aa}})}{np_{\mathrm{a}}^2}\mathrm{E}\left[\frac{1}{n}\sum_{i=1}^{n}\left\|g_i^t - h_i^t\right\|^2\right]$$

$$-\left(\frac{1}{2\gamma} - \frac{L}{2} - \frac{8\gamma\omega\,(2\omega+1)\,\widehat{L}^2}{np_{\mathrm a}^2} - \frac{2\gamma\,(p_{\mathrm a} - p_{\mathrm{aa}})\,\widehat{L}^2}{bnp_{\mathrm a}^2} - \rho\frac{2(1-p_{\mathrm a})\widehat{L}^2}{p_{\mathrm a}}\right)\mathrm{E}\left[\left\|x^{t+1} - x^t\right\|^2\right]$$

$$+\frac{\gamma}{b}\mathrm{E}\left[\left\|h^t - \nabla f(x^t)\right\|^2\right]$$

$$+\left(\frac{8b^2\gamma\omega(2\omega+1)}{np_{\mathrm a}^2} + \frac{2\gamma b\,(p_{\mathrm a} - p_{\mathrm{aa}})}{np_{\mathrm a}^2} + \rho\left(\frac{2b^2(1-p_{\mathrm a})}{p_{\mathrm a}} + (1-b)^2\right)\right)\mathrm{E}\left[\frac{1}{n}\sum_{i=1}^n\left\|h_i^t - \nabla f_i(x^t)\right\|^2\right].$$

Note that $b = \frac{p_{\mathrm a}}{2 - p_{\mathrm a}}$, thus

$$\left(\frac{8b^2\gamma\omega(2\omega+1)}{np_{\mathrm a}^2} + \frac{2\gamma b\,(p_{\mathrm a} - p_{\mathrm{aa}})}{np_{\mathrm a}^2} + \rho\left(\frac{2b^2(1-p_{\mathrm a})}{p_{\mathrm a}} + (1-b)^2\right)\right)$$

$$\leq\left(\frac{8b^2\gamma\omega(2\omega+1)}{np_{\mathrm a}^2} + \frac{2\gamma b\,(p_{\mathrm a} - p_{\mathrm{aa}})}{np_{\mathrm a}^2} + \rho\,(1-b)\right).$$

And if we take $\rho = \frac{8b\gamma\omega(2\omega+1)}{np_{\mathrm a}^2} + \frac{2\gamma(p_{\mathrm a} - p_{\mathrm{aa}})}{np_{\mathrm a}^2}$, then

$$\left(\frac{8b^2\gamma\omega(2\omega+1)}{np_{\mathrm a}^2} + \frac{2\gamma b\,(p_{\mathrm a} - p_{\mathrm{aa}})}{np_{\mathrm a}^2} + \rho\,(1-b)\right) \leq \rho,$$

and

$$\mathrm{E}\left[f(x^{t+1})\right] + \frac{\gamma(2\omega+1)}{p_{\mathrm a}}\mathrm{E}\left[\left\|g^{t+1} - h^{t+1}\right\|^2\right] + \frac{\gamma((2\omega+1)\,p_{\mathrm a} - p_{\mathrm{aa}})}{np_{\mathrm a}^2}\mathrm{E}\left[\frac{1}{n}\sum_{i=1}^n\left\|g_i^{t+1} - h_i^{t+1}\right\|^2\right]$$

$$+\frac{\gamma}{b}\mathrm{E}\left[\left\|h^{t+1} - \nabla f(x^{t+1})\right\|^2\right] + \left(\frac{8b\gamma\omega(2\omega+1)}{np_{\mathrm a}^2} + \frac{2\gamma\,(p_{\mathrm a} - p_{\mathrm{aa}})}{np_{\mathrm a}^2}\right)\mathrm{E}\left[\frac{1}{n}\sum_{i=1}^n\left\|h_i^{t+1} - \nabla f_i(x^{t+1})\right\|^2\right]$$

$$\leq \mathrm{E}\left[f(x^t)\right] - \frac{\gamma}{2}\mathrm{E}\left[\left\|\nabla f(x^t)\right\|^2\right]$$

$$+\frac{\gamma(2\omega+1)}{p_{\mathrm a}}\mathrm{E}\left[\left\|g^t - h^t\right\|^2\right] + \frac{\gamma((2\omega+1)\,p_{\mathrm a} - p_{\mathrm{aa}})}{np_{\mathrm a}^2}\mathrm{E}\left[\frac{1}{n}\sum_{i=1}^n\left\|g_i^t - h_i^t\right\|^2\right]$$

$$-\left(\frac{1}{2\gamma} - \frac{L}{2} - \frac{8\gamma\omega\,(2\omega+1)\,\widehat{L}^2}{np_{\mathrm a}^2} - \frac{2\gamma\,(p_{\mathrm a} - p_{\mathrm{aa}})\,\widehat{L}^2}{bnp_{\mathrm a}^2}\right.$$

$$\left. -\frac{16b\gamma\omega(2\omega+1)(1-p_{\mathrm a})\widehat{L}^2}{np_{\mathrm a}^3} - \frac{4\gamma\,(p_{\mathrm a} - p_{\mathrm{aa}})\,(1-p_{\mathrm a})\widehat{L}^2}{np_{\mathrm a}^3}\right)\mathrm{E}\left[\left\|x^{t+1} - x^t\right\|^2\right]$$

$$+\frac{\gamma}{b}\mathrm{E}\left[\left\|h^t - \nabla f(x^t)\right\|^2\right] + \left(\frac{8b\gamma\omega(2\omega+1)}{np_{\mathrm a}^2} + \frac{2\gamma\,(p_{\mathrm a} - p_{\mathrm{aa}})}{np_{\mathrm a}^2}\right)\mathrm{E}\left[\frac{1}{n}\sum_{i=1}^n\left\|h_i^t - \nabla f_i(x^t)\right\|^2\right].$$

Let us simplify the last inequality. First, note that

$$\frac{16b\gamma\omega(2\omega+1)(1-p_{\mathrm a})\widehat{L}^2}{np_{\mathrm a}^3} \leq \frac{16\gamma\omega(2\omega+1)\widehat{L}^2}{np_{\mathrm a}^2},$$

due to $b \leq p_{\mathrm a}$. Second,

$$\frac{2\gamma\,(p_{\mathrm a} - p_{\mathrm{aa}})\,\widehat{L}^2}{bnp_{\mathrm a}^2} \leq \frac{4\gamma\,(p_{\mathrm a} - p_{\mathrm{aa}})\,\widehat{L}^2}{np_{\mathrm a}^3},$$

due to $b \geq \frac{p_{\mathrm a}}{2}$. All in all, we have

$$\mathrm{E}\left[f(x^{t+1})\right] + \frac{\gamma(2\omega+1)}{p_{\mathrm a}}\mathrm{E}\left[\left\|g^{t+1} - h^{t+1}\right\|^2\right] + \frac{\gamma((2\omega+1)\,p_{\mathrm a} - p_{\mathrm{aa}})}{np_{\mathrm a}^2}\mathrm{E}\left[\frac{1}{n}\sum_{i=1}^n\left\|g_i^{t+1} - h_i^{t+1}\right\|^2\right]$$

$$+\frac{\gamma}{b}\mathrm{E}\left[\left\|h^{t+1} - \nabla f(x^{t+1})\right\|^2\right] + \left(\frac{8b\gamma\omega(2\omega+1)}{np_{\mathrm a}^2} + \frac{2\gamma\,(p_{\mathrm a} - p_{\mathrm{aa}})}{np_{\mathrm a}^2}\right)\mathrm{E}\left[\frac{1}{n}\sum_{i=1}^n\left\|h_i^{t+1} - \nabla f_i(x^{t+1})\right\|^2\right]$$

$$\leq \mathrm{E}\left[f(x^t)\right] - \frac{\gamma}{2}\mathrm{E}\left[\left\|\nabla f(x^t)\right\|^2\right]$$

$$+ \frac{\gamma(2\omega + 1)}{p_a}\mathrm{E}\left[\left\|g^t - h^t\right\|^2\right] + \frac{\gamma((2\omega + 1)\,p_a - p_{aa})}{np_a^2}\mathrm{E}\left[\frac{1}{n}\sum_{i=1}^n \left\|g_i^t - h_i^t\right\|^2\right]$$

$$- \left(\frac{1}{2\gamma} - \frac{L}{2} - \frac{24\gamma\omega\,(2\omega + 1)\,\widehat{L}^2}{np_a^2} - \frac{8\gamma\,(p_a - p_{aa})\,\widehat{L}^2}{np_a^3}\right)\mathrm{E}\left[\left\|x^{t+1} - x^t\right\|^2\right]$$

$$+ \frac{\gamma}{b}\mathrm{E}\left[\left\|h^t - \nabla f(x^t)\right\|^2\right] + \left(\frac{8b\gamma\omega(2\omega + 1)}{np_a^2} + \frac{2\gamma\,(p_a - p_{aa})}{np_a^2}\right)\mathrm{E}\left[\frac{1}{n}\sum_{i=1}^n \left\|h_i^t - \nabla f_i(x^t)\right\|^2\right].$$

Using Lemma 4 and the assumption about $\gamma$, we get

$$\mathrm{E}\left[f(x^{t+1})\right] + \frac{\gamma(2\omega + 1)}{p_a}\mathrm{E}\left[\left\|g^{t+1} - h^{t+1}\right\|^2\right] + \frac{\gamma((2\omega + 1)\,p_a - p_{aa})}{np_a^2}\mathrm{E}\left[\frac{1}{n}\sum_{i=1}^n \left\|g_i^{t+1} - h_i^{t+1}\right\|^2\right]$$

$$+ \frac{\gamma}{b}\mathrm{E}\left[\left\|h^{t+1} - \nabla f(x^{t+1})\right\|^2\right] + \left(\frac{8b\gamma\omega(2\omega + 1)}{np_a^2} + \frac{2\gamma\,(p_a - p_{aa})}{np_a^2}\right)\mathrm{E}\left[\frac{1}{n}\sum_{i=1}^n \left\|h_i^{t+1} - \nabla f_i(x^{t+1})\right\|^2\right]$$

$$\leq \mathrm{E}\left[f(x^t)\right] - \frac{\gamma}{2}\mathrm{E}\left[\left\|\nabla f(x^t)\right\|^2\right]$$

$$+ \frac{\gamma(2\omega + 1)}{p_a}\mathrm{E}\left[\left\|g^t - h^t\right\|^2\right] + \frac{\gamma((2\omega + 1)\,p_a - p_{aa})}{np_a^2}\mathrm{E}\left[\frac{1}{n}\sum_{i=1}^n \left\|g_i^t - h_i^t\right\|^2\right]$$

$$+ \frac{\gamma}{b}\mathrm{E}\left[\left\|h^t - \nabla f(x^t)\right\|^2\right] + \left(\frac{8b\gamma\omega(2\omega + 1)}{np_a^2} + \frac{2\gamma\,(p_a - p_{aa})}{np_a^2}\right)\mathrm{E}\left[\frac{1}{n}\sum_{i=1}^n \left\|h_i^t - \nabla f_i(x^t)\right\|^2\right].$$

It is left to apply Lemma 3 with

$$\Psi^t \;=\; \frac{(2\omega + 1)}{p_a}\mathrm{E}\left[\left\|g^t - h^t\right\|^2\right] + \frac{((2\omega + 1)\,p_a - p_{aa})}{np_a^2}\mathrm{E}\left[\frac{1}{n}\sum_{i=1}^n \left\|g_i^t - h_i^t\right\|^2\right]$$

$$+\; \frac{1}{b}\mathrm{E}\left[\left\|h^t - \nabla f(x^t)\right\|^2\right] + \left(\frac{8b\omega(2\omega + 1)}{np_a^2} + \frac{2\,(p_a - p_{aa})}{np_a^2}\right)\mathrm{E}\left[\frac{1}{n}\sum_{i=1}^n \left\|h_i^t - \nabla f_i(x^t)\right\|^2\right]$$

to conclude the proof. $\qquad\square$

### E.4 Proof for DASHA-PP-PAGE

Let us denote

$$k_{i,1}^{t+1} := \nabla f_i(x^{t+1}) - \nabla f_i(x^t) - \frac{b}{p_{\text{page}}}\left(h_i^t - \nabla f_i(x^t)\right),$$

$$k_{i,2}^{t+1} := \frac{1}{B}\sum_{j \in I_i^t}\left(\nabla f_{ij}(x^{t+1}) - \nabla f_{ij}(x^t)\right),$$

$$h_{i,1}^{t+1} := \begin{cases} h_i^t + \frac{1}{p_a}k_{i,1}^{t+1}, & i^{\text{th}} \text{ node is } \textit{participating}, \\ h_i^t, & \text{otherwise}, \end{cases}$$

$$h_{i,2}^{t+1} := \begin{cases} h_i^t + \frac{1}{p_a}k_{i,2}^{t+1}, & i^{\text{th}} \text{ node is } \textit{participating}, \\ h_i^t, & \text{otherwise}, \end{cases}$$

$h_1^{t+1} := \frac{1}{n}\sum_{i=1}^n h_{i,1}^{t+1}$, and $h_2^{t+1} := \frac{1}{n}\sum_{i=1}^n h_{i,2}^{t+1}$. Note, that

$$h^{t+1} = \begin{cases} h_1^{t+1}, & \text{with probability } p_{\text{page}}, \\ h_2^{t+1}, & \text{with probability } 1 - p_{\text{page}}. \end{cases}$$

**Lemma 8.** *Suppose that Assumptions 3, 4, and 8 hold. For $h_i^{t+1}$ and $k_i^{t+1}$ from Algorithm 1 (DASHA-PP-PAGE) we have*

*1.*

$$
\mathrm{E}_B \left[ \mathrm{E}_{p_{\mathrm{a}}} \left[ \mathrm{E}_{p_{page}} \left[ \left\| h^{t+1} - \nabla f(x^{t+1}) \right\|^2 \right] \right] \right]
$$
$$
\leq \left( \frac{2 \left( p_{\mathrm{a}} - p_{\mathrm{aa}} \right) \widehat{L}^2}{n p_{\mathrm{a}}^2} + \frac{(1 - p_{page}) L_{\max}^2}{n p_{\mathrm{a}} B} \right) \left\| x^{t+1} - x^t \right\|^2
$$
$$
+ \frac{2 \left( p_{\mathrm{a}} - p_{\mathrm{aa}} \right) b^2}{n^2 p_{\mathrm{a}}^2 p_{page}} \sum_{i=1}^n \left\| h_i^t - \nabla f_i(x^t) \right\|^2 + \left( p_{page} \left( 1 - \frac{b}{p_{page}} \right)^2 + (1 - p_{page}) \right) \left\| h^t - \nabla f(x^t) \right\|^2 .
$$

*2.*

$$
\mathrm{E}_B \left[ \mathrm{E}_{p_{\mathrm{a}}} \left[ \mathrm{E}_{p_{page}} \left[ \left\| h_i^{t+1} - \nabla f_i(x^{t+1}) \right\|^2 \right] \right] \right]
$$
$$
\leq \left( \frac{2 \left( 1 - p_{\mathrm{a}} \right) L_i^2}{p_{\mathrm{a}}} + \frac{(1 - p_{page}) L_{\max}^2}{p_{\mathrm{a}} B} \right) \left\| x^{t+1} - x^t \right\|^2
$$
$$
+ \left( \frac{2 \left( 1 - p_{\mathrm{a}} \right) b^2}{p_{\mathrm{a}} p_{page}} + p_{page} \left( 1 - \frac{b}{p_{page}} \right)^2 + (1 - p_{page}) \right) \left\| h_i^t - \nabla f_i(x^t) \right\|^2 , \quad \forall i \in [n].
$$

*3.*

$$
\mathrm{E}_B \left[ \mathrm{E}_{p_{page}} \left[ \left\| k_i^{t+1} \right\|^2 \right] \right]
$$
$$
\leq \left( 2 L_i^2 + \frac{(1 - p_{page}) L_{\max}^2}{B} \right) \left\| x^{t+1} - x^t \right\|^2 + \frac{2 b^2}{p_{page}} \left\| h_i^t - \nabla f_i(x^t) \right\|^2 , \quad \forall i \in [n].
$$

*Proof.* First, we prove the first inequality of the theorem:

$$
\mathrm{E}_B \left[ \mathrm{E}_{p_{\mathrm{a}}} \left[ \mathrm{E}_{p_{page}} \left[ \left\| h^{t+1} - \nabla f(x^{t+1}) \right\|^2 \right] \right] \right]
$$
$$
= p_{page} \mathrm{E}_B \left[ \mathrm{E}_{p_{\mathrm{a}}} \left[ \left\| h_1^{t+1} - \nabla f(x^{t+1}) \right\|^2 \right] \right] + (1 - p_{page}) \mathrm{E}_B \left[ \mathrm{E}_{p_{\mathrm{a}}} \left[ \left\| h_2^{t+1} - \nabla f(x^{t+1}) \right\|^2 \right] \right].
$$

Using

$$
\mathrm{E}_B \left[ \mathrm{E}_{p_{\mathrm{a}}} \left[ h_{i,1}^{t+1} \right] \right] =
$$
$$
= p_{\mathrm{a}} h_i^t + \nabla f_i(x^{t+1}) - \nabla f_i(x^t) - \frac{b}{p_{page}} \left( h_i^t - \nabla f_i(x^t) \right) + (1 - p_{\mathrm{a}}) h_i^t
$$
$$
= h_i^t + \nabla f_i(x^{t+1}) - \nabla f_i(x^t) - \frac{b}{p_{page}} \left( h_i^t - \nabla f_i(x^t) \right).
$$

and

$$
\mathrm{E}_B \left[ \mathrm{E}_{p_{\mathrm{a}}} \left[ h_{i,2}^{t+1} \right] \right] =
$$
$$
= p_{\mathrm{a}} h_i^t + \mathrm{E}_B \left[ \frac{1}{B} \sum_{j \in I_i^t} \left( \nabla f_{ij}(x^{t+1}) - \nabla f_{ij}(x^t) \right) \right] + (1 - p_{\mathrm{a}}) h_i^t
$$
$$
= h_i^t + \nabla f_i(x^{t+1}) - \nabla f_i(x^t),
$$

we obtain

$$
\mathrm{E}_B \left[ \mathrm{E}_{p_{\mathrm{a}}} \left[ \mathrm{E}_{p_{page}} \left[ \left\| h^{t+1} - \nabla f(x^{t+1}) \right\|^2 \right] \right] \right]
$$
$$
\stackrel{(16)}{=} p_{page} \mathrm{E}_{p_{\mathrm{a}}} \left[ \left\| h_1^{t+1} - \mathrm{E}_{p_{\mathrm{a}}} \left[ h_1^{t+1} \right] \right\|^2 \right] + (1 - p_{page}) \mathrm{E}_B \left[ \mathrm{E}_{p_{\mathrm{a}}} \left[ \left\| h_2^{t+1} - \mathrm{E}_B \left[ \mathrm{E}_{p_{\mathrm{a}}} \left[ h_2^{t+1} \right] \right] \right\|^2 \right] \right]
$$

$$+ p_{\text{page}} \left\| \mathrm{E}_{p_a} \left[ h_1^{t+1} \right] - \nabla f(x^{t+1}) \right\|^2 + (1 - p_{\text{page}}) \left\| \mathrm{E}_B \left[ \mathrm{E}_{p_a} \left[ h_2^{t+1} \right] \right] - \nabla f(x^{t+1}) \right\|^2$$

$$= p_{\text{page}} \mathrm{E}_{p_a} \left[ \left\| h_1^{t+1} - \mathrm{E}_{p_a} \left[ h_1^{t+1} \right] \right\|^2 \right] + (1 - p_{\text{page}}) \mathrm{E}_B \left[ \mathrm{E}_{p_a} \left[ \left\| h_2^{t+1} - \mathrm{E}_B \left[ \mathrm{E}_{p_a} \left[ h_2^{t+1} \right] \right] \right\|^2 \right] \right]$$

$$+ \left( p_{\text{page}} \left( 1 - \frac{b}{p_{\text{page}}} \right)^2 + (1 - p_{\text{page}}) \right) \left\| h^t - \nabla f(x^t) \right\|^2. \tag{24}$$

Next, we consider $\mathrm{E}_{p_a} \left[ \left\| h_1^{t+1} - \mathrm{E}_{p_a} \left[ h_1^{t+1} \right] \right\|^2 \right]$. We can use Lemma 1 with $r_i = h_i^t$ and $s_i = k_{i,1}^{t+1}$ to obtain

$$\mathrm{E}_{p_a} \left[ \left\| h_1^{t+1} - \mathrm{E}_{p_a} \left[ h_1^{t+1} \right] \right\|^2 \right]$$

$$\leq \frac{1}{n^2 p_a} \sum_{i=1}^n \left\| k_{i,1}^{t+1} - k_{i,1}^{t+1} \right\|^2 + \frac{p_a - p_{aa}}{n^2 p_a^2} \sum_{i=1}^n \left\| k_{i,1}^{t+1} \right\|^2$$

$$= \frac{p_a - p_{aa}}{n^2 p_a^2} \sum_{i=1}^n \left\| \nabla f_i(x^{t+1}) - \nabla f_i(x^t) - \frac{b}{p_{\text{page}}} \left( h_i^t - \nabla f_i(x^t) \right) \right\|^2$$

$$\overset{(15)}{\leq} \frac{2 (p_a - p_{aa})}{n^2 p_a^2} \sum_{i=1}^n \left\| \nabla f_i(x^{t+1}) - \nabla f_i(x^t) \right\|^2 + \frac{2 (p_a - p_{aa}) b^2}{n^2 p_a^2 p_{\text{page}}^2} \sum_{i=1}^n \left\| h_i^t - \nabla f_i(x^t) \right\|^2.$$

From Assumption 3, we have

$$\mathrm{E}_{p_a} \left[ \left\| h_1^{t+1} - \mathrm{E}_{p_a} \left[ h_1^{t+1} \right] \right\|^2 \right]$$

$$\leq \frac{2 (p_a - p_{aa}) \widehat{L}^2}{n p_a^2} \left\| x^{t+1} - x^t \right\|^2 + \frac{2 (p_a - p_{aa}) b^2}{n^2 p_a^2 p_{\text{page}}^2} \sum_{i=1}^n \left\| h_i^t - \nabla f_i(x^t) \right\|^2. \tag{25}$$

Now, we prove the bound for $\mathrm{E}_B \left[ \mathrm{E}_{p_a} \left[ \left\| h_2^{t+1} - \mathrm{E}_B \left[ \mathrm{E}_{p_a} \left[ h_2^{t+1} \right] \right] \right\|^2 \right] \right]$. Considering that mini-batches in the algorithm are independent, we can use Lemma 1 with $r_i = h_i^t$ and $s_i = k_{i,2}^{t+1}$ to obtain

$$\mathrm{E}_B \left[ \mathrm{E}_{p_a} \left[ \left\| h_2^{t+1} - \mathrm{E}_B \left[ \mathrm{E}_{p_a} \left[ h_2^{t+1} \right] \right] \right\|^2 \right] \right]$$

$$\leq \frac{1}{n^2 p_a} \sum_{i=1}^n \mathrm{E}_B \left[ \left\| k_{i,2}^{t+1} - \mathrm{E}_B \left[ k_{i,2}^{t+1} \right] \right\|^2 \right] + \frac{p_a - p_{aa}}{n^2 p_a^2} \sum_{i=1}^n \left\| \mathrm{E}_B \left[ k_{i,2}^{t+1} \right] \right\|^2$$

$$= \frac{1}{n^2 p_a} \sum_{i=1}^n \mathrm{E}_B \left[ \left\| \frac{1}{B} \sum_{j \in I_i^t} \left( \nabla f_{ij}(x^{t+1}) - \nabla f_{ij}(x^t) \right) - \left( \nabla f_i(x^{t+1}) - \nabla f_i(x^t) \right) \right\|^2 \right]$$

$$+ \frac{p_a - p_{aa}}{n^2 p_a^2} \sum_{i=1}^n \left\| \nabla f_i(x^{t+1}) - \nabla f_i(x^t) \right\|^2$$

$$= \frac{1}{n^2 p_a B^2} \sum_{i=1}^n \mathrm{E}_B \left[ \sum_{j \in I_i^t} \left\| \left( \nabla f_{ij}(x^{t+1}) - \nabla f_{ij}(x^t) \right) - \left( \nabla f_i(x^{t+1}) - \nabla f_i(x^t) \right) \right\|^2 \right]$$

$$+ \frac{p_a - p_{aa}}{n^2 p_a^2} \sum_{i=1}^n \left\| \nabla f_i(x^{t+1}) - \nabla f_i(x^t) \right\|^2$$

$$= \frac{1}{n^2 p_a B m} \sum_{i=1}^n \sum_{j=1}^m \left\| \left( \nabla f_{ij}(x^{t+1}) - \nabla f_{ij}(x^t) \right) - \left( \nabla f_i(x^{t+1}) - \nabla f_i(x^t) \right) \right\|^2$$

$$+ \frac{p_a - p_{aa}}{n^2 p_a^2} \sum_{i=1}^n \left\| \nabla f_i(x^{t+1}) - \nabla f_i(x^t) \right\|^2$$

$$\leq \frac{1}{n^2 p_a B m} \sum_{i=1}^n \sum_{j=1}^m \left\| \nabla f_{ij}(x^{t+1}) - \nabla f_{ij}(x^t) \right\|^2 + \frac{p_a - p_{aa}}{n^2 p_a^2} \sum_{i=1}^n \left\| \nabla f_i(x^{t+1}) - \nabla f_i(x^t) \right\|^2.$$

Next, we use Assumptions 3 and 4 to get

$$\mathrm{E}_B\left[\mathrm{E}_{p_\mathrm{a}}\left[\left\|h_2^{t+1} - \mathrm{E}_B\left[\mathrm{E}_{p_\mathrm{a}}\left[h_2^{t+1}\right]\right]\right\|^2\right]\right] \leq \left(\frac{L_{\max}^2}{np_\mathrm{a}B} + \frac{(p_\mathrm{a} - p_{\mathrm{aa}})\widehat{L}^2}{np_\mathrm{a}^2}\right)\left\|x^{t+1} - x^t\right\|^2. \qquad (26)$$

Applying (25) and (26) into (24), we get

$$\mathrm{E}_B\left[\mathrm{E}_{p_\mathrm{a}}\left[\mathrm{E}_{p_{\mathrm{page}}}\left[\left\|h^{t+1} - \nabla f(x^{t+1})\right\|^2\right]\right]\right]$$

$$\leq p_{\mathrm{page}}\left(\frac{2(p_\mathrm{a} - p_{\mathrm{aa}})\widehat{L}^2}{np_\mathrm{a}^2}\left\|x^{t+1} - x^t\right\|^2 + \frac{2(p_\mathrm{a} - p_{\mathrm{aa}})b^2}{n^2p_\mathrm{a}^2p_{\mathrm{page}}^2}\sum_{i=1}^n\left\|h_i^t - \nabla f_i(x^t)\right\|^2\right) +$$

$$+ (1 - p_{\mathrm{page}})\left(\frac{L_{\max}^2}{np_\mathrm{a}B} + \frac{(p_\mathrm{a} - p_{\mathrm{aa}})\widehat{L}^2}{np_\mathrm{a}^2}\right)\left\|x^{t+1} - x^t\right\|^2$$

$$+ \left(p_{\mathrm{page}}\left(1 - \frac{b}{p_{\mathrm{page}}}\right)^2 + (1 - p_{\mathrm{page}})\right)\left\|h^t - \nabla f(x^t)\right\|^2$$

$$\leq \left(\frac{2(p_\mathrm{a} - p_{\mathrm{aa}})\widehat{L}^2}{np_\mathrm{a}^2} + \frac{(1 - p_{\mathrm{page}})L_{\max}^2}{np_\mathrm{a}B}\right)\left\|x^{t+1} - x^t\right\|^2$$

$$+ \frac{2(p_\mathrm{a} - p_{\mathrm{aa}})b^2}{n^2p_\mathrm{a}^2p_{\mathrm{page}}}\sum_{i=1}^n\left\|h_i^t - \nabla f_i(x^t)\right\|^2 + \left(p_{\mathrm{page}}\left(1 - \frac{b}{p_{\mathrm{page}}}\right)^2 + (1 - p_{\mathrm{page}})\right)\left\|h^t - \nabla f(x^t)\right\|^2.$$

The proof of the second inequality almost repeats the previous one:

$$\mathrm{E}_B\left[\mathrm{E}_{p_\mathrm{a}}\left[\mathrm{E}_{p_{\mathrm{page}}}\left[\left\|h_i^{t+1} - \nabla f_i(x^{t+1})\right\|^2\right]\right]\right]$$

$$= p_{\mathrm{page}}\mathrm{E}_B\left[\mathrm{E}_{p_\mathrm{a}}\left[\left\|h_{i,1}^{t+1} - \nabla f_i(x^{t+1})\right\|^2\right]\right] + (1 - p_{\mathrm{page}})\mathrm{E}_B\left[\mathrm{E}_{p_\mathrm{a}}\left[\left\|h_{i,2}^{t+1} - \nabla f_i(x^{t+1})\right\|^2\right]\right]$$

$$\overset{(16)}{=} p_{\mathrm{page}}\mathrm{E}_B\left[\mathrm{E}_{p_\mathrm{a}}\left[\left\|h_{i,1}^{t+1} - \mathrm{E}_B\left[\mathrm{E}_{p_\mathrm{a}}\left[h_{i,1}^{t+1}\right]\right]\right\|^2\right]\right] + (1 - p_{\mathrm{page}})\mathrm{E}_B\left[\mathrm{E}_{p_\mathrm{a}}\left[\left\|h_{i,2}^{t+1} - \mathrm{E}_B\left[\mathrm{E}_{p_\mathrm{a}}\left[h_{i,2}^{t+1}\right]\right]\right\|^2\right]\right]$$

$$+ p_{\mathrm{page}}\left\|\mathrm{E}_B\left[\mathrm{E}_{p_\mathrm{a}}\left[h_{i,1}^{t+1}\right]\right] - \nabla f_i(x^{t+1})\right\|^2 + (1 - p_{\mathrm{page}})\left\|\mathrm{E}_B\left[\mathrm{E}_{p_\mathrm{a}}\left[h_{i,2}^{t+1}\right]\right] - \nabla f_i(x^{t+1})\right\|^2$$

$$= p_{\mathrm{page}}\mathrm{E}_B\left[\mathrm{E}_{p_\mathrm{a}}\left[\left\|h_{i,1}^{t+1} - \mathrm{E}_B\left[\mathrm{E}_{p_\mathrm{a}}\left[h_{i,1}^{t+1}\right]\right]\right\|^2\right]\right] + (1 - p_{\mathrm{page}})\mathrm{E}_B\left[\mathrm{E}_{p_\mathrm{a}}\left[\left\|h_{i,2}^{t+1} - \mathrm{E}_B\left[\mathrm{E}_{p_\mathrm{a}}\left[h_{i,2}^{t+1}\right]\right]\right\|^2\right]\right]$$

$$+ \left(p_{\mathrm{page}}\left(1 - \frac{b}{p_{\mathrm{page}}}\right)^2 + (1 - p_{\mathrm{page}})\right)\left\|h_i^t - \nabla f_i(x^t)\right\|^2. \qquad (27)$$

Let us consider $\mathrm{E}_B\left[\mathrm{E}_{p_\mathrm{a}}\left[\left\|h_{i,1}^{t+1} - \mathrm{E}_B\left[\mathrm{E}_{p_\mathrm{a}}\left[h_{i,1}^{t+1}\right]\right]\right\|^2\right]\right]$:

$$\mathrm{E}_B\left[\mathrm{E}_{p_\mathrm{a}}\left[\left\|h_{i,1}^{t+1} - \mathrm{E}_B\left[\mathrm{E}_{p_\mathrm{a}}\left[h_{i,1}^{t+1}\right]\right]\right\|^2\right]\right]$$

$$= \mathrm{E}_{p_\mathrm{a}}\left[\left\|h_{i,1}^{t+1} - \mathrm{E}_B\left[\mathrm{E}_{p_\mathrm{a}}\left[h_{i,1}^{t+1}\right]\right]\right\|^2\right]$$

$$= p_\mathrm{a}\left\|h_i^t + \frac{1}{p_\mathrm{a}}k_{i,1}^{t+1} - \left(h_i^t + \nabla f_i(x^{t+1}) - \nabla f_i(x^t) - \frac{b}{p_{\mathrm{page}}}\left(h_i^t - \nabla f_i(x^t)\right)\right)\right\|^2$$

$$+ (1 - p_\mathrm{a})\left\|h_i^t - \left(h_i^t + \nabla f_i(x^{t+1}) - \nabla f_i(x^t) - \frac{b}{p_{\mathrm{page}}}\left(h_i^t - \nabla f_i(x^t)\right)\right)\right\|^2$$

$$= \frac{(1 - p_\mathrm{a})^2}{p_\mathrm{a}}\left\|\nabla f_i(x^{t+1}) - \nabla f_i(x^t) - \frac{b}{p_{\mathrm{page}}}\left(h_i^t - \nabla f_i(x^t)\right)\right\|^2$$

$$+ (1 - p_\mathrm{a})\left\|\nabla f_i(x^{t+1}) - \nabla f_i(x^t) - \frac{b}{p_{\mathrm{page}}}\left(h_i^t - \nabla f_i(x^t)\right)\right\|^2$$

$$= \frac{1 - p_\mathrm{a}}{p_\mathrm{a}}\left\|\nabla f_i(x^{t+1}) - \nabla f_i(x^t) - \frac{b}{p_{\mathrm{page}}}\left(h_i^t - \nabla f_i(x^t)\right)\right\|^2.$$

Considering (15) and Assumption 3, we obtain

$$\mathrm{E}_B\left[\mathrm{E}_{p_\mathrm{a}}\left[\left\|h_{i,1}^{t+1} - \mathrm{E}_B\left[\mathrm{E}_{p_\mathrm{a}}\left[h_{i,1}^{t+1}\right]\right]\right\|^2\right]\right]$$

$$\leq \frac{2\left(1 - p_\mathrm{a}\right)L_i^2}{p_\mathrm{a}}\left\|x^{t+1} - x^t\right\|^2 + \frac{2\left(1 - p_\mathrm{a}\right)b^2}{p_\mathrm{a}p_\mathrm{page}^2}\left\|h_i^t - \nabla f_i(x^t)\right\|^2. \tag{28}$$

Next, we obtain the bound for $\mathrm{E}_B\left[\mathrm{E}_{p_\mathrm{a}}\left[\left\|h_{i,2}^{t+1} - \mathrm{E}_B\left[\mathrm{E}_{p_\mathrm{a}}\left[h_{i,2}^{t+1}\right]\right]\right\|^2\right]\right]$:

$$\mathrm{E}_B\left[\mathrm{E}_{p_\mathrm{a}}\left[\left\|h_{i,2}^{t+1} - \mathrm{E}_B\left[\mathrm{E}_{p_\mathrm{a}}\left[h_{i,2}^{t+1}\right]\right]\right\|^2\right]\right]$$

$$= p_\mathrm{a}\mathrm{E}_B\left[\left\|h_i^t + \frac{1}{p_\mathrm{a}}k_{i,2}^{t+1} - \left(h_i^t + \nabla f_i(x^{t+1}) - \nabla f_i(x^t)\right)\right\|^2\right]$$

$$+ (1 - p_\mathrm{a})\mathrm{E}_B\left[\left\|h_i^t - \left(h_i^t + \nabla f_i(x^{t+1}) - \nabla f_i(x^t)\right)\right\|^2\right]$$

$$= p_\mathrm{a}\mathrm{E}_B\left[\left\|\frac{1}{p_\mathrm{a}}k_{i,2}^{t+1} - \left(\nabla f_i(x^{t+1}) - \nabla f_i(x^t)\right)\right\|^2\right]$$

$$+ (1 - p_\mathrm{a})\left\|\nabla f_i(x^{t+1}) - \nabla f_i(x^t)\right\|^2$$

$$\overset{(16)}{=} \frac{1}{p_\mathrm{a}}\mathrm{E}_B\left[\left\|k_{i,2}^{t+1} - \left(\nabla f_i(x^{t+1}) - \nabla f_i(x^t)\right)\right\|^2\right] + \frac{(1 - p_\mathrm{a})^2}{p_\mathrm{a}}\left\|\nabla f_i(x^{t+1}) - \nabla f_i(x^t)\right\|^2$$

$$+ (1 - p_\mathrm{a})\left\|\nabla f_i(x^{t+1}) - \nabla f_i(x^t)\right\|^2$$

$$= \frac{1}{p_\mathrm{a}}\mathrm{E}_B\left[\left\|k_{i,2}^{t+1} - \left(\nabla f_i(x^{t+1}) - \nabla f_i(x^t)\right)\right\|^2\right] + \frac{1 - p_\mathrm{a}}{p_\mathrm{a}}\left\|\nabla f_i(x^{t+1}) - \nabla f_i(x^t)\right\|^2$$

$$\leq \frac{1}{p_\mathrm{a}}\mathrm{E}_B\left[\left\|k_{i,2}^{t+1} - \left(\nabla f_i(x^{t+1}) - \nabla f_i(x^t)\right)\right\|^2\right] + \frac{(1 - p_\mathrm{a})L_i^2}{p_\mathrm{a}}\left\|x^{t+1} - x^t\right\|^2, \tag{29}$$

where we used Assumption 3. By plugging (28) and (29) into (27), we get

$$\mathrm{E}_B\left[\mathrm{E}_{p_\mathrm{a}}\left[\mathrm{E}_{p_\mathrm{page}}\left[\left\|h_i^{t+1} - \nabla f_i(x^{t+1})\right\|^2\right]\right]\right]$$

$$\leq p_\mathrm{page}\left(\frac{2\left(1 - p_\mathrm{a}\right)L_i^2}{p_\mathrm{a}}\left\|x^{t+1} - x^t\right\|^2 + \frac{2\left(1 - p_\mathrm{a}\right)b^2}{p_\mathrm{a}p_\mathrm{page}^2}\left\|h_i^t - \nabla f_i(x^t)\right\|^2\right)$$

$$+ (1 - p_\mathrm{page})\left(\frac{1}{p_\mathrm{a}}\mathrm{E}_B\left[\left\|k_{i,2}^{t+1} - \left(\nabla f_i(x^{t+1}) - \nabla f_i(x^t)\right)\right\|^2\right] + \frac{(1 - p_\mathrm{a})L_i^2}{p_\mathrm{a}}\left\|x^{t+1} - x^t\right\|^2\right)$$

$$+ \left(p_\mathrm{page}\left(1 - \frac{b}{p_\mathrm{page}}\right)^2 + (1 - p_\mathrm{page})\right)\left\|h_i^t - \nabla f_i(x^t)\right\|^2$$

$$\leq \frac{2\left(1 - p_\mathrm{a}\right)L_i^2}{p_\mathrm{a}}\left\|x^{t+1} - x^t\right\|^2 + \frac{1 - p_\mathrm{page}}{p_\mathrm{a}}\mathrm{E}_B\left[\left\|k_{i,2}^{t+1} - \left(\nabla f_i(x^{t+1}) - \nabla f_i(x^t)\right)\right\|^2\right]$$

$$+ \left(\frac{2\left(1 - p_\mathrm{a}\right)b^2}{p_\mathrm{a}p_\mathrm{page}} + p_\mathrm{page}\left(1 - \frac{b}{p_\mathrm{page}}\right)^2 + (1 - p_\mathrm{page})\right)\left\|h_i^t - \nabla f_i(x^t)\right\|^2.$$

From the independence of elements in the mini-batch, we obtain

$$\mathrm{E}_B\left[\mathrm{E}_{p_\mathrm{a}}\left[\mathrm{E}_{p_\mathrm{page}}\left[\left\|h_i^{t+1} - \nabla f_i(x^{t+1})\right\|^2\right]\right]\right]$$

$$\leq \frac{2\left(1 - p_\mathrm{a}\right)L_i^2}{p_\mathrm{a}}\left\|x^{t+1} - x^t\right\|^2 + \frac{1 - p_\mathrm{page}}{p_\mathrm{a}}\mathrm{E}_B\left[\left\|\frac{1}{B}\sum_{j \in I_i^t}\left(\nabla f_{ij}(x^{t+1}) - \nabla f_{ij}(x^t)\right) - \left(\nabla f_i(x^{t+1}) - \nabla f_i(x^t)\right)\right\|^2\right]$$

$$+ \left(\frac{2\left(1 - p_\mathrm{a}\right)b^2}{p_\mathrm{a}p_\mathrm{page}} + p_\mathrm{page}\left(1 - \frac{b}{p_\mathrm{page}}\right)^2 + (1 - p_\mathrm{page})\right)\left\|h_i^t - \nabla f_i(x^t)\right\|^2$$

$$
= \frac{2\left(1-p_{\mathrm{a}}\right)L_i^2}{p_{\mathrm{a}}}\left\|x^{t+1}-x^t\right\|^2 + \frac{1-p_{\text{page}}}{p_{\mathrm{a}}B^2}\mathrm{E}_B\left[\sum_{j\in I_i^t}\left\|\left(\nabla f_{ij}(x^{t+1})-\nabla f_{ij}(x^t)\right)-\left(\nabla f_i(x^{t+1})-\nabla f_i(x^t)\right)\right\|^2\right]
$$

$$
+ \left(\frac{2\left(1-p_{\mathrm{a}}\right)b^2}{p_{\mathrm{a}}p_{\text{page}}} + p_{\text{page}}\left(1-\frac{b}{p_{\text{page}}}\right)^2 + (1-p_{\text{page}})\right)\left\|h_i^t-\nabla f_i(x^t)\right\|^2
$$

$$
= \frac{2\left(1-p_{\mathrm{a}}\right)L_i^2}{p_{\mathrm{a}}}\left\|x^{t+1}-x^t\right\|^2 + \frac{1-p_{\text{page}}}{mp_{\mathrm{a}}B}\sum_{j=1}^m\left\|\left(\nabla f_{ij}(x^{t+1})-\nabla f_{ij}(x^t)\right)-\left(\nabla f_i(x^{t+1})-\nabla f_i(x^t)\right)\right\|^2
$$

$$
+ \left(\frac{2\left(1-p_{\mathrm{a}}\right)b^2}{p_{\mathrm{a}}p_{\text{page}}} + p_{\text{page}}\left(1-\frac{b}{p_{\text{page}}}\right)^2 + (1-p_{\text{page}})\right)\left\|h_i^t-\nabla f_i(x^t)\right\|^2
$$

$$
\leq \frac{2\left(1-p_{\mathrm{a}}\right)L_i^2}{p_{\mathrm{a}}}\left\|x^{t+1}-x^t\right\|^2 + \frac{1-p_{\text{page}}}{mp_{\mathrm{a}}B}\sum_{j=1}^m\left\|\nabla f_{ij}(x^{t+1})-\nabla f_{ij}(x^t)\right\|^2
$$

$$
+ \left(\frac{2\left(1-p_{\mathrm{a}}\right)b^2}{p_{\mathrm{a}}p_{\text{page}}} + p_{\text{page}}\left(1-\frac{b}{p_{\text{page}}}\right)^2 + (1-p_{\text{page}})\right)\left\|h_i^t-\nabla f_i(x^t)\right\|^2
$$

$$
\leq \left(\frac{2\left(1-p_{\mathrm{a}}\right)L_i^2}{p_{\mathrm{a}}} + \frac{(1-p_{\text{page}})L_{\max}^2}{p_{\mathrm{a}}B}\right)\left\|x^{t+1}-x^t\right\|^2
$$

$$
+ \left(\frac{2\left(1-p_{\mathrm{a}}\right)b^2}{p_{\mathrm{a}}p_{\text{page}}} + p_{\text{page}}\left(1-\frac{b}{p_{\text{page}}}\right)^2 + (1-p_{\text{page}})\right)\left\|h_i^t-\nabla f_i(x^t)\right\|^2,
$$

where we used Assumption 4. Finally, we prove the last inequality:

$$
\mathrm{E}_B\left[\mathrm{E}_{p_{\text{page}}}\left[\left\|k_i^{t+1}\right\|^2\right]\right]
$$

$$
= p_{\text{page}}\left\|\nabla f_i(x^{t+1})-\nabla f_i(x^t)-\frac{b}{p_{\text{page}}}\left(h_i^t-\nabla f_i(x^t)\right)\right\|^2
$$

$$
+ (1-p_{\text{page}})\mathrm{E}_B\left[\left\|\frac{1}{B}\sum_{j\in I_i^t}\left(\nabla f_{ij}(x^{t+1})-\nabla f_{ij}(x^t)\right)\right\|^2\right]
$$

$$
\stackrel{(16)}{=} p_{\text{page}}\left\|\nabla f_i(x^{t+1})-\nabla f_i(x^t)-\frac{b}{p_{\text{page}}}\left(h_i^t-\nabla f_i(x^t)\right)\right\|^2
$$

$$
+ (1-p_{\text{page}})\mathrm{E}_B\left[\left\|\frac{1}{B}\sum_{j\in I_i^t}\left(\nabla f_{ij}(x^{t+1})-\nabla f_{ij}(x^t)\right)-\left(\nabla f_i(x^{t+1})-\nabla f_i(x^t)\right)\right\|^2\right]
$$

$$
+ (1-p_{\text{page}})\left\|\nabla f_i(x^{t+1})-\nabla f_i(x^t)\right\|^2
$$

$$
\stackrel{(15)}{\leq} 2p_{\text{page}}\left\|\nabla f_i(x^{t+1})-\nabla f_i(x^t)\right\|^2 + \frac{2b^2}{p_{\text{page}}}\left\|h_i^t-\nabla f_i(x^t)\right\|^2
$$

$$
+ (1-p_{\text{page}})\mathrm{E}_B\left[\left\|\frac{1}{B}\sum_{j\in I_i^t}\left(\nabla f_{ij}(x^{t+1})-\nabla f_{ij}(x^t)\right)-\left(\nabla f_i(x^{t+1})-\nabla f_i(x^t)\right)\right\|^2\right]
$$

$$
+ (1-p_{\text{page}})\left\|\nabla f_i(x^{t+1})-\nabla f_i(x^t)\right\|^2
$$

$$
\leq 2\left\|\nabla f_i(x^{t+1})-\nabla f_i(x^t)\right\|^2 + \frac{2b^2}{p_{\text{page}}}\left\|h_i^t-\nabla f_i(x^t)\right\|^2
$$

$$+ (1 - p_{\text{page}}) \mathbb{E}_B \left[ \left\| \frac{1}{B} \sum_{j \in I_i^t} \left( \nabla f_{ij}(x^{t+1}) - \nabla f_{ij}(x^t) \right) - \left( \nabla f_i(x^{t+1}) - \nabla f_i(x^t) \right) \right\|^2 \right].$$

Using the independence of elements in the mini-batch, we have

$$\mathbb{E}_B \left[ \mathbb{E}_{p_{\text{page}}} \left[ \left\| k_i^{t+1} \right\|^2 \right] \right]$$

$$\leq 2 \left\| \nabla f_i(x^{t+1}) - \nabla f_i(x^t) \right\|^2 + \frac{2b^2}{p_{\text{page}}} \left\| h_i^t - \nabla f_i(x^t) \right\|^2$$

$$+ \frac{1 - p_{\text{page}}}{B^2} \mathbb{E}_B \left[ \sum_{j \in I_i^t} \left\| \left( \nabla f_{ij}(x^{t+1}) - \nabla f_{ij}(x^t) \right) - \left( \nabla f_i(x^{t+1}) - \nabla f_i(x^t) \right) \right\|^2 \right]$$

$$= 2 \left\| \nabla f_i(x^{t+1}) - \nabla f_i(x^t) \right\|^2 + \frac{2b^2}{p_{\text{page}}} \left\| h_i^t - \nabla f_i(x^t) \right\|^2$$

$$+ \frac{1 - p_{\text{page}}}{Bm} \sum_{j=1}^m \left\| \left( \nabla f_{ij}(x^{t+1}) - \nabla f_{ij}(x^t) \right) - \left( \nabla f_i(x^{t+1}) - \nabla f_i(x^t) \right) \right\|^2$$

$$\leq 2 \left\| \nabla f_i(x^{t+1}) - \nabla f_i(x^t) \right\|^2 + \frac{2b^2}{p_{\text{page}}} \left\| h_i^t - \nabla f_i(x^t) \right\|^2$$

$$+ \frac{1 - p_{\text{page}}}{Bm} \sum_{j=1}^m \left\| \nabla f_{ij}(x^{t+1}) - \nabla f_{ij}(x^t) \right\|^2$$

.

It it left to consider Assumptions 3 and 4 to get

$$\mathbb{E}_B \left[ \mathbb{E}_{p_{\text{page}}} \left[ \left\| k_i^{t+1} \right\|^2 \right] \right]$$

$$\leq \left( 2L_i^2 + \frac{(1 - p_{\text{page}})L_{\max}^2}{B} \right) \left\| x^{t+1} - x^t \right\|^2 + \frac{2b^2}{p_{\text{page}}} \left\| h_i^t - \nabla f_i(x^t) \right\|^2.$$

$\square$

**Theorem 3.** *Suppose that Assumptions 1, 2, 3, 4, 7, and 8 hold. Let us take $a = \frac{p_a}{2\omega + 1}$, $b = \frac{p_{page} p_a}{2 - p_a}$, probability $p_{page} \in (0, 1]$,*

$$\gamma \leq \left( L + \left[ \frac{48\omega(2\omega + 1)}{np_a^2} \left( \widehat{L}^2 + \frac{(1 - p_{page})L_{\max}^2}{B} \right) + \frac{16}{np_a^2 p_{page}} \left( \left( 1 - \frac{p_{aa}}{p_a} \right) \widehat{L}^2 + \frac{(1 - p_{page})L_{\max}^2}{B} \right) \right]^{1/2} \right)^{-1}$$

*and $g_i^0 = h_i^0 = \nabla f_i(x^0)$ for all $i \in [n]$ in Algorithm 1 (DASHA-PP-PAGE) then $\mathbb{E}\left[ \left\| \nabla f(\widehat{x}^T) \right\|^2 \right] \leq \frac{2\Delta_0}{\gamma T}$.*

*Proof.* Let us fix constants $\nu, \rho \in [0, \infty)$ that we will define later. Considering Lemma 6, Lemma 8, and the law of total expectation, we obtain

$$\mathbb{E}\left[ f(x^{t+1}) \right] + \frac{\gamma(2\omega + 1)}{p_a} \mathbb{E}\left[ \left\| g^{t+1} - h^{t+1} \right\|^2 \right] + \frac{\gamma((2\omega + 1)p_a - p_{aa})}{np_a^2} \mathbb{E}\left[ \frac{1}{n} \sum_{i=1}^n \left\| g_i^{t+1} - h_i^{t+1} \right\|^2 \right]$$

$$+ \nu \mathbb{E}\left[ \left\| h^{t+1} - \nabla f(x^{t+1}) \right\|^2 \right] + \rho \mathbb{E}\left[ \frac{1}{n} \sum_{i=1}^n \left\| h_i^{t+1} - \nabla f_i(x^{t+1}) \right\|^2 \right]$$

$$\leq \mathbb{E}\left[ f(x^t) - \frac{\gamma}{2} \left\| \nabla f(x^t) \right\|^2 - \left( \frac{1}{2\gamma} - \frac{L}{2} \right) \left\| x^{t+1} - x^t \right\|^2 + \gamma \left\| h^t - \nabla f(x^t) \right\|^2 \right]$$

$$+ \frac{\gamma(2\omega + 1)}{p_a} \mathbb{E}\left[ \left\| g^t - h^t \right\|^2 \right] + \frac{\gamma((2\omega + 1)p_a - p_{aa})}{np_a^2} \mathbb{E}\left[ \frac{1}{n} \sum_{i=1}^n \left\| g_i^t - h_i^t \right\|^2 \right]$$

$$+ \frac{4\gamma\omega(2\omega+1)}{np_{\mathrm{a}}^2}\mathrm{E}\left[\frac{1}{n}\sum_{i=1}^n \left\|k_i^{t+1}\right\|^2\right]$$

$$+ \nu\mathrm{E}\left[\left\|h^{t+1}-\nabla f(x^{t+1})\right\|^2\right] + \rho\mathrm{E}\left[\frac{1}{n}\sum_{i=1}^n \left\|h_i^{t+1}-\nabla f_i(x^{t+1})\right\|^2\right]$$

$$= \mathrm{E}\left[f(x^t) - \frac{\gamma}{2}\left\|\nabla f(x^t)\right\|^2 - \left(\frac{1}{2\gamma}-\frac{L}{2}\right)\left\|x^{t+1}-x^t\right\|^2 + \gamma\left\|h^t-\nabla f(x^t)\right\|^2\right]$$

$$+ \frac{\gamma(2\omega+1)}{p_{\mathrm{a}}}\mathrm{E}\left[\left\|g^t-h^t\right\|^2\right] + \frac{\gamma((2\omega+1)\,p_{\mathrm{a}}-p_{\mathrm{aa}})}{np_{\mathrm{a}}^2}\mathrm{E}\left[\frac{1}{n}\sum_{i=1}^n\left\|g_i^t-h_i^t\right\|^2\right]$$

$$+ \frac{4\gamma\omega(2\omega+1)}{np_{\mathrm{a}}^2}\mathrm{E}\left[\mathrm{E}_B\left[\mathrm{E}_{p_{\mathrm{page}}}\left[\frac{1}{n}\sum_{i=1}^n\left\|k_i^{t+1}\right\|^2\right]\right]\right]$$

$$+ \nu\mathrm{E}\left[\mathrm{E}_B\left[\mathrm{E}_{p_{\mathrm{a}}}\left[\mathrm{E}_{p_{\mathrm{page}}}\left[\left\|h^{t+1}-\nabla f(x^{t+1})\right\|^2\right]\right]\right]\right]$$

$$+ \rho\mathrm{E}\left[\mathrm{E}_B\left[\mathrm{E}_{p_{\mathrm{a}}}\left[\mathrm{E}_{p_{\mathrm{page}}}\left[\frac{1}{n}\sum_{i=1}^n\left\|h_i^{t+1}-\nabla f_i(x^{t+1})\right\|^2\right]\right]\right]\right]$$

$$\le \mathrm{E}\left[f(x^t) - \frac{\gamma}{2}\left\|\nabla f(x^t)\right\|^2 - \left(\frac{1}{2\gamma}-\frac{L}{2}\right)\left\|x^{t+1}-x^t\right\|^2 + \gamma\left\|h^t-\nabla f(x^t)\right\|^2\right]$$

$$+ \frac{\gamma(2\omega+1)}{p_{\mathrm{a}}}\mathrm{E}\left[\left\|g^t-h^t\right\|^2\right] + \frac{\gamma((2\omega+1)\,p_{\mathrm{a}}-p_{\mathrm{aa}})}{np_{\mathrm{a}}^2}\mathrm{E}\left[\frac{1}{n}\sum_{i=1}^n\left\|g_i^t-h_i^t\right\|^2\right]$$

$$+ \frac{4\gamma\omega(2\omega+1)}{np_{\mathrm{a}}^2}\mathrm{E}\left[\left(2\widehat{L}^2 + \frac{(1-p_{\mathrm{page}})L_{\max}^2}{B}\right)\left\|x^{t+1}-x^t\right\|^2 + \frac{2b^2}{p_{\mathrm{page}}}\frac{1}{n}\sum_{i=1}^n\left\|h_i^t-\nabla f_i(x^t)\right\|^2\right]$$

$$+ \nu\mathrm{E}\Bigg(\left(\frac{2\,(p_{\mathrm{a}}-p_{\mathrm{aa}})\,\widehat{L}^2}{np_{\mathrm{a}}^2} + \frac{(1-p_{\mathrm{page}})L_{\max}^2}{np_{\mathrm{a}}B}\right)\left\|x^{t+1}-x^t\right\|^2$$

$$+ \frac{2\,(p_{\mathrm{a}}-p_{\mathrm{aa}})\,b^2}{n^2p_{\mathrm{a}}^2p_{\mathrm{page}}}\sum_{i=1}^n\left\|h_i^t-\nabla f_i(x^t)\right\|^2 + \left(p_{\mathrm{page}}\left(1-\frac{b}{p_{\mathrm{page}}}\right)^2 + (1-p_{\mathrm{page}})\right)\left\|h^t-\nabla f(x^t)\right\|^2\Bigg)$$

$$+ \rho\mathrm{E}\Bigg(\left(\frac{2\,(1-p_{\mathrm{a}})\,\widehat{L}^2}{p_{\mathrm{a}}} + \frac{(1-p_{\mathrm{page}})L_{\max}^2}{p_{\mathrm{a}}B}\right)\left\|x^{t+1}-x^t\right\|^2$$

$$+ \left(\frac{2\,(1-p_{\mathrm{a}})\,b^2}{p_{\mathrm{a}}p_{\mathrm{page}}} + p_{\mathrm{page}}\left(1-\frac{b}{p_{\mathrm{page}}}\right)^2 + (1-p_{\mathrm{page}})\right)\frac{1}{n}\sum_{i=1}^n\left\|h_i^t-\nabla f_i(x^t)\right\|^2\Bigg)$$

After rearranging the terms, we get

$$\mathrm{E}\left[f(x^{t+1})\right] + \frac{\gamma(2\omega+1)}{p_{\mathrm{a}}}\mathrm{E}\left[\left\|g^{t+1}-h^{t+1}\right\|^2\right] + \frac{\gamma((2\omega+1)\,p_{\mathrm{a}}-p_{\mathrm{aa}})}{np_{\mathrm{a}}^2}\mathrm{E}\left[\frac{1}{n}\sum_{i=1}^n\left\|g_i^{t+1}-h_i^{t+1}\right\|^2\right]$$

$$+ \nu\mathrm{E}\left[\left\|h^{t+1}-\nabla f(x^{t+1})\right\|^2\right] + \rho\mathrm{E}\left[\frac{1}{n}\sum_{i=1}^n\left\|h_i^{t+1}-\nabla f_i(x^{t+1})\right\|^2\right]$$

$$\le \mathrm{E}\left[f(x^t)\right] - \frac{\gamma}{2}\mathrm{E}\left[\left\|\nabla f(x^t)\right\|^2\right]$$

$$+ \frac{\gamma(2\omega+1)}{p_{\mathrm{a}}}\mathrm{E}\left[\left\|g^t-h^t\right\|^2\right] + \frac{\gamma((2\omega+1)\,p_{\mathrm{a}}-p_{\mathrm{aa}})}{np_{\mathrm{a}}^2}\mathrm{E}\left[\frac{1}{n}\sum_{i=1}^n\left\|g_i^t-h_i^t\right\|^2\right]$$

$$- \left(\frac{1}{2\gamma}-\frac{L}{2} - \frac{4\gamma\omega(2\omega+1)}{np_{\mathrm{a}}^2}\left(2\widehat{L}^2 + \frac{(1-p_{\mathrm{page}})L_{\max}^2}{B}\right)\right.$$

$$- \nu \left( \frac{2 \left( p_{\mathrm{a}} - p_{\mathrm{aa}} \right) \widehat{L}^2}{np_{\mathrm{a}}^2} + \frac{(1 - p_{\mathrm{page}})L_{\max}^2}{np_{\mathrm{a}}B} \right) - \rho \left( \frac{2 \left( 1 - p_{\mathrm{a}} \right) \widehat{L}^2}{p_{\mathrm{a}}} + \frac{(1 - p_{\mathrm{page}})L_{\max}^2}{p_{\mathrm{a}}B} \right) \right) \mathrm{E} \left[ \left\| x^{t+1} - x^t \right\|^2 \right]$$

$$+ \left( \gamma + \nu \left( p_{\mathrm{page}} \left( 1 - \frac{b}{p_{\mathrm{page}}} \right)^2 + (1 - p_{\mathrm{page}}) \right) \right) \mathrm{E} \left[ \left\| h^t - \nabla f(x^t) \right\|^2 \right]$$

$$+ \left( \frac{8b^2 \gamma \omega (2\omega + 1)}{np_{\mathrm{a}}^2 p_{\mathrm{page}}} + \frac{2\nu \left( p_{\mathrm{a}} - p_{\mathrm{aa}} \right) b^2}{np_{\mathrm{a}}^2 p_{\mathrm{page}}} \right.$$

$$+ \rho \left( \frac{2 \left( 1 - p_{\mathrm{a}} \right) b^2}{p_{\mathrm{a}} p_{\mathrm{page}}} + p_{\mathrm{page}} \left( 1 - \frac{b}{p_{\mathrm{page}}} \right)^2 + (1 - p_{\mathrm{page}}) \right) \right) \mathrm{E} \left[ \frac{1}{n} \sum_{i=1}^n \left\| h_i^t - \nabla f_i(x^t) \right\|^2 \right].$$

Due to $b = \frac{p_{\mathrm{page}} p_{\mathrm{a}}}{2 - p_{\mathrm{a}}} \leq p_{\mathrm{page}}$, one can show that $\left( p_{\mathrm{page}} \left( 1 - \frac{b}{p_{\mathrm{page}}} \right)^2 + (1 - p_{\mathrm{page}}) \right) \leq 1 - b$. Thus, if we take $\nu = \frac{\gamma}{b}$, then

$$\left( \gamma + \nu \left( p_{\mathrm{page}} \left( 1 - \frac{b}{p_{\mathrm{page}}} \right)^2 + (1 - p_{\mathrm{page}}) \right) \right) \leq \gamma + \nu(1 - b) = \nu,$$

therefore

$$\mathrm{E} \left[ f(x^{t+1}) \right] + \frac{\gamma(2\omega + 1)}{p_{\mathrm{a}}} \mathrm{E} \left[ \left\| g^{t+1} - h^{t+1} \right\|^2 \right] + \frac{\gamma((2\omega + 1) p_{\mathrm{a}} - p_{\mathrm{aa}})}{np_{\mathrm{a}}^2} \mathrm{E} \left[ \frac{1}{n} \sum_{i=1}^n \left\| g_i^{t+1} - h_i^{t+1} \right\|^2 \right]$$

$$+ \frac{\gamma}{b} \mathrm{E} \left[ \left\| h^{t+1} - \nabla f(x^{t+1}) \right\|^2 \right] + \rho \mathrm{E} \left[ \frac{1}{n} \sum_{i=1}^n \left\| h_i^{t+1} - \nabla f_i(x^{t+1}) \right\|^2 \right]$$

$$\leq \mathrm{E} \left[ f(x^t) \right] - \frac{\gamma}{2} \mathrm{E} \left[ \left\| \nabla f(x^t) \right\|^2 \right]$$

$$+ \frac{\gamma(2\omega + 1)}{p_{\mathrm{a}}} \mathrm{E} \left[ \left\| g^t - h^t \right\|^2 \right] + \frac{\gamma((2\omega + 1) p_{\mathrm{a}} - p_{\mathrm{aa}})}{np_{\mathrm{a}}^2} \mathrm{E} \left[ \frac{1}{n} \sum_{i=1}^n \left\| g_i^t - h_i^t \right\|^2 \right]$$

$$- \left( \frac{1}{2\gamma} - \frac{L}{2} - \frac{4\gamma\omega(2\omega + 1)}{np_{\mathrm{a}}^2} \left( 2\widehat{L}^2 + \frac{(1 - p_{\mathrm{page}})L_{\max}^2}{B} \right) \right.$$

$$- \frac{\gamma}{b} \left( \frac{2 \left( p_{\mathrm{a}} - p_{\mathrm{aa}} \right) \widehat{L}^2}{np_{\mathrm{a}}^2} + \frac{(1 - p_{\mathrm{page}})L_{\max}^2}{np_{\mathrm{a}}B} \right) - \rho \left( \frac{2 \left( 1 - p_{\mathrm{a}} \right) \widehat{L}^2}{p_{\mathrm{a}}} + \frac{(1 - p_{\mathrm{page}})L_{\max}^2}{p_{\mathrm{a}}B} \right) \right) \mathrm{E} \left[ \left\| x^{t+1} - x^t \right\|^2 \right]$$

$$+ \frac{\gamma}{b} \mathrm{E} \left[ \left\| h^t - \nabla f(x^t) \right\|^2 \right]$$

$$+ \left( \frac{8b^2 \gamma \omega (2\omega + 1)}{np_{\mathrm{a}}^2 p_{\mathrm{page}}} + \frac{2\gamma \left( p_{\mathrm{a}} - p_{\mathrm{aa}} \right) b}{np_{\mathrm{a}}^2 p_{\mathrm{page}}} \right.$$

$$+ \rho \left( \frac{2 \left( 1 - p_{\mathrm{a}} \right) b^2}{p_{\mathrm{a}} p_{\mathrm{page}}} + p_{\mathrm{page}} \left( 1 - \frac{b}{p_{\mathrm{page}}} \right)^2 + (1 - p_{\mathrm{page}}) \right) \right) \mathrm{E} \left[ \frac{1}{n} \sum_{i=1}^n \left\| h_i^t - \nabla f_i(x^t) \right\|^2 \right].$$

Next, with the choice of $b = \frac{p_{\mathrm{page}} p_{\mathrm{a}}}{2 - p_{\mathrm{a}}}$, we ensure that

$$\left( \frac{2 \left( 1 - p_{\mathrm{a}} \right) b^2}{p_{\mathrm{a}} p_{\mathrm{page}}} + p_{\mathrm{page}} \left( 1 - \frac{b}{p_{\mathrm{page}}} \right)^2 + (1 - p_{\mathrm{page}}) \right) \leq 1 - b.$$

If we take $\rho = \frac{8b\gamma\omega(2\omega + 1)}{np_{\mathrm{a}}^2 p_{\mathrm{page}}} + \frac{2\gamma(p_{\mathrm{a}} - p_{\mathrm{aa}})}{np_{\mathrm{a}}^2 p_{\mathrm{page}}}$, then

$$\left( \frac{8b^2 \gamma \omega (2\omega + 1)}{np_{\mathrm{a}}^2 p_{\mathrm{page}}} + \frac{2\gamma \left( p_{\mathrm{a}} - p_{\mathrm{aa}} \right) b}{np_{\mathrm{a}}^2 p_{\mathrm{page}}} + \rho \left( \frac{2 \left( 1 - p_{\mathrm{a}} \right) b^2}{p_{\mathrm{a}} p_{\mathrm{page}}} + p_{\mathrm{page}} \left( 1 - \frac{b}{p_{\mathrm{page}}} \right)^2 + (1 - p_{\mathrm{page}}) \right) \right) \leq \rho,$$

therefore

$$\mathrm{E} \left[ f(x^{t+1}) \right] + \frac{\gamma(2\omega + 1)}{p_{\mathrm{a}}} \mathrm{E} \left[ \left\| g^{t+1} - h^{t+1} \right\|^2 \right] + \frac{\gamma((2\omega + 1) p_{\mathrm{a}} - p_{\mathrm{aa}})}{np_{\mathrm{a}}^2} \mathrm{E} \left[ \frac{1}{n} \sum_{i=1}^n \left\| g_i^{t+1} - h_i^{t+1} \right\|^2 \right]$$

$$+ \frac{\gamma}{b} \mathrm{E}\left[\left\|h^{t+1} - \nabla f(x^{t+1})\right\|^2\right] + \left(\frac{8b\gamma\omega(2\omega+1)}{np_a^2 p_{\text{page}}} + \frac{2\gamma\left(p_a - p_{aa}\right)}{np_a^2 p_{\text{page}}}\right) \mathrm{E}\left[\frac{1}{n}\sum_{i=1}^n \left\|h_i^{t+1} - \nabla f_i(x^{t+1})\right\|^2\right]$$

$$\leq \mathrm{E}\left[f(x^t)\right] - \frac{\gamma}{2}\mathrm{E}\left[\left\|\nabla f(x^t)\right\|^2\right]$$

$$+ \frac{\gamma(2\omega+1)}{p_a}\mathrm{E}\left[\left\|g^t - h^t\right\|^2\right] + \frac{\gamma((2\omega+1)p_a - p_{aa})}{np_a^2}\mathrm{E}\left[\frac{1}{n}\sum_{i=1}^n \left\|g_i^t - h_i^t\right\|^2\right]$$

$$- \left(\frac{1}{2\gamma} - \frac{L}{2} - \frac{4\gamma\omega(2\omega+1)}{np_a^2}\left(2\widehat{L}^2 + \frac{(1-p_{\text{page}})L_{\max}^2}{B}\right)\right.$$

$$- \frac{\gamma}{bnp_a}\left(2\left(1 - \frac{p_{aa}}{p_a}\right)\widehat{L}^2 + \frac{(1-p_{\text{page}})L_{\max}^2}{B}\right)$$

$$\left.- \left(\frac{8b\gamma\omega(2\omega+1)}{np_a^3 p_{\text{page}}} + \frac{2\gamma\left(1 - \frac{p_{aa}}{p_a}\right)}{np_a^2 p_{\text{page}}}\right)\left(2(1-p_a)\widehat{L}^2 + \frac{(1-p_{\text{page}})L_{\max}^2}{B}\right)\right)\mathrm{E}\left[\left\|x^{t+1} - x^t\right\|^2\right]$$

$$+ \frac{\gamma}{b}\mathrm{E}\left[\left\|h^t - \nabla f(x^t)\right\|^2\right] + \left(\frac{8b\gamma\omega(2\omega+1)}{np_a^2 p_{\text{page}}} + \frac{2\gamma\left(p_a - p_{aa}\right)}{np_a^2 p_{\text{page}}}\right)\mathrm{E}\left[\frac{1}{n}\sum_{i=1}^n \left\|h_i^t - \nabla f_i(x^t)\right\|^2\right].$$

Let us simplify the inequality. First, due to $b \geq \frac{p_{\text{page}} p_a}{2}$, we have

$$\frac{\gamma}{bnp_a}\left(2\left(1 - \frac{p_{aa}}{p_a}\right)\widehat{L}^2 + \frac{(1-p_{\text{page}})L_{\max}^2}{B}\right) \leq \frac{4\gamma}{np_a^2 p_{\text{page}}}\left(\left(1 - \frac{p_{aa}}{p_a}\right)\widehat{L}^2 + \frac{(1-p_{\text{page}})L_{\max}^2}{B}\right).$$

Second, due to $b \leq p_a p_{\text{page}}$ and $p_{aa} \leq p_a^2$, we get

$$\left(\frac{8b\gamma\omega(2\omega+1)}{np_a^3 p_{\text{page}}} + \frac{2\gamma\left(1 - \frac{p_{aa}}{p_a}\right)}{np_a^2 p_{\text{page}}}\right)\left(2(1-p_a)\widehat{L}^2 + \frac{(1-p_{\text{page}})L_{\max}^2}{B}\right)$$

$$\leq \left(\frac{8\gamma\omega(2\omega+1)}{np_a^2} + \frac{2\gamma\left(1 - \frac{p_{aa}}{p_a}\right)}{np_a^2 p_{\text{page}}}\right)\left(2\left(1 - \frac{p_{aa}}{p_a}\right)\widehat{L}^2 + \frac{(1-p_{\text{page}})L_{\max}^2}{B}\right)$$

$$\leq \frac{16\gamma\omega(2\omega+1)}{np_a^2}\left(\left(1 - \frac{p_{aa}}{p_a}\right)\widehat{L}^2 + \frac{(1-p_{\text{page}})L_{\max}^2}{B}\right)$$

$$+ \frac{4\gamma\left(1 - \frac{p_{aa}}{p_a}\right)}{np_a^2 p_{\text{page}}}\left(\left(1 - \frac{p_{aa}}{p_a}\right)\widehat{L}^2 + \frac{(1-p_{\text{page}})L_{\max}^2}{B}\right)$$

$$\leq \frac{16\gamma\omega(2\omega+1)}{np_a^2}\left(\widehat{L}^2 + \frac{(1-p_{\text{page}})L_{\max}^2}{B}\right)$$

$$+ \frac{4\gamma}{np_a^2 p_{\text{page}}}\left(\left(1 - \frac{p_{aa}}{p_a}\right)\widehat{L}^2 + \frac{(1-p_{\text{page}})L_{\max}^2}{B}\right).$$

Combining all bounds together, we obtain the following simplified inequality:

$$\mathrm{E}\left[f(x^{t+1})\right] + \frac{\gamma(2\omega+1)}{p_a}\mathrm{E}\left[\left\|g^{t+1} - h^{t+1}\right\|^2\right] + \frac{\gamma((2\omega+1)p_a - p_{aa})}{np_a^2}\mathrm{E}\left[\frac{1}{n}\sum_{i=1}^n \left\|g_i^{t+1} - h_i^{t+1}\right\|^2\right]$$

$$+ \frac{\gamma}{b}\mathrm{E}\left[\left\|h^{t+1} - \nabla f(x^{t+1})\right\|^2\right] + \left(\frac{8b\gamma\omega(2\omega+1)}{np_a^2 p_{\text{page}}} + \frac{2\gamma\left(p_a - p_{aa}\right)}{np_a^2 p_{\text{page}}}\right)\mathrm{E}\left[\frac{1}{n}\sum_{i=1}^n \left\|h_i^{t+1} - \nabla f_i(x^{t+1})\right\|^2\right]$$

$$\leq \mathrm{E}\left[f(x^t)\right] - \frac{\gamma}{2}\mathrm{E}\left[\left\|\nabla f(x^t)\right\|^2\right]$$

$$+ \frac{\gamma(2\omega+1)}{p_a}\mathrm{E}\left[\left\|g^t - h^t\right\|^2\right] + \frac{\gamma((2\omega+1)p_a - p_{aa})}{np_a^2}\mathrm{E}\left[\frac{1}{n}\sum_{i=1}^n \left\|g_i^t - h_i^t\right\|^2\right]$$

$$-\left(\frac{1}{2\gamma} - \frac{L}{2} - \frac{24\gamma\omega(2\omega+1)}{np_{\mathrm{a}}^2}\left(\widehat{L}^2 + \frac{(1-p_{\mathrm{page}})L_{\max}^2}{B}\right)\right)$$

$$-\frac{8\gamma}{np_{\mathrm{a}}^2 p_{\mathrm{page}}}\left(\left(1-\frac{p_{\mathrm{aa}}}{p_{\mathrm{a}}}\right)\widehat{L}^2 + \frac{(1-p_{\mathrm{page}})L_{\max}^2}{B}\right)\right)\mathrm{E}\left[\left\|x^{t+1}-x^t\right\|^2\right]$$

$$+\frac{\gamma}{b}\mathrm{E}\left[\left\|h^t - \nabla f(x^t)\right\|^2\right] + \left(\frac{8b\gamma\omega(2\omega+1)}{np_{\mathrm{a}}^2 p_{\mathrm{page}}} + \frac{2\gamma(p_{\mathrm{a}}-p_{\mathrm{aa}})}{np_{\mathrm{a}}^2 p_{\mathrm{page}}}\right)\mathrm{E}\left[\frac{1}{n}\sum_{i=1}^n\left\|h_i^t - \nabla f_i(x^t)\right\|^2\right].$$

Using Lemma 4 and the assumption about $\gamma$, we get

$$\mathrm{E}\left[f(x^{t+1})\right] + \frac{\gamma(2\omega+1)}{p_{\mathrm{a}}}\mathrm{E}\left[\left\|g^{t+1}-h^{t+1}\right\|^2\right] + \frac{\gamma((2\omega+1)p_{\mathrm{a}}-p_{\mathrm{aa}})}{np_{\mathrm{a}}^2}\mathrm{E}\left[\frac{1}{n}\sum_{i=1}^n\left\|g_i^{t+1}-h_i^{t+1}\right\|^2\right]$$

$$+\frac{\gamma}{b}\mathrm{E}\left[\left\|h^{t+1}-\nabla f(x^{t+1})\right\|^2\right] + \left(\frac{8b\gamma\omega(2\omega+1)}{np_{\mathrm{a}}^2 p_{\mathrm{page}}} + \frac{2\gamma(p_{\mathrm{a}}-p_{\mathrm{aa}})}{np_{\mathrm{a}}^2 p_{\mathrm{page}}}\right)\mathrm{E}\left[\frac{1}{n}\sum_{i=1}^n\left\|h_i^{t+1}-\nabla f_i(x^{t+1})\right\|^2\right]$$

$$\leq \mathrm{E}\left[f(x^t)\right] - \frac{\gamma}{2}\mathrm{E}\left[\left\|\nabla f(x^t)\right\|^2\right]$$

$$+\frac{\gamma(2\omega+1)}{p_{\mathrm{a}}}\mathrm{E}\left[\left\|g^t-h^t\right\|^2\right] + \frac{\gamma((2\omega+1)p_{\mathrm{a}}-p_{\mathrm{aa}})}{np_{\mathrm{a}}^2}\mathrm{E}\left[\frac{1}{n}\sum_{i=1}^n\left\|g_i^t-h_i^t\right\|^2\right]$$

$$+\frac{\gamma}{b}\mathrm{E}\left[\left\|h^t-\nabla f(x^t)\right\|^2\right] + \left(\frac{8b\gamma\omega(2\omega+1)}{np_{\mathrm{a}}^2 p_{\mathrm{page}}} + \frac{2\gamma(p_{\mathrm{a}}-p_{\mathrm{aa}})}{np_{\mathrm{a}}^2 p_{\mathrm{page}}}\right)\mathrm{E}\left[\frac{1}{n}\sum_{i=1}^n\left\|h_i^t-\nabla f_i(x^t)\right\|^2\right].$$

It is left to apply Lemma 3 with

$$\Psi^t \;=\; \frac{(2\omega+1)}{p_{\mathrm{a}}}\mathrm{E}\left[\left\|g^t-h^t\right\|^2\right] + \frac{((2\omega+1)p_{\mathrm{a}}-p_{\mathrm{aa}})}{np_{\mathrm{a}}^2}\mathrm{E}\left[\frac{1}{n}\sum_{i=1}^n\left\|g_i^t-h_i^t\right\|^2\right]$$

$$+\frac{1}{b}\mathrm{E}\left[\left\|h^t-\nabla f(x^t)\right\|^2\right] + \left(\frac{8b\omega(2\omega+1)}{np_{\mathrm{a}}^2 p_{\mathrm{page}}} + \frac{2(p_{\mathrm{a}}-p_{\mathrm{aa}})}{np_{\mathrm{a}}^2 p_{\mathrm{page}}}\right)\mathrm{E}\left[\frac{1}{n}\sum_{i=1}^n\left\|h_i^t-\nabla f_i(x^t)\right\|^2\right]$$

to conclude the proof. $\qquad\square$

**Corollary 1.** *Let the assumptions from Theorem 3 hold and $p_{page} = {}^{B}/(m+B)$. Then* DASHA-PP-PAGE *needs*

$$T := \mathcal{O}\left(\frac{\Delta_0}{\varepsilon}\left[L + \frac{\omega}{p_{\mathrm{a}}\sqrt{n}}\left(\widehat{L} + \frac{L_{\max}}{\sqrt{B}}\right) + \frac{1}{p_{\mathrm{a}}}\sqrt{\frac{m}{n}}\left(\frac{\mathbb{1}_{p_{\mathrm{a}}}\widehat{L}}{\sqrt{B}} + \frac{L_{\max}}{B}\right)\right]\right) \qquad (10)$$

*communication rounds to get an $\varepsilon$-solution and the expected number of gradient calculations per node equals $\mathcal{O}\left(m + BT\right)$.*

*Proof.* In the view of Theorem 3, it is enough to do

$$T := \mathcal{O}\left(\frac{\Delta_0}{\varepsilon}\left[L + \sqrt{\frac{\omega^2}{np_{\mathrm{a}}^2}\left(\widehat{L}^2 + \frac{(1-p_{\mathrm{page}})L_{\max}^2}{B}\right) + \frac{1}{np_{\mathrm{a}}^2 p_{\mathrm{page}}}\left(\left(1-\frac{p_{\mathrm{aa}}}{p_{\mathrm{a}}}\right)\widehat{L}^2 + \frac{(1-p_{\mathrm{page}})L_{\max}^2}{B}\right)}\right]\right)$$

steps to get $\varepsilon$-solution. Using the choice of $p_{\mathrm{mega}}$ and the definition of $\mathbb{1}_{p_{\mathrm{a}}}$, we can get (10).

Note that the expected number of gradients calculations at each communication round equals $p_{\mathrm{mega}}m + (1-p_{\mathrm{mega}})B = \frac{2mB}{m+B} \leq 2B$. $\qquad\square$

**Corollary 2.** *Suppose that assumptions of Corollary 1 hold, $B \leq \min\left\{\frac{1}{p_{\mathrm{a}}}\sqrt{\frac{m}{n}}, \frac{L_{\max}^2}{\mathbb{1}_{p_{\mathrm{a}}}^2\widehat{L}^2}\right\}$[8], and we use the unbiased compressor RandK with $K = \Theta\left({}^{Bd}/\sqrt{m}\right)$. Then the communication complexity of*

---

[8] If $\mathbb{1}_{p_{\mathrm{a}}} = 0$, then $\frac{L_{\max}^2}{\mathbb{1}_{p_{\mathrm{a}}}^2\widehat{L}^2} = +\infty$

*Algorithm 1 is $\mathcal{O}\left(d + \frac{L_{\max}\Delta_0 d}{p_a \varepsilon \sqrt{n}}\right)$, and the expected number of gradient calculations per node equals $\mathcal{O}\left(m + \frac{L_{\max}\Delta_0\sqrt{m}}{p_a \varepsilon \sqrt{n}}\right)$.*

*Proof.* The communication complexity equals

$$\mathcal{O}\left(d + KT\right) = \mathcal{O}\left(d + \frac{\Delta_0}{\varepsilon}\left[KL + K\frac{\omega}{p_a\sqrt{n}}\left(\widehat{L} + \frac{L_{\max}}{\sqrt{B}}\right) + K\frac{1}{p_a}\sqrt{\frac{m}{n}}\left(\frac{\mathbb{1}_{p_a}\widehat{L}}{\sqrt{B}} + \frac{L_{\max}}{B}\right)\right]\right).$$

Since $B \le \frac{L_{\max}^2}{\mathbb{1}_{p_a}^2 \widehat{L}^2}$, we have $\frac{\mathbb{1}_{p_a}\widehat{L}}{\sqrt{B}} + \frac{L_{\max}}{B} \le \frac{2L_{\max}}{B}$ and

$$\mathcal{O}\left(d + KT\right) = \mathcal{O}\left(d + \frac{\Delta_0}{\varepsilon}\left[KL + K\frac{\omega}{p_a\sqrt{n}}\left(\widehat{L} + \frac{L_{\max}}{\sqrt{B}}\right) + K\frac{1}{p_a}\sqrt{\frac{m}{n}}\frac{L_{\max}}{B}\right]\right).$$

Note that $K = \Theta\left(\frac{Bd}{\sqrt{m}}\right) = \mathcal{O}\left(\frac{d}{p_a\sqrt{n}}\right)$ and $\omega + 1 = \frac{d}{K}$ due to Theorem 6, thus

$$\mathcal{O}\left(d + KT\right) = \mathcal{O}\left(d + \frac{\Delta_0}{\varepsilon}\left[\frac{d}{p_a\sqrt{n}}L + \frac{d}{p_a\sqrt{n}}\left(\widehat{L} + \frac{L_{\max}}{\sqrt{B}}\right) + \frac{d}{p_a\sqrt{n}}L_{\max}\right]\right)$$

$$= \mathcal{O}\left(d + \frac{L_{\max}\Delta_0 d}{p_a\varepsilon\sqrt{n}}\right).$$

Using the same reasoning, the expected number of gradient calculations per node equals

$$\mathcal{O}\left(m + BT\right) = \mathcal{O}\left(m + \frac{\Delta_0}{\varepsilon}\left[BL + B\frac{\omega}{p_a\sqrt{n}}\left(\widehat{L} + \frac{L_{\max}}{\sqrt{B}}\right) + B\frac{1}{p_a}\sqrt{\frac{m}{n}}\left(\frac{\mathbb{1}_{p_a}\widehat{L}}{\sqrt{B}} + \frac{L_{\max}}{B}\right)\right]\right)$$

$$= \mathcal{O}\left(m + \frac{\Delta_0}{\varepsilon}\left[BL + B\frac{d}{Kp_a\sqrt{n}}\left(\widehat{L} + \frac{L_{\max}}{\sqrt{B}}\right) + B\frac{1}{p_a}\sqrt{\frac{m}{n}}\frac{L_{\max}}{B}\right]\right)$$

$$= \mathcal{O}\left(m + \frac{\Delta_0}{\varepsilon}\left[\frac{1}{p_a}\sqrt{\frac{m}{n}}L + \frac{\sqrt{m}}{p_a\sqrt{n}}\left(\widehat{L} + \frac{L_{\max}}{\sqrt{B}}\right) + \frac{1}{p_a}\sqrt{\frac{m}{n}}L_{\max}\right]\right)$$

$$= \mathcal{O}\left(m + \frac{L_{\max}\Delta_0\sqrt{m}}{p_a\varepsilon\sqrt{n}}\right).$$

$\square$

### E.5 Proof for DASHA-PP-FINITE-MVR

**Lemma 9.** *Suppose that Assumptions 3, 4, and 8 hold. For $h_i^{t+1}$, $h_{ij}^{t+1}$ and $k_i^{t+1}$ from Algorithm 1 (DASHA-PP-FINITE-MVR) we have*

*1.*

$$\mathrm{E}_B\left[\mathrm{E}_{p_a}\left[\left\|h^{t+1} - \nabla f(x^{t+1})\right\|^2\right]\right]$$

$$\le \left(\frac{2L_{\max}^2}{np_aB} + \frac{2(p_a - p_{aa})\widehat{L}^2}{np_a^2}\right)\left\|x^{t+1} - x^t\right\|^2$$

$$+ \frac{2(p_a - p_{aa})b^2}{n^2p_a^2}\sum_{i=1}^n\left\|h_i^t - \nabla f_i(x^t)\right\|^2 + \frac{2b^2}{n^2p_aBm}\sum_{i=1}^n\sum_{j=1}^m\left\|h_{ij}^t - \nabla f_{ij}(x^t)\right\|^2$$

$$+ (1-b)^2\left\|h^t - \nabla f(x^t)\right\|^2.$$

*2.*

$$\mathrm{E}_B\left[\mathrm{E}_{p_a}\left[\left\|h_i^{t+1} - \nabla f_i(x^{t+1})\right\|^2\right]\right]$$

$$\le \left(\frac{2L_{\max}^2}{p_aB} + \frac{2(1-p_a)L_i^2}{p_a}\right)\left\|x^{t+1} - x^t\right\|^2$$

$$+ \frac{2b^2}{p_aBm}\sum_{j=1}^m\left\|h_{ij}^t - \nabla f_{ij}(x^t)\right\|^2 + \left(\frac{2(1-p_a)b^2}{p_a} + (1-b)^2\right)\left\|h_i^t - \nabla f_i(x^t)\right\|^2, \quad \forall i \in [n].$$

*3.*

$$\mathrm{E}_B\left[\mathrm{E}_{p_\mathrm{a}}\left[\left\|h_{ij}^{t+1} - \nabla f_{ij}(x^{t+1})\right\|^2\right]\right]$$

$$\leq \frac{2\left(1 - \frac{p_\mathrm{a}B}{m}\right)L_{\max}^2}{\frac{p_\mathrm{a}B}{m}}\left\|x^{t+1} - x^t\right\|^2$$

$$+ \left(\frac{2\left(1 - \frac{p_\mathrm{a}B}{m}\right)b^2}{\frac{p_\mathrm{a}B}{m}} + (1-b)^2\right)\left\|h_{ij}^t - \nabla f_{ij}(x^t)\right\|^2, \quad \forall i \in [n], \forall j \in [m].$$

*4.*

$$\mathrm{E}_B\left[\left\|k_i^{t+1}\right\|^2\right]$$

$$\leq \left(\frac{2L_{\max}^2}{B} + 2L_i^2\right)\left\|x^{t+1} - x^t\right\|^2$$

$$+ \frac{2b^2}{Bm}\sum_{j=1}^m\left\|h_{ij}^t - \nabla f_{ij}(x^t)\right\|^2 + 2b^2\left\|h_i^t - \nabla f_i(x^t)\right\|^2, \quad \forall i \in [n].$$

*Proof.* We start by proving the first inequality. Note that

$$\mathrm{E}_B\left[\mathrm{E}_{p_\mathrm{a}}\left[h_i^{t+1}\right]\right]$$

$$= p_\mathrm{a}\left(h_i^t + \frac{1}{p_\mathrm{a}}\mathrm{E}_B\left[k_i^{t+1}\right]\right) + (1 - p_\mathrm{a})h_i^t$$

$$= h_i^t + \frac{1}{m}\sum_{j=1}^m\frac{B}{m}\cdot\frac{m}{B}\left(\nabla f_{ij}(x^{t+1}) - \nabla f_{ij}(x^t) - b\left(h_{ij}^t - \nabla f_{ij}(x^t)\right)\right) + \left(1 - \frac{B}{m}\right)\cdot 0$$

$$= \nabla f_i(x^{t+1}) + (1 - b)\left(h_i^t - \nabla f_i(x^t)\right),$$

thus

$$\mathrm{E}_B\left[\mathrm{E}_{p_\mathrm{a}}\left[\left\|h^{t+1} - \nabla f(x^{t+1})\right\|^2\right]\right]$$

$$\stackrel{(16)}{=} \mathrm{E}_B\left[\mathrm{E}_{p_\mathrm{a}}\left[\left\|h^{t+1} - \mathrm{E}_B\left[\mathrm{E}_{p_\mathrm{a}}\left[h^{t+1}\right]\right]\right\|^2\right]\right] + (1-b)^2\left\|h^t - \nabla f(x^t)\right\|^2.$$

We can use Lemma 1 with $r_i = h_i^t$ and $s_i = k_i^{t+1}$ to obtain

$$\mathrm{E}_B\left[\mathrm{E}_{p_\mathrm{a}}\left[\left\|h^{t+1} - \nabla f(x^{t+1})\right\|^2\right]\right]$$

$$\leq \frac{1}{n^2 p_\mathrm{a}}\sum_{i=1}^n\mathrm{E}_B\left[\left\|k_i^{t+1} - \mathrm{E}_B\left[k_i^{t+1}\right]\right\|^2\right] + \frac{p_\mathrm{a} - p_\mathrm{aa}}{n^2 p_\mathrm{a}^2}\sum_{i=1}^n\left\|\mathrm{E}_B\left[k_i^{t+1}\right]\right\|^2$$

$$+ (1-b)^2\left\|h^t - \nabla f(x^t)\right\|^2$$

$$= \frac{1}{n^2 p_\mathrm{a}}\sum_{i=1}^n\mathrm{E}_B\left[\left\|\frac{1}{m}\sum_{j=1}^m k_{ij}^{t+1} - \left(\nabla f_i(x^{t+1}) - \nabla f_i(x^t) - b\left(h_i^t - \nabla f_i(x^t)\right)\right)\right\|^2\right]$$

$$+ \frac{p_\mathrm{a} - p_\mathrm{aa}}{n^2 p_\mathrm{a}^2}\sum_{i=1}^n\left\|\nabla f_i(x^{t+1}) - \nabla f_i(x^t) - b\left(h_i^t - \nabla f_i(x^t)\right)\right\|^2$$

$$+ (1-b)^2\left\|h^t - \nabla f(x^t)\right\|^2.$$

Next, we again use Lemma 1 with $r_i = 0$, $s_i = \nabla f_{ij}(x^{t+1}) - \nabla f_{ij}(x^t) - b\left(h_{ij}^t - \nabla f_{ij}(x^t)\right)$, $p_\mathrm{a} = \frac{B}{m}$, and $p_\mathrm{aa} = \frac{B(B-1)}{m(m-1)}$:

$$\mathrm{E}_B\left[\mathrm{E}_{p_\mathrm{a}}\left[\left\|h^{t+1} - \nabla f(x^{t+1})\right\|^2\right]\right]$$

$$\leq \frac{1}{n^2 p_{\mathrm{a}}} \sum_{i=1}^{n} \left( \frac{m-B}{Bm(m-1)} \sum_{j=1}^{m} \left\| \nabla f_{ij}(x^{t+1}) - \nabla f_{ij}(x^t) - b\left( h_{ij}^t - \nabla f_{ij}(x^t) \right) \right\|^2 \right)$$

$$+ \frac{p_{\mathrm{a}} - p_{\mathrm{aa}}}{n^2 p_{\mathrm{a}}^2} \sum_{i=1}^{n} \left\| \nabla f_i(x^{t+1}) - \nabla f_i(x^t) - b\left( h_i^t - \nabla f_i(x^t) \right) \right\|^2$$

$$+ (1-b)^2 \left\| h^t - \nabla f(x^t) \right\|^2$$

$$\leq \frac{1}{n^2 p_{\mathrm{a}} Bm} \sum_{i=1}^{n} \sum_{j=1}^{m} \left\| \nabla f_{ij}(x^{t+1}) - \nabla f_{ij}(x^t) - b\left( h_{ij}^t - \nabla f_{ij}(x^t) \right) \right\|^2$$

$$+ \frac{p_{\mathrm{a}} - p_{\mathrm{aa}}}{n^2 p_{\mathrm{a}}^2} \sum_{i=1}^{n} \left\| \nabla f_i(x^{t+1}) - \nabla f_i(x^t) - b\left( h_i^t - \nabla f_i(x^t) \right) \right\|^2$$

$$+ (1-b)^2 \left\| h^t - \nabla f(x^t) \right\|^2$$

$$\overset{(15)}{\leq} \frac{2}{n^2 p_{\mathrm{a}} Bm} \sum_{i=1}^{n} \sum_{j=1}^{m} \left\| \nabla f_{ij}(x^{t+1}) - \nabla f_{ij}(x^t) \right\|^2 + \frac{2b^2}{n^2 p_{\mathrm{a}} Bm} \sum_{i=1}^{n} \sum_{j=1}^{m} \left\| h_{ij}^t - \nabla f_{ij}(x^t) \right\|^2$$

$$+ \frac{2(p_{\mathrm{a}} - p_{\mathrm{aa}})}{n^2 p_{\mathrm{a}}^2} \sum_{i=1}^{n} \left\| \nabla f_i(x^{t+1}) - \nabla f_i(x^t) \right\|^2 + \frac{2(p_{\mathrm{a}} - p_{\mathrm{aa}})b^2}{n^2 p_{\mathrm{a}}^2} \sum_{i=1}^{n} \left\| h_i^t - \nabla f_i(x^t) \right\|^2$$

$$+ (1-b)^2 \left\| h^t - \nabla f(x^t) \right\|^2.$$

Due to Assumptions 3 and 4, we have

$$\mathrm{E}_B \left[ \mathrm{E}_{p_{\mathrm{a}}} \left[ \left\| h^{t+1} - \nabla f(x^{t+1}) \right\|^2 \right] \right]$$

$$\leq \left( \frac{2L_{\max}^2}{n p_{\mathrm{a}} B} + \frac{2(p_{\mathrm{a}} - p_{\mathrm{aa}}) \widehat{L}^2}{n p_{\mathrm{a}}^2} \right) \left\| x^{t+1} - x^t \right\|^2$$

$$+ \frac{2(p_{\mathrm{a}} - p_{\mathrm{aa}})b^2}{n^2 p_{\mathrm{a}}^2} \sum_{i=1}^{n} \left\| h_i^t - \nabla f_i(x^t) \right\|^2 + \frac{2b^2}{n^2 p_{\mathrm{a}} Bm} \sum_{i=1}^{n} \sum_{j=1}^{m} \left\| h_{ij}^t - \nabla f_{ij}(x^t) \right\|^2$$

$$+ (1-b)^2 \left\| h^t - \nabla f(x^t) \right\|^2.$$

Let us get the bound for the second inequality:

$$\mathrm{E}_B \left[ \mathrm{E}_{p_{\mathrm{a}}} \left[ \left\| h_i^{t+1} - \nabla f_i(x^{t+1}) \right\|^2 \right] \right]$$

$$\overset{(16)}{=} \mathrm{E}_B \left[ \mathrm{E}_{p_{\mathrm{a}}} \left[ \left\| h_i^{t+1} - \left( \nabla f_i(x^{t+1}) + (1-b)(h_i^t - \nabla f_i(x^t)) \right) \right\|^2 \right] \right]$$

$$+ (1-b)^2 \left\| h_i^t - \nabla f_i(x^t) \right\|^2$$

$$= p_{\mathrm{a}} \mathrm{E}_B \left[ \left\| h_i^t + \frac{1}{p_{\mathrm{a}}} k_i^{t+1} - \left( \nabla f_i(x^{t+1}) + (1-b)(h_i^t - \nabla f_i(x^t)) \right) \right\|^2 \right]$$

$$+ (1-p_{\mathrm{a}}) \left\| h_i^t - \left( \nabla f_i(x^{t+1}) + (1-b)(h_i^t - \nabla f_i(x^t)) \right) \right\|^2$$

$$+ (1-b)^2 \left\| h_i^t - \nabla f_i(x^t) \right\|^2$$

$$\overset{(16)}{=} \frac{1}{p_{\mathrm{a}}} \mathrm{E}_B \left[ \left\| k_i^{t+1} - \mathrm{E}_B \left[ k_i^{t+1} \right] \right\|^2 \right]$$

$$+ \frac{1 - p_{\mathrm{a}}}{p_{\mathrm{a}}} \left\| \nabla f_i(x^{t+1}) - \nabla f_i(x^t) - b(h_i^t - \nabla f_i(x^t)) \right\|^2$$

$$+ (1-b)^2 \left\| h_i^t - \nabla f_i(x^t) \right\|^2.$$

Let us use Lemma 1 with $r_i = 0$, $s_i = \nabla f_{ij}(x^{t+1}) - \nabla f_{ij}(x^t) - b\left( h_{ij}^t - \nabla f_{ij}(x^t) \right)$, $p_{\mathrm{a}} = \frac{B}{m}$, and $p_{\mathrm{aa}} = \frac{B(B-1)}{m(m-1)}$:

$$\mathrm{E}_B \left[ \mathrm{E}_{p_{\mathrm{a}}} \left[ \left\| h_i^{t+1} - \nabla f_i(x^{t+1}) \right\|^2 \right] \right]$$

$$\leq \frac{1}{p_{\mathrm{a}}} \left( \frac{m-B}{Bm(m-1)} \sum_{j=1}^{m} \left\| \nabla f_{ij}(x^{t+1}) - \nabla f_{ij}(x^t) - b\left(h_{ij}^t - \nabla f_{ij}(x^t)\right) \right\|^2 \right)$$

$$+ \frac{1-p_{\mathrm{a}}}{p_{\mathrm{a}}} \left\| \nabla f_i(x^{t+1}) - \nabla f_i(x^t) - b(h_i^t - \nabla f_i(x^t)) \right\|^2$$

$$+ (1-b)^2 \left\| h_i^t - \nabla f_i(x^t) \right\|^2$$

$$\leq \frac{1}{p_{\mathrm{a}}Bm} \sum_{j=1}^{m} \left\| \nabla f_{ij}(x^{t+1}) - \nabla f_{ij}(x^t) - b\left(h_{ij}^t - \nabla f_{ij}(x^t)\right) \right\|^2$$

$$+ \frac{1-p_{\mathrm{a}}}{p_{\mathrm{a}}} \left\| \nabla f_i(x^{t+1}) - \nabla f_i(x^t) - b(h_i^t - \nabla f_i(x^t)) \right\|^2$$

$$+ (1-b)^2 \left\| h_i^t - \nabla f_i(x^t) \right\|^2$$

$$\overset{(15)}{\leq} \frac{2}{p_{\mathrm{a}}Bm} \sum_{j=1}^{m} \left\| \nabla f_{ij}(x^{t+1}) - \nabla f_{ij}(x^t) \right\|^2 + \frac{2(1-p_{\mathrm{a}})}{p_{\mathrm{a}}} \left\| \nabla f_i(x^{t+1}) - \nabla f_i(x^t) \right\|^2$$

$$+ \frac{2b^2}{p_{\mathrm{a}}Bm} \sum_{j=1}^{m} \left\| h_{ij}^t - \nabla f_{ij}(x^t) \right\|^2 + \left( \frac{2(1-p_{\mathrm{a}})b^2}{p_{\mathrm{a}}} + (1-b)^2 \right) \left\| h_i^t - \nabla f_i(x^t) \right\|^2$$

$$\leq \left( \frac{2L_{\max}^2}{p_{\mathrm{a}}B} + \frac{2(1-p_{\mathrm{a}})L_i^2}{p_{\mathrm{a}}} \right) \left\| x^{t+1} - x^t \right\|^2$$

$$+ \frac{2b^2}{p_{\mathrm{a}}Bm} \sum_{j=1}^{m} \left\| h_{ij}^t - \nabla f_{ij}(x^t) \right\|^2 + \left( \frac{2(1-p_{\mathrm{a}})b^2}{p_{\mathrm{a}}} + (1-b)^2 \right) \left\| h_i^t - \nabla f_i(x^t) \right\|^2,$$

where we used Assumptions 3 and 4. We continue the proof by considering $\mathrm{E}_B \left[ \mathrm{E}_{p_{\mathrm{a}}} \left[ \left\| h_{ij}^{t+1} - \nabla f_{ij}(x^{t+1}) \right\|^2 \right] \right]$:

$$\mathrm{E}_B \left[ \mathrm{E}_{p_{\mathrm{a}}} \left[ \left\| h_{ij}^{t+1} - \nabla f_{ij}(x^{t+1}) \right\|^2 \right] \right]$$

$$\overset{(16)}{=} \mathrm{E}_B \left[ \mathrm{E}_{p_{\mathrm{a}}} \left[ \left\| h_{ij}^{t+1} - \left( \nabla f_{ij}(x^{t+1}) + (1-b)(h_{ij}^t - \nabla f_{ij}(x^t)) \right) \right\|^2 \right] \right]$$

$$+ (1-b)^2 \left\| h_{ij}^t - \nabla f_{ij}(x^t) \right\|^2$$

$$= \frac{p_{\mathrm{a}}B}{m} \mathrm{E}_B \left[ \left\| h_{ij}^t + \frac{m}{Bp_{\mathrm{a}}} \left( \nabla f_{ij}(x^{t+1}) - \nabla f_{ij}(x^t) - b\left(h_{ij}^t - \nabla f_{ij}(x^t)\right) \right) - \left( \nabla f_{ij}(x^{t+1}) + (1-b)(h_{ij}^t - \nabla f_{ij}(x^t)) \right) \right\|^2 \right]$$

$$+ \left( 1 - \frac{p_{\mathrm{a}}B}{m} \right) \left\| h_{ij}^t - \left( \nabla f_{ij}(x^{t+1}) + (1-b)(h_{ij}^t - \nabla f_{ij}(x^t)) \right) \right\|^2$$

$$+ (1-b)^2 \left\| h_{ij}^t - \nabla f_{ij}(x^t) \right\|^2$$

$$= \frac{\left( 1 - \frac{p_{\mathrm{a}}B}{m} \right)^2}{\frac{p_{\mathrm{a}}B}{m}} \left\| \nabla f_{ij}(x^{t+1}) - \nabla f_{ij}(x^t) - b(h_{ij}^t - \nabla f_{ij}(x^t)) \right\|^2$$

$$+ \left( 1 - \frac{p_{\mathrm{a}}B}{m} \right) \left\| \nabla f_{ij}(x^{t+1}) - \nabla f_{ij}(x^t) - b(h_{ij}^t - \nabla f_{ij}(x^t)) \right\|^2$$

$$+ (1-b)^2 \left\| h_{ij}^t - \nabla f_{ij}(x^t) \right\|^2$$

$$= \frac{\left( 1 - \frac{p_{\mathrm{a}}B}{m} \right)}{\frac{p_{\mathrm{a}}B}{m}} \left\| \nabla f_{ij}(x^{t+1}) - \nabla f_{ij}(x^t) - b(h_{ij}^t - \nabla f_{ij}(x^t)) \right\|^2$$

$$+ (1-b)^2 \left\| h_{ij}^t - \nabla f_{ij}(x^t) \right\|^2$$

$$\overset{(15)}{\leq} \frac{2\left( 1 - \frac{p_{\mathrm{a}}B}{m} \right)}{\frac{p_{\mathrm{a}}B}{m}} \left\| \nabla f_{ij}(x^{t+1}) - \nabla f_{ij}(x^t) \right\|^2 + \left( \frac{2\left( 1 - \frac{p_{\mathrm{a}}B}{m} \right)b^2}{\frac{p_{\mathrm{a}}B}{m}} + (1-b)^2 \right) \left\| h_{ij}^t - \nabla f_{ij}(x^t) \right\|^2.$$

It is left to consider Assumption 4:

$$\mathrm{E}_B\left[\mathrm{E}_{p_\mathrm{a}}\left[\left\|h_{ij}^{t+1} - \nabla f_{ij}(x^{t+1})\right\|^2\right]\right]$$

$$\leq \frac{2\left(1 - \frac{p_\mathrm{a} B}{m}\right) L_{\max}^2}{\frac{p_\mathrm{a} B}{m}}\left\|x^{t+1} - x^t\right\|^2 + \left(\frac{2\left(1 - \frac{p_\mathrm{a} B}{m}\right) b^2}{\frac{p_\mathrm{a} B}{m}} + (1-b)^2\right)\left\|h_{ij}^t - \nabla f_{ij}(x^t)\right\|^2.$$

Finally, we obtain the bound for the last inequality of the lemma:

$$\mathrm{E}_B\left[\left\|k_i^{t+1}\right\|^2\right]$$

$$\overset{(16)}{=} \mathrm{E}_B\left[\left\|k_i^{t+1} - \mathrm{E}_B\left[k_i^{t+1}\right]\right\|^2\right]$$

$$+ \left\|\nabla f_i(x^{t+1}) - \nabla f_i(x^t) - b(h_i^t - \nabla f_i(x^t))\right\|^2.$$

Using Lemma 1, we get

$$\mathrm{E}_B\left[\left\|k_i^{t+1}\right\|^2\right]$$

$$\leq \frac{m-B}{Bm(m-1)}\sum_{j=1}^m\left\|\nabla f_{ij}(x^{t+1}) - \nabla f_{ij}(x^t) - b\left(h_{ij}^t - \nabla f_{ij}(x^t)\right)\right\|^2$$

$$+ \left\|\nabla f_i(x^{t+1}) - \nabla f_i(x^t) - b(h_i^t - \nabla f_i(x^t))\right\|^2$$

$$\leq \frac{1}{Bm}\sum_{j=1}^m\left\|\nabla f_{ij}(x^{t+1}) - \nabla f_{ij}(x^t) - b\left(h_{ij}^t - \nabla f_{ij}(x^t)\right)\right\|^2$$

$$+ \left\|\nabla f_i(x^{t+1}) - \nabla f_i(x^t) - b(h_i^t - \nabla f_i(x^t))\right\|^2$$

$$\overset{(15)}{\leq} \frac{2}{Bm}\sum_{j=1}^m\left\|\nabla f_{ij}(x^{t+1}) - \nabla f_{ij}(x^t)\right\|^2 + 2\left\|\nabla f_i(x^{t+1}) - \nabla f_i(x^t)\right\|^2$$

$$+ \frac{2b^2}{Bm}\sum_{j=1}^m\left\|h_{ij}^t - \nabla f_{ij}(x^t)\right\|^2 + 2b^2\left\|h_i^t - \nabla f_i(x^t)\right\|^2$$

$$\leq \left(\frac{2L_{\max}^2}{B} + 2L_i^2\right)\left\|x^{t+1} - x^t\right\|^2$$

$$+ \frac{2b^2}{Bm}\sum_{j=1}^m\left\|h_{ij}^t - \nabla f_{ij}(x^t)\right\|^2 + 2b^2\left\|h_i^t - \nabla f_i(x^t)\right\|^2,$$

where we used Assumptions 3 and 4. $\qquad\square$

**Theorem 7.** *Suppose that Assumptions 1, 2, 3, 4, 7, and 8 hold. Let us take* $a = \frac{p_\mathrm{a}}{2\omega+1}$, $b = \frac{\frac{p_\mathrm{a} B}{m}}{2 - \frac{p_\mathrm{a} B}{m}}$,

$$\gamma \leq \left(L + \sqrt{\frac{148\omega(2\omega+1)}{np_\mathrm{a}^2}\left(\widehat{L}^2 + \frac{L_{\max}^2}{B}\right) + \frac{72m}{np_\mathrm{a}^2 B}\left(\left(1 - \frac{p_\mathrm{aa}}{p_\mathrm{a}}\right)\widehat{L}^2 + \frac{L_{\max}^2}{B}\right)}\right)^{-1},$$

$g_i^0 = h_i^0 = \nabla f_i(x^0)$ *for all* $i \in [n]$ *and* $h_{ij}^0 = \nabla f_{ij}(x^0)$ *for all* $i \in [n], j \in [m]$ *in Algorithm 1*
(DASHA-PP-FINITE-MVR) *then* $\mathrm{E}\left[\left\|\nabla f(\widehat{x}^T)\right\|^2\right] \leq \frac{2\Delta_0}{\gamma T}.$

*Proof.* Let us fix constants $\nu, \rho, \delta \in [0, \infty)$ that we will define later. Considering Lemma 6, Lemma 9, and the law of total expectation, we obtain

$$\mathrm{E}\left[f(x^{t+1})\right] + \frac{\gamma(2\omega+1)}{p_\mathrm{a}}\mathrm{E}\left[\left\|g^{t+1} - h^{t+1}\right\|^2\right] + \frac{\gamma((2\omega+1)p_\mathrm{a} - p_\mathrm{aa})}{np_\mathrm{a}^2}\mathrm{E}\left[\frac{1}{n}\sum_{i=1}^n\left\|g_i^{t+1} - h_i^{t+1}\right\|^2\right]$$

$$
+ \nu \mathrm{E}\left[\left\|h^{t+1} - \nabla f(x^{t+1})\right\|^2\right] + \rho \mathrm{E}\left[\frac{1}{n}\sum_{i=1}^{n}\left\|h_i^{t+1} - \nabla f_i(x^{t+1})\right\|^2\right]
$$

$$
+ \delta \mathrm{E}\left[\frac{1}{nm}\sum_{i=1}^{n}\sum_{j=1}^{m}\left\|h_{ij}^{t+1} - \nabla f_{ij}(x^{t+1})\right\|^2\right]
$$

$$
\leq \mathrm{E}\left[f(x^t) - \frac{\gamma}{2}\left\|\nabla f(x^t)\right\|^2 - \left(\frac{1}{2\gamma} - \frac{L}{2}\right)\left\|x^{t+1} - x^t\right\|^2 + \gamma\left\|h^t - \nabla f(x^t)\right\|^2\right]
$$

$$
+ \frac{\gamma(2\omega+1)}{p_{\mathrm{a}}}\mathrm{E}\left[\left\|g^t - h^t\right\|^2\right] + \frac{\gamma((2\omega+1)\,p_{\mathrm{a}} - p_{\mathrm{aa}})}{np_{\mathrm{a}}^2}\mathrm{E}\left[\frac{1}{n}\sum_{i=1}^{n}\left\|g_i^t - h_i^t\right\|^2\right]
$$

$$
+ \frac{4\gamma\omega(2\omega+1)}{np_{\mathrm{a}}^2}\mathrm{E}\left[\frac{1}{n}\sum_{i=1}^{n}\left\|k_i^{t+1}\right\|^2\right]
$$

$$
+ \nu \mathrm{E}\left[\left\|h^{t+1} - \nabla f(x^{t+1})\right\|^2\right] + \rho \mathrm{E}\left[\frac{1}{n}\sum_{i=1}^{n}\left\|h_i^{t+1} - \nabla f_i(x^{t+1})\right\|^2\right]
$$

$$
+ \delta \mathrm{E}\left[\frac{1}{nm}\sum_{i=1}^{n}\sum_{j=1}^{m}\left\|h_{ij}^{t+1} - \nabla f_{ij}(x^{t+1})\right\|^2\right]
$$

$$
= \mathrm{E}\left[f(x^t) - \frac{\gamma}{2}\left\|\nabla f(x^t)\right\|^2 - \left(\frac{1}{2\gamma} - \frac{L}{2}\right)\left\|x^{t+1} - x^t\right\|^2 + \gamma\left\|h^t - \nabla f(x^t)\right\|^2\right]
$$

$$
+ \frac{\gamma(2\omega+1)}{p_{\mathrm{a}}}\mathrm{E}\left[\left\|g^t - h^t\right\|^2\right] + \frac{\gamma((2\omega+1)\,p_{\mathrm{a}} - p_{\mathrm{aa}})}{np_{\mathrm{a}}^2}\mathrm{E}\left[\frac{1}{n}\sum_{i=1}^{n}\left\|g_i^t - h_i^t\right\|^2\right]
$$

$$
+ \frac{4\gamma\omega(2\omega+1)}{np_{\mathrm{a}}^2}\mathrm{E}\left[\mathrm{E}_B\left[\frac{1}{n}\sum_{i=1}^{n}\left\|k_i^{t+1}\right\|^2\right]\right]
$$

$$
+ \nu \mathrm{E}\left[\mathrm{E}_B\left[\mathrm{E}_{p_{\mathrm{a}}}\left[\left\|h^{t+1} - \nabla f(x^{t+1})\right\|^2\right]\right]\right]
$$

$$
+ \rho \mathrm{E}\left[\mathrm{E}_B\left[\mathrm{E}_{p_{\mathrm{a}}}\left[\frac{1}{n}\sum_{i=1}^{n}\left\|h_i^{t+1} - \nabla f_i(x^{t+1})\right\|^2\right]\right]\right]
$$

$$
+ \delta \mathrm{E}\left[\mathrm{E}_B\left[\mathrm{E}_{p_{\mathrm{a}}}\left[\frac{1}{nm}\sum_{i=1}^{n}\sum_{j=1}^{m}\left\|h_{ij}^{t+1} - \nabla f_{ij}(x^{t+1})\right\|^2\right]\right]\right]
$$

$$
\leq \mathrm{E}\left[f(x^t) - \frac{\gamma}{2}\left\|\nabla f(x^t)\right\|^2 - \left(\frac{1}{2\gamma} - \frac{L}{2}\right)\left\|x^{t+1} - x^t\right\|^2 + \gamma\left\|h^t - \nabla f(x^t)\right\|^2\right]
$$

$$
+ \frac{\gamma(2\omega+1)}{p_{\mathrm{a}}}\mathrm{E}\left[\left\|g^t - h^t\right\|^2\right] + \frac{\gamma((2\omega+1)\,p_{\mathrm{a}} - p_{\mathrm{aa}})}{np_{\mathrm{a}}^2}\mathrm{E}\left[\frac{1}{n}\sum_{i=1}^{n}\left\|g_i^t - h_i^t\right\|^2\right]
$$

$$
+ \frac{4\gamma\omega(2\omega+1)}{np_{\mathrm{a}}^2}\mathrm{E}\left[\left(\frac{2L_{\max}^2}{B} + 2\widehat{L}^2\right)\left\|x^{t+1} - x^t\right\|^2 + \frac{2b^2}{Bmn}\sum_{i=1}^{n}\sum_{j=1}^{m}\left\|h_{ij}^t - \nabla f_{ij}(x^t)\right\|^2 + \frac{2b^2}{n}\sum_{i=1}^{n}\left\|h_i^t - \nabla f_i(x^t)\right\|^2\right]
$$

$$
+ \nu\mathrm{E}\Bigg(\left(\frac{2L_{\max}^2}{np_{\mathrm{a}}B} + \frac{2\,(p_{\mathrm{a}} - p_{\mathrm{aa}})\,\widehat{L}^2}{np_{\mathrm{a}}^2}\right)\left\|x^{t+1} - x^t\right\|^2
$$

$$
+ \frac{2\,(p_{\mathrm{a}} - p_{\mathrm{aa}})\,b^2}{n^2 p_{\mathrm{a}}^2}\sum_{i=1}^{n}\left\|h_i^t - \nabla f_i(x^t)\right\|^2 + \frac{2b^2}{n^2 p_{\mathrm{a}}Bm}\sum_{i=1}^{n}\sum_{j=1}^{m}\left\|h_{ij}^t - \nabla f_{ij}(x^t)\right\|^2
$$

$$
+ (1-b)^2\left\|h^t - \nabla f(x^t)\right\|^2\Bigg)
$$

$$+ \rho \mathrm{E}\Bigg( \left( \frac{2L_{\max}^2}{p_{\mathrm{a}}B} + \frac{2(1-p_{\mathrm{a}})\widehat{L}^2}{p_{\mathrm{a}}} \right) \left\| x^{t+1} - x^t \right\|^2$$

$$+ \frac{2b^2}{p_{\mathrm{a}}Bnm} \sum_{i=1}^{n} \sum_{j=1}^{m} \left\| h_{ij}^t - \nabla f_{ij}(x^t) \right\|^2 + \left( \frac{2\left(1-p_{\mathrm{a}}\right)b^2}{p_{\mathrm{a}}} + (1-b)^2 \right) \frac{1}{n} \sum_{i=1}^{n} \left\| h_i^t - \nabla f_i(x^t) \right\|^2 \Bigg)$$

$$+ \delta \mathrm{E}\Bigg( \frac{2\left(1 - \frac{p_{\mathrm{a}}B}{m}\right) L_{\max}^2}{\frac{p_{\mathrm{a}}B}{m}} \left\| x^{t+1} - x^t \right\|^2$$

$$+ \left( \frac{2\left(1 - \frac{p_{\mathrm{a}}B}{m}\right)b^2}{\frac{p_{\mathrm{a}}B}{m}} + (1-b)^2 \right) \frac{1}{nm} \sum_{i=1}^{n} \sum_{j=1}^{m} \left\| h_{ij}^t - \nabla f_{ij}(x^t) \right\|^2 \Bigg).$$

Due to $b = \frac{\frac{p_{\mathrm{a}}B}{m}}{2 - \frac{p_{\mathrm{a}}B}{m}} \leq \frac{p_{\mathrm{a}}}{2-p_{\mathrm{a}}}$, we have

$$\left( \frac{2\left(1 - \frac{p_{\mathrm{a}}B}{m}\right)b^2}{\frac{p_{\mathrm{a}}B}{m}} + (1-b)^2 \right) \leq 1 - b$$

and

$$\left( \frac{2\left(1 - p_{\mathrm{a}}\right)b^2}{p_{\mathrm{a}}} + (1-b)^2 \right) \leq 1 - b.$$

Moreover, we consider that $1 - \frac{p_{\mathrm{a}}B}{m} \leq 1$, therefore

$$\mathrm{E}\left[ f(x^{t+1}) \right] + \frac{\gamma(2\omega + 1)}{p_{\mathrm{a}}} \mathrm{E}\left[ \left\| g^{t+1} - h^{t+1} \right\|^2 \right] + \frac{\gamma((2\omega + 1)p_{\mathrm{a}} - p_{\mathrm{aa}})}{np_{\mathrm{a}}^2} \mathrm{E}\left[ \frac{1}{n} \sum_{i=1}^{n} \left\| g_i^{t+1} - h_i^{t+1} \right\|^2 \right]$$

$$+ \nu \mathrm{E}\left[ \left\| h^{t+1} - \nabla f(x^{t+1}) \right\|^2 \right] + \rho \mathrm{E}\left[ \frac{1}{n} \sum_{i=1}^{n} \left\| h_i^{t+1} - \nabla f_i(x^{t+1}) \right\|^2 \right]$$

$$+ \delta \mathrm{E}\left[ \frac{1}{nm} \sum_{i=1}^{n} \sum_{j=1}^{m} \left\| h_{ij}^{t+1} - \nabla f_{ij}(x^{t+1}) \right\|^2 \right]$$

$$\leq \mathrm{E}\left[ f(x^t) - \frac{\gamma}{2} \left\| \nabla f(x^t) \right\|^2 - \left( \frac{1}{2\gamma} - \frac{L}{2} \right) \left\| x^{t+1} - x^t \right\|^2 + \gamma \left\| h^t - \nabla f(x^t) \right\|^2 \right]$$

$$+ \frac{\gamma(2\omega + 1)}{p_{\mathrm{a}}} \mathrm{E}\left[ \left\| g^t - h^t \right\|^2 \right] + \frac{\gamma((2\omega + 1)p_{\mathrm{a}} - p_{\mathrm{aa}})}{np_{\mathrm{a}}^2} \mathrm{E}\left[ \frac{1}{n} \sum_{i=1}^{n} \left\| g_i^t - h_i^t \right\|^2 \right]$$

$$+ \frac{4\gamma\omega(2\omega + 1)}{np_{\mathrm{a}}^2} \mathrm{E}\left[ \left( \frac{2L_{\max}^2}{B} + 2\widehat{L}^2 \right) \left\| x^{t+1} - x^t \right\|^2 + \frac{2b^2}{Bmn} \sum_{i=1}^{n} \sum_{j=1}^{m} \left\| h_{ij}^t - \nabla f_{ij}(x^t) \right\|^2 + \frac{2b^2}{n} \sum_{i=1}^{n} \left\| h_i^t - \nabla f_i(x^t) \right\|^2 \right]$$

$$+ \nu \mathrm{E}\Bigg( \left( \frac{2L_{\max}^2}{np_{\mathrm{a}}B} + \frac{2\left(p_{\mathrm{a}} - p_{\mathrm{aa}}\right)\widehat{L}^2}{np_{\mathrm{a}}^2} \right) \left\| x^{t+1} - x^t \right\|^2$$

$$+ \frac{2\left(p_{\mathrm{a}} - p_{\mathrm{aa}}\right)b^2}{n^2 p_{\mathrm{a}}^2} \sum_{i=1}^{n} \left\| h_i^t - \nabla f_i(x^t) \right\|^2 + \frac{2b^2}{n^2 p_{\mathrm{a}}Bm} \sum_{i=1}^{n} \sum_{j=1}^{m} \left\| h_{ij}^t - \nabla f_{ij}(x^t) \right\|^2$$

$$+ (1-b)^2 \left\| h^t - \nabla f(x^t) \right\|^2 \Bigg)$$

$$+ \rho \mathrm{E}\Bigg( \left( \frac{2L_{\max}^2}{p_{\mathrm{a}}B} + \frac{2(1-p_{\mathrm{a}})\widehat{L}^2}{p_{\mathrm{a}}} \right) \left\| x^{t+1} - x^t \right\|^2$$

$$+ \frac{2b^2}{p_{\mathrm{a}}Bnm} \sum_{i=1}^{n} \sum_{j=1}^{m} \left\| h_{ij}^t - \nabla f_{ij}(x^t) \right\|^2 + (1-b) \frac{1}{n} \sum_{i=1}^{n} \left\| h_i^t - \nabla f_i(x^t) \right\|^2 \Bigg)$$

$$+ \delta \mathrm{E} \left( \frac{2mL_{\mathrm{max}}^2}{p_{\mathrm{a}}B} \left\| x^{t+1} - x^t \right\|^2 + (1-b) \frac{1}{nm} \sum_{i=1}^{n} \sum_{j=1}^{m} \left\| h_{ij}^t - \nabla f_{ij}(x^t) \right\|^2 \right).$$

After rearranging the terms, we get

$$\mathrm{E}\left[f(x^{t+1})\right] + \frac{\gamma(2\omega+1)}{p_{\mathrm{a}}} \mathrm{E}\left[\left\|g^{t+1} - h^{t+1}\right\|^2\right] + \frac{\gamma((2\omega+1)p_{\mathrm{a}} - p_{\mathrm{aa}})}{np_{\mathrm{a}}^2} \mathrm{E}\left[\frac{1}{n}\sum_{i=1}^{n}\left\|g_i^{t+1} - h_i^{t+1}\right\|^2\right]$$

$$+ \nu \mathrm{E}\left[\left\|h^{t+1} - \nabla f(x^{t+1})\right\|^2\right] + \rho \mathrm{E}\left[\frac{1}{n}\sum_{i=1}^{n}\left\|h_i^{t+1} - \nabla f_i(x^{t+1})\right\|^2\right]$$

$$+ \delta \mathrm{E}\left[\frac{1}{nm}\sum_{i=1}^{n}\sum_{j=1}^{m}\left\|h_{ij}^{t+1} - \nabla f_{ij}(x^{t+1})\right\|^2\right]$$

$$\leq \mathrm{E}\left[f(x^t)\right] - \frac{\gamma}{2}\mathrm{E}\left[\left\|\nabla f(x^t)\right\|^2\right]$$

$$+ \frac{\gamma(2\omega+1)}{p_{\mathrm{a}}}\mathrm{E}\left[\left\|g^t - h^t\right\|^2\right] + \frac{\gamma((2\omega+1)p_{\mathrm{a}} - p_{\mathrm{aa}})}{np_{\mathrm{a}}^2}\mathrm{E}\left[\frac{1}{n}\sum_{i=1}^{n}\left\|g_i^t - h_i^t\right\|^2\right]$$

$$- \left(\frac{1}{2\gamma} - \frac{L}{2} - \frac{4\gamma\omega(2\omega+1)}{np_{\mathrm{a}}^2}\left(\frac{2L_{\mathrm{max}}^2}{B} + 2\widehat{L}^2\right)\right.$$

$$\left. - \nu\left(\frac{2L_{\mathrm{max}}^2}{np_{\mathrm{a}}B} + \frac{2(p_{\mathrm{a}} - p_{\mathrm{aa}})\widehat{L}^2}{np_{\mathrm{a}}^2}\right) - \rho\left(\frac{2L_{\mathrm{max}}^2}{p_{\mathrm{a}}B} + \frac{2(1-p_{\mathrm{a}})\widehat{L}^2}{p_{\mathrm{a}}}\right) - \delta\frac{2mL_{\mathrm{max}}^2}{p_{\mathrm{a}}B}\right)\mathrm{E}\left[\left\|x^{t+1} - x^t\right\|^2\right]$$

$$+ \left(\gamma + \nu(1-b)^2\right)\mathrm{E}\left[\left\|h^t - \nabla f(x^t)\right\|^2\right]$$

$$+ \left(\frac{8b^2\gamma\omega(2\omega+1)}{np_{\mathrm{a}}^2} + \frac{2\nu(p_{\mathrm{a}} - p_{\mathrm{aa}})b^2}{np_{\mathrm{a}}^2} + \rho(1-b)\right)\mathrm{E}\left[\frac{1}{n}\sum_{i=1}^{n}\left\|h_i^t - \nabla f_i(x^t)\right\|^2\right]$$

$$+ \left(\frac{8b^2\gamma\omega(2\omega+1)}{np_{\mathrm{a}}^2B} + \frac{2\nu b^2}{np_{\mathrm{a}}B} + \frac{2\rho b^2}{p_{\mathrm{a}}B} + \delta(1-b)\right)\mathrm{E}\left[\frac{1}{nm}\sum_{i=1}^{n}\sum_{j=1}^{m}\left\|h_{ij}^t - \nabla f_{ij}(x^t)\right\|^2\right].$$

Thus, if we take $\nu = \frac{\gamma}{b}$, then $\gamma + \nu(1-b)^2 \leq \nu$ and

$$\mathrm{E}\left[f(x^{t+1})\right] + \frac{\gamma(2\omega+1)}{p_{\mathrm{a}}}\mathrm{E}\left[\left\|g^{t+1} - h^{t+1}\right\|^2\right] + \frac{\gamma((2\omega+1)p_{\mathrm{a}} - p_{\mathrm{aa}})}{np_{\mathrm{a}}^2}\mathrm{E}\left[\frac{1}{n}\sum_{i=1}^{n}\left\|g_i^{t+1} - h_i^{t+1}\right\|^2\right]$$

$$+ \frac{\gamma}{b}\mathrm{E}\left[\left\|h^{t+1} - \nabla f(x^{t+1})\right\|^2\right] + \rho\mathrm{E}\left[\frac{1}{n}\sum_{i=1}^{n}\left\|h_i^{t+1} - \nabla f_i(x^{t+1})\right\|^2\right]$$

$$+ \delta\mathrm{E}\left[\frac{1}{nm}\sum_{i=1}^{n}\sum_{j=1}^{m}\left\|h_{ij}^{t+1} - \nabla f_{ij}(x^{t+1})\right\|^2\right]$$

$$\leq \mathrm{E}\left[f(x^t)\right] - \frac{\gamma}{2}\mathrm{E}\left[\left\|\nabla f(x^t)\right\|^2\right]$$

$$+ \frac{\gamma(2\omega+1)}{p_{\mathrm{a}}}\mathrm{E}\left[\left\|g^t - h^t\right\|^2\right] + \frac{\gamma((2\omega+1)p_{\mathrm{a}} - p_{\mathrm{aa}})}{np_{\mathrm{a}}^2}\mathrm{E}\left[\frac{1}{n}\sum_{i=1}^{n}\left\|g_i^t - h_i^t\right\|^2\right]$$

$$- \left(\frac{1}{2\gamma} - \frac{L}{2} - \frac{4\gamma\omega(2\omega+1)}{np_{\mathrm{a}}^2}\left(\frac{2L_{\mathrm{max}}^2}{B} + 2\widehat{L}^2\right)\right.$$

$$
- \left( \frac{2\gamma L_{\max}^2}{bnp_{\mathrm{a}}B} + \frac{2\gamma\left(p_{\mathrm{a}} - p_{\mathrm{aa}}\right)\widehat{L}^2}{bnp_{\mathrm{a}}^2} \right) - \rho \left( \frac{2L_{\max}^2}{p_{\mathrm{a}}B} + \frac{2(1 - p_{\mathrm{a}})\widehat{L}^2}{p_{\mathrm{a}}} \right) - \delta\frac{2mL_{\max}^2}{p_{\mathrm{a}}B} \right) \mathrm{E}\left[ \left\| x^{t+1} - x^t \right\|^2 \right]
$$

$$
+ \frac{\gamma}{b}\mathrm{E}\left[ \left\| h^t - \nabla f(x^t) \right\|^2 \right]
$$

$$
+ \left( \frac{8b^2\gamma\omega(2\omega + 1)}{np_{\mathrm{a}}^2} + \frac{2\gamma\left(p_{\mathrm{a}} - p_{\mathrm{aa}}\right)b}{np_{\mathrm{a}}^2} + \rho\left(1 - b\right) \right)\mathrm{E}\left[ \frac{1}{n}\sum_{i=1}^n \left\| h_i^t - \nabla f_i(x^t) \right\|^2 \right]
$$

$$
+ \left( \frac{8b^2\gamma\omega(2\omega + 1)}{np_{\mathrm{a}}^2 B} + \frac{2\gamma b}{np_{\mathrm{a}}B} + \frac{2\rho b^2}{p_{\mathrm{a}}B} + \delta\left(1 - b\right) \right)\mathrm{E}\left[ \frac{1}{nm}\sum_{i=1}^n\sum_{j=1}^m \left\| h_{ij}^t - \nabla f_{ij}(x^t) \right\|^2 \right].
$$

Next, if we take $\rho = \frac{8b\gamma\omega(2\omega+1)}{np_{\mathrm{a}}^2} + \frac{2\gamma(p_{\mathrm{a}} - p_{\mathrm{aa}})}{np_{\mathrm{a}}^2}$, then

$$
\left( \frac{8b^2\gamma\omega(2\omega + 1)}{np_{\mathrm{a}}^2} + \frac{2\gamma\left(p_{\mathrm{a}} - p_{\mathrm{aa}}\right)b}{np_{\mathrm{a}}^2} + \rho\left(1 - b\right) \right) = \rho,
$$

therefore

$$
\mathrm{E}\left[ f(x^{t+1}) \right] + \frac{\gamma(2\omega + 1)}{p_{\mathrm{a}}}\mathrm{E}\left[ \left\| g^{t+1} - h^{t+1} \right\|^2 \right] + \frac{\gamma((2\omega + 1)p_{\mathrm{a}} - p_{\mathrm{aa}})}{np_{\mathrm{a}}^2}\mathrm{E}\left[ \frac{1}{n}\sum_{i=1}^n \left\| g_i^{t+1} - h_i^{t+1} \right\|^2 \right]
$$

$$
+ \frac{\gamma}{b}\mathrm{E}\left[ \left\| h^{t+1} - \nabla f(x^{t+1}) \right\|^2 \right] + \left( \frac{8b\gamma\omega(2\omega + 1)}{np_{\mathrm{a}}^2} + \frac{2\gamma\left(p_{\mathrm{a}} - p_{\mathrm{aa}}\right)}{np_{\mathrm{a}}^2} \right)\mathrm{E}\left[ \frac{1}{n}\sum_{i=1}^n \left\| h_i^{t+1} - \nabla f_i(x^{t+1}) \right\|^2 \right]
$$

$$
+ \delta\mathrm{E}\left[ \frac{1}{nm}\sum_{i=1}^n\sum_{j=1}^m \left\| h_{ij}^{t+1} - \nabla f_{ij}(x^{t+1}) \right\|^2 \right]
$$

$$
\leq \mathrm{E}\left[ f(x^t) \right] - \frac{\gamma}{2}\mathrm{E}\left[ \left\| \nabla f(x^t) \right\|^2 \right]
$$

$$
+ \frac{\gamma(2\omega + 1)}{p_{\mathrm{a}}}\mathrm{E}\left[ \left\| g^t - h^t \right\|^2 \right] + \frac{\gamma((2\omega + 1)p_{\mathrm{a}} - p_{\mathrm{aa}})}{np_{\mathrm{a}}^2}\mathrm{E}\left[ \frac{1}{n}\sum_{i=1}^n \left\| g_i^t - h_i^t \right\|^2 \right]
$$

$$
- \left( \frac{1}{2\gamma} - \frac{L}{2} - \frac{4\gamma\omega(2\omega + 1)}{np_{\mathrm{a}}^2}\left( \frac{2L_{\max}^2}{B} + 2\widehat{L}^2 \right) \right.
$$

$$
- \left( \frac{2\gamma L_{\max}^2}{bnp_{\mathrm{a}}B} + \frac{2\gamma\left(p_{\mathrm{a}} - p_{\mathrm{aa}}\right)\widehat{L}^2}{bnp_{\mathrm{a}}^2} \right) - \left( \frac{8b\gamma\omega(2\omega + 1)}{np_{\mathrm{a}}^2} + \frac{2\gamma\left(p_{\mathrm{a}} - p_{\mathrm{aa}}\right)}{np_{\mathrm{a}}^2} \right)\left( \frac{2L_{\max}^2}{p_{\mathrm{a}}B} + \frac{2(1 - p_{\mathrm{a}})\widehat{L}^2}{p_{\mathrm{a}}} \right)
$$

$$
\left. - \delta\frac{2mL_{\max}^2}{p_{\mathrm{a}}B} \right)\mathrm{E}\left[ \left\| x^{t+1} - x^t \right\|^2 \right]
$$

$$
+ \frac{\gamma}{b}\mathrm{E}\left[ \left\| h^t - \nabla f(x^t) \right\|^2 \right]
$$

$$
+ \left( \frac{8b\gamma\omega(2\omega + 1)}{np_{\mathrm{a}}^2} + \frac{2\gamma\left(p_{\mathrm{a}} - p_{\mathrm{aa}}\right)}{np_{\mathrm{a}}^2} \right)\mathrm{E}\left[ \frac{1}{n}\sum_{i=1}^n \left\| h_i^t - \nabla f_i(x^t) \right\|^2 \right]
$$

$$
+ \left( \frac{8b^2\gamma\omega(2\omega + 1)}{np_{\mathrm{a}}^2 B} + \frac{2\gamma b}{np_{\mathrm{a}}B} + \frac{16b^3\gamma\omega(2\omega + 1)}{np_{\mathrm{a}}^3 B} + \frac{4b^2\gamma\left(p_{\mathrm{a}} - p_{\mathrm{aa}}\right)}{nBp_{\mathrm{a}}^3} + \delta\left(1 - b\right) \right)\mathrm{E}\left[ \frac{1}{nm}\sum_{i=1}^n\sum_{j=1}^m \left\| h_{ij}^t - \nabla f_{ij}(x^t) \right\|^2 \right].
$$

Due to $b \leq p_{\mathrm{a}}$ and $\frac{p_{\mathrm{a}} - p_{\mathrm{aa}}}{p_{\mathrm{a}}} \leq 1$, we have

$$
\frac{8b^2\gamma\omega(2\omega + 1)}{np_{\mathrm{a}}^2 B} + \frac{2\gamma b}{np_{\mathrm{a}}B} + \frac{16b^3\gamma\omega(2\omega + 1)}{np_{\mathrm{a}}^3 B} + \frac{4b^2\gamma\left(p_{\mathrm{a}} - p_{\mathrm{aa}}\right)}{nBp_{\mathrm{a}}^3}
$$

$$
\leq \frac{8b^2\gamma\omega(2\omega + 1)}{np_{\mathrm{a}}^2 B} + \frac{2\gamma b}{np_{\mathrm{a}}B} + \frac{16b^2\gamma\omega(2\omega + 1)}{np_{\mathrm{a}}^2 B} + \frac{4\gamma b}{np_{\mathrm{a}}B}
$$

$$= \frac{24b^2\gamma\omega(2\omega+1)}{np_a^2B} + \frac{6\gamma b}{np_aB}.$$

Let us take $\delta = \frac{24b\gamma\omega(2\omega+1)}{np_a^2B} + \frac{6\gamma}{np_aB}$. Thus

$$\left(\frac{8b^2\gamma\omega(2\omega+1)}{np_a^2B} + \frac{2\gamma b}{np_aB} + \frac{16b^3\gamma\omega(2\omega+1)}{np_a^3B} + \frac{4b^2\gamma\left(p_a-p_{aa}\right)}{nBp_a^3} + \delta\left(1-b\right)\right) \le \delta$$

and

$$\mathrm{E}\left[f(x^{t+1})\right] + \frac{\gamma(2\omega+1)}{p_a}\mathrm{E}\left[\left\|g^{t+1}-h^{t+1}\right\|^2\right] + \frac{\gamma((2\omega+1)\,p_a-p_{aa})}{np_a^2}\mathrm{E}\left[\frac{1}{n}\sum_{i=1}^{n}\left\|g_i^{t+1}-h_i^{t+1}\right\|^2\right]$$

$$+ \frac{\gamma}{b}\mathrm{E}\left[\left\|h^{t+1}-\nabla f(x^{t+1})\right\|^2\right] + \left(\frac{8b\gamma\omega(2\omega+1)}{np_a^2} + \frac{2\gamma\left(p_a-p_{aa}\right)}{np_a^2}\right)\mathrm{E}\left[\frac{1}{n}\sum_{i=1}^{n}\left\|h_i^{t+1}-\nabla f_i(x^{t+1})\right\|^2\right]$$

$$+ \left(\frac{24b\gamma\omega(2\omega+1)}{np_a^2B} + \frac{6\gamma}{np_aB}\right)\mathrm{E}\left[\frac{1}{nm}\sum_{i=1}^{n}\sum_{j=1}^{m}\left\|h_{ij}^{t+1}-\nabla f_{ij}(x^{t+1})\right\|^2\right]$$

$$\le \mathrm{E}\left[f(x^t)\right] - \frac{\gamma}{2}\mathrm{E}\left[\left\|\nabla f(x^t)\right\|^2\right]$$

$$+ \frac{\gamma(2\omega+1)}{p_a}\mathrm{E}\left[\left\|g^t-h^t\right\|^2\right] + \frac{\gamma((2\omega+1)\,p_a-p_{aa})}{np_a^2}\mathrm{E}\left[\frac{1}{n}\sum_{i=1}^{n}\left\|g_i^t-h_i^t\right\|^2\right]$$

$$- \left(\frac{1}{2\gamma} - \frac{L}{2} - \frac{4\gamma\omega(2\omega+1)}{np_a^2}\left(\frac{2L_{\max}^2}{B} + 2\widehat{L}^2\right)\right.$$

$$- \left(\frac{2\gamma L_{\max}^2}{bnp_aB} + \frac{2\gamma\left(p_a-p_{aa}\right)\widehat{L}^2}{bnp_a^2}\right) - \left(\frac{8b\gamma\omega(2\omega+1)}{np_a^2} + \frac{2\gamma\left(p_a-p_{aa}\right)}{np_a^2}\right)\left(\frac{2L_{\max}^2}{p_aB} + \frac{2(1-p_a)\widehat{L}^2}{p_a}\right)$$

$$\left. - \left(\frac{24b\gamma\omega(2\omega+1)}{np_a^2B} + \frac{6\gamma}{np_aB}\right)\frac{2mL_{\max}^2}{p_aB}\right)\mathrm{E}\left[\left\|x^{t+1}-x^t\right\|^2\right]$$

$$+ \frac{\gamma}{b}\mathrm{E}\left[\left\|h^t-\nabla f(x^t)\right\|^2\right]$$

$$+ \left(\frac{8b\gamma\omega(2\omega+1)}{np_a^2} + \frac{2\gamma\left(p_a-p_{aa}\right)}{np_a^2}\right)\mathrm{E}\left[\frac{1}{n}\sum_{i=1}^{n}\left\|h_i^t-\nabla f_i(x^t)\right\|^2\right]$$

$$+ \left(\frac{24b\gamma\omega(2\omega+1)}{np_a^2B} + \frac{6\gamma}{np_aB}\right)\mathrm{E}\left[\frac{1}{nm}\sum_{i=1}^{n}\sum_{j=1}^{m}\left\|h_{ij}^t-\nabla f_{ij}(x^t)\right\|^2\right].$$

Let us simplify the term near $\mathrm{E}\left[\left\|x^{t+1}-x^t\right\|^2\right]$. Due to $b \le p_a$, $\frac{p_a-p_{aa}}{p_a} \le 1$, and $1-p_a \le 1$, we have

$$\frac{4\gamma\omega(2\omega+1)}{np_a^2}\left(\frac{2L_{\max}^2}{B} + 2\widehat{L}^2\right)$$

$$+ \left(\frac{2\gamma L_{\max}^2}{bnp_aB} + \frac{2\gamma\left(p_a-p_{aa}\right)\widehat{L}^2}{bnp_a^2}\right)$$

$$+ \left(\frac{8b\gamma\omega(2\omega+1)}{np_a^2} + \frac{2\gamma\left(p_a-p_{aa}\right)}{np_a^2}\right)\left(\frac{2L_{\max}^2}{p_aB} + \frac{2(1-p_a)\widehat{L}^2}{p_a}\right)$$

$$+ \left(\frac{24b\gamma\omega(2\omega+1)}{np_a^2B} + \frac{6\gamma}{np_aB}\right)\frac{2mL_{\max}^2}{p_aB}$$

$$\le \frac{12\gamma\omega(2\omega+1)}{np_a^2}\left(\frac{2L_{\max}^2}{B} + 2\widehat{L}^2\right)$$

$$+\left(\frac{6\gamma L_{\max}^2}{bnp_{\mathrm{a}}B}+\frac{6\gamma\left(p_{\mathrm{a}}-p_{\mathrm{aa}}\right)\widehat{L}^2}{bnp_{\mathrm{a}}^2}\right)$$

$$+\left(\frac{24b\gamma\omega(2\omega+1)}{np_{\mathrm{a}}^2B}+\frac{6\gamma}{np_{\mathrm{a}}B}\right)\frac{2mL_{\max}^2}{p_{\mathrm{a}}B}$$

Considering that $b\leq\frac{p_{\mathrm{a}}B}{m}$ and $b\geq\frac{p_{\mathrm{a}}B}{2m}$, we obtain

$$\frac{4\gamma\omega(2\omega+1)}{np_{\mathrm{a}}^2}\left(\frac{2L_{\max}^2}{B}+2\widehat{L}^2\right)$$

$$+\left(\frac{2\gamma L_{\max}^2}{bnp_{\mathrm{a}}B}+\frac{2\gamma\left(p_{\mathrm{a}}-p_{\mathrm{aa}}\right)\widehat{L}^2}{bnp_{\mathrm{a}}^2}\right)$$

$$+\left(\frac{8b\gamma\omega(2\omega+1)}{np_{\mathrm{a}}^2}+\frac{2\gamma\left(p_{\mathrm{a}}-p_{\mathrm{aa}}\right)}{np_{\mathrm{a}}^2}\right)\left(\frac{2L_{\max}^2}{p_{\mathrm{a}}B}+\frac{2(1-p_{\mathrm{a}})\widehat{L}^2}{p_{\mathrm{a}}}\right)$$

$$+\left(\frac{24b\gamma\omega(2\omega+1)}{np_{\mathrm{a}}^2B}+\frac{6\gamma}{np_{\mathrm{a}}B}\right)\frac{2mL_{\max}^2}{p_{\mathrm{a}}B}$$

$$\leq\frac{36\gamma\omega(2\omega+1)}{np_{\mathrm{a}}^2}\left(\frac{2L_{\max}^2}{B}+2\widehat{L}^2\right)+\left(\frac{18\gamma L_{\max}^2}{bnp_{\mathrm{a}}B}+\frac{6\gamma\left(p_{\mathrm{a}}-p_{\mathrm{aa}}\right)\widehat{L}^2}{bnp_{\mathrm{a}}^2}\right)$$

$$\leq\frac{36\gamma\omega(2\omega+1)}{np_{\mathrm{a}}^2}\left(\frac{2L_{\max}^2}{B}+2\widehat{L}^2\right)+\left(\frac{36m\gamma L_{\max}^2}{np_{\mathrm{a}}^2B^2}+\frac{12m\gamma\left(p_{\mathrm{a}}-p_{\mathrm{aa}}\right)\widehat{L}^2}{Bnp_{\mathrm{a}}^3}\right).$$

All in all, we have

$$\mathrm{E}\left[f(x^{t+1})\right]+\frac{\gamma(2\omega+1)}{p_{\mathrm{a}}}\mathrm{E}\left[\left\|g^{t+1}-h^{t+1}\right\|^2\right]+\frac{\gamma((2\omega+1)\,p_{\mathrm{a}}-p_{\mathrm{aa}})}{np_{\mathrm{a}}^2}\mathrm{E}\left[\frac{1}{n}\sum_{i=1}^n\left\|g_i^{t+1}-h_i^{t+1}\right\|^2\right]$$

$$+\frac{\gamma}{b}\mathrm{E}\left[\left\|h^{t+1}-\nabla f(x^{t+1})\right\|^2\right]+\left(\frac{8b\gamma\omega(2\omega+1)}{np_{\mathrm{a}}^2}+\frac{2\gamma\left(p_{\mathrm{a}}-p_{\mathrm{aa}}\right)}{np_{\mathrm{a}}^2}\right)\mathrm{E}\left[\frac{1}{n}\sum_{i=1}^n\left\|h_i^{t+1}-\nabla f_i(x^{t+1})\right\|^2\right]$$

$$+\left(\frac{24b\gamma\omega(2\omega+1)}{np_{\mathrm{a}}^2B}+\frac{6\gamma}{np_{\mathrm{a}}B}\right)\mathrm{E}\left[\frac{1}{nm}\sum_{i=1}^n\sum_{j=1}^m\left\|h_{ij}^{t+1}-\nabla f_{ij}(x^{t+1})\right\|^2\right]$$

$$\leq\mathrm{E}\left[f(x^t)\right]-\frac{\gamma}{2}\mathrm{E}\left[\left\|\nabla f(x^t)\right\|^2\right]$$

$$+\frac{\gamma(2\omega+1)}{p_{\mathrm{a}}}\mathrm{E}\left[\left\|g^t-h^t\right\|^2\right]+\frac{\gamma((2\omega+1)\,p_{\mathrm{a}}-p_{\mathrm{aa}})}{np_{\mathrm{a}}^2}\mathrm{E}\left[\frac{1}{n}\sum_{i=1}^n\left\|g_i^t-h_i^t\right\|^2\right]$$

$$-\left(\frac{1}{2\gamma}-\frac{L}{2}-\frac{36\gamma\omega(2\omega+1)}{np_{\mathrm{a}}^2}\left(\frac{2L_{\max}^2}{B}+2\widehat{L}^2\right)-\left(\frac{36m\gamma L_{\max}^2}{np_{\mathrm{a}}^2B^2}+\frac{12m\gamma\left(p_{\mathrm{a}}-p_{\mathrm{aa}}\right)\widehat{L}^2}{Bnp_{\mathrm{a}}^3}\right)\right)\mathrm{E}\left[\left\|x^{t+1}-x^t\right\|^2\right]$$

$$+\frac{\gamma}{b}\mathrm{E}\left[\left\|h^t-\nabla f(x^t)\right\|^2\right]$$

$$+\left(\frac{8b\gamma\omega(2\omega+1)}{np_{\mathrm{a}}^2}+\frac{2\gamma\left(p_{\mathrm{a}}-p_{\mathrm{aa}}\right)}{np_{\mathrm{a}}^2}\right)\mathrm{E}\left[\frac{1}{n}\sum_{i=1}^n\left\|h_i^t-\nabla f_i(x^t)\right\|^2\right]$$

$$+\left(\frac{24b\gamma\omega(2\omega+1)}{np_{\mathrm{a}}^2B}+\frac{6\gamma}{np_{\mathrm{a}}B}\right)\mathrm{E}\left[\frac{1}{nm}\sum_{i=1}^n\sum_{j=1}^m\left\|h_{ij}^t-\nabla f_{ij}(x^t)\right\|^2\right].$$

Using Lemma 4 and the assumption about $\gamma$, we get

$$\mathrm{E}\left[f(x^{t+1})\right]+\frac{\gamma(2\omega+1)}{p_{\mathrm{a}}}\mathrm{E}\left[\left\|g^{t+1}-h^{t+1}\right\|^2\right]+\frac{\gamma((2\omega+1)\,p_{\mathrm{a}}-p_{\mathrm{aa}})}{np_{\mathrm{a}}^2}\mathrm{E}\left[\frac{1}{n}\sum_{i=1}^n\left\|g_i^{t+1}-h_i^{t+1}\right\|^2\right]$$

$$+ \frac{\gamma}{b} \mathrm{E}\left[\left\|h^{t+1} - \nabla f(x^{t+1})\right\|^2\right] + \left(\frac{8b\gamma\omega(2\omega+1)}{np_{\mathrm{a}}^2} + \frac{2\gamma(p_{\mathrm{a}} - p_{\mathrm{aa}})}{np_{\mathrm{a}}^2}\right) \mathrm{E}\left[\frac{1}{n}\sum_{i=1}^n \left\|h_i^{t+1} - \nabla f_i(x^{t+1})\right\|^2\right]$$

$$+ \left(\frac{24b\gamma\omega(2\omega+1)}{np_{\mathrm{a}}^2 B} + \frac{6\gamma}{np_{\mathrm{a}} B}\right) \mathrm{E}\left[\frac{1}{nm}\sum_{i=1}^n\sum_{j=1}^m \left\|h_{ij}^{t+1} - \nabla f_{ij}(x^{t+1})\right\|^2\right]$$

$$\leq \mathrm{E}\left[f(x^t)\right] - \frac{\gamma}{2}\mathrm{E}\left[\left\|\nabla f(x^t)\right\|^2\right]$$

$$+ \frac{\gamma(2\omega+1)}{p_{\mathrm{a}}}\mathrm{E}\left[\left\|g^t - h^t\right\|^2\right] + \frac{\gamma((2\omega+1)p_{\mathrm{a}} - p_{\mathrm{aa}})}{np_{\mathrm{a}}^2}\mathrm{E}\left[\frac{1}{n}\sum_{i=1}^n \left\|g_i^t - h_i^t\right\|^2\right]$$

$$+ \frac{\gamma}{b}\mathrm{E}\left[\left\|h^t - \nabla f(x^t)\right\|^2\right]$$

$$+ \left(\frac{8b\gamma\omega(2\omega+1)}{np_{\mathrm{a}}^2} + \frac{2\gamma(p_{\mathrm{a}} - p_{\mathrm{aa}})}{np_{\mathrm{a}}^2}\right)\mathrm{E}\left[\frac{1}{n}\sum_{i=1}^n \left\|h_i^t - \nabla f_i(x^t)\right\|^2\right]$$

$$+ \left(\frac{24b\gamma\omega(2\omega+1)}{np_{\mathrm{a}}^2 B} + \frac{6\gamma}{np_{\mathrm{a}} B}\right)\mathrm{E}\left[\frac{1}{nm}\sum_{i=1}^n\sum_{j=1}^m \left\|h_{ij}^t - \nabla f_{ij}(x^t)\right\|^2\right].$$

It is left to apply Lemma 3 with

$$\Psi^t = \frac{(2\omega+1)}{p_{\mathrm{a}}}\mathrm{E}\left[\left\|g^t - h^t\right\|^2\right] + \frac{((2\omega+1)p_{\mathrm{a}} - p_{\mathrm{aa}})}{np_{\mathrm{a}}^2}\mathrm{E}\left[\frac{1}{n}\sum_{i=1}^n \left\|g_i^t - h_i^t\right\|^2\right]$$

$$+ \frac{1}{b}\mathrm{E}\left[\left\|h^t - \nabla f(x^t)\right\|^2\right]$$

$$+ \left(\frac{8b\omega(2\omega+1)}{np_{\mathrm{a}}^2} + \frac{2(p_{\mathrm{a}} - p_{\mathrm{aa}})}{np_{\mathrm{a}}^2}\right)\mathrm{E}\left[\frac{1}{n}\sum_{i=1}^n \left\|h_i^t - \nabla f_i(x^t)\right\|^2\right]$$

$$+ \left(\frac{24b\omega(2\omega+1)}{np_{\mathrm{a}}^2 B} + \frac{6}{np_{\mathrm{a}} B}\right)\mathrm{E}\left[\frac{1}{nm}\sum_{i=1}^n\sum_{j=1}^m \left\|h_{ij}^t - \nabla f_{ij}(x^t)\right\|^2\right]$$

to conclude the proof. $\qquad\square$

### E.6 Proof for DASHA-PP-MVR

Let us denote $\nabla f_i(x^{t+1}; \xi_i^{t+1}) := \frac{1}{B}\sum_{j=1}^B \nabla f_i(x^{t+1}; \xi_{ij}^{t+1})$.

**Lemma 10.** *Suppose that Assumptions 3, 5, 6 and 8 hold. For $h_i^{t+1}$ and $k_i^{t+1}$ from Algorithm 1 (DASHA-PP-MVR) we have*

1.

$$\mathrm{E}_k\left[\mathrm{E}_{p_{\mathrm{a}}}\left[\left\|h^{t+1} - \nabla f(x^{t+1})\right\|^2\right]\right]$$

$$\leq \frac{2b^2\sigma^2}{np_{\mathrm{a}} B} + \left(\frac{2(1-b)^2 L_\sigma^2}{np_{\mathrm{a}} B} + \frac{2(p_{\mathrm{a}} - p_{\mathrm{aa}})\widehat{L}^2}{np_{\mathrm{a}}^2}\right)\left\|x^{t+1} - x^t\right\|^2$$

$$+ \frac{2(p_{\mathrm{a}} - p_{\mathrm{aa}})b^2}{n^2 p_{\mathrm{a}}^2}\sum_{i=1}^n \left\|h_i^t - \nabla f_i(x^t)\right\|^2 + (1-b)^2\left\|h^t - \nabla f(x^t)\right\|^2.$$

2.

$$\mathrm{E}_k\left[\mathrm{E}_{p_{\mathrm{a}}}\left[\left\|h_i^{t+1} - \nabla f_i(x^{t+1})\right\|^2\right]\right]$$

$$\leq \frac{2b^2\sigma^2}{p_{\mathrm{a}} B} + \left(\frac{2(1-b)^2 L_\sigma^2}{p_{\mathrm{a}} B} + \frac{2(1-p_{\mathrm{a}})L_i^2}{p_{\mathrm{a}}}\right)\left\|x^{t+1} - x^t\right\|^2$$

$$+ \left(\frac{2(1-p_{\mathrm{a}})b^2}{p_{\mathrm{a}}} + (1-b)^2\right)\left\|h_i^t - \nabla f_i(x^t)\right\|^2, \quad \forall i \in [n].$$

*3.*

$$\mathrm{E}_k\left[\left\|k_i^{t+1}\right\|^2\right] \leq \frac{2b^2\sigma^2}{B} + \left(\frac{2(1-b)^2 L_\sigma^2}{B} + 2L_i^2\right)\left\|x^{t+1} - x^t\right\|^2 + 2b^2\left\|h_i^t - \nabla f_i(x^t)\right\|^2, \quad \forall i \in [n].$$

*Proof.* First, let us proof the bound for $\mathrm{E}_k\left[\mathrm{E}_{p_\mathrm{a}}\left[\left\|h^{t+1} - \nabla f(x^{t+1})\right\|^2\right]\right]$:

$$\mathrm{E}_k\left[\mathrm{E}_{p_\mathrm{a}}\left[\left\|h^{t+1} - \nabla f(x^{t+1})\right\|^2\right]\right]$$
$$= \mathrm{E}_k\left[\mathrm{E}_{p_\mathrm{a}}\left[\left\|h^{t+1} - \mathrm{E}_k\left[\mathrm{E}_{p_\mathrm{a}}\left[h^{t+1}\right]\right]\right\|^2\right]\right] + \left\|\mathrm{E}_k\left[\mathrm{E}_{p_\mathrm{a}}\left[h^{t+1}\right]\right] - \nabla f(x^{t+1})\right\|^2.$$

Using

$$\mathrm{E}_k\left[\mathrm{E}_{p_\mathrm{a}}\left[h_i^{t+1}\right]\right] = h_i^t + \mathrm{E}_k\left[k_i^{t+1}\right] = h_i^t + \nabla f_i(x^{t+1}) - \nabla f_i(x^t) - b(h_i^t - \nabla f_i(x^t))$$

and (16), we have

$$\mathrm{E}_k\left[\mathrm{E}_{p_\mathrm{a}}\left[\left\|h^{t+1} - \nabla f(x^{t+1})\right\|^2\right]\right]$$
$$= \mathrm{E}_k\left[\mathrm{E}_{p_\mathrm{a}}\left[\left\|h^{t+1} - \mathrm{E}_k\left[\mathrm{E}_{p_\mathrm{a}}\left[h^{t+1}\right]\right]\right\|^2\right]\right] + (1-b)^2\left\|h^t - \nabla f(x^t)\right\|^2.$$

We can use Lemma 1 with $r_i = h_i^t$ and $s_i = k_i^{t+1}$ to obtain

$$\mathrm{E}_k\left[\mathrm{E}_{p_\mathrm{a}}\left[\left\|h^{t+1} - \nabla f(x^{t+1})\right\|^2\right]\right]$$
$$\leq \frac{1}{n^2 p_\mathrm{a}}\sum_{i=1}^{n}\mathrm{E}_k\left[\left\|k_i^{t+1} - \mathrm{E}_k\left[k_i^{t+1}\right]\right\|^2\right] + \frac{p_\mathrm{a} - p_\mathrm{aa}}{n^2 p_\mathrm{a}^2}\sum_{i=1}^{n}\left\|\mathrm{E}_k\left[k_i^{t+1}\right]\right\|^2 + (1-b)^2\left\|h^t - \nabla f(x^t)\right\|^2$$
$$= \frac{1}{n^2 p_\mathrm{a}}\sum_{i=1}^{n}\mathrm{E}_k\left[\left\|\nabla f_i(x^{t+1};\xi_i^{t+1}) - \nabla f_i(x^t;\xi_i^{t+1}) - b\left(h_i^t - \nabla f_i(x^t;\xi_i^{t+1})\right)\right.\right.$$
$$\left.\left. - \left(\nabla f_i(x^{t+1}) - \nabla f_i(x^t) - b\left(h_i^t - \nabla f_i(x^t)\right)\right)\right\|^2\right]$$
$$+ \frac{p_\mathrm{a} - p_\mathrm{aa}}{n^2 p_\mathrm{a}^2}\sum_{i=1}^{n}\left\|\nabla f_i(x^{t+1}) - \nabla f_i(x^t) - b\left(h_i^t - \nabla f_i(x^t)\right)\right\|^2$$
$$+ (1-b)^2\left\|h^t - \nabla f(x^t)\right\|^2$$
$$\overset{(15)}{\leq} \frac{2}{n^2 p_\mathrm{a}}\sum_{i=1}^{n}\mathrm{E}_k\left[\left\|b\left(\nabla f_i(x^{t+1};\xi_i^{t+1}) - \nabla f_i(x^{t+1})\right)\right\|^2\right]$$
$$+ \frac{2}{n^2 p_\mathrm{a}}\sum_{i=1}^{n}\mathrm{E}_k\left[\left\|(1-b)\left(\nabla f_i(x^{t+1};\xi_i^{t+1}) - \nabla f_i(x^t;\xi_i^{t+1}) - \left(\nabla f_i(x^{t+1}) - \nabla f_i(x^t)\right)\right)\right\|^2\right]$$
$$+ \frac{p_\mathrm{a} - p_\mathrm{aa}}{n^2 p_\mathrm{a}^2}\sum_{i=1}^{n}\left\|\nabla f_i(x^{t+1}) - \nabla f_i(x^t) - b(h_i^t - \nabla f_i(x^t))\right\|^2$$
$$+ (1-b)^2\left\|h^t - \nabla f(x^t)\right\|^2$$
$$= \frac{2b^2}{n^2 p_\mathrm{a}}\sum_{i=1}^{n}\mathrm{E}_k\left[\left\|\nabla f_i(x^{t+1};\xi_i^{t+1}) - \nabla f_i(x^{t+1})\right\|^2\right]$$
$$+ \frac{2(1-b)^2}{n^2 p_\mathrm{a}}\sum_{i=1}^{n}\mathrm{E}_k\left[\left\|\nabla f_i(x^{t+1};\xi_i^{t+1}) - \nabla f_i(x^t;\xi_i^{t+1}) - \left(\nabla f_i(x^{t+1}) - \nabla f_i(x^t)\right)\right\|^2\right]$$
$$+ \frac{p_\mathrm{a} - p_\mathrm{aa}}{n^2 p_\mathrm{a}^2}\sum_{i=1}^{n}\left\|\nabla f_i(x^{t+1}) - \nabla f_i(x^t) - b(h_i^t - \nabla f_i(x^t))\right\|^2$$
$$+ (1-b)^2\left\|h^t - \nabla f(x^t)\right\|^2.$$

$$= \frac{2b^2}{n^2 p_{\mathrm{a}} B^2} \sum_{i=1}^{n} \sum_{j=1}^{B} \mathrm{E}_k \left[ \left\| \nabla f_i(x^{t+1}; \xi_{ij}^{t+1}) - \nabla f_i(x^{t+1}) \right\|^2 \right]$$

$$+ \frac{2(1-b)^2}{n^2 p_{\mathrm{a}}} \sum_{i=1}^{n} \mathrm{E}_k \left[ \left\| \nabla f_i(x^{t+1}; \xi_i^{t+1}) - \nabla f_i(x^t; \xi_i^{t+1}) - \left( \nabla f_i(x^{t+1}) - \nabla f_i(x^t) \right) \right\|^2 \right]$$

$$+ \frac{p_{\mathrm{a}} - p_{\mathrm{aa}}}{n^2 p_{\mathrm{a}}^2} \sum_{i=1}^{n} \left\| \nabla f_i(x^{t+1}) - \nabla f_i(x^t) - b(h_i^t - \nabla f_i(x^t)) \right\|^2$$

$$+ (1-b)^2 \left\| h^t - \nabla f(x^t) \right\|^2.$$

In the last equality, we use the independence of elements in the mini-batches. Due to Assumption 5, we get

$$\mathrm{E}_k \left[ \mathrm{E}_{p_{\mathrm{a}}} \left[ \left\| h^{t+1} - \nabla f(x^{t+1}) \right\|^2 \right] \right]$$

$$\leq \frac{2b^2 \sigma^2}{n p_{\mathrm{a}} B}$$

$$+ \frac{2(1-b)^2}{n^2 p_{\mathrm{a}}} \sum_{i=1}^{n} \mathrm{E}_k \left[ \left\| \nabla f_i(x^{t+1}; \xi_i^{t+1}) - \nabla f_i(x^t; \xi_i^{t+1}) - \left( \nabla f_i(x^{t+1}) - \nabla f_i(x^t) \right) \right\|^2 \right]$$

$$+ \frac{p_{\mathrm{a}} - p_{\mathrm{aa}}}{n^2 p_{\mathrm{a}}^2} \sum_{i=1}^{n} \left\| \nabla f_i(x^{t+1}) - \nabla f_i(x^t) - b(h_i^t - \nabla f_i(x^t)) \right\|^2$$

$$+ (1-b)^2 \left\| h^t - \nabla f(x^t) \right\|^2$$

$$\overset{(15)}{\leq} \frac{2b^2 \sigma^2}{n p_{\mathrm{a}} B}$$

$$+ \frac{2(1-b)^2}{n^2 p_{\mathrm{a}}} \sum_{i=1}^{n} \mathrm{E}_k \left[ \left\| \nabla f_i(x^{t+1}; \xi_i^{t+1}) - \nabla f_i(x^t; \xi_i^{t+1}) - \left( \nabla f_i(x^{t+1}) - \nabla f_i(x^t) \right) \right\|^2 \right]$$

$$+ \frac{2(p_{\mathrm{a}} - p_{\mathrm{aa}})}{n^2 p_{\mathrm{a}}^2} \sum_{i=1}^{n} \left\| \nabla f_i(x^{t+1}) - \nabla f_i(x^t) \right\|^2 + \frac{2(p_{\mathrm{a}} - p_{\mathrm{aa}}) b^2}{n^2 p_{\mathrm{a}}^2} \sum_{i=1}^{n} \left\| h_i^t - \nabla f_i(x^t) \right\|^2$$

$$+ (1-b)^2 \left\| h^t - \nabla f(x^t) \right\|^2.$$

$$= \frac{2b^2 \sigma^2}{n p_{\mathrm{a}} B}$$

$$+ \frac{2(1-b)^2}{n^2 p_{\mathrm{a}} B^2} \sum_{i=1}^{n} \sum_{j=1}^{B} \mathrm{E}_k \left[ \left\| \nabla f_i(x^{t+1}; \xi_{ij}^{t+1}) - \nabla f_i(x^t; \xi_{ij}^{t+1}) - \left( \nabla f_i(x^{t+1}) - \nabla f_i(x^t) \right) \right\|^2 \right]$$

$$+ \frac{2(p_{\mathrm{a}} - p_{\mathrm{aa}})}{n^2 p_{\mathrm{a}}^2} \sum_{i=1}^{n} \left\| \nabla f_i(x^{t+1}) - \nabla f_i(x^t) \right\|^2 + \frac{2(p_{\mathrm{a}} - p_{\mathrm{aa}}) b^2}{n^2 p_{\mathrm{a}}^2} \sum_{i=1}^{n} \left\| h_i^t - \nabla f_i(x^t) \right\|^2$$

$$+ (1-b)^2 \left\| h^t - \nabla f(x^t) \right\|^2,$$

where we use the independence of elements in the mini-batches. Using Assumptions 3 and 6, we obtain

$$\mathrm{E}_k \left[ \mathrm{E}_{p_{\mathrm{a}}} \left[ \left\| h^{t+1} - \nabla f(x^{t+1}) \right\|^2 \right] \right]$$

$$\leq \frac{2b^2 \sigma^2}{n p_{\mathrm{a}} B} + \left( \frac{2(1-b)^2 L_\sigma^2}{n p_{\mathrm{a}} B} + \frac{2(p_{\mathrm{a}} - p_{\mathrm{aa}}) \widehat{L}^2}{n p_{\mathrm{a}}^2} \right) \left\| x^{t+1} - x^t \right\|^2$$

$$+ \frac{2(p_{\mathrm{a}} - p_{\mathrm{aa}}) b^2}{n^2 p_{\mathrm{a}}^2} \sum_{i=1}^{n} \left\| h_i^t - \nabla f_i(x^t) \right\|^2 + (1-b)^2 \left\| h^t - \nabla f(x^t) \right\|^2.$$

Now, we prove the second inequality:

$$\mathrm{E}_k\left[\mathrm{E}_{p_\mathrm{a}}\left[\left\|h_i^{t+1} - \nabla f_i(x^{t+1})\right\|^2\right]\right]$$

$$= \mathrm{E}_k\left[\mathrm{E}_{p_\mathrm{a}}\left[\left\|h_i^{t+1} - \mathrm{E}_k\left[\mathrm{E}_{p_\mathrm{a}}\left[h_i^{t+1}\right]\right]\right\|^2\right]\right]$$
$$\quad + \left\|\mathrm{E}_k\left[\mathrm{E}_{p_\mathrm{a}}\left[h_i^{t+1}\right]\right] - \nabla f_i(x^{t+1})\right\|^2$$

$$= \mathrm{E}_k\left[\mathrm{E}_{p_\mathrm{a}}\left[\left\|h_i^{t+1} - \left(h_i^t + \nabla f_i(x^{t+1}) - \nabla f_i(x^t) - b(h_i^t - \nabla f_i(x^t))\right)\right\|^2\right]\right]$$
$$\quad + \left\|h_i^t + \nabla f_i(x^{t+1}) - \nabla f_i(x^t) - b(h_i^t - \nabla f_i(x^t)) - \nabla f_i(x^{t+1})\right\|^2$$

$$= \mathrm{E}_k\left[\mathrm{E}_{p_\mathrm{a}}\left[\left\|h_i^{t+1} - \left(h_i^t + \nabla f_i(x^{t+1}) - \nabla f_i(x^t) - b(h_i^t - \nabla f_i(x^t))\right)\right\|^2\right]\right]$$
$$\quad + (1-b)^2 \left\|h_i^t - \nabla f_i(x^t)\right\|^2$$

$$= p_\mathrm{a}\mathrm{E}_k\left[\left\|h_i^t + \frac{1}{p_\mathrm{a}}k_i^{t+1} - \left(h_i^t + \nabla f_i(x^{t+1}) - \nabla f_i(x^t) - b(h_i^t - \nabla f_i(x^t))\right)\right\|^2\right]$$
$$\quad + (1-p_\mathrm{a})\left\|h_i^t - \left(h_i^t + \nabla f_i(x^{t+1}) - \nabla f_i(x^t) - b(h_i^t - \nabla f_i(x^t))\right)\right\|^2$$
$$\quad + (1-b)^2\left\|h_i^t - \nabla f_i(x^t)\right\|^2$$

$$= p_\mathrm{a}\mathrm{E}_k\left[\left\|\frac{1}{p_\mathrm{a}}k_i^{t+1} - \left(\nabla f_i(x^{t+1}) - \nabla f_i(x^t) - b(h_i^t - \nabla f_i(x^t))\right)\right\|^2\right]$$
$$\quad + (1-p_\mathrm{a})\left\|\nabla f_i(x^{t+1}) - \nabla f_i(x^t) - b(h_i^t - \nabla f_i(x^t))\right\|^2$$
$$\quad + (1-b)^2\left\|h_i^t - \nabla f_i(x^t)\right\|^2$$

$$\overset{(16)}{=} \frac{1}{p_\mathrm{a}}\mathrm{E}_k\left[\left\|k_i^{t+1} - \left(\nabla f_i(x^{t+1}) - \nabla f_i(x^t) - b(h_i^t - \nabla f_i(x^t))\right)\right\|^2\right]$$
$$\quad + \frac{(1-p_\mathrm{a})^2}{p_\mathrm{a}}\left\|\nabla f_i(x^{t+1}) - \nabla f_i(x^t) - b(h_i^t - \nabla f_i(x^t))\right\|^2$$
$$\quad + (1-p_\mathrm{a})\left\|\nabla f_i(x^{t+1}) - \nabla f_i(x^t) - b(h_i^t - \nabla f_i(x^t))\right\|^2$$
$$\quad + (1-b)^2\left\|h_i^t - \nabla f_i(x^t)\right\|^2$$

$$= \frac{1}{p_\mathrm{a}}\mathrm{E}_k\left[\left\|\nabla f_i(x^{t+1};\xi_i^{t+1}) - \nabla f_i(x^t;\xi_i^{t+1}) - b\left(h_i^t - \nabla f_i(x^t;\xi_i^{t+1})\right) - \left(\nabla f_i(x^{t+1}) - \nabla f_i(x^t) - b(h_i^t - \nabla f_i(x^t))\right)\right\|^2\right]$$
$$\quad + \frac{1-p_\mathrm{a}}{p_\mathrm{a}}\left\|\nabla f_i(x^{t+1}) - \nabla f_i(x^t) - b(h_i^t - \nabla f_i(x^t))\right\|^2$$
$$\quad + (1-b)^2\left\|h_i^t - \nabla f_i(x^t)\right\|^2$$

$$= \frac{1}{p_\mathrm{a}}\mathrm{E}_k\left[\left\|b\left(\nabla f_i(x^{t+1};\xi_i^{t+1}) - \nabla f_i(x^{t+1})\right) + (1-b)\left(\nabla f_i(x^{t+1};\xi_i^{t+1}) - \nabla f_i(x^t;\xi_i^{t+1}) - \left(\nabla f_i(x^{t+1}) - \nabla f_i(x^t)\right)\right)\right\|^2\right]$$
$$\quad + \frac{1-p_\mathrm{a}}{p_\mathrm{a}}\left\|\nabla f_i(x^{t+1}) - \nabla f_i(x^t) - b(h_i^t - \nabla f_i(x^t))\right\|^2$$
$$\quad + (1-b)^2\left\|h_i^t - \nabla f_i(x^t)\right\|^2$$

$$\overset{(15)}{\leq} \frac{2b^2}{p_\mathrm{a}}\mathrm{E}_k\left[\left\|\nabla f_i(x^{t+1};\xi_i^{t+1}) - \nabla f_i(x^{t+1})\right\|^2\right]$$
$$\quad + \frac{2(1-b)^2}{p_\mathrm{a}}\mathrm{E}_k\left[\left\|\nabla f_i(x^{t+1};\xi_i^{t+1}) - \nabla f_i(x^t;\xi_i^{t+1}) - \left(\nabla f_i(x^{t+1}) - \nabla f_i(x^t)\right)\right\|^2\right]$$
$$\quad + \frac{1-p_\mathrm{a}}{p_\mathrm{a}}\left\|\nabla f_i(x^{t+1}) - \nabla f_i(x^t) - b(h_i^t - \nabla f_i(x^t))\right\|^2$$
$$\quad + (1-b)^2\left\|h_i^t - \nabla f_i(x^t)\right\|^2.$$

Considering the independence of elements in the mini-batch, we obtain

$$
\mathrm{E}_k\left[\mathrm{E}_{p_\mathrm{a}}\left[\left\|h_i^{t+1} - \nabla f_i(x^{t+1})\right\|^2\right]\right]
$$

$$
= \frac{2b^2}{p_\mathrm{a}B^2}\sum_{j=1}^{B}\mathrm{E}_k\left[\left\|\nabla f_i(x^{t+1};\xi_{ij}^{t+1}) - \nabla f_i(x^{t+1})\right\|^2\right]
$$

$$
+ \frac{2(1-b)^2}{p_\mathrm{a}B^2}\sum_{j=1}^{B}\mathrm{E}_k\left[\left\|\nabla f_i(x^{t+1};\xi_{ij}^{t+1}) - \nabla f_i(x^t;\xi_{ij}^{t+1}) - \left(\nabla f_i(x^{t+1}) - \nabla f_i(x^t)\right)\right\|^2\right]
$$

$$
+ \frac{1-p_\mathrm{a}}{p_\mathrm{a}}\left\|\nabla f_i(x^{t+1}) - \nabla f_i(x^t) - b(h_i^t - \nabla f_i(x^t))\right\|^2
$$

$$
+ (1-b)^2\left\|h_i^t - \nabla f_i(x^t)\right\|^2.
$$

$$
\overset{(15)}{\leq} \frac{2b^2}{p_\mathrm{a}B^2}\sum_{j=1}^{B}\mathrm{E}_k\left[\left\|\nabla f_i(x^{t+1};\xi_{ij}^{t+1}) - \nabla f_i(x^{t+1})\right\|^2\right]
$$

$$
+ \frac{2(1-b)^2}{p_\mathrm{a}B^2}\sum_{j=1}^{B}\mathrm{E}_k\left[\left\|\nabla f_i(x^{t+1};\xi_{ij}^{t+1}) - \nabla f_i(x^t;\xi_{ij}^{t+1}) - \left(\nabla f_i(x^{t+1}) - \nabla f_i(x^t)\right)\right\|^2\right]
$$

$$
+ \frac{2(1-p_\mathrm{a})}{p_\mathrm{a}}\left\|\nabla f_i(x^{t+1}) - \nabla f_i(x^t)\right\|^2 + \left(\frac{2(1-p_\mathrm{a})b^2}{p_\mathrm{a}} + (1-b)^2\right)\left\|h_i^t - \nabla f_i(x^t)\right\|^2
$$

Next, we use Assumptions 3, 6, 5, to get

$$
\mathrm{E}_k\left[\mathrm{E}_{p_\mathrm{a}}\left[\left\|h_i^{t+1} - \nabla f_i(x^{t+1})\right\|^2\right]\right]
$$

$$
\leq \frac{2b^2\sigma^2}{p_\mathrm{a}B} + \left(\frac{2(1-b)^2L_\sigma^2}{p_\mathrm{a}B} + \frac{2(1-p_\mathrm{a})L_i^2}{p_\mathrm{a}}\right)\left\|x^{t+1} - x^t\right\|^2
$$

$$
+ \left(\frac{2(1-p_\mathrm{a})b^2}{p_\mathrm{a}} + (1-b)^2\right)\left\|h_i^t - \nabla f_i(x^t)\right\|^2.
$$

It is left to prove the bound for $\mathrm{E}_k\left[\left\|k_i^{t+1}\right\|^2\right]$:

$$
\mathrm{E}_k\left[\left\|k_i^{t+1}\right\|^2\right]
$$

$$
= \mathrm{E}_k\left[\left\|\nabla f_i(x^{t+1};\xi_i^{t+1}) - \nabla f_i(x^t;\xi_i^{t+1}) - b\left(h_i^t - \nabla f_i(x^t;\xi_i^{t+1})\right)\right\|^2\right]
$$

$$
\overset{(16)}{=} \mathrm{E}_k\left[\left\|\nabla f_i(x^{t+1};\xi_i^{t+1}) - \nabla f_i(x^t;\xi_i^{t+1}) - b\left(h_i^t - \nabla f_i(x^t;\xi_i^{t+1})\right) - \left(\nabla f_i(x^{t+1}) - \nabla f_i(x^t) - b(h_i^t - \nabla f_i(x^t))\right)\right\|^2\right]
$$

$$
+ \left\|\nabla f_i(x^{t+1}) - \nabla f_i(x^t) - b(h_i^t - \nabla f_i(x^t))\right\|^2
$$

$$
= \mathrm{E}_k\left[\left\|b\left(\nabla f_i(x^{t+1};\xi_i^{t+1}) - \nabla f_i(x^{t+1})\right) + (1-b)\left(\nabla f_i(x^{t+1};\xi_i^{t+1}) - \nabla f_i(x^t;\xi_i^{t+1}) - \left(\nabla f_i(x^{t+1}) - \nabla f_i(x^t)\right)\right)\right\|^2\right]
$$

$$
+ \left\|\nabla f_i(x^{t+1}) - \nabla f_i(x^t) - b(h_i^t - \nabla f_i(x^t))\right\|^2
$$

$$
\overset{(15)}{\leq} 2b^2\mathrm{E}_k\left[\left\|\nabla f_i(x^{t+1};\xi_i^{t+1}) - \nabla f_i(x^{t+1})\right\|^2\right]
$$

$$
+ 2(1-b)^2\mathrm{E}_k\left[\left\|\nabla f_i(x^{t+1};\xi_i^{t+1}) - \nabla f_i(x^t;\xi_i^{t+1}) - \left(\nabla f_i(x^{t+1}) - \nabla f_i(x^t)\right)\right\|^2\right]
$$

$$
+ 2\left\|\nabla f_i(x^{t+1}) - \nabla f_i(x^t)\right\|^2 + 2b^2\left\|h_i^t - \nabla f_i(x^t)\right\|^2.
$$

Using Assumptions 3, 6, 5 and the independence of elements in the mini-batch, we get

$$
\mathrm{E}_k\left[\left\|k_i^{t+1}\right\|^2\right]
$$

$$\leq \frac{2b^2\sigma^2}{B} + \left(\frac{2(1-b)^2 L_\sigma^2}{B} + 2L_i^2\right) \left\|x^{t+1} - x^t\right\|^2 + 2b^2 \left\|h_i^t - \nabla f_i(x^t)\right\|^2.$$

□

**Theorem 4.** *Suppose that Assumptions* *1, 2, 3, 5, 6, 7 and 8 hold. Let us take* $a = \frac{p_a}{2\omega+1}$, $b \in \left(0, \frac{p_a}{2-p_a}\right]$, $\gamma \leq \left(L + \left[\frac{48\omega(2\omega+1)}{np_a^2}\left(\widehat{L}^2 + \frac{(1-b)^2 L_\sigma^2}{B}\right) + \frac{12}{np_a b}\left(\left(1 - \frac{p_{aa}}{p_a}\right)\widehat{L}^2 + \frac{(1-b)^2 L_\sigma^2}{B}\right)\right]^{1/2}\right)^{-1}$, *and* $g_i^0 = h_i^0$ *for all* $i \in [n]$ *in Algorithm* *1* (DASHA-PP-MVR). *Then*

$$\mathrm{E}\left[\left\|\nabla f(\widehat{x}^T)\right\|^2\right] \leq \frac{1}{T}\left[\frac{2\Delta_0}{\gamma} + \frac{2}{b}\left\|h^0 - \nabla f(x^0)\right\|^2 + \left(\frac{32b\omega(2\omega+1)}{np_a^2} + \frac{4\left(1 - \frac{p_{aa}}{p_a}\right)}{np_a}\right)\left(\frac{1}{n}\sum_{i=1}^n\left\|h_i^0 - \nabla f_i(x^0)\right\|^2\right)\right]$$

$$+ \left(\frac{48b^2\omega(2\omega+1)}{p_a^2} + \frac{12b}{p_a}\right)\frac{\sigma^2}{nB}.$$

*Proof.* Let us fix constants $\nu, \rho \in [0, \infty)$ that we will define later. Considering Lemma 6, Lemma 10, and the law of total expectation, we obtain

$$\mathrm{E}\left[f(x^{t+1})\right] + \frac{\gamma(2\omega+1)}{p_a}\mathrm{E}\left[\left\|g^{t+1} - h^{t+1}\right\|^2\right] + \frac{\gamma((2\omega+1)p_a - p_{aa})}{np_a^2}\mathrm{E}\left[\frac{1}{n}\sum_{i=1}^n\left\|g_i^{t+1} - h_i^{t+1}\right\|^2\right]$$

$$+ \nu\mathrm{E}\left[\left\|h^{t+1} - \nabla f(x^{t+1})\right\|^2\right] + \rho\mathrm{E}\left[\frac{1}{n}\sum_{i=1}^n\left\|h_i^{t+1} - \nabla f_i(x^{t+1})\right\|^2\right]$$

$$\leq \mathrm{E}\left[f(x^t) - \frac{\gamma}{2}\left\|\nabla f(x^t)\right\|^2 - \left(\frac{1}{2\gamma} - \frac{L}{2}\right)\left\|x^{t+1} - x^t\right\|^2 + \gamma\left\|h^t - \nabla f(x^t)\right\|^2\right]$$

$$+ \frac{\gamma(2\omega+1)}{p_a}\mathrm{E}\left[\left\|g^t - h^t\right\|^2\right] + \frac{\gamma((2\omega+1)p_a - p_{aa})}{np_a^2}\mathrm{E}\left[\frac{1}{n}\sum_{i=1}^n\left\|g_i^t - h_i^t\right\|^2\right]$$

$$+ \frac{4\gamma\omega(2\omega+1)}{np_a^2}\mathrm{E}\left[\frac{1}{n}\sum_{i=1}^n\left\|k_i^{t+1}\right\|^2\right]$$

$$+ \nu\mathrm{E}\left[\left\|h^{t+1} - \nabla f(x^{t+1})\right\|^2\right] + \rho\mathrm{E}\left[\frac{1}{n}\sum_{i=1}^n\left\|h_i^{t+1} - \nabla f_i(x^{t+1})\right\|^2\right]$$

$$= \mathrm{E}\left[f(x^t) - \frac{\gamma}{2}\left\|\nabla f(x^t)\right\|^2 - \left(\frac{1}{2\gamma} - \frac{L}{2}\right)\left\|x^{t+1} - x^t\right\|^2 + \gamma\left\|h^t - \nabla f(x^t)\right\|^2\right]$$

$$+ \frac{\gamma(2\omega+1)}{p_a}\mathrm{E}\left[\left\|g^t - h^t\right\|^2\right] + \frac{\gamma((2\omega+1)p_a - p_{aa})}{np_a^2}\mathrm{E}\left[\frac{1}{n}\sum_{i=1}^n\left\|g_i^t - h_i^t\right\|^2\right]$$

$$+ \frac{4\gamma\omega(2\omega+1)}{np_a^2}\mathrm{E}\left[\mathrm{E}_k\left[\frac{1}{n}\sum_{i=1}^n\left\|k_i^{t+1}\right\|^2\right]\right]$$

$$+ \nu\mathrm{E}\left[\mathrm{E}_B\left[\mathrm{E}_{p_a}\left[\left\|h^{t+1} - \nabla f(x^{t+1})\right\|^2\right]\right]\right]$$

$$+ \rho\mathrm{E}\left[\mathrm{E}_B\left[\mathrm{E}_{p_a}\left[\frac{1}{n}\sum_{i=1}^n\left\|h_i^{t+1} - \nabla f_i(x^{t+1})\right\|^2\right]\right]\right]$$

$$\leq \mathrm{E}\left[f(x^t) - \frac{\gamma}{2}\left\|\nabla f(x^t)\right\|^2 - \left(\frac{1}{2\gamma} - \frac{L}{2}\right)\left\|x^{t+1} - x^t\right\|^2 + \gamma\left\|h^t - \nabla f(x^t)\right\|^2\right]$$

$$+ \frac{\gamma(2\omega+1)}{p_a}\mathrm{E}\left[\left\|g^t - h^t\right\|^2\right] + \frac{\gamma((2\omega+1)p_a - p_{aa})}{np_a^2}\mathrm{E}\left[\frac{1}{n}\sum_{i=1}^n\left\|g_i^t - h_i^t\right\|^2\right]$$

$$+ \frac{4\gamma\omega(2\omega+1)}{np_a^2}\mathrm{E}\left[\frac{2b^2\sigma^2}{B} + \left(\frac{2(1-b)^2 L_\sigma^2}{B} + 2\widehat{L}^2\right)\left\|x^{t+1} - x^t\right\|^2 + 2b^2\frac{1}{n}\sum_{i=1}^n\left\|h_i^t - \nabla f_i(x^t)\right\|^2\right]$$

$$+ \nu \mathrm{E}\Bigg( \frac{2b^2\sigma^2}{np_\mathrm{a}B} + \left( \frac{2(1-b)^2 L_\sigma^2}{np_\mathrm{a}B} + \frac{2\left(p_\mathrm{a} - p_\mathrm{aa}\right)\widehat{L}^2}{np_\mathrm{a}^2} \right) \left\| x^{t+1} - x^t \right\|^2$$

$$+ \frac{2\left(p_\mathrm{a} - p_\mathrm{aa}\right)b^2}{n^2 p_\mathrm{a}^2} \sum_{i=1}^n \left\| h_i^t - \nabla f_i(x^t) \right\|^2 + (1-b)^2 \left\| h^t - \nabla f(x^t) \right\|^2 \Bigg)$$

$$+ \rho \mathrm{E}\Bigg( \frac{2b^2\sigma^2}{p_\mathrm{a}B} + \left( \frac{2(1-b)^2 L_\sigma^2}{p_\mathrm{a}B} + \frac{2(1-p_\mathrm{a})\widehat{L}^2}{p_\mathrm{a}} \right) \left\| x^{t+1} - x^t \right\|^2$$

$$+ \left( \frac{2(1-p_\mathrm{a})b^2}{p_\mathrm{a}} + (1-b)^2 \right) \frac{1}{n}\sum_{i=1}^n \left\| h_i^t - \nabla f_i(x^t) \right\|^2 \Bigg).$$

After rearranging the terms, we get

$$\mathrm{E}\left[ f(x^{t+1}) \right] + \frac{\gamma(2\omega+1)}{p_\mathrm{a}} \mathrm{E}\left[ \left\| g^{t+1} - h^{t+1} \right\|^2 \right] + \frac{\gamma((2\omega+1)\, p_\mathrm{a} - p_\mathrm{aa})}{np_\mathrm{a}^2} \mathrm{E}\left[ \frac{1}{n}\sum_{i=1}^n \left\| g_i^{t+1} - h_i^{t+1} \right\|^2 \right]$$

$$+ \nu \mathrm{E}\left[ \left\| h^{t+1} - \nabla f(x^{t+1}) \right\|^2 \right] + \rho \mathrm{E}\left[ \frac{1}{n}\sum_{i=1}^n \left\| h_i^{t+1} - \nabla f_i(x^{t+1}) \right\|^2 \right]$$

$$\leq \mathrm{E}\left[ f(x^t) \right] - \frac{\gamma}{2}\mathrm{E}\left[ \left\| \nabla f(x^t) \right\|^2 \right]$$

$$+ \frac{\gamma(2\omega+1)}{p_\mathrm{a}} \mathrm{E}\left[ \left\| g^t - h^t \right\|^2 \right] + \frac{\gamma((2\omega+1)\, p_\mathrm{a} - p_\mathrm{aa})}{np_\mathrm{a}^2} \mathrm{E}\left[ \frac{1}{n}\sum_{i=1}^n \left\| g_i^t - h_i^t \right\|^2 \right]$$

$$- \Bigg( \frac{1}{2\gamma} - \frac{L}{2} - \frac{4\gamma\omega(2\omega+1)}{np_\mathrm{a}^2}\left( \frac{2(1-b)^2 L_\sigma^2}{B} + 2\widehat{L}^2 \right)$$

$$- \nu \left( \frac{2(1-b)^2 L_\sigma^2}{np_\mathrm{a}B} + \frac{2\left(p_\mathrm{a} - p_\mathrm{aa}\right)\widehat{L}^2}{np_\mathrm{a}^2} \right) - \rho \left( \frac{2(1-b)^2 L_\sigma^2}{p_\mathrm{a}B} + \frac{2(1-p_\mathrm{a})\widehat{L}^2}{p_\mathrm{a}} \right) \Bigg) \mathrm{E}\left[ \left\| x^{t+1} - x^t \right\|^2 \right]$$

$$+ \left( \gamma + \nu\left(1-b\right)^2 \right)\mathrm{E}\left[ \left\| h^t - \nabla f(x^t) \right\|^2 \right]$$

$$+ \left( \frac{8b^2\gamma\omega(2\omega+1)}{np_\mathrm{a}^2} + \frac{2\nu\left(p_\mathrm{a} - p_\mathrm{aa}\right)b^2}{np_\mathrm{a}^2} + \rho\left( \frac{2(1-p_\mathrm{a})b^2}{p_\mathrm{a}} + (1-b)^2 \right) \right)\mathrm{E}\left[ \frac{1}{n}\sum_{i=1}^n \left\| h_i^t - \nabla f_i(x^t) \right\|^2 \right]$$

$$+ \left( \frac{8b^2\gamma\omega(2\omega+1)}{np_\mathrm{a}^2} + \nu\frac{2b^2}{np_\mathrm{a}} + \rho\frac{2b^2}{p_\mathrm{a}} \right)\frac{\sigma^2}{B}.$$

By taking $\nu = \frac{\gamma}{b}$, one can show that $\left( \gamma + \nu(1-b)^2 \right) \leq \nu$, and

$$\mathrm{E}\left[ f(x^{t+1}) \right] + \frac{\gamma(2\omega+1)}{p_\mathrm{a}} \mathrm{E}\left[ \left\| g^{t+1} - h^{t+1} \right\|^2 \right] + \frac{\gamma((2\omega+1)\, p_\mathrm{a} - p_\mathrm{aa})}{np_\mathrm{a}^2} \mathrm{E}\left[ \frac{1}{n}\sum_{i=1}^n \left\| g_i^{t+1} - h_i^{t+1} \right\|^2 \right]$$

$$+ \frac{\gamma}{b}\mathrm{E}\left[ \left\| h^{t+1} - \nabla f(x^{t+1}) \right\|^2 \right] + \rho \mathrm{E}\left[ \frac{1}{n}\sum_{i=1}^n \left\| h_i^{t+1} - \nabla f_i(x^{t+1}) \right\|^2 \right]$$

$$\leq \mathrm{E}\left[ f(x^t) \right] - \frac{\gamma}{2}\mathrm{E}\left[ \left\| \nabla f(x^t) \right\|^2 \right]$$

$$+ \frac{\gamma(2\omega+1)}{p_\mathrm{a}} \mathrm{E}\left[ \left\| g^t - h^t \right\|^2 \right] + \frac{\gamma((2\omega+1)\, p_\mathrm{a} - p_\mathrm{aa})}{np_\mathrm{a}^2} \mathrm{E}\left[ \frac{1}{n}\sum_{i=1}^n \left\| g_i^t - h_i^t \right\|^2 \right]$$

$$- \Bigg( \frac{1}{2\gamma} - \frac{L}{2} - \frac{4\gamma\omega(2\omega+1)}{np_\mathrm{a}^2}\left( \frac{2(1-b)^2 L_\sigma^2}{B} + 2\widehat{L}^2 \right)$$

$$- \frac{\gamma}{b}\left( \frac{2(1-b)^2 L_\sigma^2}{np_\mathrm{a}B} + \frac{2\left(p_\mathrm{a} - p_\mathrm{aa}\right)\widehat{L}^2}{np_\mathrm{a}^2} \right) - \rho \left( \frac{2(1-b)^2 L_\sigma^2}{p_\mathrm{a}B} + \frac{2(1-p_\mathrm{a})\widehat{L}^2}{p_\mathrm{a}} \right) \Bigg) \mathrm{E}\left[ \left\| x^{t+1} - x^t \right\|^2 \right]$$

$$+ \frac{\gamma}{b}\mathrm{E}\left[ \left\| h^t - \nabla f(x^t) \right\|^2 \right]$$

$$+ \left( \frac{8b^2\gamma\omega(2\omega+1)}{np_{\mathrm{a}}^2} + \frac{2\gamma\left(p_{\mathrm{a}}-p_{\mathrm{aa}}\right)b}{np_{\mathrm{a}}^2} + \rho\left( \frac{2(1-p_{\mathrm{a}})b^2}{p_{\mathrm{a}}} + (1-b)^2 \right) \right) \mathrm{E}\left[ \frac{1}{n}\sum_{i=1}^{n}\left\| h_i^t - \nabla f_i(x^t) \right\|^2 \right]$$

$$+ \left( \frac{8b^2\gamma\omega(2\omega+1)}{np_{\mathrm{a}}^2} + \frac{2\gamma b}{np_{\mathrm{a}}} + \rho\frac{2b^2}{p_{\mathrm{a}}} \right) \frac{\sigma^2}{B}.$$

Note that $b \le \frac{p_{\mathrm{a}}}{2-p_{\mathrm{a}}}$, thus

$$\left( \frac{8b^2\gamma\omega(2\omega+1)}{np_{\mathrm{a}}^2} + \frac{2\gamma\left(p_{\mathrm{a}}-p_{\mathrm{aa}}\right)b}{np_{\mathrm{a}}^2} + \rho\left( \frac{2(1-p_{\mathrm{a}})b^2}{p_{\mathrm{a}}} + (1-b)^2 \right) \right)$$

$$\le \left( \frac{8b^2\gamma\omega(2\omega+1)}{np_{\mathrm{a}}^2} + \frac{2\gamma\left(p_{\mathrm{a}}-p_{\mathrm{aa}}\right)b}{np_{\mathrm{a}}^2} + \rho\left(1-b\right) \right).$$

And if we take $\rho = \frac{8b\gamma\omega(2\omega+1)}{np_{\mathrm{a}}^2} + \frac{2\gamma(p_{\mathrm{a}}-p_{\mathrm{aa}})}{np_{\mathrm{a}}^2}$, then

$$\left( \frac{8b^2\gamma\omega(2\omega+1)}{np_{\mathrm{a}}^2} + \frac{2\gamma\left(p_{\mathrm{a}}-p_{\mathrm{aa}}\right)b}{np_{\mathrm{a}}^2} + \rho\left(1-b\right) \right) \le \rho,$$

and

$$\mathrm{E}\left[ f(x^{t+1}) \right] + \frac{\gamma(2\omega+1)}{p_{\mathrm{a}}}\mathrm{E}\left[ \left\| g^{t+1} - h^{t+1} \right\|^2 \right] + \frac{\gamma((2\omega+1)p_{\mathrm{a}}-p_{\mathrm{aa}})}{np_{\mathrm{a}}^2}\mathrm{E}\left[ \frac{1}{n}\sum_{i=1}^{n}\left\| g_i^{t+1} - h_i^{t+1} \right\|^2 \right]$$

$$+ \frac{\gamma}{b}\mathrm{E}\left[ \left\| h^{t+1} - \nabla f(x^{t+1}) \right\|^2 \right] + \left( \frac{8b\gamma\omega(2\omega+1)}{np_{\mathrm{a}}^2} + \frac{2\gamma\left(p_{\mathrm{a}}-p_{\mathrm{aa}}\right)}{np_{\mathrm{a}}^2} \right) \mathrm{E}\left[ \frac{1}{n}\sum_{i=1}^{n}\left\| h_i^{t+1} - \nabla f_i(x^{t+1}) \right\|^2 \right]$$

$$\le \mathrm{E}\left[ f(x^t) \right] - \frac{\gamma}{2}\mathrm{E}\left[ \left\| \nabla f(x^t) \right\|^2 \right]$$

$$+ \frac{\gamma(2\omega+1)}{p_{\mathrm{a}}}\mathrm{E}\left[ \left\| g^t - h^t \right\|^2 \right] + \frac{\gamma((2\omega+1)p_{\mathrm{a}}-p_{\mathrm{aa}})}{np_{\mathrm{a}}^2}\mathrm{E}\left[ \frac{1}{n}\sum_{i=1}^{n}\left\| g_i^t - h_i^t \right\|^2 \right]$$

$$- \left( \frac{1}{2\gamma} - \frac{L}{2} - \frac{4\gamma\omega(2\omega+1)}{np_{\mathrm{a}}^2}\left( \frac{2(1-b)^2 L_\sigma^2}{B} + 2\widehat{L}^2 \right) \right)$$

$$- \frac{\gamma}{np_{\mathrm{a}}b}\left( \frac{2(1-b)^2 L_\sigma^2}{B} + 2\left(1-\frac{p_{\mathrm{aa}}}{p_{\mathrm{a}}}\right)\widehat{L}^2 \right)$$

$$- \left( \frac{8b\gamma\omega(2\omega+1)}{np_{\mathrm{a}}^3} + \frac{2\gamma\left(1-\frac{p_{\mathrm{aa}}}{p_{\mathrm{a}}}\right)}{np_{\mathrm{a}}^2} \right)\left( \frac{2(1-b)^2 L_\sigma^2}{B} + 2(1-p_{\mathrm{a}})\widehat{L}^2 \right) \mathrm{E}\left[ \left\| x^{t+1} - x^t \right\|^2 \right]$$

$$+ \frac{\gamma}{b}\mathrm{E}\left[ \left\| h^t - \nabla f(x^t) \right\|^2 \right] + \left( \frac{8b\gamma\omega(2\omega+1)}{np_{\mathrm{a}}^2} + \frac{2\gamma\left(p_{\mathrm{a}}-p_{\mathrm{aa}}\right)}{np_{\mathrm{a}}^2} \right)\mathrm{E}\left[ \frac{1}{n}\sum_{i=1}^{n}\left\| h_i^t - \nabla f_i(x^t) \right\|^2 \right]$$

$$+ \left( \frac{8b^2\gamma\omega(2\omega+1)}{np_{\mathrm{a}}^2} + \frac{2\gamma b}{np_{\mathrm{a}}} + \left( \frac{8b\gamma\omega(2\omega+1)}{np_{\mathrm{a}}^2} + \frac{2\gamma\left(p_{\mathrm{a}}-p_{\mathrm{aa}}\right)}{np_{\mathrm{a}}^2} \right)\frac{2b^2}{p_{\mathrm{a}}} \right)\frac{\sigma^2}{B}.$$

Let us simplify the inequality. First, due to $b \le p_{\mathrm{a}}$ and $(1-p_{\mathrm{a}}) \le \left(1-\frac{p_{\mathrm{aa}}}{p_{\mathrm{a}}}\right)$, we have

$$\left( \frac{8b\gamma\omega(2\omega+1)}{np_{\mathrm{a}}^3} + \frac{2\gamma\left(1-\frac{p_{\mathrm{aa}}}{p_{\mathrm{a}}}\right)}{np_{\mathrm{a}}^2} \right)\left( \frac{2(1-b)^2 L_\sigma^2}{B} + 2(1-p_{\mathrm{a}})\widehat{L}^2 \right)$$

$$= \frac{8b\gamma\omega(2\omega+1)}{np_{\mathrm{a}}^3}\left( \frac{2(1-b)^2 L_\sigma^2}{B} + 2(1-p_{\mathrm{a}})\widehat{L}^2 \right)$$

$$+ \frac{2\gamma\left(1-\frac{p_{\mathrm{aa}}}{p_{\mathrm{a}}}\right)}{np_{\mathrm{a}}^2}\left( \frac{2(1-b)^2 L_\sigma^2}{B} + 2(1-p_{\mathrm{a}})\widehat{L}^2 \right)$$

$$\leq \frac{8\gamma\omega(2\omega+1)}{np_{\mathrm{a}}^2}\left(\frac{2(1-b)^2L_\sigma^2}{B}+2\widehat{L}^2\right)$$
$$+\frac{2\gamma}{np_{\mathrm{a}}b}\left(\frac{2(1-b)^2L_\sigma^2}{B}+2\left(1-\frac{p_{\mathrm{aa}}}{p_{\mathrm{a}}}\right)\widehat{L}^2\right),$$

therefore

$$\mathrm{E}\left[f(x^{t+1})\right]+\frac{\gamma(2\omega+1)}{p_{\mathrm{a}}}\mathrm{E}\left[\left\|g^{t+1}-h^{t+1}\right\|^2\right]+\frac{\gamma((2\omega+1)\,p_{\mathrm{a}}-p_{\mathrm{aa}})}{np_{\mathrm{a}}^2}\mathrm{E}\left[\frac{1}{n}\sum_{i=1}^n\left\|g_i^{t+1}-h_i^{t+1}\right\|^2\right]$$
$$+\frac{\gamma}{b}\mathrm{E}\left[\left\|h^{t+1}-\nabla f(x^{t+1})\right\|^2\right]+\left(\frac{8b\gamma\omega(2\omega+1)}{np_{\mathrm{a}}^2}+\frac{2\gamma\,(p_{\mathrm{a}}-p_{\mathrm{aa}})}{np_{\mathrm{a}}^2}\right)\mathrm{E}\left[\frac{1}{n}\sum_{i=1}^n\left\|h_i^{t+1}-\nabla f_i(x^{t+1})\right\|^2\right]$$
$$\leq \mathrm{E}\left[f(x^t)\right]-\frac{\gamma}{2}\mathrm{E}\left[\left\|\nabla f(x^t)\right\|^2\right]$$
$$+\frac{\gamma(2\omega+1)}{p_{\mathrm{a}}}\mathrm{E}\left[\left\|g^t-h^t\right\|^2\right]+\frac{\gamma((2\omega+1)\,p_{\mathrm{a}}-p_{\mathrm{aa}})}{np_{\mathrm{a}}^2}\mathrm{E}\left[\frac{1}{n}\sum_{i=1}^n\left\|g_i^t-h_i^t\right\|^2\right]$$
$$-\left(\frac{1}{2\gamma}-\frac{L}{2}-\frac{12\gamma\omega(2\omega+1)}{np_{\mathrm{a}}^2}\left(\frac{2(1-b)^2L_\sigma^2}{B}+2\widehat{L}^2\right)\right.$$
$$\left.-\frac{3\gamma}{np_{\mathrm{a}}b}\left(\frac{2(1-b)^2L_\sigma^2}{B}+2\left(1-\frac{p_{\mathrm{aa}}}{p_{\mathrm{a}}}\right)\widehat{L}^2\right)\right)\mathrm{E}\left[\left\|x^{t+1}-x^t\right\|^2\right]$$
$$+\frac{\gamma}{b}\mathrm{E}\left[\left\|h^t-\nabla f(x^t)\right\|^2\right]+\left(\frac{8b\gamma\omega(2\omega+1)}{np_{\mathrm{a}}^2}+\frac{2\gamma\,(p_{\mathrm{a}}-p_{\mathrm{aa}})}{np_{\mathrm{a}}^2}\right)\mathrm{E}\left[\frac{1}{n}\sum_{i=1}^n\left\|h_i^t-\nabla f_i(x^t)\right\|^2\right]$$
$$+\left(\frac{8b^2\gamma\omega(2\omega+1)}{np_{\mathrm{a}}^2}+\frac{2\gamma b}{np_{\mathrm{a}}}+\left(\frac{8b\gamma\omega(2\omega+1)}{np_{\mathrm{a}}^2}+\frac{2\gamma\,(p_{\mathrm{a}}-p_{\mathrm{aa}})}{np_{\mathrm{a}}^2}\right)\frac{2b^2}{p_{\mathrm{a}}}\right)\frac{\sigma^2}{B}$$
$$= \mathrm{E}\left[f(x^t)\right]-\frac{\gamma}{2}\mathrm{E}\left[\left\|\nabla f(x^t)\right\|^2\right]$$
$$+\frac{\gamma(2\omega+1)}{p_{\mathrm{a}}}\mathrm{E}\left[\left\|g^t-h^t\right\|^2\right]+\frac{\gamma((2\omega+1)\,p_{\mathrm{a}}-p_{\mathrm{aa}})}{np_{\mathrm{a}}^2}\mathrm{E}\left[\frac{1}{n}\sum_{i=1}^n\left\|g_i^t-h_i^t\right\|^2\right]$$
$$-\left(\frac{1}{2\gamma}-\frac{L}{2}-\frac{24\gamma\omega(2\omega+1)}{np_{\mathrm{a}}^2}\left(\frac{(1-b)^2L_\sigma^2}{B}+\widehat{L}^2\right)\right.$$
$$\left.-\frac{6\gamma}{np_{\mathrm{a}}b}\left(\frac{(1-b)^2L_\sigma^2}{B}+\left(1-\frac{p_{\mathrm{aa}}}{p_{\mathrm{a}}}\right)\widehat{L}^2\right)\right)\mathrm{E}\left[\left\|x^{t+1}-x^t\right\|^2\right]$$
$$+\frac{\gamma}{b}\mathrm{E}\left[\left\|h^t-\nabla f(x^t)\right\|^2\right]+\left(\frac{8b\gamma\omega(2\omega+1)}{np_{\mathrm{a}}^2}+\frac{2\gamma\,(p_{\mathrm{a}}-p_{\mathrm{aa}})}{np_{\mathrm{a}}^2}\right)\mathrm{E}\left[\frac{1}{n}\sum_{i=1}^n\left\|h_i^t-\nabla f_i(x^t)\right\|^2\right]$$
$$+\left(\frac{8b^2\gamma\omega(2\omega+1)}{np_{\mathrm{a}}^2}+\frac{2\gamma b}{np_{\mathrm{a}}}+\left(\frac{8b\gamma\omega(2\omega+1)}{np_{\mathrm{a}}^2}+\frac{2\gamma\,(p_{\mathrm{a}}-p_{\mathrm{aa}})}{np_{\mathrm{a}}^2}\right)\frac{2b^2}{p_{\mathrm{a}}}\right)\frac{\sigma^2}{B}.$$

Also, we can simplify the last term:

$$\left(\frac{8b\gamma\omega(2\omega+1)}{np_{\mathrm{a}}^2}+\frac{2\gamma\,(p_{\mathrm{a}}-p_{\mathrm{aa}})}{np_{\mathrm{a}}^2}\right)\frac{2b^2}{p_{\mathrm{a}}}$$
$$=\frac{16b^3\gamma\omega(2\omega+1)}{np_{\mathrm{a}}^3}+\frac{4b^2\gamma\left(1-\frac{p_{\mathrm{aa}}}{p_{\mathrm{a}}}\right)}{np_{\mathrm{a}}^2}$$
$$\leq\frac{16b^2\gamma\omega(2\omega+1)}{np_{\mathrm{a}}^2}+\frac{4b\gamma}{np_{\mathrm{a}}},$$

thus

$$\mathrm{E}\left[f(x^{t+1})\right]+\frac{\gamma(2\omega+1)}{p_{\mathrm{a}}}\mathrm{E}\left[\left\|g^{t+1}-h^{t+1}\right\|^2\right]+\frac{\gamma((2\omega+1)\,p_{\mathrm{a}}-p_{\mathrm{aa}})}{np_{\mathrm{a}}^2}\mathrm{E}\left[\frac{1}{n}\sum_{i=1}^n\left\|g_i^{t+1}-h_i^{t+1}\right\|^2\right]$$

$$+ \frac{\gamma}{b} \mathrm{E}\left[\left\|h^{t+1} - \nabla f(x^{t+1})\right\|^2\right] + \left(\frac{8b\gamma\omega(2\omega+1)}{np_{\mathrm{a}}^2} + \frac{2\gamma(p_{\mathrm{a}} - p_{\mathrm{aa}})}{np_{\mathrm{a}}^2}\right) \mathrm{E}\left[\frac{1}{n}\sum_{i=1}^{n}\left\|h_i^{t+1} - \nabla f_i(x^{t+1})\right\|^2\right]$$

$$\leq \mathrm{E}\left[f(x^t)\right] - \frac{\gamma}{2}\mathrm{E}\left[\left\|\nabla f(x^t)\right\|^2\right]$$

$$+ \frac{\gamma(2\omega+1)}{p_{\mathrm{a}}}\mathrm{E}\left[\left\|g^t - h^t\right\|^2\right] + \frac{\gamma((2\omega+1)p_{\mathrm{a}} - p_{\mathrm{aa}})}{np_{\mathrm{a}}^2}\mathrm{E}\left[\frac{1}{n}\sum_{i=1}^{n}\left\|g_i^t - h_i^t\right\|^2\right]$$

$$- \left(\frac{1}{2\gamma} - \frac{L}{2} - \frac{24\gamma\omega(2\omega+1)}{np_{\mathrm{a}}^2}\left(\frac{(1-b)^2 L_\sigma^2}{B} + \widehat{L}^2\right)\right.$$

$$\left. - \frac{6\gamma}{np_{\mathrm{a}}b}\left(\frac{(1-b)^2 L_\sigma^2}{B} + \left(1 - \frac{p_{\mathrm{aa}}}{p_{\mathrm{a}}}\right)\widehat{L}^2\right)\right)\mathrm{E}\left[\left\|x^{t+1} - x^t\right\|^2\right]$$

$$+ \frac{\gamma}{b}\mathrm{E}\left[\left\|h^t - \nabla f(x^t)\right\|^2\right] + \left(\frac{8b\gamma\omega(2\omega+1)}{np_{\mathrm{a}}^2} + \frac{2\gamma(p_{\mathrm{a}} - p_{\mathrm{aa}})}{np_{\mathrm{a}}^2}\right)\mathrm{E}\left[\frac{1}{n}\sum_{i=1}^{n}\left\|h_i^t - \nabla f_i(x^t)\right\|^2\right]$$

$$+ \left(\frac{24b^2\gamma\omega(2\omega+1)}{np_{\mathrm{a}}^2} + \frac{6\gamma b}{np_{\mathrm{a}}}\right)\frac{\sigma^2}{B}.$$

Using Lemma 4 and the assumption about $\gamma$, we get

$$\mathrm{E}\left[f(x^{t+1})\right] + \frac{\gamma(2\omega+1)}{p_{\mathrm{a}}}\mathrm{E}\left[\left\|g^{t+1} - h^{t+1}\right\|^2\right] + \frac{\gamma((2\omega+1)p_{\mathrm{a}} - p_{\mathrm{aa}})}{np_{\mathrm{a}}^2}\mathrm{E}\left[\frac{1}{n}\sum_{i=1}^{n}\left\|g_i^{t+1} - h_i^{t+1}\right\|^2\right]$$

$$+ \frac{\gamma}{b}\mathrm{E}\left[\left\|h^{t+1} - \nabla f(x^{t+1})\right\|^2\right] + \left(\frac{8b\gamma\omega(2\omega+1)}{np_{\mathrm{a}}^2} + \frac{2\gamma(p_{\mathrm{a}} - p_{\mathrm{aa}})}{np_{\mathrm{a}}^2}\right)\mathrm{E}\left[\frac{1}{n}\sum_{i=1}^{n}\left\|h_i^{t+1} - \nabla f_i(x^{t+1})\right\|^2\right]$$

$$\leq \mathrm{E}\left[f(x^t)\right] - \frac{\gamma}{2}\mathrm{E}\left[\left\|\nabla f(x^t)\right\|^2\right]$$

$$+ \frac{\gamma(2\omega+1)}{p_{\mathrm{a}}}\mathrm{E}\left[\left\|g^t - h^t\right\|^2\right] + \frac{\gamma((2\omega+1)p_{\mathrm{a}} - p_{\mathrm{aa}})}{np_{\mathrm{a}}^2}\mathrm{E}\left[\frac{1}{n}\sum_{i=1}^{n}\left\|g_i^t - h_i^t\right\|^2\right]$$

$$+ \frac{\gamma}{b}\mathrm{E}\left[\left\|h^t - \nabla f(x^t)\right\|^2\right] + \left(\frac{8b\gamma\omega(2\omega+1)}{np_{\mathrm{a}}^2} + \frac{2\gamma(p_{\mathrm{a}} - p_{\mathrm{aa}})}{np_{\mathrm{a}}^2}\right)\mathrm{E}\left[\frac{1}{n}\sum_{i=1}^{n}\left\|h_i^t - \nabla f_i(x^t)\right\|^2\right]$$

$$+ \left(\frac{24b^2\gamma\omega(2\omega+1)}{np_{\mathrm{a}}^2} + \frac{6\gamma b}{np_{\mathrm{a}}}\right)\frac{\sigma^2}{B}.$$

It is left to apply Lemma 3 with

$$\Psi^t = \frac{(2\omega+1)}{p_{\mathrm{a}}}\mathrm{E}\left[\left\|g^t - h^t\right\|^2\right] + \frac{((2\omega+1)p_{\mathrm{a}} - p_{\mathrm{aa}})}{np_{\mathrm{a}}^2}\mathrm{E}\left[\frac{1}{n}\sum_{i=1}^{n}\left\|g_i^t - h_i^t\right\|^2\right]$$

$$+ \frac{1}{b}\mathrm{E}\left[\left\|h^t - \nabla f(x^t)\right\|^2\right] + \left(\frac{8b\omega(2\omega+1)}{np_{\mathrm{a}}^2} + \frac{2(p_{\mathrm{a}} - p_{\mathrm{aa}})}{np_{\mathrm{a}}^2}\right)\mathrm{E}\left[\frac{1}{n}\sum_{i=1}^{n}\left\|h_i^t - \nabla f_i(x^t)\right\|^2\right]$$

and $C = \left(\frac{24b^2\omega(2\omega+1)}{p_{\mathrm{a}}^2} + \frac{6b}{p_{\mathrm{a}}}\right)\frac{\sigma^2}{nB}$ to conclude the proof. $\qquad\square$

**Corollary 3.** *Suppose that assumptions from Theorem 4 hold, momentum* $b = \Theta\left(\min\left\{\frac{p_{\mathrm{a}}}{\omega}\sqrt{\frac{n\varepsilon B}{\sigma^2}}, \frac{p_{\mathrm{a}}n\varepsilon B}{\sigma^2}\right\}\right)$, $\frac{\sigma^2}{n\varepsilon B} \geq 1$, *and* $h_i^0 = \frac{1}{B_{\mathrm{init}}}\sum_{k=1}^{B_{\mathrm{init}}}\nabla f_i(x^0; \xi_{ik}^0)$ *for all* $i \in [n]$, *and batch size* $B_{\mathrm{init}} = \Theta\left(\frac{\sqrt{p_{\mathrm{a}}}B}{b}\right)$, *then Algorithm 1 (DASHA-PP-MVR) needs*

$$T := \mathcal{O}\left(\frac{\Delta_0}{\varepsilon}\left[L + \frac{\omega}{p_{\mathrm{a}}\sqrt{n}}\left(\widehat{L} + \frac{L_\sigma}{\sqrt{B}}\right) + \frac{\sigma}{p_{\mathrm{a}}\sqrt{\varepsilon}n}\left(\frac{\mathbb{1}_{p_{\mathrm{a}}}\widehat{L}}{\sqrt{B}} + \frac{L_\sigma}{B}\right)\right] + \frac{\sigma^2}{\sqrt{p_{\mathrm{a}}}n\varepsilon B}\right)$$

*communication rounds to get an* $\varepsilon$*-solution and the number of stochastic gradient calculations per node equals* $\mathcal{O}(B_{\mathrm{init}} + BT)$.

*Proof.* Using the result from Theorem 4, we have

$$\mathrm{E}\left[\left\|\nabla f(\widehat{x}^T)\right\|^2\right]$$

$$\leq \frac{1}{T}\left[2\Delta_0\left(L + \sqrt{\frac{48\omega(2\omega+1)}{np_{\mathrm{a}}^2}\left(\widehat{L}^2 + \frac{(1-b)^2 L_\sigma^2}{B}\right) + \frac{12}{np_{\mathrm{a}}b}\left(\left(1 - \frac{p_{\mathrm{aa}}}{p_{\mathrm{a}}}\right)\widehat{L}^2 + \frac{(1-b)^2 L_\sigma^2}{B}\right)}\right)\right.$$

$$\left.+ \frac{2}{b}\left\|h^0 - \nabla f(x^0)\right\|^2 + \left(\frac{32b\omega(2\omega+1)}{np_{\mathrm{a}}^2} + \frac{4\left(1 - \frac{p_{\mathrm{aa}}}{p_{\mathrm{a}}}\right)}{np_{\mathrm{a}}}\right)\left(\frac{1}{n}\sum_{i=1}^{n}\left\|h_i^0 - \nabla f_i(x^0)\right\|^2\right)\right]$$

$$+ \left(\frac{48b^2\omega(2\omega+1)}{p_{\mathrm{a}}^2} + \frac{12b}{p_{\mathrm{a}}}\right)\frac{\sigma^2}{nB}$$

We choose $b$ to ensure $\left(\frac{48b^2\omega(2\omega+1)}{p_{\mathrm{a}}^2} + \frac{12b}{p_{\mathrm{a}}}\right)\frac{\sigma^2}{nB} = \Theta(\varepsilon)$. Note that $\frac{1}{b} = \Theta\left(\max\left\{\frac{\omega}{p_{\mathrm{a}}}\sqrt{\frac{\sigma^2}{n\varepsilon B}}, \frac{\sigma^2}{p_{\mathrm{a}}n\varepsilon B}\right\}\right) \leq \Theta\left(\max\left\{\frac{\omega^2}{p_{\mathrm{a}}}, \frac{\sigma^2}{p_{\mathrm{a}}n\varepsilon B}\right\}\right)$, thus

$$\mathrm{E}\left[\left\|\nabla f(\widehat{x}^T)\right\|^2\right]$$

$$= \mathcal{O}\left(\frac{1}{T}\left[\Delta_0\left(L + \frac{\omega}{p_{\mathrm{a}}\sqrt{n}}\left(\widehat{L} + \frac{L_\sigma}{\sqrt{B}}\right) + \sqrt{\frac{\sigma^2}{p_{\mathrm{a}}^2\varepsilon n^2 B}}\left(\mathbb{1}_{p_{\mathrm{a}}}\widehat{L} + \frac{L_\sigma}{\sqrt{B}}\right)\right)\right.\right.$$

$$\left.\left.+ \frac{1}{b}\left\|h^0 - \nabla f(x^0)\right\|^2 + \left(\frac{b\omega^2}{np_{\mathrm{a}}^2} + \frac{1}{np_{\mathrm{a}}}\right)\left(\frac{1}{n}\sum_{i=1}^{n}\left\|h_i^0 - \nabla f_i(x^0)\right\|^2\right)\right] + \varepsilon\right),$$

where $\mathbb{1}_{p_{\mathrm{a}}} = \sqrt{1 - \frac{p_{\mathrm{aa}}}{p_{\mathrm{a}}}}$. It enough to take the following $T$ to get $\varepsilon$-solution.

$$T = \mathcal{O}\left(\frac{1}{\varepsilon}\left[\Delta_0\left(L + \frac{\omega}{p_{\mathrm{a}}\sqrt{n}}\left(\widehat{L} + \frac{L_\sigma}{\sqrt{B}}\right) + \sqrt{\frac{\sigma^2}{p_{\mathrm{a}}^2\varepsilon n^2 B}}\left(\mathbb{1}_{p_{\mathrm{a}}}\widehat{L} + \frac{L_\sigma}{\sqrt{B}}\right)\right)\right.\right.$$

$$\left.\left.+ \frac{1}{b}\left\|h^0 - \nabla f(x^0)\right\|^2 + \left(\frac{b\omega^2}{np_{\mathrm{a}}^2} + \frac{1}{np_{\mathrm{a}}}\right)\left(\frac{1}{n}\sum_{i=1}^{n}\left\|h_i^0 - \nabla f_i(x^0)\right\|^2\right)\right]\right).$$

Let us bound the norms:

$$\mathrm{E}\left[\left\|h^0 - \nabla f(x^0)\right\|^2\right] = \mathrm{E}\left[\left\|\frac{1}{n}\sum_{i=1}^{n}\frac{1}{B_{\mathrm{init}}}\sum_{k=1}^{B_{\mathrm{init}}}\nabla f_i(x^0;\xi_{ik}^0) - \nabla f(x^0)\right\|^2\right]$$

$$= \frac{1}{n^2 B_{\mathrm{init}}^2}\sum_{i=1}^{n}\sum_{k=1}^{B_{\mathrm{init}}}\mathrm{E}\left[\left\|\nabla f_i(x^0;\xi_{ik}^0) - \nabla f_i(x^0)\right\|^2\right]$$

$$\leq \frac{\sigma^2}{nB_{\mathrm{init}}}.$$

Using the same reasoning, one cat get $\frac{1}{n}\sum_{i=1}^{n}\mathrm{E}\left[\left\|h_i^0 - \nabla f_i(x^0)\right\|^2\right] \leq \frac{\sigma^2}{B_{\mathrm{init}}}$. Combining all inequalities, we have

$$T = \mathcal{O}\left(\frac{1}{\varepsilon}\left[\Delta_0\left(L + \frac{\omega}{p_a\sqrt{n}}\left(\widehat{L} + \frac{L_\sigma}{\sqrt{B}}\right) + \sqrt{\frac{\sigma^2}{p_a^2\varepsilon n^2 B}}\left(\mathbb{1}_{p_a}\widehat{L} + \frac{L_\sigma}{\sqrt{B}}\right)\right)\right.\right.$$

$$\left.\left. + \frac{\sigma^2}{bnB_{\text{init}}} + \frac{b\omega^2\sigma^2}{np_a^2 B_{\text{init}}} + \frac{\sigma^2}{np_a B_{\text{init}}}\right]\right).$$

Using the choice of $B_{\text{init}}$ and $b$, we obtain

$$T = \mathcal{O}\left(\frac{1}{\varepsilon}\left[\Delta_0\left(L + \frac{\omega}{p_a\sqrt{n}}\left(\widehat{L} + \frac{L_\sigma}{\sqrt{B}}\right) + \sqrt{\frac{\sigma^2}{p_a^2\varepsilon n^2 B}}\left(\mathbb{1}_{p_a}\widehat{L} + \frac{L_\sigma}{\sqrt{B}}\right)\right)\right.\right.$$

$$\left.\left. + \frac{\sigma^2}{\sqrt{p_a}nB} + \frac{b^2\omega^2\sigma^2}{np_a^{5/2}B} + \frac{b\sigma^2}{p_a^{3/2}nB}\right]\right)$$

$$= \mathcal{O}\left(\frac{1}{\varepsilon}\left[\Delta_0\left(L + \frac{\omega}{p_a\sqrt{n}}\left(\widehat{L} + \frac{L_\sigma}{\sqrt{B}}\right) + \sqrt{\frac{\sigma^2}{p_a^2\varepsilon n^2 B}}\left(\mathbb{1}_{p_a}\widehat{L} + \frac{L_\sigma}{\sqrt{B}}\right)\right)\right.\right.$$

$$\left.\left. + \frac{\sigma^2}{\sqrt{p_a}nB} + \frac{\varepsilon}{\sqrt{p_a}}\right]\right)$$

$$= \mathcal{O}\left(\frac{\Delta_0}{\varepsilon}\left[L + \frac{\omega}{p_a\sqrt{n}}\left(\widehat{L} + \frac{L_\sigma}{\sqrt{B}}\right) + \sqrt{\frac{\sigma^2}{p_a^2\varepsilon n^2 B}}\left(\mathbb{1}_{p_a}\widehat{L} + \frac{L_\sigma}{\sqrt{B}}\right)\right] + \frac{\sigma^2}{\sqrt{p_a}n\varepsilon B} + \frac{1}{\sqrt{p_a}}\right).$$

Using $\frac{\sigma^2}{n\varepsilon B} \geq 1$, we can conclude the proof of the inequality. The number of stochastic gradients that each node calculates equals $B_{\text{init}} + 2BT = \mathcal{O}(B_{\text{init}} + BT)$. □

**Corollary 4.** *Suppose that assumptions of Corollary 3 hold, batch size $B \leq \min\left\{\frac{\sigma}{p_a\sqrt{\varepsilon}n}, \frac{L_\sigma^2}{\mathbb{1}_{p_a}^2\widehat{L}^2}\right\}$, we take RandK compressors with $K = \Theta\left(\frac{Bd\sqrt{\varepsilon n}}{\sigma}\right)$. Then the communication complexity equals $\mathcal{O}\left(\frac{d\sigma}{\sqrt{p_a}\sqrt{n}\varepsilon} + \frac{L_\sigma\Delta_0 d}{p_a\sqrt{n}\varepsilon}\right)$, and the expected number of stochastic gradient calculations per node equals $\mathcal{O}\left(\frac{\sigma^2}{\sqrt{p_a}n\varepsilon} + \frac{L_\sigma\Delta_0\sigma}{p_a\varepsilon^{3/2}n}\right)$.*

*Proof.* The communication complexity equals

$$\mathcal{O}\left(d + KT\right) = \mathcal{O}\left(d + \frac{\Delta_0}{\varepsilon}\left[KL + K\frac{\omega}{p_a\sqrt{n}}\left(\widehat{L} + \frac{L_\sigma}{\sqrt{B}}\right) + K\sqrt{\frac{\sigma^2}{p_a^2\varepsilon n^2 B}}\left(\mathbb{1}_{p_a}\widehat{L} + \frac{L_\sigma}{\sqrt{B}}\right)\right] + K\frac{\sigma^2}{\sqrt{p_a}n\varepsilon B}\right).$$

Due to $B \le \frac{L_\sigma^2}{\mathbb{1}_{p_a}^2 \widehat{L}^2}$, we have $\mathbb{1}_{p_a}\widehat{L} + \frac{L_\sigma}{\sqrt{B}} \le \frac{2L_\sigma}{\sqrt{B}}$ and

$$\mathcal{O}\left(d + KT\right) \;=\; \mathcal{O}\left(d + \frac{\Delta_0}{\varepsilon}\left[KL + K\frac{\omega}{p_a\sqrt{n}}\left(\widehat{L} + \frac{L_\sigma}{\sqrt{B}}\right) + K\sqrt{\frac{\sigma^2}{p_a^2\varepsilon n^2 B}}\frac{L_\sigma}{\sqrt{B}}\right] + K\frac{\sigma^2}{\sqrt{p_a}n\varepsilon B}\right).$$

From Theorem 6, we have $\omega + 1 = \frac{d}{K}$. Since $K = \Theta\left(\frac{Bd\sqrt{\varepsilon n}}{\sigma}\right) = \mathcal{O}\left(\frac{d}{p_a\sqrt{n}}\right)$, the communication complexity equals

$$\mathcal{O}\left(d + KT\right) \;=\; \mathcal{O}\left(d + \frac{\Delta_0}{\varepsilon}\left[\frac{d}{p_a\sqrt{n}}L + \frac{d}{p_a\sqrt{n}}\left(\widehat{L} + \frac{L_\sigma}{\sqrt{B}}\right) + \frac{d}{p_a\sqrt{n}}L_\sigma\right] + \frac{d\sigma}{\sqrt{p_a}\sqrt{n}\varepsilon}\right)$$

$$\;=\; \mathcal{O}\left(\frac{d\sigma}{\sqrt{p_a}\sqrt{n}\varepsilon} + \frac{L_\sigma\Delta_0 d}{p_a\sqrt{n}\varepsilon}\right)$$

And the expected number of stochastic gradient calculations per node equals

$\mathcal{O}\left(B_{\text{init}} + BT\right)$

$$= \mathcal{O}\left(\frac{\sigma^2}{\sqrt{p_a}n\varepsilon} + \frac{B\omega}{\sqrt{p_a}}\sqrt{\frac{\sigma^2}{n\varepsilon B}} + \frac{\Delta_0}{\varepsilon}\left[BL + B\frac{\omega}{p_a\sqrt{n}}\left(\widehat{L} + \frac{L_\sigma}{\sqrt{B}}\right) + B\sqrt{\frac{\sigma^2}{p_a^2\varepsilon n^2 B}}\left(\mathbb{1}_{p_a}\widehat{L} + \frac{L_\sigma}{\sqrt{B}}\right)\right] + B\frac{\sigma^2}{\sqrt{p_a}n\varepsilon B}\right)$$

$$= \mathcal{O}\left(\frac{\sigma^2}{\sqrt{p_a}n\varepsilon} + \frac{Bd}{K\sqrt{p_a}}\sqrt{\frac{\sigma^2}{n\varepsilon B}} + \frac{\Delta_0}{\varepsilon}\left[BL + B\frac{d}{Kp_a\sqrt{n}}\left(\widehat{L} + \frac{L_\sigma}{\sqrt{B}}\right) + B\sqrt{\frac{\sigma^2}{p_a^2\varepsilon n^2 B}}\frac{L_\sigma}{\sqrt{B}}\right] + \frac{\sigma^2}{\sqrt{p_a}n\varepsilon}\right)$$

$$= \mathcal{O}\left(\frac{\sigma^2}{\sqrt{p_a}n\varepsilon} + \frac{\sigma^2}{\sqrt{p_a}n\varepsilon\sqrt{B}} + \frac{\Delta_0}{\varepsilon}\left[\frac{\sigma}{p_a\sqrt{\varepsilon n}}L + \frac{\sigma}{p_a\sqrt{\varepsilon n}}\left(\widehat{L} + \frac{L_\sigma}{\sqrt{B}}\right) + \frac{\sigma}{p_a\sqrt{\varepsilon n}}L_\sigma\right]\right)$$

$$= \mathcal{O}\left(\frac{\sigma^2}{\sqrt{p_a}n\varepsilon} + \frac{L_\sigma\Delta_0\sigma}{p_a\varepsilon^{3/2}n}\right).$$

$\square$

# F Analysis of DASHA-PP under Polyak-Łojasiewicz Condition

In this section, we provide the theoretical convergence rates of DASHA-PP under Polyak-Łojasiewiczc Condition.

**Assumption 9.** *The function $f$ satisfy (Polyak-Łojasiewicz) PŁ-condition:*

$$\|\nabla f(x)\|^2 \geq 2\mu(f(x) - f^*), \quad \forall x \in \mathbb{R}, \tag{30}$$

*where $f^* = \inf_{x \in \mathbb{R}^d} f(x) > -\infty$.*

Under Polyak-Łojasiewicz condition, a (random) point $\widehat{x}$ is $\varepsilon$-solution, if $\mathrm{E}\left[f(\widehat{x})\right] - f^* \leq \varepsilon$.

We now provide the convergence rates of DASHA-PP under PŁ-condition.

## F.1 Gradient Setting

**Theorem 8.** *Suppose that Assumption 1, 2, 3, 7, 8 and 9 hold. Let us take $a = \frac{p_{\mathrm{a}}}{2\omega+1}$, $b = \frac{p_{\mathrm{a}}}{2-p_{\mathrm{a}}}$,*

$$\gamma \leq \min\left\{ \left( L + \sqrt{\frac{200\omega(2\omega+1)}{np_{\mathrm{a}}^2} + \frac{48}{np_{\mathrm{a}}^2}\left(1 - \frac{p_{\mathrm{aa}}}{p_{\mathrm{a}}}\right)\widehat{L}} \right)^{-1}, \frac{a}{4\mu} \right\},$$

*and $h_i^0 = g_i^0 = \nabla f_i(x^0)$ for all $i \in [n]$ in Algorithm 1 (DASHA-PP), then $\mathrm{E}\left[f(x^T)\right] - f^* \leq (1 - \gamma\mu)^T \Delta_0$.*

Let us provide bounds up to logarithmic factors and use $\widetilde{\mathcal{O}}(\cdot)$ notation. The provided theorem states that to get $\varepsilon$-solution DASHA-PP have to run

$$\widetilde{\mathcal{O}}\left( \frac{\omega+1}{p_{\mathrm{a}}} + \frac{L}{\mu} + \frac{\omega\widehat{L}}{p_{\mathrm{a}}\mu\sqrt{n}} + \frac{\widehat{L}}{p_{\mathrm{a}}\mu\sqrt{n}} \right),$$

communication rounds. The method DASHA from (Tyurin and Richtárik, 2023), have to run

$$\widetilde{\mathcal{O}}\left( \omega + \frac{L}{\mu} + \frac{\omega\widehat{L}}{\mu\sqrt{n}} \right),$$

communication rounds to get $\varepsilon$-solution. The difference is the same as in the general nonconvex case (see Section 6.1). Up to Lipschitz constants factors, we get the degeneration up to $1/p_{\mathrm{a}}$ factor due to the partial participation.

## F.2 Finite-Sum Setting

**Theorem 9.** *Suppose that Assumption 1, 2, 3, 7, 4, 8, and 9 hold. Let us take $a = \frac{p_{\mathrm{a}}}{2\omega+1}$, probability $p_{page} = \frac{B}{m+B}$, $b = \frac{p_{page}p_{\mathrm{a}}}{2-p_{\mathrm{a}}}$,*

$$\gamma \leq \min\left\{ \left( L + \sqrt{\frac{200\omega(2\omega+1)}{np_{\mathrm{a}}^2}\left(\widehat{L}^2 + \frac{(1-p_{page})L_{\max}^2}{B}\right) + \frac{48}{np_{\mathrm{a}}^2 p_{page}}\left(\left(1 - \frac{p_{\mathrm{aa}}}{p_{\mathrm{a}}}\right)\widehat{L}^2 + \frac{(1-p_{page})L_{\max}^2}{B}\right)} \right)^{-1}, \frac{a}{2\mu}, \frac{b}{2\mu} \right\},$$

*and $h_i^0 = g_i^0 = \nabla f_i(x^0)$ for all $i \in [n]$ in Algorithm 1 (DASHA-PP-PAGE), then $\mathrm{E}\left[f(x^T)\right] - f^* \leq (1 - \gamma\mu)^T \Delta_0$.*

The provided theorem states that to get $\varepsilon$-solution DASHA-PP have to run

$$\widetilde{\mathcal{O}}\left( \frac{\omega+1}{p_{\mathrm{a}}} + \frac{m}{p_{\mathrm{a}}B} + \frac{L}{\mu} + \frac{\omega}{p_{\mathrm{a}}\mu\sqrt{n}}\left(\widehat{L} + \frac{L_{\max}}{\sqrt{B}}\right) + \frac{\sqrt{m}}{p_{\mathrm{a}}\mu\sqrt{nB}}\left(\widehat{L} + \frac{L_{\max}}{\sqrt{B}}\right) \right),$$

communication rounds. The method DASHA-PAGE from (Tyurin and Richtárik, 2023), have to run

$$\widetilde{\mathcal{O}}\left( \omega + \frac{m}{B} + \frac{L}{\mu} + \frac{\omega}{\mu\sqrt{n}}\left(\widehat{L} + \frac{L_{\max}}{\sqrt{B}}\right) + \frac{\sqrt{m}}{\mu\sqrt{nB}}\left(\frac{L_{\max}}{\sqrt{B}}\right) \right),$$

communication rounds to get $\varepsilon$-solution. We can guarantee the degeneration up to $1/p_{\mathrm{a}}$ factor due to the partial participation only if $B = \mathcal{O}\left(\frac{L_{\max}^2}{\widehat{L}^2}\right)$. The same conclusion we have in Section 6.2.

### F.3 Stochastic Setting

**Theorem 10.** *Suppose that Assumption 1, 2, 3, 7, 5, 6, 8 and 9 hold. Let us take* $a = \frac{p_{\mathrm{a}}}{2\omega + 1}$, $b \in \left(0, \frac{p_{\mathrm{a}}}{2 - p_{\mathrm{a}}}\right]$,

$$\gamma \leq \min\left\{\left(L + \sqrt{\frac{200\omega(2\omega+1)}{np_{\mathrm{a}}^2}\left(\frac{(1-b)^2 L_\sigma^2}{B} + \widehat{L}^2\right) + \frac{40}{np_{\mathrm{a}}b}\left(\frac{(1-b)^2 L_\sigma^2}{B} + \left(1 - \frac{p_{\mathrm{aa}}}{p_{\mathrm{a}}}\right)\widehat{L}^2\right)}\right)^{-1}, \frac{a}{2\mu}, \frac{b}{2\mu}\right\},$$

*and* $h_i^0 = g_i^0$ *for all* $i \in [n]$ *in Algorithm 1* (DASHA-PP-MVR), *then*
$$\mathrm{E}\left[f(x^T) - f^*\right]$$
$$\leq (1 - \gamma\mu)^T\left(\Delta_0 + \frac{2\gamma}{b}\left\|h^0 - \nabla f(x^0)\right\|^2 + \left(\frac{40\gamma b\omega(2\omega+1)}{np_{\mathrm{a}}^2} + \frac{8\gamma(p_{\mathrm{a}} - p_{\mathrm{aa}})}{np_{\mathrm{a}}^2}\right)\frac{1}{n}\sum_{i=1}^n\left\|h_i^0 - \nabla f_i(x^0)\right\|^2\right)$$
$$+ \frac{1}{\mu}\left(\frac{100b^2\omega(2\omega+1)}{p_{\mathrm{a}}^2} + \frac{20b}{p_{\mathrm{a}}}\right)\frac{\sigma^2}{nB}.$$

The provided theorems states that to get $\varepsilon$-solution DASHA-PP have to run

$$\widetilde{\mathcal{O}}\left(\frac{\omega + 1}{p_{\mathrm{a}}} + \underbrace{\frac{\omega}{p_{\mathrm{a}}}\sqrt{\frac{\sigma^2}{\mu n\varepsilon B}}}_{\mathcal{P}_2} + \frac{\sigma^2}{p_{\mathrm{a}}\mu n\varepsilon B} + \frac{L}{\mu} + \frac{\omega}{p_{\mathrm{a}}\mu\sqrt{n}}\left(\widehat{L} + \frac{L_\sigma}{\sqrt{B}}\right) + \underbrace{\frac{\sigma}{p_{\mathrm{a}}n\mu^{3/2}\sqrt{\varepsilon B}}\left(\widehat{L} + \frac{L_\sigma}{\sqrt{B}}\right)}_{\mathcal{P}_1}\right)$$
(31)

communication rounds. We take $b = \Theta\left(\min\left\{\frac{p_{\mathrm{a}}}{\omega}\sqrt{\frac{\mu n\varepsilon B}{\sigma^2}}, \frac{p_{\mathrm{a}}\mu n\varepsilon B}{\sigma^2}\right\}\right) \geq \Theta\left(\min\left\{\frac{p_{\mathrm{a}}}{\omega^2}, \frac{p_{\mathrm{a}}\mu n\varepsilon B}{\sigma^2}\right\}\right)$.

The method DASHA-SYNC-MVR from (Tyurin and Richtárik, 2023), have to run
$$\widetilde{\mathcal{O}}\left(\omega + \frac{\sigma^2}{\mu n\varepsilon B} + \frac{L}{\mu} + \frac{\omega}{\mu\sqrt{n}}\left(\widehat{L} + \frac{L_\sigma}{\sqrt{B}}\right) + \frac{\sigma}{n\mu^{3/2}\sqrt{\varepsilon B}}\left(\frac{L_\sigma}{\sqrt{B}}\right)\right)$$
(32)

communication rounds to get $\varepsilon$-solution[9].

In the stochastic setting, the comparison is a little bit more complicated. As in the finite-sum setting, we have to take $B = \mathcal{O}\left(\frac{L_\sigma^2}{\widehat{L}^2}\right)$ to guarantee the degeneration up to $1/p_{\mathrm{a}}$ of the term $\mathcal{P}_1$ from (31). However, DASHA-PP-MVR has also suboptimal term $\mathcal{P}_2$. This suboptimality is tightly connected with the suboptimality of $B_{\mathrm{init}}$ in the general nonconvex case, which we discuss in Section 6.3, and it also appears in the analysis of DASHA-MVR (Tyurin and Richtárik, 2023). Let us provide the counterpart of Corollary 4. The corollary reveals that we can escape regimes when $\mathcal{P}_2$ is the bottleneck by choosing the parameters of the compressors.

**Corollary 5.** *Suppose that assumptions of Theorem 10 hold, batch size* $B \leq \min\left\{\frac{\sigma}{p_{\mathrm{a}}\sqrt{\mu\varepsilon}n}, \frac{L_\sigma^2}{\widehat{L}^2}\right\}$, *we take RandK compressors with* $K = \Theta\left(\frac{Bd\sqrt{\mu\varepsilon n}}{\sigma}\right)$. *Then the communication complexity equals*
$$\widetilde{\mathcal{O}}\left(\frac{d\sigma}{p_{\mathrm{a}}\sqrt{\mu\varepsilon n}} + \frac{dL_\sigma}{p_{\mathrm{a}}\mu\sqrt{n}}\right),$$
*and the expected number of stochastic gradient calculations per node equals*
$$\widetilde{\mathcal{O}}\left(\frac{\sigma^2}{p_{\mathrm{a}}\mu n\varepsilon} + \frac{\sigma L_\sigma}{p_{\mathrm{a}}n\mu^{3/2}\sqrt{\varepsilon}}\right).$$

Up to Lipschitz constants, DASHA-PP-MVR has the state-of-the-art oracle complexity under PŁ-condition (see (Li et al., 2021a)). Moreover, DASHA-PP-MVR has the state-of-the-art communication complexity of DASHA for a small enough $\mu$.

---

[9]For simplicity, we omitted $\frac{d}{\zeta_{\mathcal{C}}}$ term from the complexity in the stochastic setting, where $\zeta_{\mathcal{C}}$ is defined in Definition 12. For instance, for the RandK compressor (see Definition 5 and Theorem 6), $\zeta_{\mathcal{C}} = K$ and $\frac{d}{\zeta_{\mathcal{C}}} = \Theta(\omega)$.

## F.4 Proofs of Theorems

The following proofs almost repeat the proofs from Section E. And one of the main changes is that instead of Lemma 3, we use the following lemma.

### F.4.1 Standard Lemma under Polyak-Łojasiewicz Condition

**Lemma 11.** *Suppose that Assumptions 1 and 9 hold and*

$$\mathrm{E}\left[f(x^{t+1})\right] + \gamma \Psi^{t+1} \leq \mathrm{E}\left[f(x^t)\right] - \frac{\gamma}{2}\mathrm{E}\left[\left\|\nabla f(x^t)\right\|^2\right] + (1 - \gamma\mu)\gamma\Psi^t + \gamma C,$$

*where $\Psi^t$ is a sequence of numbers, $\Psi^t \geq 0$ for all $t \in [T]$, constant $C \geq 0$, constant $\mu > 0$, and constant $\gamma \in (0, 1/\mu)$. Then*

$$\mathrm{E}\left[f(x^T) - f^*\right] \leq (1 - \gamma\mu)^T\left(\left(f(x^0) - f^*\right) + \gamma\Psi^0\right) + \frac{C}{\mu}. \tag{33}$$

*Proof.* We subtract $f^*$ and use PŁ-condition (30) to get

$$
\begin{aligned}
\mathrm{E}\left[f(x^{t+1}) - f^*\right] + \gamma\Psi^{t+1} &\leq \mathrm{E}\left[f(x^t) - f^*\right] - \frac{\gamma}{2}\mathrm{E}\left[\left\|\nabla f(x^t)\right\|^2\right] + \gamma\Psi^t + \gamma C \\
&\leq (1 - \gamma\mu)\mathrm{E}\left[f(x^t) - f^*\right] + (1 - \gamma\mu)\gamma\Psi^t + \gamma C \\
&= (1 - \gamma\mu)\left(\mathrm{E}\left[f(x^t) - f^*\right] + \gamma\Psi^t\right) + \gamma C.
\end{aligned}
$$

Unrolling the inequality, we have

$$
\begin{aligned}
\mathrm{E}\left[f(x^{t+1}) - f^*\right] + \gamma\Psi^{t+1} &\leq (1 - \gamma\mu)^{t+1}\left(\left(f(x^0) - f^*\right) + \gamma\Psi^0\right) + \gamma C\sum_{i=0}^{t}(1 - \gamma\mu)^i \\
&\leq (1 - \gamma\mu)^{t+1}\left(\left(f(x^0) - f^*\right) + \gamma\Psi^0\right) + \frac{C}{\mu}.
\end{aligned}
$$

It is left to note that $\Psi^t \geq 0$ for all $t \in [T]$. □

### F.4.2 Generic Lemma

We now provide the counterpart of Lemma 6.

**Lemma 12.** *Suppose that Assumptions 2, 7, 8 and 9 hold and let us take $a = \frac{p_{\mathrm{a}}}{2\omega+1}$, then*

$$
\begin{aligned}
&\mathrm{E}\left[f(x^{t+1})\right] + \frac{2\gamma(2\omega + 1)}{p_{\mathrm{a}}}\mathrm{E}\left[\left\|g^{t+1} - h^{t+1}\right\|^2\right] + \frac{4\gamma((2\omega + 1)p_{\mathrm{a}} - p_{\mathrm{aa}})}{np_{\mathrm{a}}^2}\mathrm{E}\left[\frac{1}{n}\sum_{i=1}^{n}\left\|g_i^{t+1} - h_i^{t+1}\right\|^2\right] \\
&\leq \mathrm{E}\left[f(x^t) - \frac{\gamma}{2}\left\|\nabla f(x^t)\right\|^2 - \left(\frac{1}{2\gamma} - \frac{L}{2}\right)\left\|x^{t+1} - x^t\right\|^2 + \gamma\left\|h^t - \nabla f(x^t)\right\|^2\right] \\
&\quad + (1 - \gamma\mu)\frac{2\gamma(2\omega + 1)}{p_{\mathrm{a}}}\mathrm{E}\left[\left\|g^t - h^t\right\|^2\right] + (1 - \gamma\mu)\frac{4\gamma((2\omega + 1)p_{\mathrm{a}} - p_{\mathrm{aa}})}{np_{\mathrm{a}}^2}\mathrm{E}\left[\frac{1}{n}\sum_{i=1}^{n}\left\|g_i^t - h_i^t\right\|^2\right] \\
&\quad + \frac{10\gamma(2\omega + 1)\omega}{np_{\mathrm{a}}^2}\mathrm{E}\left[\frac{1}{n}\sum_{i=1}^{n}\left\|k_i^{t+1}\right\|^2\right].
\end{aligned}
$$

*Proof.* Let us fix some constants $\kappa, \eta \in [0, \infty)$ that we will define later. Using the same reasoning as in Lemma 6, we can get

$$
\begin{aligned}
&\mathrm{E}\left[f(x^{t+1})\right] \\
&\quad + \kappa\mathrm{E}\left[\left\|g^{t+1} - h^{t+1}\right\|^2\right] + \eta\mathrm{E}\left[\frac{1}{n}\sum_{i=1}^{n}\left\|g_i^{t+1} - h_i^{t+1}\right\|^2\right] \\
&\leq \mathrm{E}\left[f(x^t) - \frac{\gamma}{2}\left\|\nabla f(x^t)\right\|^2 - \left(\frac{1}{2\gamma} - \frac{L}{2}\right)\left\|x^{t+1} - x^t\right\|^2 + \gamma\left\|h^t - \nabla f(x^t)\right\|^2\right]
\end{aligned}
$$

$$+ \left( \gamma + \kappa \left( 1 - a \right)^2 \right) \mathrm{E} \left[ \left\| g^t - h^t \right\|^2 \right]$$

$$+ \left( \frac{\kappa a^2 ((2\omega + 1) p_\mathrm{a} - p_\mathrm{aa})}{n p_\mathrm{a}^2} + \eta \left( \frac{a^2 (2\omega + 1 - p_\mathrm{a})}{p_\mathrm{a}} + (1 - a)^2 \right) \right) \mathrm{E} \left[ \frac{1}{n} \sum_{i=1}^n \left\| g_i^t - h_i^t \right\|^2 \right]$$

$$+ \left( \frac{2\kappa\omega}{n p_\mathrm{a}} + \frac{2\eta\omega}{p_\mathrm{a}} \right) \mathrm{E} \left[ \frac{1}{n} \sum_{i=1}^n \left\| k_i^{t+1} \right\|^2 \right].$$

Let us take $\kappa = \frac{2\gamma}{a}$. One can show that $\gamma + \kappa \left( 1 - a \right)^2 \le \left( 1 - \frac{a}{2} \right) \kappa$, and thus

$$\mathrm{E} \left[ f(x^{t+1}) \right]$$

$$+ \frac{2\gamma}{a} \mathrm{E} \left[ \left\| g^{t+1} - h^{t+1} \right\|^2 \right] + \eta \mathrm{E} \left[ \frac{1}{n} \sum_{i=1}^n \left\| g_i^{t+1} - h_i^{t+1} \right\|^2 \right]$$

$$\le \mathrm{E} \left[ f(x^t) - \frac{\gamma}{2} \left\| \nabla f(x^t) \right\|^2 - \left( \frac{1}{2\gamma} - \frac{L}{2} \right) \left\| x^{t+1} - x^t \right\|^2 + \gamma \left\| h^t - \nabla f(x^t) \right\|^2 \right]$$

$$+ \left( 1 - \frac{a}{2} \right) \frac{2\gamma}{a} \mathrm{E} \left[ \left\| g^t - h^t \right\|^2 \right]$$

$$+ \left( \frac{2\gamma a((2\omega + 1) p_\mathrm{a} - p_\mathrm{aa})}{n p_\mathrm{a}^2} + \eta \left( \frac{a^2 (2\omega + 1 - p_\mathrm{a})}{p_\mathrm{a}} + (1 - a)^2 \right) \right) \mathrm{E} \left[ \frac{1}{n} \sum_{i=1}^n \left\| g_i^t - h_i^t \right\|^2 \right]$$

$$+ \left( \frac{4\gamma\omega}{a n p_\mathrm{a}} + \frac{2\eta\omega}{p_\mathrm{a}} \right) \mathrm{E} \left[ \frac{1}{n} \sum_{i=1}^n \left\| k_i^{t+1} \right\|^2 \right].$$

Considering the choice of $a$, one can show that $\left( \frac{a^2 (2\omega + 1 - p_\mathrm{a})}{p_\mathrm{a}} + (1 - a)^2 \right) \le 1 - a$. If we take $\eta = \frac{4\gamma((2\omega + 1) p_\mathrm{a} - p_\mathrm{aa})}{n p_\mathrm{a}^2}$, then $\left( \frac{2\gamma a((2\omega + 1) p_\mathrm{a} - p_\mathrm{aa})}{n p_\mathrm{a}^2} + \eta \left( \frac{a^2 (2\omega + 1 - p_\mathrm{a})}{p_\mathrm{a}} + (1 - a)^2 \right) \right) \le \left( 1 - \frac{a}{2} \right) \eta$ and

$$\mathrm{E} \left[ f(x^{t+1}) \right]$$

$$+ \frac{2\gamma(2\omega + 1)}{p_\mathrm{a}} \mathrm{E} \left[ \left\| g^{t+1} - h^{t+1} \right\|^2 \right] + \frac{4\gamma((2\omega + 1) p_\mathrm{a} - p_\mathrm{aa})}{n p_\mathrm{a}^2} \mathrm{E} \left[ \frac{1}{n} \sum_{i=1}^n \left\| g_i^{t+1} - h_i^{t+1} \right\|^2 \right]$$

$$\le \mathrm{E} \left[ f(x^t) - \frac{\gamma}{2} \left\| \nabla f(x^t) \right\|^2 - \left( \frac{1}{2\gamma} - \frac{L}{2} \right) \left\| x^{t+1} - x^t \right\|^2 + \gamma \left\| h^t - \nabla f(x^t) \right\|^2 \right]$$

$$+ \left( 1 - \frac{a}{2} \right) \frac{2\gamma(2\omega + 1)}{p_\mathrm{a}} \mathrm{E} \left[ \left\| g^t - h^t \right\|^2 \right] + \left( 1 - \frac{a}{2} \right) \frac{4\gamma((2\omega + 1) p_\mathrm{a} - p_\mathrm{aa})}{n p_\mathrm{a}^2} \mathrm{E} \left[ \frac{1}{n} \sum_{i=1}^n \left\| g_i^t - h_i^t \right\|^2 \right]$$

$$+ \left( \frac{2\gamma(2\omega + 1)\omega}{n p_\mathrm{a}^2} + \frac{8\gamma((2\omega + 1) p_\mathrm{a} - p_\mathrm{aa})\omega}{n p_\mathrm{a}^3} \right) \mathrm{E} \left[ \frac{1}{n} \sum_{i=1}^n \left\| k_i^{t+1} \right\|^2 \right]$$

$$\le \mathrm{E} \left[ f(x^t) - \frac{\gamma}{2} \left\| \nabla f(x^t) \right\|^2 - \left( \frac{1}{2\gamma} - \frac{L}{2} \right) \left\| x^{t+1} - x^t \right\|^2 + \gamma \left\| h^t - \nabla f(x^t) \right\|^2 \right]$$

$$+ \left( 1 - \frac{a}{2} \right) \frac{2\gamma(2\omega + 1)}{p_\mathrm{a}} \mathrm{E} \left[ \left\| g^t - h^t \right\|^2 \right] + \left( 1 - \frac{a}{2} \right) \frac{4\gamma((2\omega + 1) p_\mathrm{a} - p_\mathrm{aa})}{n p_\mathrm{a}^2} \mathrm{E} \left[ \frac{1}{n} \sum_{i=1}^n \left\| g_i^t - h_i^t \right\|^2 \right]$$

$$+ \frac{10\gamma(2\omega + 1)\omega}{n p_\mathrm{a}^2} \mathrm{E} \left[ \frac{1}{n} \sum_{i=1}^n \left\| k_i^{t+1} \right\|^2 \right].$$

It it left to consider that $\gamma \le \frac{a}{2\mu}$, and therefore $1 - \frac{a}{2} \le 1 - \gamma\mu$. $\qquad \square$

### F.4.3  Proof for DASHA-PP under PŁ-condition

**Theorem 8.** *Suppose that Assumption 1, 2, 3, 7, 8 and 9 hold. Let us take $a = \frac{p_{\mathrm{a}}}{2\omega+1}$, $b = \frac{p_{\mathrm{a}}}{2-p_{\mathrm{a}}}$,*

$$\gamma \leq \min\left\{ \left(L + \sqrt{\frac{200\omega(2\omega+1)}{np_{\mathrm{a}}^2} + \frac{48}{np_{\mathrm{a}}^2}\left(1 - \frac{p_{\mathrm{aa}}}{p_{\mathrm{a}}}\right)\widehat{L}}\right)^{-1}, \frac{a}{4\mu}\right\},$$

*and $h_i^0 = g_i^0 = \nabla f_i(x^0)$ for all $i \in [n]$ in Algorithm 1 (DASHA-PP), then $\mathrm{E}\left[f(x^T)\right] - f^* \leq (1 - \gamma\mu)^T \Delta_0$.*

*Proof.* Let us fix constants $\nu, \rho \in [0, \infty)$ that we will define later. Considering Lemma 12, Lemma 7, and the law of total expectation, we obtain

$$\mathrm{E}\left[f(x^{t+1})\right] + \frac{2\gamma(2\omega+1)}{p_{\mathrm{a}}}\mathrm{E}\left[\left\|g^{t+1} - h^{t+1}\right\|^2\right] + \frac{4\gamma((2\omega+1)p_{\mathrm{a}} - p_{\mathrm{aa}})}{np_{\mathrm{a}}^2}\mathrm{E}\left[\frac{1}{n}\sum_{i=1}^n \left\|g_i^{t+1} - h_i^{t+1}\right\|^2\right]$$

$$+ \nu\mathrm{E}\left[\left\|h^{t+1} - \nabla f(x^{t+1})\right\|^2\right] + \rho\mathrm{E}\left[\frac{1}{n}\sum_{i=1}^n \left\|h_i^{t+1} - \nabla f_i(x^{t+1})\right\|^2\right]$$

$$\leq \mathrm{E}\left[f(x^t) - \frac{\gamma}{2}\left\|\nabla f(x^t)\right\|^2 - \left(\frac{1}{2\gamma} - \frac{L}{2}\right)\left\|x^{t+1} - x^t\right\|^2 + \gamma\left\|h^t - \nabla f(x^t)\right\|^2\right]$$

$$+ (1 - \gamma\mu)\frac{2\gamma(2\omega+1)}{p_{\mathrm{a}}}\mathrm{E}\left[\left\|g^t - h^t\right\|^2\right] + (1 - \gamma\mu)\frac{4\gamma((2\omega+1)p_{\mathrm{a}} - p_{\mathrm{aa}})}{np_{\mathrm{a}}^2}\mathrm{E}\left[\frac{1}{n}\sum_{i=1}^n \left\|g_i^t - h_i^t\right\|^2\right]$$

$$+ \frac{10\gamma\omega(2\omega+1)}{np_{\mathrm{a}}^2}\mathrm{E}\left[2\widehat{L}^2\left\|x^{t+1} - x^t\right\|^2 + 2b^2\frac{1}{n}\sum_{i=1}^n \left\|h_i^t - \nabla f_i(x^t)\right\|^2\right]$$

$$+ \nu\mathrm{E}\left[\frac{2(p_{\mathrm{a}} - p_{\mathrm{aa}})\widehat{L}^2}{np_{\mathrm{a}}^2}\left\|x^{t+1} - x^t\right\|^2 + \frac{2b^2(p_{\mathrm{a}} - p_{\mathrm{aa}})}{n^2p_{\mathrm{a}}^2}\sum_{i=1}^n \left\|h_i^t - \nabla f_i(x^t)\right\|^2 + (1-b)^2\left\|h^t - \nabla f(x^t)\right\|^2\right]$$

$$+ \rho\mathrm{E}\left[\frac{2(1-p_{\mathrm{a}})}{p_{\mathrm{a}}}\widehat{L}^2\left\|x^{t+1} - x^t\right\|^2 + \left(\frac{2b^2(1-p_{\mathrm{a}})}{p_{\mathrm{a}}} + (1-b)^2\right)\frac{1}{n}\sum_{i=1}^n \left\|h_i^t - \nabla f_i(x^t)\right\|^2\right].$$

After rearranging the terms, we get

$$\mathrm{E}\left[f(x^{t+1})\right] + \frac{2\gamma(2\omega+1)}{p_{\mathrm{a}}}\mathrm{E}\left[\left\|g^{t+1} - h^{t+1}\right\|^2\right] + \frac{4\gamma((2\omega+1)p_{\mathrm{a}} - p_{\mathrm{aa}})}{np_{\mathrm{a}}^2}\mathrm{E}\left[\frac{1}{n}\sum_{i=1}^n \left\|g_i^{t+1} - h_i^{t+1}\right\|^2\right]$$

$$+ \nu\mathrm{E}\left[\left\|h^{t+1} - \nabla f(x^{t+1})\right\|^2\right] + \rho\mathrm{E}\left[\frac{1}{n}\sum_{i=1}^n \left\|h_i^{t+1} - \nabla f_i(x^{t+1})\right\|^2\right]$$

$$\leq \mathrm{E}\left[f(x^t)\right] - \frac{\gamma}{2}\mathrm{E}\left[\left\|\nabla f(x^t)\right\|^2\right]$$

$$+ (1 - \gamma\mu)\frac{2\gamma(2\omega+1)}{p_{\mathrm{a}}}\mathrm{E}\left[\left\|g^t - h^t\right\|^2\right] + (1 - \gamma\mu)\frac{4\gamma((2\omega+1)p_{\mathrm{a}} - p_{\mathrm{aa}})}{np_{\mathrm{a}}^2}\mathrm{E}\left[\frac{1}{n}\sum_{i=1}^n \left\|g_i^t - h_i^t\right\|^2\right]$$

$$- \left(\frac{1}{2\gamma} - \frac{L}{2} - \frac{20\gamma\omega(2\omega+1)\widehat{L}^2}{np_{\mathrm{a}}^2} - \nu\frac{2(p_{\mathrm{a}} - p_{\mathrm{aa}})\widehat{L}^2}{np_{\mathrm{a}}^2} - \rho\frac{2(1-p_{\mathrm{a}})\widehat{L}^2}{p_{\mathrm{a}}}\right)\mathrm{E}\left[\left\|x^{t+1} - x^t\right\|^2\right]$$

$$+ \left(\gamma + \nu(1-b)^2\right)\mathrm{E}\left[\left\|h^t - \nabla f(x^t)\right\|^2\right]$$

$$+ \left(\frac{20b^2\gamma\omega(2\omega+1)}{np_{\mathrm{a}}^2} + \nu\frac{2b^2(p_{\mathrm{a}} - p_{\mathrm{aa}})}{np_{\mathrm{a}}^2} + \rho\left(\frac{2b^2(1-p_{\mathrm{a}})}{p_{\mathrm{a}}} + (1-b)^2\right)\right)\mathrm{E}\left[\frac{1}{n}\sum_{i=1}^n \left\|h_i^t - \nabla f_i(x^t)\right\|^2\right].$$

By taking $\nu = \frac{2\gamma}{b}$, one can show that $\left(\gamma + \nu(1-b)^2\right) \leq \left(1 - \frac{b}{2}\right)\nu$, and

$$\mathrm{E}\left[f(x^{t+1})\right] + \frac{2\gamma(2\omega+1)}{p_{\mathrm{a}}}\mathrm{E}\left[\left\|g^{t+1} - h^{t+1}\right\|^2\right] + \frac{4\gamma((2\omega+1)p_{\mathrm{a}} - p_{\mathrm{aa}})}{np_{\mathrm{a}}^2}\mathrm{E}\left[\frac{1}{n}\sum_{i=1}^n \left\|g_i^{t+1} - h_i^{t+1}\right\|^2\right]$$

$$
+ \frac{2\gamma}{b} \mathrm{E}\left[\left\|h^{t+1} - \nabla f(x^{t+1})\right\|^2\right] + \rho \mathrm{E}\left[\frac{1}{n}\sum_{i=1}^{n}\left\|h_i^{t+1} - \nabla f_i(x^{t+1})\right\|^2\right]
$$

$$
\leq \mathrm{E}\left[f(x^t)\right] - \frac{\gamma}{2}\mathrm{E}\left[\left\|\nabla f(x^t)\right\|^2\right]
$$

$$
+ (1 - \gamma\mu)\frac{2\gamma(2\omega + 1)}{p_{\mathrm{a}}}\mathrm{E}\left[\left\|g^t - h^t\right\|^2\right] + (1 - \gamma\mu)\frac{4\gamma((2\omega + 1)\,p_{\mathrm{a}} - p_{\mathrm{aa}})}{np_{\mathrm{a}}^2}\mathrm{E}\left[\frac{1}{n}\sum_{i=1}^{n}\left\|g_i^t - h_i^t\right\|^2\right]
$$

$$
- \left(\frac{1}{2\gamma} - \frac{L}{2} - \frac{20\gamma\omega\,(2\omega + 1)\,\widehat{L}^2}{np_{\mathrm{a}}^2} - \frac{4\gamma\,(p_{\mathrm{a}} - p_{\mathrm{aa}})\,\widehat{L}^2}{bnp_{\mathrm{a}}^2} - \rho\frac{2(1 - p_{\mathrm{a}})\widehat{L}^2}{p_{\mathrm{a}}}\right)\mathrm{E}\left[\left\|x^{t+1} - x^t\right\|^2\right]
$$

$$
+ \left(1 - \frac{b}{2}\right)\frac{2\gamma}{b}\mathrm{E}\left[\left\|h^t - \nabla f(x^t)\right\|^2\right]
$$

$$
+ \left(\frac{20b^2\gamma\omega(2\omega + 1)}{np_{\mathrm{a}}^2} + \frac{4\gamma b\,(p_{\mathrm{a}} - p_{\mathrm{aa}})}{np_{\mathrm{a}}^2} + \rho\left(\frac{2b^2(1 - p_{\mathrm{a}})}{p_{\mathrm{a}}} + (1 - b)^2\right)\right)\mathrm{E}\left[\frac{1}{n}\sum_{i=1}^{n}\left\|h_i^t - \nabla f_i(x^t)\right\|^2\right].
$$

Note that $b = \frac{p_{\mathrm{a}}}{2 - p_{\mathrm{a}}}$, thus

$$
\left(\frac{20b^2\gamma\omega(2\omega + 1)}{np_{\mathrm{a}}^2} + \frac{4\gamma b\,(p_{\mathrm{a}} - p_{\mathrm{aa}})}{np_{\mathrm{a}}^2} + \rho\left(\frac{2b^2(1 - p_{\mathrm{a}})}{p_{\mathrm{a}}} + (1 - b)^2\right)\right)
$$

$$
\leq \left(\frac{20b^2\gamma\omega(2\omega + 1)}{np_{\mathrm{a}}^2} + \frac{4\gamma b\,(p_{\mathrm{a}} - p_{\mathrm{aa}})}{np_{\mathrm{a}}^2} + \rho\,(1 - b)\right).
$$

And if we take $\rho = \frac{40b\gamma\omega(2\omega+1)}{np_{\mathrm{a}}^2} + \frac{8\gamma(p_{\mathrm{a}}-p_{\mathrm{aa}})}{np_{\mathrm{a}}^2}$, then

$$
\left(\frac{20b^2\gamma\omega(2\omega + 1)}{np_{\mathrm{a}}^2} + \frac{4\gamma b\,(p_{\mathrm{a}} - p_{\mathrm{aa}})}{np_{\mathrm{a}}^2} + \rho\,(1 - b)\right) \leq \left(1 - \frac{b}{2}\right)\rho,
$$

and

$$
\mathrm{E}\left[f(x^{t+1})\right] + \frac{2\gamma(2\omega + 1)}{p_{\mathrm{a}}}\mathrm{E}\left[\left\|g^{t+1} - h^{t+1}\right\|^2\right] + \frac{4\gamma((2\omega + 1)\,p_{\mathrm{a}} - p_{\mathrm{aa}})}{np_{\mathrm{a}}^2}\mathrm{E}\left[\frac{1}{n}\sum_{i=1}^{n}\left\|g_i^{t+1} - h_i^{t+1}\right\|^2\right]
$$

$$
+ \frac{2\gamma}{b}\mathrm{E}\left[\left\|h^{t+1} - \nabla f(x^{t+1})\right\|^2\right] + \left(\frac{40b\gamma\omega(2\omega + 1)}{np_{\mathrm{a}}^2} + \frac{8\gamma\,(p_{\mathrm{a}} - p_{\mathrm{aa}})}{np_{\mathrm{a}}^2}\right)\mathrm{E}\left[\frac{1}{n}\sum_{i=1}^{n}\left\|h_i^{t+1} - \nabla f_i(x^{t+1})\right\|^2\right]
$$

$$
\leq \mathrm{E}\left[f(x^t)\right] - \frac{\gamma}{2}\mathrm{E}\left[\left\|\nabla f(x^t)\right\|^2\right]
$$

$$
+ (1 - \gamma\mu)\frac{2\gamma(2\omega + 1)}{p_{\mathrm{a}}}\mathrm{E}\left[\left\|g^t - h^t\right\|^2\right] + (1 - \gamma\mu)\frac{4\gamma((2\omega + 1)\,p_{\mathrm{a}} - p_{\mathrm{aa}})}{np_{\mathrm{a}}^2}\mathrm{E}\left[\frac{1}{n}\sum_{i=1}^{n}\left\|g_i^t - h_i^t\right\|^2\right]
$$

$$
- \left(\frac{1}{2\gamma} - \frac{L}{2} - \frac{20\gamma\omega\,(2\omega + 1)\,\widehat{L}^2}{np_{\mathrm{a}}^2} - \frac{4\gamma\,(p_{\mathrm{a}} - p_{\mathrm{aa}})\,\widehat{L}^2}{bnp_{\mathrm{a}}^2}\right.
$$

$$
\left. - \frac{80b\gamma\omega(2\omega + 1)(1 - p_{\mathrm{a}})\widehat{L}^2}{np_{\mathrm{a}}^3} - \frac{16\gamma\,(p_{\mathrm{a}} - p_{\mathrm{aa}})\,(1 - p_{\mathrm{a}})\widehat{L}^2}{np_{\mathrm{a}}^3}\right)\mathrm{E}\left[\left\|x^{t+1} - x^t\right\|^2\right]
$$

$$
+ \left(1 - \frac{b}{2}\right)\frac{2\gamma}{b}\mathrm{E}\left[\left\|h^t - \nabla f(x^t)\right\|^2\right] + \left(1 - \frac{b}{2}\right)\left(\frac{40b\gamma\omega(2\omega + 1)}{np_{\mathrm{a}}^2} + \frac{8\gamma\,(p_{\mathrm{a}} - p_{\mathrm{aa}})}{np_{\mathrm{a}}^2}\right)\mathrm{E}\left[\frac{1}{n}\sum_{i=1}^{n}\left\|h_i^t - \nabla f_i(x^t)\right\|^2\right].
$$

Due to $\frac{p_{\mathrm{a}}}{2} \leq b \leq p_{\mathrm{a}}$, we have

$$
\mathrm{E}\left[f(x^{t+1})\right] + \frac{2\gamma(2\omega + 1)}{p_{\mathrm{a}}}\mathrm{E}\left[\left\|g^{t+1} - h^{t+1}\right\|^2\right] + \frac{4\gamma((2\omega + 1)\,p_{\mathrm{a}} - p_{\mathrm{aa}})}{np_{\mathrm{a}}^2}\mathrm{E}\left[\frac{1}{n}\sum_{i=1}^{n}\left\|g_i^{t+1} - h_i^{t+1}\right\|^2\right]
$$

$$
+ \frac{2\gamma}{b}\mathrm{E}\left[\left\|h^{t+1} - \nabla f(x^{t+1})\right\|^2\right] + \left(\frac{40b\gamma\omega(2\omega + 1)}{np_{\mathrm{a}}^2} + \frac{8\gamma\,(p_{\mathrm{a}} - p_{\mathrm{aa}})}{np_{\mathrm{a}}^2}\right)\mathrm{E}\left[\frac{1}{n}\sum_{i=1}^{n}\left\|h_i^{t+1} - \nabla f_i(x^{t+1})\right\|^2\right]
$$

$$\leq \mathrm{E}\left[f(x^t)\right] - \frac{\gamma}{2}\mathrm{E}\left[\|\nabla f(x^t)\|^2\right]$$

$$+ (1-\gamma\mu)\frac{2\gamma(2\omega+1)}{p_{\mathrm{a}}}\mathrm{E}\left[\|g^t - h^t\|^2\right] + (1-\gamma\mu)\frac{4\gamma((2\omega+1)\,p_{\mathrm{a}} - p_{\mathrm{aa}})}{np_{\mathrm{a}}^2}\mathrm{E}\left[\frac{1}{n}\sum_{i=1}^{n}\|g_i^t - h_i^t\|^2\right]$$

$$- \left(\frac{1}{2\gamma} - \frac{L}{2} - \frac{100\gamma\omega\,(2\omega+1)\,\widehat{L}^2}{np_{\mathrm{a}}^2} - \frac{24\gamma\,(p_{\mathrm{a}} - p_{\mathrm{aa}})\,\widehat{L}^2}{np_{\mathrm{a}}^3}\right)\mathrm{E}\left[\|x^{t+1} - x^t\|^2\right]$$

$$+ \left(1 - \frac{b}{2}\right)\frac{2\gamma}{b}\mathrm{E}\left[\|h^t - \nabla f(x^t)\|^2\right] + \left(1 - \frac{b}{2}\right)\left(\frac{40b\gamma\omega(2\omega+1)}{np_{\mathrm{a}}^2} + \frac{8\gamma\,(p_{\mathrm{a}} - p_{\mathrm{aa}})}{np_{\mathrm{a}}^2}\right)\mathrm{E}\left[\frac{1}{n}\sum_{i=1}^{n}\|h_i^t - \nabla f_i(x^t)\|^2\right].$$

Using Lemma 4 and the assumption about $\gamma$, we get

$$\mathrm{E}\left[f(x^{t+1})\right] + \frac{2\gamma(2\omega+1)}{p_{\mathrm{a}}}\mathrm{E}\left[\|g^{t+1} - h^{t+1}\|^2\right] + \frac{4\gamma((2\omega+1)\,p_{\mathrm{a}} - p_{\mathrm{aa}})}{np_{\mathrm{a}}^2}\mathrm{E}\left[\frac{1}{n}\sum_{i=1}^{n}\|g_i^{t+1} - h_i^{t+1}\|^2\right]$$

$$+ \frac{2\gamma}{b}\mathrm{E}\left[\|h^{t+1} - \nabla f(x^{t+1})\|^2\right] + \left(\frac{40b\gamma\omega(2\omega+1)}{np_{\mathrm{a}}^2} + \frac{8\gamma\,(p_{\mathrm{a}} - p_{\mathrm{aa}})}{np_{\mathrm{a}}^2}\right)\mathrm{E}\left[\frac{1}{n}\sum_{i=1}^{n}\|h_i^{t+1} - \nabla f_i(x^{t+1})\|^2\right]$$

$$\leq \mathrm{E}\left[f(x^t)\right] - \frac{\gamma}{2}\mathrm{E}\left[\|\nabla f(x^t)\|^2\right]$$

$$+ (1-\gamma\mu)\frac{2\gamma(2\omega+1)}{p_{\mathrm{a}}}\mathrm{E}\left[\|g^t - h^t\|^2\right] + (1-\gamma\mu)\frac{4\gamma((2\omega+1)\,p_{\mathrm{a}} - p_{\mathrm{aa}})}{np_{\mathrm{a}}^2}\mathrm{E}\left[\frac{1}{n}\sum_{i=1}^{n}\|g_i^t - h_i^t\|^2\right]$$

$$+ \left(1 - \frac{b}{2}\right)\frac{2\gamma}{b}\mathrm{E}\left[\|h^t - \nabla f(x^t)\|^2\right] + \left(1 - \frac{b}{2}\right)\left(\frac{40b\gamma\omega(2\omega+1)}{np_{\mathrm{a}}^2} + \frac{8\gamma\,(p_{\mathrm{a}} - p_{\mathrm{aa}})}{np_{\mathrm{a}}^2}\right)\mathrm{E}\left[\frac{1}{n}\sum_{i=1}^{n}\|h_i^t - \nabla f_i(x^t)\|^2\right].$$

Note that $\gamma \leq \frac{a}{4\mu} \leq \frac{p_{\mathrm{a}}}{4\mu} \leq \frac{b}{2\mu}$, thus $1 - \frac{b}{2} \leq 1 - \gamma\mu$ and

$$\mathrm{E}\left[f(x^{t+1})\right] + \frac{2\gamma(2\omega+1)}{p_{\mathrm{a}}}\mathrm{E}\left[\|g^{t+1} - h^{t+1}\|^2\right] + \frac{4\gamma((2\omega+1)\,p_{\mathrm{a}} - p_{\mathrm{aa}})}{np_{\mathrm{a}}^2}\mathrm{E}\left[\frac{1}{n}\sum_{i=1}^{n}\|g_i^{t+1} - h_i^{t+1}\|^2\right]$$

$$+ \frac{2\gamma}{b}\mathrm{E}\left[\|h^{t+1} - \nabla f(x^{t+1})\|^2\right] + \left(\frac{40b\gamma\omega(2\omega+1)}{np_{\mathrm{a}}^2} + \frac{8\gamma\,(p_{\mathrm{a}} - p_{\mathrm{aa}})}{np_{\mathrm{a}}^2}\right)\mathrm{E}\left[\frac{1}{n}\sum_{i=1}^{n}\|h_i^{t+1} - \nabla f_i(x^{t+1})\|^2\right]$$

$$\leq \mathrm{E}\left[f(x^t)\right] - \frac{\gamma}{2}\mathrm{E}\left[\|\nabla f(x^t)\|^2\right]$$

$$+ (1-\gamma\mu)\frac{2\gamma(2\omega+1)}{p_{\mathrm{a}}}\mathrm{E}\left[\|g^t - h^t\|^2\right] + (1-\gamma\mu)\frac{4\gamma((2\omega+1)\,p_{\mathrm{a}} - p_{\mathrm{aa}})}{np_{\mathrm{a}}^2}\mathrm{E}\left[\frac{1}{n}\sum_{i=1}^{n}\|g_i^t - h_i^t\|^2\right]$$

$$+ (1-\gamma\mu)\frac{2\gamma}{b}\mathrm{E}\left[\|h^t - \nabla f(x^t)\|^2\right] + (1-\gamma\mu)\left(\frac{40b\gamma\omega(2\omega+1)}{np_{\mathrm{a}}^2} + \frac{8\gamma\,(p_{\mathrm{a}} - p_{\mathrm{aa}})}{np_{\mathrm{a}}^2}\right)\mathrm{E}\left[\frac{1}{n}\sum_{i=1}^{n}\|h_i^t - \nabla f_i(x^t)\|^2\right].$$

In the view of Lemma 11 with

$$\Psi^t = \frac{2(2\omega+1)}{p_{\mathrm{a}}}\mathrm{E}\left[\|g^t - h^t\|^2\right] + \frac{4((2\omega+1)\,p_{\mathrm{a}} - p_{\mathrm{aa}})}{np_{\mathrm{a}}^2}\mathrm{E}\left[\frac{1}{n}\sum_{i=1}^{n}\|g_i^t - h_i^t\|^2\right]$$

$$+ \frac{2}{b}\mathrm{E}\left[\|h^t - \nabla f(x^t)\|^2\right] + \left(\frac{40b\omega(2\omega+1)}{np_{\mathrm{a}}^2} + \frac{8\gamma\,(p_{\mathrm{a}} - p_{\mathrm{aa}})}{np_{\mathrm{a}}^2}\right)\mathrm{E}\left[\frac{1}{n}\sum_{i=1}^{n}\|h_i^t - \nabla f_i(x^t)\|^2\right],$$

we can conclude the proof of the theorem. $\qquad\square$

### F.4.4  Proof for DASHA-PP-PAGE under PŁ-condition

**Theorem 9.** *Suppose that Assumption 1, 2, 3, 7, 4, 8, and 9 hold. Let us take* $a = \frac{p_{\mathrm{a}}}{2\omega+1}$, *probability* $p_{page} = \frac{B}{m+B}, b = \frac{p_{page}p_{\mathrm{a}}}{2 - p_{\mathrm{a}}}$,

$$\gamma \leq \min\left\{\left(L + \sqrt{\frac{200\omega(2\omega+1)}{np_{\mathrm{a}}^2}\left(\widehat{L}^2 + \frac{(1 - p_{page})L_{\max}^2}{B}\right) + \frac{48}{np_{\mathrm{a}}^2 p_{page}}\left(\left(1 - \frac{p_{\mathrm{aa}}}{p_{\mathrm{a}}}\right)\widehat{L}^2 + \frac{(1 - p_{page})L_{\max}^2}{B}\right)}\right)^{-1}, \frac{a}{2\mu}, \frac{b}{2\mu}\right\},$$

*and $h_i^0 = g_i^0 = \nabla f_i(x^0)$ for all $i \in [n]$ in Algorithm 1* (DASHA-PP-PAGE)*, then* $\mathrm{E}\left[f(x^T)\right] - f^* \le$
$(1 - \gamma\mu)^T \Delta_0.$

*Proof.* Let us fix constants $\nu, \rho \in [0, \infty)$ that we will define later. Considering Lemma 12, Lemma 8, and the law of total expectation, we obtain

$$\mathrm{E}\left[f(x^{t+1})\right] + \frac{2\gamma(2\omega+1)}{p_\mathrm{a}}\mathrm{E}\left[\left\|g^{t+1} - h^{t+1}\right\|^2\right] + \frac{4\gamma((2\omega+1)\,p_\mathrm{a} - p_\mathrm{aa})}{np_\mathrm{a}^2}\mathrm{E}\left[\frac{1}{n}\sum_{i=1}^n\left\|g_i^{t+1} - h_i^{t+1}\right\|^2\right]$$

$$+ \nu\mathrm{E}\left[\left\|h^{t+1} - \nabla f(x^{t+1})\right\|^2\right] + \rho\mathrm{E}\left[\frac{1}{n}\sum_{i=1}^n\left\|h_i^{t+1} - \nabla f_i(x^{t+1})\right\|^2\right]$$

$$\le \mathrm{E}\left[f(x^t) - \frac{\gamma}{2}\left\|\nabla f(x^t)\right\|^2 - \left(\frac{1}{2\gamma} - \frac{L}{2}\right)\left\|x^{t+1} - x^t\right\|^2 + \gamma\left\|h^t - \nabla f(x^t)\right\|^2\right]$$

$$+ (1 - \gamma\mu)\frac{2\gamma(2\omega+1)}{p_\mathrm{a}}\mathrm{E}\left[\left\|g^t - h^t\right\|^2\right] + (1 - \gamma\mu)\frac{4\gamma((2\omega+1)\,p_\mathrm{a} - p_\mathrm{aa})}{np_\mathrm{a}^2}\mathrm{E}\left[\frac{1}{n}\sum_{i=1}^n\left\|g_i^t - h_i^t\right\|^2\right]$$

$$+ \frac{10\gamma(2\omega+1)\omega}{np_\mathrm{a}^2}\mathrm{E}\left[\frac{1}{n}\sum_{i=1}^n\left\|k_i^{t+1}\right\|^2\right]$$

$$+ \nu\mathrm{E}\left[\left\|h^{t+1} - \nabla f(x^{t+1})\right\|^2\right] + \rho\mathrm{E}\left[\frac{1}{n}\sum_{i=1}^n\left\|h_i^{t+1} - \nabla f_i(x^{t+1})\right\|^2\right]$$

$$\le \mathrm{E}\left[f(x^t) - \frac{\gamma}{2}\left\|\nabla f(x^t)\right\|^2 - \left(\frac{1}{2\gamma} - \frac{L}{2}\right)\left\|x^{t+1} - x^t\right\|^2 + \gamma\left\|h^t - \nabla f(x^t)\right\|^2\right]$$

$$+ (1 - \gamma\mu)\frac{2\gamma(2\omega+1)}{p_\mathrm{a}}\mathrm{E}\left[\left\|g^t - h^t\right\|^2\right] + (1 - \gamma\mu)\frac{4\gamma((2\omega+1)\,p_\mathrm{a} - p_\mathrm{aa})}{np_\mathrm{a}^2}\mathrm{E}\left[\frac{1}{n}\sum_{i=1}^n\left\|g_i^t - h_i^t\right\|^2\right]$$

$$+ \frac{10\gamma(2\omega+1)\omega}{np_\mathrm{a}^2}\mathrm{E}\left[\left(2\widehat{L}^2 + \frac{(1 - p_\mathrm{page})L_\mathrm{max}^2}{B}\right)\left\|x^{t+1} - x^t\right\|^2 + \frac{2b^2}{p_\mathrm{page}}\frac{1}{n}\sum_{i=1}^n\left\|h_i^t - \nabla f_i(x^t)\right\|^2\right]$$

$$+ \nu\mathrm{E}\Bigg(\left(\frac{2\,(p_\mathrm{a} - p_\mathrm{aa})\,\widehat{L}^2}{np_\mathrm{a}^2} + \frac{(1 - p_\mathrm{page})L_\mathrm{max}^2}{np_\mathrm{a}B}\right)\left\|x^{t+1} - x^t\right\|^2$$

$$+ \frac{2\,(p_\mathrm{a} - p_\mathrm{aa})\,b^2}{n^2p_\mathrm{a}^2 p_\mathrm{page}}\sum_{i=1}^n\left\|h_i^t - \nabla f_i(x^t)\right\|^2 + \left(p_\mathrm{page}\left(1 - \frac{b}{p_\mathrm{page}}\right)^2 + (1 - p_\mathrm{page})\right)\left\|h^t - \nabla f(x^t)\right\|^2\Bigg)$$

$$+ \rho\mathrm{E}\Bigg(\left(\frac{2\,(1 - p_\mathrm{a})\,\widehat{L}^2}{p_\mathrm{a}} + \frac{(1 - p_\mathrm{page})L_\mathrm{max}^2}{p_\mathrm{a}B}\right)\left\|x^{t+1} - x^t\right\|^2$$

$$+ \left(\frac{2\,(1 - p_\mathrm{a})\,b^2}{p_\mathrm{a}p_\mathrm{page}} + p_\mathrm{page}\left(1 - \frac{b}{p_\mathrm{page}}\right)^2 + (1 - p_\mathrm{page})\right)\frac{1}{n}\sum_{i=1}^n\left\|h_i^t - \nabla f_i(x^t)\right\|^2\Bigg).$$

After rearranging the terms, we get

$$\mathrm{E}\left[f(x^{t+1})\right] + \frac{2\gamma(2\omega+1)}{p_\mathrm{a}}\mathrm{E}\left[\left\|g^{t+1} - h^{t+1}\right\|^2\right] + \frac{4\gamma((2\omega+1)\,p_\mathrm{a} - p_\mathrm{aa})}{np_\mathrm{a}^2}\mathrm{E}\left[\frac{1}{n}\sum_{i=1}^n\left\|g_i^{t+1} - h_i^{t+1}\right\|^2\right]$$

$$+ \nu\mathrm{E}\left[\left\|h^{t+1} - \nabla f(x^{t+1})\right\|^2\right] + \rho\mathrm{E}\left[\frac{1}{n}\sum_{i=1}^n\left\|h_i^{t+1} - \nabla f_i(x^{t+1})\right\|^2\right]$$

$$\le \mathrm{E}\left[f(x^t)\right] - \frac{\gamma}{2}\mathrm{E}\left[\left\|\nabla f(x^t)\right\|^2\right]$$

$$+ (1 - \gamma\mu)\frac{2\gamma(2\omega+1)}{p_\mathrm{a}}\mathrm{E}\left[\left\|g^t - h^t\right\|^2\right] + (1 - \gamma\mu)\frac{4\gamma((2\omega+1)\,p_\mathrm{a} - p_\mathrm{aa})}{np_\mathrm{a}^2}\mathrm{E}\left[\frac{1}{n}\sum_{i=1}^n\left\|g_i^t - h_i^t\right\|^2\right]$$

$$-\left(\frac{1}{2\gamma}-\frac{L}{2}-\frac{10\gamma\omega(2\omega+1)}{np_{\mathrm{a}}^2}\left(2\widehat{L}^2+\frac{(1-p_{\mathrm{page}})L_{\max}^2}{B}\right)\right.$$

$$\left.-\nu\left(\frac{2\,(p_{\mathrm{a}}-p_{\mathrm{aa}})\,\widehat{L}^2}{np_{\mathrm{a}}^2}+\frac{(1-p_{\mathrm{page}})L_{\max}^2}{np_{\mathrm{a}}B}\right)-\rho\left(\frac{2\,(1-p_{\mathrm{a}})\,\widehat{L}^2}{p_{\mathrm{a}}}+\frac{(1-p_{\mathrm{page}})L_{\max}^2}{p_{\mathrm{a}}B}\right)\right)\mathrm{E}\left[\left\|x^{t+1}-x^t\right\|^2\right]$$

$$+\left(\gamma+\nu\left(p_{\mathrm{page}}\left(1-\frac{b}{p_{\mathrm{page}}}\right)^2+(1-p_{\mathrm{page}})\right)\right)\mathrm{E}\left[\left\|h^t-\nabla f(x^t)\right\|^2\right]$$

$$+\left(\frac{20b^2\gamma\omega(2\omega+1)}{np_{\mathrm{a}}^2p_{\mathrm{page}}}+\frac{2\nu\,(p_{\mathrm{a}}-p_{\mathrm{aa}})\,b^2}{np_{\mathrm{a}}^2p_{\mathrm{page}}}\right.$$

$$\left.+\rho\left(\frac{2\,(1-p_{\mathrm{a}})\,b^2}{p_{\mathrm{a}}p_{\mathrm{page}}}+p_{\mathrm{page}}\left(1-\frac{b}{p_{\mathrm{page}}}\right)^2+(1-p_{\mathrm{page}})\right)\right)\mathrm{E}\left[\frac{1}{n}\sum_{i=1}^{n}\left\|h_i^t-\nabla f_i(x^t)\right\|^2\right].$$

Due to $b=\frac{p_{\mathrm{page}}p_{\mathrm{a}}}{2-p_{\mathrm{a}}}\leq p_{\mathrm{page}}$, one can show that $\left(p_{\mathrm{page}}\left(1-\frac{b}{p_{\mathrm{page}}}\right)^2+(1-p_{\mathrm{page}})\right)\leq 1-b$. Thus, if we take $\nu=\frac{2\gamma}{b}$, then

$$\left(\gamma+\nu\left(p_{\mathrm{page}}\left(1-\frac{b}{p_{\mathrm{page}}}\right)^2+(1-p_{\mathrm{page}})\right)\right)\leq\gamma+\nu(1-b)=\left(1-\frac{b}{2}\right)\nu,$$

therefore

$$\mathrm{E}\left[f(x^{t+1})\right]+\frac{2\gamma(2\omega+1)}{p_{\mathrm{a}}}\mathrm{E}\left[\left\|g^{t+1}-h^{t+1}\right\|^2\right]+\frac{4\gamma((2\omega+1)\,p_{\mathrm{a}}-p_{\mathrm{aa}})}{np_{\mathrm{a}}^2}\mathrm{E}\left[\frac{1}{n}\sum_{i=1}^{n}\left\|g_i^{t+1}-h_i^{t+1}\right\|^2\right]$$

$$+\frac{2\gamma}{b}\mathrm{E}\left[\left\|h^{t+1}-\nabla f(x^{t+1})\right\|^2\right]+\rho\mathrm{E}\left[\frac{1}{n}\sum_{i=1}^{n}\left\|h_i^{t+1}-\nabla f_i(x^{t+1})\right\|^2\right]$$

$$\leq\mathrm{E}\left[f(x^t)\right]-\frac{\gamma}{2}\mathrm{E}\left[\left\|\nabla f(x^t)\right\|^2\right]$$

$$+(1-\gamma\mu)\frac{2\gamma(2\omega+1)}{p_{\mathrm{a}}}\mathrm{E}\left[\left\|g^t-h^t\right\|^2\right]+(1-\gamma\mu)\frac{4\gamma((2\omega+1)\,p_{\mathrm{a}}-p_{\mathrm{aa}})}{np_{\mathrm{a}}^2}\mathrm{E}\left[\frac{1}{n}\sum_{i=1}^{n}\left\|g_i^t-h_i^t\right\|^2\right]$$

$$-\left(\frac{1}{2\gamma}-\frac{L}{2}-\frac{10\gamma\omega(2\omega+1)}{np_{\mathrm{a}}^2}\left(2\widehat{L}^2+\frac{(1-p_{\mathrm{page}})L_{\max}^2}{B}\right)\right.$$

$$\left.-\frac{2\gamma}{bnp_{\mathrm{a}}}\left(2\left(1-\frac{p_{\mathrm{aa}}}{p_{\mathrm{a}}}\right)\widehat{L}^2+\frac{(1-p_{\mathrm{page}})L_{\max}^2}{B}\right)-\rho\left(\frac{2\,(1-p_{\mathrm{a}})\,\widehat{L}^2}{p_{\mathrm{a}}}+\frac{(1-p_{\mathrm{page}})L_{\max}^2}{p_{\mathrm{a}}B}\right)\right)\mathrm{E}\left[\left\|x^{t+1}-x^t\right\|^2\right]$$

$$+\left(1-\frac{b}{2}\right)\frac{2\gamma}{b}\mathrm{E}\left[\left\|h^t-\nabla f(x^t)\right\|^2\right]$$

$$+\left(\frac{20b^2\gamma\omega(2\omega+1)}{np_{\mathrm{a}}^2p_{\mathrm{page}}}+\frac{4\gamma\,(p_{\mathrm{a}}-p_{\mathrm{aa}})\,b}{np_{\mathrm{a}}^2p_{\mathrm{page}}}\right.$$

$$\left.+\rho\left(\frac{2\,(1-p_{\mathrm{a}})\,b^2}{p_{\mathrm{a}}p_{\mathrm{page}}}+p_{\mathrm{page}}\left(1-\frac{b}{p_{\mathrm{page}}}\right)^2+(1-p_{\mathrm{page}})\right)\right)\mathrm{E}\left[\frac{1}{n}\sum_{i=1}^{n}\left\|h_i^t-\nabla f_i(x^t)\right\|^2\right].$$

Next, with the choice of $b=\frac{p_{\mathrm{page}}p_{\mathrm{a}}}{2-p_{\mathrm{a}}}$, we ensure that

$$\left(\frac{2\,(1-p_{\mathrm{a}})\,b^2}{p_{\mathrm{a}}p_{\mathrm{page}}}+p_{\mathrm{page}}\left(1-\frac{b}{p_{\mathrm{page}}}\right)^2+(1-p_{\mathrm{page}})\right)\leq 1-b.$$

If we take $\rho=\frac{40b\gamma\omega(2\omega+1)}{np_{\mathrm{a}}^2p_{\mathrm{page}}}+\frac{8\gamma(p_{\mathrm{a}}-p_{\mathrm{aa}})}{np_{\mathrm{a}}^2p_{\mathrm{page}}}$, then

$$\left(\frac{20b^2\gamma\omega(2\omega+1)}{np_{\mathrm{a}}^2p_{\mathrm{page}}}+\frac{4\gamma\,(p_{\mathrm{a}}-p_{\mathrm{aa}})\,b}{np_{\mathrm{a}}^2p_{\mathrm{page}}}+\rho\left(\frac{2\,(1-p_{\mathrm{a}})\,b^2}{p_{\mathrm{a}}p_{\mathrm{page}}}+p_{\mathrm{page}}\left(1-\frac{b}{p_{\mathrm{page}}}\right)^2+(1-p_{\mathrm{page}})\right)\right)\leq\left(1-\frac{b}{2}\right)\rho,$$

therefore

$$
\mathrm{E}\left[f(x^{t+1})\right] + \frac{2\gamma(2\omega+1)}{p_{\mathrm{a}}}\mathrm{E}\left[\left\|g^{t+1}-h^{t+1}\right\|^2\right] + \frac{4\gamma((2\omega+1)\,p_{\mathrm{a}}-p_{\mathrm{aa}})}{np_{\mathrm{a}}^2}\mathrm{E}\left[\frac{1}{n}\sum_{i=1}^{n}\left\|g_i^{t+1}-h_i^{t+1}\right\|^2\right]
$$

$$
+ \frac{2\gamma}{b}\mathrm{E}\left[\left\|h^{t+1}-\nabla f(x^{t+1})\right\|^2\right] + \left(\frac{40b\gamma\omega(2\omega+1)}{np_{\mathrm{a}}^2 p_{\mathrm{page}}} + \frac{8\gamma\,(p_{\mathrm{a}}-p_{\mathrm{aa}})}{np_{\mathrm{a}}^2 p_{\mathrm{page}}}\right)\mathrm{E}\left[\frac{1}{n}\sum_{i=1}^{n}\left\|h_i^{t+1}-\nabla f_i(x^{t+1})\right\|^2\right]
$$

$$
\leq \mathrm{E}\left[f(x^{t})\right] - \frac{\gamma}{2}\mathrm{E}\left[\left\|\nabla f(x^{t})\right\|^2\right]
$$

$$
+ (1-\gamma\mu)\frac{2\gamma(2\omega+1)}{p_{\mathrm{a}}}\mathrm{E}\left[\left\|g^{t}-h^{t}\right\|^2\right] + (1-\gamma\mu)\frac{4\gamma((2\omega+1)\,p_{\mathrm{a}}-p_{\mathrm{aa}})}{np_{\mathrm{a}}^2}\mathrm{E}\left[\frac{1}{n}\sum_{i=1}^{n}\left\|g_i^{t}-h_i^{t}\right\|^2\right]
$$

$$
- \left(\frac{1}{2\gamma} - \frac{L}{2} - \frac{10\gamma\omega(2\omega+1)}{np_{\mathrm{a}}^2}\left(2\widehat{L}^2 + \frac{(1-p_{\mathrm{page}})L_{\max}^2}{B}\right)\right)
$$

$$
- \frac{2\gamma}{bnp_{\mathrm{a}}}\left(2\left(1-\frac{p_{\mathrm{aa}}}{p_{\mathrm{a}}}\right)\widehat{L}^2 + \frac{(1-p_{\mathrm{page}})L_{\max}^2}{B}\right)
$$

$$
- \left(\frac{40b\gamma\omega(2\omega+1)}{np_{\mathrm{a}}^3 p_{\mathrm{page}}} + \frac{8\gamma\left(1-\frac{p_{\mathrm{aa}}}{p_{\mathrm{a}}}\right)}{np_{\mathrm{a}}^2 p_{\mathrm{page}}}\right)\left(2\left(1-p_{\mathrm{a}}\right)\widehat{L}^2 + \frac{(1-p_{\mathrm{page}})L_{\max}^2}{B}\right)\right)\mathrm{E}\left[\left\|x^{t+1}-x^{t}\right\|^2\right]
$$

$$
+ \left(1-\frac{b}{2}\right)\frac{2\gamma}{b}\mathrm{E}\left[\left\|h^{t}-\nabla f(x^{t})\right\|^2\right] + \left(1-\frac{b}{2}\right)\left(\frac{40b\gamma\omega(2\omega+1)}{np_{\mathrm{a}}^2 p_{\mathrm{page}}} + \frac{8\gamma\,(p_{\mathrm{a}}-p_{\mathrm{aa}})}{np_{\mathrm{a}}^2 p_{\mathrm{page}}}\right)\mathrm{E}\left[\frac{1}{n}\sum_{i=1}^{n}\left\|h_i^{t}-\nabla f_i(x^{t})\right\|^2\right].
$$

Let us simplify the inequality. First, due to $b \geq \frac{p_{\mathrm{page}}p_{\mathrm{a}}}{2}$, we have

$$
\frac{2\gamma}{bnp_{\mathrm{a}}}\left(2\left(1-\frac{p_{\mathrm{aa}}}{p_{\mathrm{a}}}\right)\widehat{L}^2 + \frac{(1-p_{\mathrm{page}})L_{\max}^2}{B}\right) \leq \frac{8\gamma}{np_{\mathrm{a}}^2 p_{\mathrm{page}}}\left(\left(1-\frac{p_{\mathrm{aa}}}{p_{\mathrm{a}}}\right)\widehat{L}^2 + \frac{(1-p_{\mathrm{page}})L_{\max}^2}{B}\right).
$$

Second, due to $b \leq p_{\mathrm{a}}p_{\mathrm{page}}$ and $p_{\mathrm{aa}} \leq p_{\mathrm{a}}^2$, we get

$$
\left(\frac{40b\gamma\omega(2\omega+1)}{np_{\mathrm{a}}^3 p_{\mathrm{page}}} + \frac{8\gamma\left(1-\frac{p_{\mathrm{aa}}}{p_{\mathrm{a}}}\right)}{np_{\mathrm{a}}^2 p_{\mathrm{page}}}\right)\left(2\left(1-p_{\mathrm{a}}\right)\widehat{L}^2 + \frac{(1-p_{\mathrm{page}})L_{\max}^2}{B}\right)
$$

$$
\leq \left(\frac{40\gamma\omega(2\omega+1)}{np_{\mathrm{a}}^2} + \frac{8\gamma\left(1-\frac{p_{\mathrm{aa}}}{p_{\mathrm{a}}}\right)}{np_{\mathrm{a}}^2 p_{\mathrm{page}}}\right)\left(2\left(1-\frac{p_{\mathrm{aa}}}{p_{\mathrm{a}}}\right)\widehat{L}^2 + \frac{(1-p_{\mathrm{page}})L_{\max}^2}{B}\right)
$$

$$
\leq \frac{80\gamma\omega(2\omega+1)}{np_{\mathrm{a}}^2}\left(\left(1-\frac{p_{\mathrm{aa}}}{p_{\mathrm{a}}}\right)\widehat{L}^2 + \frac{(1-p_{\mathrm{page}})L_{\max}^2}{B}\right)
$$

$$
+ \frac{16\gamma\left(1-\frac{p_{\mathrm{aa}}}{p_{\mathrm{a}}}\right)}{np_{\mathrm{a}}^2 p_{\mathrm{page}}}\left(\left(1-\frac{p_{\mathrm{aa}}}{p_{\mathrm{a}}}\right)\widehat{L}^2 + \frac{(1-p_{\mathrm{page}})L_{\max}^2}{B}\right)
$$

$$
\leq \frac{80\gamma\omega(2\omega+1)}{np_{\mathrm{a}}^2}\left(\widehat{L}^2 + \frac{(1-p_{\mathrm{page}})L_{\max}^2}{B}\right)
$$

$$
+ \frac{16\gamma}{np_{\mathrm{a}}^2 p_{\mathrm{page}}}\left(\left(1-\frac{p_{\mathrm{aa}}}{p_{\mathrm{a}}}\right)\widehat{L}^2 + \frac{(1-p_{\mathrm{page}})L_{\max}^2}{B}\right).
$$

Combining all bounds together, we obtain the following inequality:

$$
\mathrm{E}\left[f(x^{t+1})\right] + \frac{2\gamma(2\omega+1)}{p_{\mathrm{a}}}\mathrm{E}\left[\left\|g^{t+1}-h^{t+1}\right\|^2\right] + \frac{4\gamma((2\omega+1)\,p_{\mathrm{a}}-p_{\mathrm{aa}})}{np_{\mathrm{a}}^2}\mathrm{E}\left[\frac{1}{n}\sum_{i=1}^{n}\left\|g_i^{t+1}-h_i^{t+1}\right\|^2\right]
$$

$$
+ \frac{2\gamma}{b}\mathrm{E}\left[\left\|h^{t+1}-\nabla f(x^{t+1})\right\|^2\right] + \left(\frac{40b\gamma\omega(2\omega+1)}{np_{\mathrm{a}}^2 p_{\mathrm{page}}} + \frac{8\gamma\,(p_{\mathrm{a}}-p_{\mathrm{aa}})}{np_{\mathrm{a}}^2 p_{\mathrm{page}}}\right)\mathrm{E}\left[\frac{1}{n}\sum_{i=1}^{n}\left\|h_i^{t+1}-\nabla f_i(x^{t+1})\right\|^2\right]
$$

$$
\leq \mathrm{E}\left[f(x^{t})\right] - \frac{\gamma}{2}\mathrm{E}\left[\left\|\nabla f(x^{t})\right\|^2\right]
$$

$$
+ (1 - \gamma\mu) \frac{2\gamma(2\omega + 1)}{p_{\mathrm{a}}} \mathrm{E}\left[\left\|g^t - h^t\right\|^2\right] + (1 - \gamma\mu) \frac{4\gamma((2\omega + 1)p_{\mathrm{a}} - p_{\mathrm{aa}})}{np_{\mathrm{a}}^2} \mathrm{E}\left[\frac{1}{n}\sum_{i=1}^{n}\left\|g_i^t - h_i^t\right\|^2\right]
$$

$$
- \left(\frac{1}{2\gamma} - \frac{L}{2} - \frac{100\gamma\omega(2\omega + 1)}{np_{\mathrm{a}}^2}\left(\widehat{L}^2 + \frac{(1 - p_{\mathrm{page}})L_{\max}^2}{B}\right)\right.
$$

$$
\left. - \frac{24\gamma}{np_{\mathrm{a}}^2 p_{\mathrm{page}}}\left(\left(1 - \frac{p_{\mathrm{aa}}}{p_{\mathrm{a}}}\right)\widehat{L}^2 + \frac{(1 - p_{\mathrm{page}})L_{\max}^2}{B}\right)\right)\mathrm{E}\left[\left\|x^{t+1} - x^t\right\|^2\right]
$$

$$
+ \left(1 - \frac{b}{2}\right)\frac{2\gamma}{b}\mathrm{E}\left[\left\|h^t - \nabla f(x^t)\right\|^2\right] + \left(1 - \frac{b}{2}\right)\left(\frac{40b\gamma\omega(2\omega + 1)}{np_{\mathrm{a}}^2 p_{\mathrm{page}}} + \frac{8\gamma(p_{\mathrm{a}} - p_{\mathrm{aa}})}{np_{\mathrm{a}}^2 p_{\mathrm{page}}}\right)\mathrm{E}\left[\frac{1}{n}\sum_{i=1}^{n}\left\|h_i^t - \nabla f_i(x^t)\right\|^2\right].
$$

Using Lemma 4 and the assumption about $\gamma$, we get

$$
\mathrm{E}\left[f(x^{t+1})\right] + \frac{2\gamma(2\omega + 1)}{p_{\mathrm{a}}}\mathrm{E}\left[\left\|g^{t+1} - h^{t+1}\right\|^2\right] + \frac{4\gamma((2\omega + 1)p_{\mathrm{a}} - p_{\mathrm{aa}})}{np_{\mathrm{a}}^2}\mathrm{E}\left[\frac{1}{n}\sum_{i=1}^{n}\left\|g_i^{t+1} - h_i^{t+1}\right\|^2\right]
$$

$$
+ \frac{2\gamma}{b}\mathrm{E}\left[\left\|h^{t+1} - \nabla f(x^{t+1})\right\|^2\right] + \left(\frac{40b\gamma\omega(2\omega + 1)}{np_{\mathrm{a}}^2 p_{\mathrm{page}}} + \frac{8\gamma(p_{\mathrm{a}} - p_{\mathrm{aa}})}{np_{\mathrm{a}}^2 p_{\mathrm{page}}}\right)\mathrm{E}\left[\frac{1}{n}\sum_{i=1}^{n}\left\|h_i^{t+1} - \nabla f_i(x^{t+1})\right\|^2\right]
$$

$$
\leq \mathrm{E}\left[f(x^t)\right] - \frac{\gamma}{2}\mathrm{E}\left[\left\|\nabla f(x^t)\right\|^2\right]
$$

$$
+ (1 - \gamma\mu)\frac{2\gamma(2\omega + 1)}{p_{\mathrm{a}}}\mathrm{E}\left[\left\|g^t - h^t\right\|^2\right] + (1 - \gamma\mu)\frac{4\gamma((2\omega + 1)p_{\mathrm{a}} - p_{\mathrm{aa}})}{np_{\mathrm{a}}^2}\mathrm{E}\left[\frac{1}{n}\sum_{i=1}^{n}\left\|g_i^t - h_i^t\right\|^2\right]
$$

$$
+ \left(1 - \frac{b}{2}\right)\frac{2\gamma}{b}\mathrm{E}\left[\left\|h^t - \nabla f(x^t)\right\|^2\right] + \left(1 - \frac{b}{2}\right)\left(\frac{40b\gamma\omega(2\omega + 1)}{np_{\mathrm{a}}^2 p_{\mathrm{page}}} + \frac{8\gamma(p_{\mathrm{a}} - p_{\mathrm{aa}})}{np_{\mathrm{a}}^2 p_{\mathrm{page}}}\right)\mathrm{E}\left[\frac{1}{n}\sum_{i=1}^{n}\left\|h_i^t - \nabla f_i(x^t)\right\|^2\right].
$$

Note that $\gamma \leq \frac{b}{2\mu}$, thus $1 - \frac{b}{2} \leq 1 - \gamma\mu$ and

$$
\mathrm{E}\left[f(x^{t+1})\right] + \frac{2\gamma(2\omega + 1)}{p_{\mathrm{a}}}\mathrm{E}\left[\left\|g^{t+1} - h^{t+1}\right\|^2\right] + \frac{4\gamma((2\omega + 1)p_{\mathrm{a}} - p_{\mathrm{aa}})}{np_{\mathrm{a}}^2}\mathrm{E}\left[\frac{1}{n}\sum_{i=1}^{n}\left\|g_i^{t+1} - h_i^{t+1}\right\|^2\right]
$$

$$
+ \frac{2\gamma}{b}\mathrm{E}\left[\left\|h^{t+1} - \nabla f(x^{t+1})\right\|^2\right] + \left(\frac{40b\gamma\omega(2\omega + 1)}{np_{\mathrm{a}}^2 p_{\mathrm{page}}} + \frac{8\gamma(p_{\mathrm{a}} - p_{\mathrm{aa}})}{np_{\mathrm{a}}^2 p_{\mathrm{page}}}\right)\mathrm{E}\left[\frac{1}{n}\sum_{i=1}^{n}\left\|h_i^{t+1} - \nabla f_i(x^{t+1})\right\|^2\right]
$$

$$
\leq \mathrm{E}\left[f(x^t)\right] - \frac{\gamma}{2}\mathrm{E}\left[\left\|\nabla f(x^t)\right\|^2\right]
$$

$$
+ (1 - \gamma\mu)\frac{2\gamma(2\omega + 1)}{p_{\mathrm{a}}}\mathrm{E}\left[\left\|g^t - h^t\right\|^2\right] + (1 - \gamma\mu)\frac{4\gamma((2\omega + 1)p_{\mathrm{a}} - p_{\mathrm{aa}})}{np_{\mathrm{a}}^2}\mathrm{E}\left[\frac{1}{n}\sum_{i=1}^{n}\left\|g_i^t - h_i^t\right\|^2\right]
$$

$$
+ (1 - \gamma\mu)\frac{2\gamma}{b}\mathrm{E}\left[\left\|h^t - \nabla f(x^t)\right\|^2\right] + (1 - \gamma\mu)\left(\frac{40b\gamma\omega(2\omega + 1)}{np_{\mathrm{a}}^2 p_{\mathrm{page}}} + \frac{8\gamma(p_{\mathrm{a}} - p_{\mathrm{aa}})}{np_{\mathrm{a}}^2 p_{\mathrm{page}}}\right)\mathrm{E}\left[\frac{1}{n}\sum_{i=1}^{n}\left\|h_i^t - \nabla f_i(x^t)\right\|^2\right].
$$

It is left to apply Lemma 11 with

$$
\Psi^t = \frac{2(2\omega + 1)}{p_{\mathrm{a}}}\mathrm{E}\left[\left\|g^t - h^t\right\|^2\right] + \frac{4((2\omega + 1)p_{\mathrm{a}} - p_{\mathrm{aa}})}{np_{\mathrm{a}}^2}\mathrm{E}\left[\frac{1}{n}\sum_{i=1}^{n}\left\|g_i^t - h_i^t\right\|^2\right]
$$

$$
+ \frac{2}{b}\mathrm{E}\left[\left\|h^t - \nabla f(x^t)\right\|^2\right] + \left(\frac{40b\omega(2\omega + 1)}{np_{\mathrm{a}}^2 p_{\mathrm{page}}} + \frac{8(p_{\mathrm{a}} - p_{\mathrm{aa}})}{np_{\mathrm{a}}^2 p_{\mathrm{page}}}\right)\mathrm{E}\left[\frac{1}{n}\sum_{i=1}^{n}\left\|h_i^t - \nabla f_i(x^t)\right\|^2\right]
$$

to conclude the proof. $\qquad\square$

### F.4.5 Proof for DASHA-PP-MVR under PŁ-condition

**Theorem 10.** *Suppose that Assumption 1, 2, 3, 7, 5, 6, 8 and 9 hold. Let us take $a = \frac{p_{\mathrm{a}}}{2\omega+1}$, $b \in \left(0, \frac{p_{\mathrm{a}}}{2-p_{\mathrm{a}}}\right]$,*

$$
\gamma \leq \min\left\{\left(L + \sqrt{\frac{200\omega(2\omega+1)}{np_{\mathrm{a}}^2}\left(\frac{(1-b)^2 L_\sigma^2}{B} + \widehat{L}^2\right) + \frac{40}{np_{\mathrm{a}}b}\left(\frac{(1-b)^2 L_\sigma^2}{B} + \left(1 - \frac{p_{\mathrm{aa}}}{p_{\mathrm{a}}}\right)\widehat{L}^2\right)}\right)^{-1}, \frac{a}{2\mu}, \frac{b}{2\mu}\right\},
$$

*and $h_i^0 = g_i^0$ for all $i \in [n]$ in Algorithm 1 (DASHA-PP-MVR), then*

$$
\mathrm{E}\left[f(x^T) - f^*\right]
$$
$$
\leq (1 - \gamma\mu)^T\left(\Delta_0 + \frac{2\gamma}{b}\left\|h^0 - \nabla f(x^0)\right\|^2 + \left(\frac{40\gamma b\omega(2\omega+1)}{np_{\mathrm{a}}^2} + \frac{8\gamma(p_{\mathrm{a}} - p_{\mathrm{aa}})}{np_{\mathrm{a}}^2}\right)\frac{1}{n}\sum_{i=1}^n \left\|h_i^0 - \nabla f_i(x^0)\right\|^2\right)
$$
$$
+ \frac{1}{\mu}\left(\frac{100b^2\omega(2\omega+1)}{p_{\mathrm{a}}^2} + \frac{20b}{p_{\mathrm{a}}}\right)\frac{\sigma^2}{nB}.
$$

$$
\mathrm{E}\left[f(x^{t+1})\right] + \frac{2\gamma(2\omega+1)}{p_{\mathrm{a}}}\mathrm{E}\left[\left\|g^{t+1} - h^{t+1}\right\|^2\right] + \frac{4\gamma((2\omega+1)p_{\mathrm{a}} - p_{\mathrm{aa}})}{np_{\mathrm{a}}^2}\mathrm{E}\left[\frac{1}{n}\sum_{i=1}^n \left\|g_i^{t+1} - h_i^{t+1}\right\|^2\right]
$$
$$
\leq \mathrm{E}\left[f(x^t) - \frac{\gamma}{2}\left\|\nabla f(x^t)\right\|^2 - \left(\frac{1}{2\gamma} - \frac{L}{2}\right)\left\|x^{t+1} - x^t\right\|^2 + \gamma\left\|h^t - \nabla f(x^t)\right\|^2\right]
$$
$$
+ (1 - \gamma\mu)\frac{2\gamma(2\omega+1)}{p_{\mathrm{a}}}\mathrm{E}\left[\left\|g^t - h^t\right\|^2\right] + (1 - \gamma\mu)\frac{4\gamma((2\omega+1)p_{\mathrm{a}} - p_{\mathrm{aa}})}{np_{\mathrm{a}}^2}\mathrm{E}\left[\frac{1}{n}\sum_{i=1}^n \left\|g_i^t - h_i^t\right\|^2\right]
$$
$$
+ \frac{10\gamma(2\omega+1)\omega}{np_{\mathrm{a}}^2}\mathrm{E}\left[\frac{1}{n}\sum_{i=1}^n \left\|k_i^{t+1}\right\|^2\right].
$$

*Proof.* Let us fix constants $\nu, \rho \in [0, \infty)$ that we will define later. Considering Lemma 12, Lemma 10, and the law of total expectation, we obtain

$$
\mathrm{E}\left[f(x^{t+1})\right] + \frac{2\gamma(2\omega+1)}{p_{\mathrm{a}}}\mathrm{E}\left[\left\|g^{t+1} - h^{t+1}\right\|^2\right] + \frac{4\gamma((2\omega+1)p_{\mathrm{a}} - p_{\mathrm{aa}})}{np_{\mathrm{a}}^2}\mathrm{E}\left[\frac{1}{n}\sum_{i=1}^n \left\|g_i^{t+1} - h_i^{t+1}\right\|^2\right]
$$
$$
+ \nu\mathrm{E}\left[\left\|h^{t+1} - \nabla f(x^{t+1})\right\|^2\right] + \rho\mathrm{E}\left[\frac{1}{n}\sum_{i=1}^n \left\|h_i^{t+1} - \nabla f_i(x^{t+1})\right\|^2\right]
$$
$$
\leq \mathrm{E}\left[f(x^t) - \frac{\gamma}{2}\left\|\nabla f(x^t)\right\|^2 - \left(\frac{1}{2\gamma} - \frac{L}{2}\right)\left\|x^{t+1} - x^t\right\|^2 + \gamma\left\|h^t - \nabla f(x^t)\right\|^2\right]
$$
$$
+ (1 - \gamma\mu)\frac{2\gamma(2\omega+1)}{p_{\mathrm{a}}}\mathrm{E}\left[\left\|g^t - h^t\right\|^2\right] + (1 - \gamma\mu)\frac{4\gamma((2\omega+1)p_{\mathrm{a}} - p_{\mathrm{aa}})}{np_{\mathrm{a}}^2}\mathrm{E}\left[\frac{1}{n}\sum_{i=1}^n \left\|g_i^t - h_i^t\right\|^2\right]
$$
$$
+ \frac{10\gamma(2\omega+1)\omega}{np_{\mathrm{a}}^2}\mathrm{E}\left[\frac{1}{n}\sum_{i=1}^n \left\|k_i^{t+1}\right\|^2\right]
$$
$$
+ \nu\mathrm{E}\left[\left\|h^{t+1} - \nabla f(x^{t+1})\right\|^2\right] + \rho\mathrm{E}\left[\frac{1}{n}\sum_{i=1}^n \left\|h_i^{t+1} - \nabla f_i(x^{t+1})\right\|^2\right]
$$
$$
\leq \mathrm{E}\left[f(x^t) - \frac{\gamma}{2}\left\|\nabla f(x^t)\right\|^2 - \left(\frac{1}{2\gamma} - \frac{L}{2}\right)\left\|x^{t+1} - x^t\right\|^2 + \gamma\left\|h^t - \nabla f(x^t)\right\|^2\right]
$$
$$
+ (1 - \gamma\mu)\frac{2\gamma(2\omega+1)}{p_{\mathrm{a}}}\mathrm{E}\left[\left\|g^t - h^t\right\|^2\right] + (1 - \gamma\mu)\frac{4\gamma((2\omega+1)p_{\mathrm{a}} - p_{\mathrm{aa}})}{np_{\mathrm{a}}^2}\mathrm{E}\left[\frac{1}{n}\sum_{i=1}^n \left\|g_i^t - h_i^t\right\|^2\right]
$$

$$+ \frac{10\gamma\omega(2\omega+1)}{np_{\mathrm{a}}^2}\mathrm{E}\left[\frac{2b^2\sigma^2}{B} + \left(\frac{2(1-b)^2 L_\sigma^2}{B} + 2\widehat{L}^2\right)\left\|x^{t+1} - x^t\right\|^2 + 2b^2\frac{1}{n}\sum_{i=1}^n \left\|h_i^t - \nabla f_i(x^t)\right\|^2\right]$$

$$+ \nu\mathrm{E}\left(\frac{2b^2\sigma^2}{np_{\mathrm{a}}B} + \left(\frac{2(1-b)^2 L_\sigma^2}{np_{\mathrm{a}}B} + \frac{2\left(p_{\mathrm{a}} - p_{\mathrm{aa}}\right)\widehat{L}^2}{np_{\mathrm{a}}^2}\right)\left\|x^{t+1} - x^t\right\|^2\right.$$

$$\left.+ \frac{2\left(p_{\mathrm{a}} - p_{\mathrm{aa}}\right)b^2}{n^2 p_{\mathrm{a}}^2}\sum_{i=1}^n \left\|h_i^t - \nabla f_i(x^t)\right\|^2 + (1-b)^2\left\|h^t - \nabla f(x^t)\right\|^2\right)$$

$$+ \rho\mathrm{E}\left(\frac{2b^2\sigma^2}{p_{\mathrm{a}}B} + \left(\frac{2(1-b)^2 L_\sigma^2}{p_{\mathrm{a}}B} + \frac{2(1-p_{\mathrm{a}})\widehat{L}^2}{p_{\mathrm{a}}}\right)\left\|x^{t+1} - x^t\right\|^2\right.$$

$$\left.+ \left(\frac{2(1-p_{\mathrm{a}})b^2}{p_{\mathrm{a}}} + (1-b)^2\right)\frac{1}{n}\sum_{i=1}^n \left\|h_i^t - \nabla f_i(x^t)\right\|^2\right).$$

After rearranging the terms, we get

$$\mathrm{E}\left[f(x^{t+1})\right] + (1-\gamma\mu)\frac{2\gamma(2\omega+1)}{p_{\mathrm{a}}}\mathrm{E}\left[\left\|g^{t+1} - h^{t+1}\right\|^2\right] + (1-\gamma\mu)\frac{4\gamma((2\omega+1)p_{\mathrm{a}} - p_{\mathrm{aa}})}{np_{\mathrm{a}}^2}\mathrm{E}\left[\frac{1}{n}\sum_{i=1}^n \left\|g_i^{t+1} - h_i^{t+1}\right\|^2\right]$$

$$+ \nu\mathrm{E}\left[\left\|h^{t+1} - \nabla f(x^{t+1})\right\|^2\right] + \rho\mathrm{E}\left[\frac{1}{n}\sum_{i=1}^n \left\|h_i^{t+1} - \nabla f_i(x^{t+1})\right\|^2\right]$$

$$\leq \mathrm{E}\left[f(x^t)\right] - \frac{\gamma}{2}\mathrm{E}\left[\left\|\nabla f(x^t)\right\|^2\right]$$

$$+ (1-\gamma\mu)\frac{2\gamma(2\omega+1)}{p_{\mathrm{a}}}\mathrm{E}\left[\left\|g^t - h^t\right\|^2\right] + (1-\gamma\mu)\frac{4\gamma((2\omega+1)p_{\mathrm{a}} - p_{\mathrm{aa}})}{np_{\mathrm{a}}^2}\mathrm{E}\left[\frac{1}{n}\sum_{i=1}^n \left\|g_i^t - h_i^t\right\|^2\right]$$

$$- \left(\frac{1}{2\gamma} - \frac{L}{2} - \frac{10\gamma\omega(2\omega+1)}{np_{\mathrm{a}}^2}\left(\frac{2(1-b)^2 L_\sigma^2}{B} + 2\widehat{L}^2\right)\right.$$

$$\left.- \nu\left(\frac{2(1-b)^2 L_\sigma^2}{np_{\mathrm{a}}B} + \frac{2\left(p_{\mathrm{a}} - p_{\mathrm{aa}}\right)\widehat{L}^2}{np_{\mathrm{a}}^2}\right) - \rho\left(\frac{2(1-b)^2 L_\sigma^2}{p_{\mathrm{a}}B} + \frac{2(1-p_{\mathrm{a}})\widehat{L}^2}{p_{\mathrm{a}}}\right)\right)\mathrm{E}\left[\left\|x^{t+1} - x^t\right\|^2\right]$$

$$+ \left(\gamma + \nu\left(1-b\right)^2\right)\mathrm{E}\left[\left\|h^t - \nabla f(x^t)\right\|^2\right]$$

$$+ \left(\frac{20b^2\gamma\omega(2\omega+1)}{np_{\mathrm{a}}^2} + \frac{2\nu\left(p_{\mathrm{a}} - p_{\mathrm{aa}}\right)b^2}{np_{\mathrm{a}}^2} + \rho\left(\frac{2(1-p_{\mathrm{a}})b^2}{p_{\mathrm{a}}} + (1-b)^2\right)\right)\mathrm{E}\left[\frac{1}{n}\sum_{i=1}^n \left\|h_i^t - \nabla f_i(x^t)\right\|^2\right]$$

$$+ \left(\frac{20b^2\gamma\omega(2\omega+1)}{np_{\mathrm{a}}^2} + \nu\frac{2b^2}{np_{\mathrm{a}}} + \rho\frac{2b^2}{p_{\mathrm{a}}}\right)\frac{\sigma^2}{B}.$$

By taking $\nu = \frac{2\gamma}{b}$, one can show that $\left(\gamma + \nu(1-b)^2\right) \leq \left(1 - \frac{b}{2}\right)\nu$, and

$$\mathrm{E}\left[f(x^{t+1})\right] + (1-\gamma\mu)\frac{2\gamma(2\omega+1)}{p_{\mathrm{a}}}\mathrm{E}\left[\left\|g^{t+1} - h^{t+1}\right\|^2\right] + (1-\gamma\mu)\frac{4\gamma((2\omega+1)p_{\mathrm{a}} - p_{\mathrm{aa}})}{np_{\mathrm{a}}^2}\mathrm{E}\left[\frac{1}{n}\sum_{i=1}^n \left\|g_i^{t+1} - h_i^{t+1}\right\|^2\right]$$

$$+ \frac{2\gamma}{b}\mathrm{E}\left[\left\|h^{t+1} - \nabla f(x^{t+1})\right\|^2\right] + \rho\mathrm{E}\left[\frac{1}{n}\sum_{i=1}^n \left\|h_i^{t+1} - \nabla f_i(x^{t+1})\right\|^2\right]$$

$$\leq \mathrm{E}\left[f(x^t)\right] - \frac{\gamma}{2}\mathrm{E}\left[\left\|\nabla f(x^t)\right\|^2\right]$$

$$+ (1-\gamma\mu)\frac{2\gamma(2\omega+1)}{p_{\mathrm{a}}}\mathrm{E}\left[\left\|g^t - h^t\right\|^2\right] + (1-\gamma\mu)\frac{4\gamma((2\omega+1)p_{\mathrm{a}} - p_{\mathrm{aa}})}{np_{\mathrm{a}}^2}\mathrm{E}\left[\frac{1}{n}\sum_{i=1}^n \left\|g_i^t - h_i^t\right\|^2\right]$$

$$- \left(\frac{1}{2\gamma} - \frac{L}{2} - \frac{10\gamma\omega(2\omega+1)}{np_{\mathrm{a}}^2}\left(\frac{2(1-b)^2 L_\sigma^2}{B} + 2\widehat{L}^2\right)\right.$$

$$
-\frac{2\gamma}{b}\left(\frac{2(1-b)^2 L_\sigma^2}{np_{\mathrm{a}}B} + \frac{2\left(p_{\mathrm{a}} - p_{\mathrm{aa}}\right)\widehat{L}^2}{np_{\mathrm{a}}^2}\right) - \rho\left(\frac{2(1-b)^2 L_\sigma^2}{p_{\mathrm{a}}B} + \frac{2(1-p_{\mathrm{a}})\widehat{L}^2}{p_{\mathrm{a}}}\right)\right)\mathrm{E}\left[\left\|x^{t+1} - x^t\right\|^2\right]
$$

$$
+\left(1 - \frac{b}{2}\right)\frac{2\gamma}{b}\mathrm{E}\left[\left\|h^t - \nabla f(x^t)\right\|^2\right]
$$

$$
+\left(\frac{20b^2\gamma\omega(2\omega+1)}{np_{\mathrm{a}}^2} + \frac{4\gamma\left(p_{\mathrm{a}} - p_{\mathrm{aa}}\right)b}{np_{\mathrm{a}}^2} + \rho\left(\frac{2(1-p_{\mathrm{a}})b^2}{p_{\mathrm{a}}} + (1-b)^2\right)\right)\mathrm{E}\left[\frac{1}{n}\sum_{i=1}^{n}\left\|h_i^t - \nabla f_i(x^t)\right\|^2\right]
$$

$$
+\left(\frac{20b^2\gamma\omega(2\omega+1)}{np_{\mathrm{a}}^2} + \frac{4\gamma b}{np_{\mathrm{a}}} + \rho\frac{2b^2}{p_{\mathrm{a}}}\right)\frac{\sigma^2}{B}.
$$

Note that $b \le \frac{p_{\mathrm{a}}}{2 - p_{\mathrm{a}}}$, thus

$$
\left(\frac{20b^2\gamma\omega(2\omega+1)}{np_{\mathrm{a}}^2} + \frac{4\gamma\left(p_{\mathrm{a}} - p_{\mathrm{aa}}\right)b}{np_{\mathrm{a}}^2} + \rho\left(\frac{2(1-p_{\mathrm{a}})b^2}{p_{\mathrm{a}}} + (1-b)^2\right)\right)
$$

$$
\le \left(\frac{20b^2\gamma\omega(2\omega+1)}{np_{\mathrm{a}}^2} + \frac{4\gamma\left(p_{\mathrm{a}} - p_{\mathrm{aa}}\right)b}{np_{\mathrm{a}}^2} + \rho\left(1-b\right)\right).
$$

And if we take $\rho = \frac{40b\gamma\omega(2\omega+1)}{np_{\mathrm{a}}^2} + \frac{8\gamma(p_{\mathrm{a}} - p_{\mathrm{aa}})}{np_{\mathrm{a}}^2}$, then

$$
\left(\frac{20b^2\gamma\omega(2\omega+1)}{np_{\mathrm{a}}^2} + \frac{4\gamma\left(p_{\mathrm{a}} - p_{\mathrm{aa}}\right)b}{np_{\mathrm{a}}^2} + \rho\left(1-b\right)\right) \le \rho,
$$

and

$$
\mathrm{E}\left[f(x^{t+1})\right] + (1 - \gamma\mu)\frac{2\gamma(2\omega+1)}{p_{\mathrm{a}}}\mathrm{E}\left[\left\|g^{t+1} - h^{t+1}\right\|^2\right] + (1 - \gamma\mu)\frac{4\gamma((2\omega+1)p_{\mathrm{a}} - p_{\mathrm{aa}})}{np_{\mathrm{a}}^2}\mathrm{E}\left[\frac{1}{n}\sum_{i=1}^{n}\left\|g_i^{t+1} - h_i^{t+1}\right\|^2\right]
$$

$$
+\frac{2\gamma}{b}\mathrm{E}\left[\left\|h^{t+1} - \nabla f(x^{t+1})\right\|^2\right] + \left(\frac{40b\gamma\omega(2\omega+1)}{np_{\mathrm{a}}^2} + \frac{8\gamma\left(p_{\mathrm{a}} - p_{\mathrm{aa}}\right)}{np_{\mathrm{a}}^2}\right)\mathrm{E}\left[\frac{1}{n}\sum_{i=1}^{n}\left\|h_i^{t+1} - \nabla f_i(x^{t+1})\right\|^2\right]
$$

$$
\le \mathrm{E}\left[f(x^t)\right] - \frac{\gamma}{2}\mathrm{E}\left[\left\|\nabla f(x^t)\right\|^2\right]
$$

$$
+ (1 - \gamma\mu)\frac{2\gamma(2\omega+1)}{p_{\mathrm{a}}}\mathrm{E}\left[\left\|g^t - h^t\right\|^2\right] + (1 - \gamma\mu)\frac{4\gamma((2\omega+1)p_{\mathrm{a}} - p_{\mathrm{aa}})}{np_{\mathrm{a}}^2}\mathrm{E}\left[\frac{1}{n}\sum_{i=1}^{n}\left\|g_i^t - h_i^t\right\|^2\right]
$$

$$
- \left(\frac{1}{2\gamma} - \frac{L}{2} - \frac{10\gamma\omega(2\omega+1)}{np_{\mathrm{a}}^2}\left(\frac{2(1-b)^2 L_\sigma^2}{B} + 2\widehat{L}^2\right)\right.
$$

$$
- \frac{2\gamma}{np_{\mathrm{a}}b}\left(\frac{2(1-b)^2 L_\sigma^2}{B} + 2\left(1 - \frac{p_{\mathrm{aa}}}{p_{\mathrm{a}}}\right)\widehat{L}^2\right)
$$

$$
- \left(\frac{40b\gamma\omega(2\omega+1)}{np_{\mathrm{a}}^3} + \frac{8\gamma\left(1 - \frac{p_{\mathrm{aa}}}{p_{\mathrm{a}}}\right)}{np_{\mathrm{a}}^2}\right)\left(\frac{2(1-b)^2 L_\sigma^2}{B} + 2(1-p_{\mathrm{a}})\widehat{L}^2\right)\right)\mathrm{E}\left[\left\|x^{t+1} - x^t\right\|^2\right]
$$

$$
+\left(1 - \frac{b}{2}\right)\frac{2\gamma}{b}\mathrm{E}\left[\left\|h^t - \nabla f(x^t)\right\|^2\right] + \left(1 - \frac{b}{2}\right)\left(\frac{40b\gamma\omega(2\omega+1)}{np_{\mathrm{a}}^2} + \frac{8\gamma\left(p_{\mathrm{a}} - p_{\mathrm{aa}}\right)}{np_{\mathrm{a}}^2}\right)\mathrm{E}\left[\frac{1}{n}\sum_{i=1}^{n}\left\|h_i^t - \nabla f_i(x^t)\right\|^2\right]
$$

$$
+\left(\frac{20b^2\gamma\omega(2\omega+1)}{np_{\mathrm{a}}^2} + \frac{4\gamma b}{np_{\mathrm{a}}} + \left(\frac{40b\gamma\omega(2\omega+1)}{np_{\mathrm{a}}^2} + \frac{8\gamma\left(p_{\mathrm{a}} - p_{\mathrm{aa}}\right)}{np_{\mathrm{a}}^2}\right)\frac{2b^2}{p_{\mathrm{a}}}\right)\frac{\sigma^2}{B}.
$$

Let us simplify the inequality. First, due to $b \le p_{\mathrm{a}}$ and $(1 - p_{\mathrm{a}}) \le \left(1 - \frac{p_{\mathrm{aa}}}{p_{\mathrm{a}}}\right)$, we have

$$
\left(\frac{40b\gamma\omega(2\omega+1)}{np_{\mathrm{a}}^3} + \frac{2\gamma\left(1 - \frac{p_{\mathrm{aa}}}{p_{\mathrm{a}}}\right)}{np_{\mathrm{a}}^2}\right)\left(\frac{2(1-b)^2 L_\sigma^2}{B} + 8(1 - p_{\mathrm{a}})\widehat{L}^2\right)
$$

$$= \frac{40b\gamma\omega(2\omega+1)}{np_{\mathrm{a}}^3}\left(\frac{2(1-b)^2 L_\sigma^2}{B}+2(1-p_{\mathrm{a}})\widehat{L}^2\right)$$

$$+\frac{8\gamma\left(1-\frac{p_{\mathrm{aa}}}{p_{\mathrm{a}}}\right)}{np_{\mathrm{a}}^2}\left(\frac{2(1-b)^2 L_\sigma^2}{B}+2(1-p_{\mathrm{a}})\widehat{L}^2\right)$$

$$\leq \frac{40\gamma\omega(2\omega+1)}{np_{\mathrm{a}}^2}\left(\frac{2(1-b)^2 L_\sigma^2}{B}+2\widehat{L}^2\right)$$

$$+\frac{8\gamma}{np_{\mathrm{a}}b}\left(\frac{2(1-b)^2 L_\sigma^2}{B}+2\left(1-\frac{p_{\mathrm{aa}}}{p_{\mathrm{a}}}\right)\widehat{L}^2\right),$$

therefore

$$\mathrm{E}\left[f(x^{t+1})\right]+(1-\gamma\mu)\frac{2\gamma(2\omega+1)}{p_{\mathrm{a}}}\mathrm{E}\left[\left\|g^{t+1}-h^{t+1}\right\|^2\right]+(1-\gamma\mu)\frac{4\gamma((2\omega+1)p_{\mathrm{a}}-p_{\mathrm{aa}})}{np_{\mathrm{a}}^2}\mathrm{E}\left[\frac{1}{n}\sum_{i=1}^n\left\|g_i^{t+1}-h_i^{t+1}\right\|^2\right]$$

$$+\frac{2\gamma}{b}\mathrm{E}\left[\left\|h^{t+1}-\nabla f(x^{t+1})\right\|^2\right]+\left(\frac{40b\gamma\omega(2\omega+1)}{np_{\mathrm{a}}^2}+\frac{8\gamma(p_{\mathrm{a}}-p_{\mathrm{aa}})}{np_{\mathrm{a}}^2}\right)\mathrm{E}\left[\frac{1}{n}\sum_{i=1}^n\left\|h_i^{t+1}-\nabla f_i(x^{t+1})\right\|^2\right]$$

$$\leq \mathrm{E}\left[f(x^t)\right]-\frac{\gamma}{2}\mathrm{E}\left[\left\|\nabla f(x^t)\right\|^2\right]$$

$$+(1-\gamma\mu)\frac{2\gamma(2\omega+1)}{p_{\mathrm{a}}}\mathrm{E}\left[\left\|g^t-h^t\right\|^2\right]+(1-\gamma\mu)\frac{4\gamma((2\omega+1)p_{\mathrm{a}}-p_{\mathrm{aa}})}{np_{\mathrm{a}}^2}\mathrm{E}\left[\frac{1}{n}\sum_{i=1}^n\left\|g_i^t-h_i^t\right\|^2\right]$$

$$-\left(\frac{1}{2\gamma}-\frac{L}{2}-\frac{50\gamma\omega(2\omega+1)}{np_{\mathrm{a}}^2}\left(\frac{2(1-b)^2 L_\sigma^2}{B}+2\widehat{L}^2\right)\right.$$

$$\left.-\frac{10\gamma}{np_{\mathrm{a}}b}\left(\frac{2(1-b)^2 L_\sigma^2}{B}+2\left(1-\frac{p_{\mathrm{aa}}}{p_{\mathrm{a}}}\right)\widehat{L}^2\right)\right)\mathrm{E}\left[\left\|x^{t+1}-x^t\right\|^2\right]$$

$$+\left(1-\frac{b}{2}\right)\frac{2\gamma}{b}\mathrm{E}\left[\left\|h^t-\nabla f(x^t)\right\|^2\right]+\left(1-\frac{b}{2}\right)\left(\frac{40b\gamma\omega(2\omega+1)}{np_{\mathrm{a}}^2}+\frac{8\gamma(p_{\mathrm{a}}-p_{\mathrm{aa}})}{np_{\mathrm{a}}^2}\right)\mathrm{E}\left[\frac{1}{n}\sum_{i=1}^n\left\|h_i^t-\nabla f_i(x^t)\right\|^2\right]$$

$$+\left(\frac{20b^2\gamma\omega(2\omega+1)}{np_{\mathrm{a}}^2}+\frac{4\gamma b}{np_{\mathrm{a}}}+\left(\frac{40b\gamma\omega(2\omega+1)}{np_{\mathrm{a}}^2}+\frac{8\gamma(p_{\mathrm{a}}-p_{\mathrm{aa}})}{np_{\mathrm{a}}^2}\right)\frac{2b^2}{p_{\mathrm{a}}}\right)\frac{\sigma^2}{B}$$

$$\leq \mathrm{E}\left[f(x^t)\right]-\frac{\gamma}{2}\mathrm{E}\left[\left\|\nabla f(x^t)\right\|^2\right]$$

$$+(1-\gamma\mu)\frac{2\gamma(2\omega+1)}{p_{\mathrm{a}}}\mathrm{E}\left[\left\|g^t-h^t\right\|^2\right]+(1-\gamma\mu)\frac{4\gamma((2\omega+1)p_{\mathrm{a}}-p_{\mathrm{aa}})}{np_{\mathrm{a}}^2}\mathrm{E}\left[\frac{1}{n}\sum_{i=1}^n\left\|g_i^t-h_i^t\right\|^2\right]$$

$$-\left(\frac{1}{2\gamma}-\frac{L}{2}-\frac{100\gamma\omega(2\omega+1)}{np_{\mathrm{a}}^2}\left(\frac{(1-b)^2 L_\sigma^2}{B}+\widehat{L}^2\right)\right.$$

$$\left.-\frac{20\gamma}{np_{\mathrm{a}}b}\left(\frac{(1-b)^2 L_\sigma^2}{B}+\left(1-\frac{p_{\mathrm{aa}}}{p_{\mathrm{a}}}\right)\widehat{L}^2\right)\right)\mathrm{E}\left[\left\|x^{t+1}-x^t\right\|^2\right]$$

$$+\left(1-\frac{b}{2}\right)\frac{2\gamma}{b}\mathrm{E}\left[\left\|h^t-\nabla f(x^t)\right\|^2\right]+\left(1-\frac{b}{2}\right)\left(\frac{40b\gamma\omega(2\omega+1)}{np_{\mathrm{a}}^2}+\frac{8\gamma(p_{\mathrm{a}}-p_{\mathrm{aa}})}{np_{\mathrm{a}}^2}\right)\mathrm{E}\left[\frac{1}{n}\sum_{i=1}^n\left\|h_i^t-\nabla f_i(x^t)\right\|^2\right]$$

$$+\left(\frac{20b^2\gamma\omega(2\omega+1)}{np_{\mathrm{a}}^2}+\frac{4\gamma b}{np_{\mathrm{a}}}+\left(\frac{40b\gamma\omega(2\omega+1)}{np_{\mathrm{a}}^2}+\frac{8\gamma(p_{\mathrm{a}}-p_{\mathrm{aa}})}{np_{\mathrm{a}}^2}\right)\frac{2b^2}{p_{\mathrm{a}}}\right)\frac{\sigma^2}{B}.$$

Also, we can simplify the last term:

$$\left(\frac{40b\gamma\omega(2\omega+1)}{np_{\mathrm{a}}^2}+\frac{8\gamma(p_{\mathrm{a}}-p_{\mathrm{aa}})}{np_{\mathrm{a}}^2}\right)\frac{2b^2}{p_{\mathrm{a}}}$$

$$=\frac{80b^3\gamma\omega(2\omega+1)}{np_{\mathrm{a}}^3}+\frac{16b^2\gamma\left(1-\frac{p_{\mathrm{aa}}}{p_{\mathrm{a}}}\right)}{np_{\mathrm{a}}^2}$$

$$\leq \frac{80b^2\gamma\omega(2\omega+1)}{np_{\mathrm{a}}^2} + \frac{16b\gamma}{np_{\mathrm{a}}},$$

thus

$$\mathrm{E}\left[f(x^{t+1})\right] + (1-\gamma\mu)\frac{2\gamma(2\omega+1)}{p_{\mathrm{a}}}\mathrm{E}\left[\left\|g^{t+1} - h^{t+1}\right\|^2\right] + (1-\gamma\mu)\frac{4\gamma((2\omega+1)p_{\mathrm{a}} - p_{\mathrm{aa}})}{np_{\mathrm{a}}^2}\mathrm{E}\left[\frac{1}{n}\sum_{i=1}^{n}\left\|g_i^{t+1} - h_i^{t+1}\right\|^2\right]$$

$$+ \frac{2\gamma}{b}\mathrm{E}\left[\left\|h^{t+1} - \nabla f(x^{t+1})\right\|^2\right] + \left(\frac{40b\gamma\omega(2\omega+1)}{np_{\mathrm{a}}^2} + \frac{8\gamma(p_{\mathrm{a}} - p_{\mathrm{aa}})}{np_{\mathrm{a}}^2}\right)\mathrm{E}\left[\frac{1}{n}\sum_{i=1}^{n}\left\|h_i^{t+1} - \nabla f_i(x^{t+1})\right\|^2\right]$$

$$\leq \mathrm{E}\left[f(x^t)\right] - \frac{\gamma}{2}\mathrm{E}\left[\left\|\nabla f(x^t)\right\|^2\right]$$

$$+ (1-\gamma\mu)\frac{2\gamma(2\omega+1)}{p_{\mathrm{a}}}\mathrm{E}\left[\left\|g^t - h^t\right\|^2\right] + (1-\gamma\mu)\frac{4\gamma((2\omega+1)p_{\mathrm{a}} - p_{\mathrm{aa}})}{np_{\mathrm{a}}^2}\mathrm{E}\left[\frac{1}{n}\sum_{i=1}^{n}\left\|g_i^t - h_i^t\right\|^2\right]$$

$$- \left(\frac{1}{2\gamma} - \frac{L}{2} - \frac{100\gamma\omega(2\omega+1)}{np_{\mathrm{a}}^2}\left(\frac{(1-b)^2 L_\sigma^2}{B} + \widehat{L}^2\right)\right.$$

$$\left. - \frac{20\gamma}{np_{\mathrm{a}}b}\left(\frac{(1-b)^2 L_\sigma^2}{B} + \left(1 - \frac{p_{\mathrm{aa}}}{p_{\mathrm{a}}}\right)\widehat{L}^2\right)\right)\mathrm{E}\left[\left\|x^{t+1} - x^t\right\|^2\right]$$

$$+ \left(1 - \frac{b}{2}\right)\frac{2\gamma}{b}\mathrm{E}\left[\left\|h^t - \nabla f(x^t)\right\|^2\right] + \left(1 - \frac{b}{2}\right)\left(\frac{40b\gamma\omega(2\omega+1)}{np_{\mathrm{a}}^2} + \frac{8\gamma(p_{\mathrm{a}} - p_{\mathrm{aa}})}{np_{\mathrm{a}}^2}\right)\mathrm{E}\left[\frac{1}{n}\sum_{i=1}^{n}\left\|h_i^t - \nabla f_i(x^t)\right\|^2\right]$$

$$+ \left(\frac{100b^2\gamma\omega(2\omega+1)}{np_{\mathrm{a}}^2} + \frac{20\gamma b}{np_{\mathrm{a}}}\right)\frac{\sigma^2}{B}.$$

Using Lemma [4] and the assumption about $\gamma$, we get

$$\mathrm{E}\left[f(x^{t+1})\right] + (1-\gamma\mu)\frac{2\gamma(2\omega+1)}{p_{\mathrm{a}}}\mathrm{E}\left[\left\|g^{t+1} - h^{t+1}\right\|^2\right] + (1-\gamma\mu)\frac{4\gamma((2\omega+1)p_{\mathrm{a}} - p_{\mathrm{aa}})}{np_{\mathrm{a}}^2}\mathrm{E}\left[\frac{1}{n}\sum_{i=1}^{n}\left\|g_i^{t+1} - h_i^{t+1}\right\|^2\right]$$

$$+ \frac{2\gamma}{b}\mathrm{E}\left[\left\|h^{t+1} - \nabla f(x^{t+1})\right\|^2\right] + \left(\frac{40b\gamma\omega(2\omega+1)}{np_{\mathrm{a}}^2} + \frac{8\gamma(p_{\mathrm{a}} - p_{\mathrm{aa}})}{np_{\mathrm{a}}^2}\right)\mathrm{E}\left[\frac{1}{n}\sum_{i=1}^{n}\left\|h_i^{t+1} - \nabla f_i(x^{t+1})\right\|^2\right]$$

$$\leq \mathrm{E}\left[f(x^t)\right] - \frac{\gamma}{2}\mathrm{E}\left[\left\|\nabla f(x^t)\right\|^2\right]$$

$$+ (1-\gamma\mu)\frac{2\gamma(2\omega+1)}{p_{\mathrm{a}}}\mathrm{E}\left[\left\|g^t - h^t\right\|^2\right] + (1-\gamma\mu)\frac{4\gamma((2\omega+1)p_{\mathrm{a}} - p_{\mathrm{aa}})}{np_{\mathrm{a}}^2}\mathrm{E}\left[\frac{1}{n}\sum_{i=1}^{n}\left\|g_i^t - h_i^t\right\|^2\right]$$

$$+ \left(1 - \frac{b}{2}\right)\frac{2\gamma}{b}\mathrm{E}\left[\left\|h^t - \nabla f(x^t)\right\|^2\right] + \left(1 - \frac{b}{2}\right)\left(\frac{40b\gamma\omega(2\omega+1)}{np_{\mathrm{a}}^2} + \frac{8\gamma(p_{\mathrm{a}} - p_{\mathrm{aa}})}{np_{\mathrm{a}}^2}\right)\mathrm{E}\left[\frac{1}{n}\sum_{i=1}^{n}\left\|h_i^t - \nabla f_i(x^t)\right\|^2\right]$$

$$+ \left(\frac{100b^2\gamma\omega(2\omega+1)}{np_{\mathrm{a}}^2} + \frac{20\gamma b}{np_{\mathrm{a}}}\right)\frac{\sigma^2}{B}.$$

Note that $\gamma \leq \frac{b}{2\mu}$, thus $1 - \frac{b}{2} \leq 1 - \gamma\mu$ and

$$\mathrm{E}\left[f(x^{t+1})\right] + (1-\gamma\mu)\frac{2\gamma(2\omega+1)}{p_{\mathrm{a}}}\mathrm{E}\left[\left\|g^{t+1} - h^{t+1}\right\|^2\right] + (1-\gamma\mu)\frac{4\gamma((2\omega+1)p_{\mathrm{a}} - p_{\mathrm{aa}})}{np_{\mathrm{a}}^2}\mathrm{E}\left[\frac{1}{n}\sum_{i=1}^{n}\left\|g_i^{t+1} - h_i^{t+1}\right\|^2\right]$$

$$+ \frac{2\gamma}{b}\mathrm{E}\left[\left\|h^{t+1} - \nabla f(x^{t+1})\right\|^2\right] + \left(\frac{40b\gamma\omega(2\omega+1)}{np_{\mathrm{a}}^2} + \frac{8\gamma(p_{\mathrm{a}} - p_{\mathrm{aa}})}{np_{\mathrm{a}}^2}\right)\mathrm{E}\left[\frac{1}{n}\sum_{i=1}^{n}\left\|h_i^{t+1} - \nabla f_i(x^{t+1})\right\|^2\right]$$

$$\leq \mathrm{E}\left[f(x^t)\right] - \frac{\gamma}{2}\mathrm{E}\left[\left\|\nabla f(x^t)\right\|^2\right]$$

$$+ (1-\gamma\mu)\frac{2\gamma(2\omega+1)}{p_{\mathrm{a}}}\mathrm{E}\left[\left\|g^t - h^t\right\|^2\right] + (1-\gamma\mu)\frac{4\gamma((2\omega+1)p_{\mathrm{a}} - p_{\mathrm{aa}})}{np_{\mathrm{a}}^2}\mathrm{E}\left[\frac{1}{n}\sum_{i=1}^{n}\left\|g_i^t - h_i^t\right\|^2\right]$$

$$+ (1 - \gamma\mu) \frac{2\gamma}{b} \mathrm{E}\left[ \left\| h^t - \nabla f(x^t) \right\|^2 \right] + (1 - \gamma\mu) \left( \frac{40 b\gamma\omega(2\omega + 1)}{np_\mathrm{a}^2} + \frac{8\gamma\left(p_\mathrm{a} - p_\mathrm{aa}\right)}{np_\mathrm{a}^2} \right) \mathrm{E}\left[ \frac{1}{n} \sum_{i=1}^{n} \left\| h_i^t - \nabla f_i(x^t) \right\|^2 \right]$$

$$+ \left( \frac{100 b^2 \gamma\omega(2\omega + 1)}{np_\mathrm{a}^2} + \frac{20\gamma b}{np_\mathrm{a}} \right) \frac{\sigma^2}{B}.$$

It is left to apply Lemma 11 with

$$
\begin{aligned}
\Psi^t \quad &= \quad \frac{2(2\omega + 1)}{p_\mathrm{a}} \mathrm{E}\left[ \left\| g^t - h^t \right\|^2 \right] + \frac{4((2\omega + 1)\, p_\mathrm{a} - p_\mathrm{aa})}{np_\mathrm{a}^2} \mathrm{E}\left[ \frac{1}{n} \sum_{i=1}^{n} \left\| g_i^t - h_i^t \right\|^2 \right] \\
&+ \quad \frac{2}{b} \mathrm{E}\left[ \left\| h^t - \nabla f(x^t) \right\|^2 \right] + \left( \frac{40 b\omega(2\omega + 1)}{np_\mathrm{a}^2} + \frac{8\left(p_\mathrm{a} - p_\mathrm{aa}\right)}{np_\mathrm{a}^2} \right) \mathrm{E}\left[ \frac{1}{n} \sum_{i=1}^{n} \left\| h_i^t - \nabla f_i(x^t) \right\|^2 \right]
\end{aligned}
$$

and $C = \left( \frac{100 b^2 \omega(2\omega + 1)}{p_\mathrm{a}^2} + \frac{20 b}{p_\mathrm{a}} \right) \frac{\sigma^2}{nB}$ to conclude the proof. $\qquad\square$

**Corollary 5.** *Suppose that assumptions of Theorem 10 hold, batch size $B \le \min\left\{ \frac{\sigma}{p_\mathrm{a}\sqrt{\mu\varepsilon n}}, \frac{L_\sigma^2}{\widehat{L}^2} \right\}$, we take RandK compressors with $K = \Theta\left( \frac{Bd\sqrt{\mu\varepsilon n}}{\sigma} \right)$. Then the communication complexity equals*

$$\widetilde{\mathcal{O}}\left( \frac{d\sigma}{p_\mathrm{a}\sqrt{\mu\varepsilon n}} + \frac{dL_\sigma}{p_\mathrm{a}\mu\sqrt{n}} \right),$$

*and the expected number of stochastic gradient calculations per node equals*

$$\widetilde{\mathcal{O}}\left( \frac{\sigma^2}{p_\mathrm{a}\mu n\varepsilon} + \frac{\sigma L_\sigma}{p_\mathrm{a} n\mu^{3/2}\sqrt{\varepsilon}} \right).$$

*Proof.* In the view of Theorem 10, DASHA-PP have to run

$$\widetilde{\mathcal{O}}\left( \frac{\omega + 1}{p_\mathrm{a}} + \frac{\omega}{p_\mathrm{a}} \sqrt{\frac{\sigma^2}{\mu n\varepsilon B}} + \frac{\sigma^2}{p_\mathrm{a}\mu n\varepsilon B} + \frac{L}{\mu} + \frac{\omega}{p_\mathrm{a}\mu\sqrt{n}}\left( \widehat{L} + \frac{L_\sigma}{\sqrt{B}} \right) + \frac{\sigma}{p_\mathrm{a} n\mu^{3/2}\sqrt{\varepsilon B}}\left( \widehat{L} + \frac{L_\sigma}{\sqrt{B}} \right) \right)$$

communication rounds in the stochastic settings to get $\varepsilon$-solution. Note that $K = \mathcal{O}\left( \frac{d}{p_\mathrm{a}\sqrt{n}} \right)$. Moreover, we can skip the initialization procedure and initialize $h_i^0$ and $g_i^0$, for instance, with zeros because the initialization error is under a logarithm. Considering Theorem 6, the communication complexity equals

$$\widetilde{\mathcal{O}}\left( K\frac{\omega + 1}{p_\mathrm{a}} + K\frac{\omega}{p_\mathrm{a}} \sqrt{\frac{\sigma^2}{\mu n\varepsilon B}} + K\frac{\sigma^2}{p_\mathrm{a}\mu n\varepsilon B} + K\frac{L}{\mu} + K\frac{\omega}{p_\mathrm{a}\mu\sqrt{n}}\left( \widehat{L} + \frac{L_\sigma}{\sqrt{B}} \right) + K\frac{\sigma}{p_\mathrm{a} n\mu^{3/2}\sqrt{\varepsilon B}}\left( \widehat{L} + \frac{L_\sigma}{\sqrt{B}} \right) \right)$$

$$= \widetilde{\mathcal{O}}\left( K\frac{\omega + 1}{p_\mathrm{a}} + K\frac{\omega}{p_\mathrm{a}} \sqrt{\frac{\sigma^2}{\mu n\varepsilon B}} + K\frac{\sigma^2}{p_\mathrm{a}\mu n\varepsilon B} + K\frac{L}{\mu} + K\frac{\omega}{p_\mathrm{a}\mu\sqrt{n}}\left( \widehat{L} + \frac{L_\sigma}{\sqrt{B}} \right) + K\frac{\sigma L_\sigma}{p_\mathrm{a} n\mu^{3/2}\sqrt{\varepsilon B}} \right)$$

$$= \widetilde{\mathcal{O}}\left( \frac{d}{p_\mathrm{a}} + \frac{d}{p_\mathrm{a}} \sqrt{\frac{\sigma^2}{\mu n\varepsilon B}} + \frac{K\sigma^2}{p_\mathrm{a}\mu n\varepsilon B} + \frac{dL}{p_\mathrm{a}\mu\sqrt{n}} + \frac{d}{p_\mathrm{a}\mu\sqrt{n}}\left( \widehat{L} + \frac{L_\sigma}{\sqrt{B}} \right) + \frac{K\sigma L_\sigma}{p_\mathrm{a} n\mu^{3/2}\sqrt{\varepsilon B}} \right)$$

$$= \widetilde{\mathcal{O}}\left( \frac{d}{p_\mathrm{a}} + \frac{d\sigma}{p_\mathrm{a}\sqrt{\mu n\varepsilon B}} + \frac{d\sigma}{p_\mathrm{a}\sqrt{\mu\varepsilon n}} + \frac{dL}{p_\mathrm{a}\mu\sqrt{n}} + \frac{d}{p_\mathrm{a}\mu\sqrt{n}}\left( \widehat{L} + \frac{L_\sigma}{\sqrt{B}} \right) + \frac{dL_\sigma}{p_\mathrm{a}\mu\sqrt{n}} \right)$$

$$= \widetilde{\mathcal{O}}\left( \frac{d\sigma}{p_\mathrm{a}\sqrt{\mu\varepsilon n}} + \frac{dL_\sigma}{p_\mathrm{a}\mu\sqrt{n}} \right).$$

The expected number of stochastic gradient calculations per node equals

$$\widetilde{\mathcal{O}}\left( B\frac{\omega + 1}{p_\mathrm{a}} + B\frac{\omega}{p_\mathrm{a}} \sqrt{\frac{\sigma^2}{\mu n\varepsilon B}} + B\frac{\sigma^2}{p_\mathrm{a}\mu n\varepsilon B} + B\frac{L}{\mu} + B\frac{\omega}{p_\mathrm{a}\mu\sqrt{n}}\left( \widehat{L} + \frac{L_\sigma}{\sqrt{B}} \right) + B\frac{\sigma}{p_\mathrm{a} n\mu^{3/2}\sqrt{\varepsilon B}}\left( \widehat{L} + \frac{L_\sigma}{\sqrt{B}} \right) \right)$$

$$= \widetilde{\mathcal{O}}\left( B\frac{\omega+1}{p_{\mathrm{a}}} + B\frac{\omega}{p_{\mathrm{a}}}\sqrt{\frac{\sigma^2}{\mu n \varepsilon B}} + B\frac{\sigma^2}{p_{\mathrm{a}}\mu n \varepsilon B} + B\frac{L}{\mu} + B\frac{\omega}{p_{\mathrm{a}}\mu\sqrt{n}}\left(\widehat{L} + \frac{L_\sigma}{\sqrt{B}}\right) + B\frac{\sigma}{p_{\mathrm{a}}n\mu^{3/2}\sqrt{\varepsilon B}}\left(\frac{L_\sigma}{\sqrt{B}}\right)\right)$$

$$= \widetilde{\mathcal{O}}\left( \frac{Bd}{Kp_{\mathrm{a}}} + \frac{Bd}{Kp_{\mathrm{a}}}\sqrt{\frac{\sigma^2}{\mu n \varepsilon B}} + \frac{\sigma^2}{p_{\mathrm{a}}\mu n \varepsilon} + B\frac{L}{\mu} + \frac{Bd}{Kp_{\mathrm{a}}\mu\sqrt{n}}\left(\widehat{L} + \frac{L_\sigma}{\sqrt{B}}\right) + \frac{\sigma L_\sigma}{p_{\mathrm{a}}n\mu^{3/2}\sqrt{\varepsilon}}\right)$$

$$= \widetilde{\mathcal{O}}\left( \frac{\sigma}{p_{\mathrm{a}}\sqrt{\mu\varepsilon n}} + \frac{\sigma^2}{p_{\mathrm{a}}\mu\varepsilon n\sqrt{B}} + \frac{\sigma^2}{p_{\mathrm{a}}\mu n \varepsilon} + \frac{\sigma L}{p_{\mathrm{a}}\mu^{3/2}\sqrt{\varepsilon}n} + \frac{\sigma}{p_{\mathrm{a}}\mu^{3/2}\sqrt{\varepsilon}n}\left(\widehat{L} + \frac{L_\sigma}{\sqrt{B}}\right) + \frac{\sigma L_\sigma}{p_{\mathrm{a}}n\mu^{3/2}\sqrt{\varepsilon}}\right)$$

$$= \widetilde{\mathcal{O}}\left( \frac{\sigma^2}{p_{\mathrm{a}}\mu n \varepsilon} + \frac{\sigma L_\sigma}{p_{\mathrm{a}}n\mu^{3/2}\sqrt{\varepsilon}}\right).$$

$\square$

# G   Description of DASHA-PP-SYNC-MVR

By analogy to (Tyurin and Richtárik, 2023), we provide a "synchronized" version of the algorithm. With a small probability, participating nodes calculate and send a mega batch without compression. This helps us to resolve the suboptimality of DASHA-PP-MVR w.r.t. $\omega$. Note that this suboptimality is not a problem. We show in Corollary 4 that DASHA-PP-MVR can have the optimal oracle complexity and SOTA communication complexity with the particular choices of parameters of the compressors.

---

**Algorithm 8** DASHA-PP-SYNC-MVR

1: **Input:** starting point $x^0 \in \mathbb{R}^d$, stepsize $\gamma > 0$, momentum $a \in (0,1]$, momentum $b \in (0,1]$, probability $p_{\text{mega}} \in (0,1]$, batch size $B'$ and $B$, probability $p_{\text{a}} \in (0,1]$ that a node is *participating*[(a)], number of iterations $T \geq 1$.
2: Initialize $g_i^0, h_i^0$ on the nodes and $g^0 = \frac{1}{n}\sum_{i=1}^n g_i^0$ on the server
3: **for** $t = 0, 1, \ldots, T-1$ **do**
4:     $x^{t+1} = x^t - \gamma g^t$
5:     $c^{t+1} = \begin{cases} 1, \text{with probability } p_{\text{mega}}, \\ 0, \text{with probability } 1 - p_{\text{mega}} \end{cases}$
6:     Broadcast $x^{t+1}, x^t$ to all *participating*[(a)] nodes
7:     **for** $i = 1, \ldots, n$ in parallel **do**
8:         **if** $i^{\text{th}}$ node is *participating*[(a)] **then**
9:             **if** $c^{t+1} = 1$ **then**
10:                 Generate i.i.d. samples $\{\xi_{ik}^{t+1}\}_{k=1}^{B'}$ of size $B'$ from $\mathcal{D}_i$.
11:                 $k_i^{t+1} = \frac{1}{B'}\sum_{k=1}^{B'} \nabla f_i(x^{t+1};\xi_{ik}^{t+1}) - \frac{1}{B'}\sum_{k=1}^{B'} \nabla f_i(x^t;\xi_{ik}^{t+1}) - \frac{b}{p_{\text{mega}}}\left(h_i^t - \frac{1}{B'}\sum_{k=1}^{B'}\nabla f_i(x^t;\xi_{ik}^{t+1})\right)$
12:                 $m_i^{t+1} = \frac{1}{p_{\text{a}}}k_i^{t+1} - \frac{a}{p_{\text{a}}}(g_i^t - h_i^t)$
13:             **else**
14:                 Generate i.i.d. samples $\{\xi_{ij}^{t+1}\}_{j=1}^{B}$ of size $B$ from $\mathcal{D}_i$.
15:                 $k_i^{t+1} = \frac{1}{B}\sum_{j=1}^{B} \nabla f_i(x^{t+1};\xi_{ij}^{t+1}) - \frac{1}{B}\sum_{j=1}^{B}\nabla f_i(x^t;\xi_{ij}^{t+1})$
16:                 $m_i^{t+1} = \mathcal{C}_i\left(\frac{1}{p_{\text{a}}}k_i^{t+1} - \frac{a}{p_{\text{a}}}(g_i^t - h_i^t)\right)$
17:             **end if**
18:             $h_i^{t+1} = h_i^t + \frac{1}{p_{\text{a}}}k_i^{t+1}$
19:             $g_i^{t+1} = g_i^t + m_i^{t+1}$
20:             Send $m_i^{t+1}$ to the server
21:         **else**
22:             $h_i^{t+1} = h_i^t$
23:             $m_i^{t+1} = 0$
24:             $g_i^{t+1} = g_i^t$
25:         **end if**
26:     **end for**
27:     $g^{t+1} = g^t + \frac{1}{n}\sum_{i=1}^n m_i^{t+1}$
28: **end for**
29: **Output:** $\hat{x}^T$ chosen uniformly at random from $\{x^t\}_{k=0}^{T-1}$
   (a): For the formal description see Section 2.2.

---

In the following theorem, we provide the convergence rate of DASHA-PP-SYNC-MVR.

**Theorem 11.** *Suppose that Assumptions 1, 2, 3, 5, 6, 7 and 8 hold. Let us take* $a = \frac{p_{\text{a}}}{2\omega+1}$, $b = \frac{p_{mega}p_{\text{a}}}{2-p_{\text{a}}}$, *probability* $p_{mega} \in (0,1]$, *batch size* $B' \geq B \geq 1$

$$\gamma \leq \left(L + \sqrt{\frac{8(2\omega+1)\omega}{np_{\text{a}}^2}\left(\widehat{L}^2 + \frac{L_\sigma^2}{B}\right) + \frac{16}{np_{mega}p_{\text{a}}^2}\left(\left(1 - \frac{p_{\text{aa}}}{p_{\text{a}}}\right)\widehat{L}^2 + \frac{L_\sigma^2}{B}\right)}\right)^{-1},$$

*and $h_i^0 = g_i^0$ for all $i \in [n]$ in Algorithm 8. Then*

$$\mathrm{E}\left[\left\|\nabla f(\widehat{x}^T)\right\|^2\right] \leq \frac{1}{T}\left[\frac{2\Delta_0}{\gamma} + \frac{4}{p_{mega}p_{\mathrm{a}}}\left\|h^0 - \nabla f(x^0)\right\|^2 + \frac{4\left(1 - \frac{p_{\mathrm{aa}}}{p_{\mathrm{a}}}\right)}{np_{mega}p_{\mathrm{a}}}\frac{1}{n}\sum_{i=1}^{n}\left\|h_i^0 - \nabla f_i(x^0)\right\|^2\right]$$

$$+ \frac{12\sigma^2}{nB'}.$$

First, we introduce the expected density of compressors (Gorbunov et al., 2021; Tyurin and Richtárik, 2023).

**Definition 12.** The expected density of the compressor $\mathcal{C}_i$ is $\zeta_{\mathcal{C}_i} := \sup_{x \in \mathbb{R}^d} \mathrm{E}\left[\left\|\mathcal{C}_i(x)\right\|_0\right]$, where $\|x\|_0$ is the number of nonzero components of $x \in \mathbb{R}^d$. Let $\zeta_{\mathcal{C}} = \max_{i \in [n]} \zeta_{\mathcal{C}_i}$.

Note that $\zeta_{\mathcal{C}}$ is finite and $\zeta_{\mathcal{C}} \leq d$.

In the next corollary, we choose particular algorithm parameters to reveal the communication and oracle complexity.

**Corollary 6.** *Suppose that assumptions from Theorem 11 hold, probability $p_{mega} = \min\left\{\frac{\zeta_{\mathcal{C}}}{d}, \frac{n\varepsilon B}{\sigma^2}\right\}$, batch size $B' = \Theta\left(\frac{\sigma^2}{n\varepsilon}\right)$, and $h_i^0 = g_i^0 = \frac{1}{B_{\mathrm{init}}}\sum_{k=1}^{B_{\mathrm{init}}}\nabla f_i(x^0; \xi_{ik}^0)$ for all $i \in [n]$, initial batch size $B_{\mathrm{init}} = \Theta\left(\frac{B}{p_{mega}\sqrt{p_{\mathrm{a}}}}\right) = \Theta\left(\max\left\{\frac{Bd}{\sqrt{p_{\mathrm{a}}}\zeta_{\mathcal{C}}}, \frac{\sigma^2}{\sqrt{p_{\mathrm{a}}}n\varepsilon}\right\}\right)$, then DASHA-PP-SYNC-MVR needs*

$$T := \mathcal{O}\left(\frac{\Delta_0}{\varepsilon}\left[L + \left(\frac{\omega}{p_{\mathrm{a}}\sqrt{n}} + \sqrt{\frac{d}{p_{\mathrm{a}}^2\zeta_{\mathcal{C}}n}}\right)\left(\widehat{L} + \frac{L_\sigma}{\sqrt{B}}\right) + \frac{\sigma}{p_{\mathrm{a}}\sqrt{\varepsilon n}}\left(\frac{\widehat{L}}{\sqrt{B}} + \frac{L_\sigma}{B}\right)\right] + \frac{\sigma^2}{\sqrt{p_{\mathrm{a}}}n\varepsilon B}\right).$$

*communication rounds to get an $\varepsilon$-solution, the expected communication complexity is equal to $\mathcal{O}\left(d + \zeta_{\mathcal{C}}T\right)$, and the expected number of stochastic gradient calculations per node equals $\mathcal{O}(B_{\mathrm{init}} + BT)$, where $\zeta_{\mathcal{C}}$ is the expected density from Definition 12.*

The main improvement of Corollary 6 over Corollary 3 is the size of the initial batch size $B_{\mathrm{init}}$. However, Corollary 4 reveals that we can avoid regimes when DASHA-PP-MVR is suboptimal.

We also provide a theorem under PŁ-condition (see Assumption 9).

**Theorem 13.** *Suppose that Assumptions 1, 2, 3, 5, 6, 7, 8 and 9 hold. Let us take $a = \frac{p_{\mathrm{a}}}{2\omega+1}$, $b = \frac{p_{mega}p_{\mathrm{a}}}{2-p_{\mathrm{a}}}$, probability $p_{mega} \in (0,1]$, batch size $B' \geq B \geq 1$,*

$$\gamma \leq \min\left\{\left(\left(L + \sqrt{\frac{16\left(2\omega+1\right)\omega}{np_{\mathrm{a}}^2}\left(\frac{L_\sigma^2}{B} + \widehat{L}^2\right) + \left(\frac{48L_\sigma^2}{np_{mega}p_{\mathrm{a}}^2B} + \frac{24\left(1 - \frac{p_{\mathrm{aa}}}{p_{\mathrm{a}}}\right)\widehat{L}^2}{np_{mega}p_{\mathrm{a}}^2}\right)}\right)^{-1}, \frac{a}{2\mu}, \frac{b}{2\mu}\right\},$$

*and $h_i^0 = g_i^0$ for all $i \in [n]$ in Algorithm 8. Then*

$$\mathrm{E}\left[f(x^T) - f^*\right]$$

$$\leq (1 - \gamma\mu)^T\left(\Delta_0 + \frac{2\gamma}{b}\left\|h^0 - \nabla f(x^0)\right\|^2 + \frac{8\gamma\left(p_{\mathrm{a}} - p_{\mathrm{aa}}\right)}{np_{\mathrm{a}}^2p_{mega}}\frac{1}{n}\sum_{i=1}^{n}\left\|h_i^0 - \nabla f_i(x^0)\right\|^2\right) + \frac{20\sigma^2}{\mu nB'}.$$

Let us provide bounds up to logarithmic factors and use $\widetilde{\mathcal{O}}(\cdot)$ notation.

**Corollary 7.** *Suppose that assumptions from Theorem 13 hold, probability $p_{mega} = \min\left\{\frac{\zeta_{\mathcal{C}}}{d}, \frac{\mu n\varepsilon B}{\sigma^2}\right\}$, batch size $B' = \Theta\left(\frac{\sigma^2}{\mu n\varepsilon}\right)$ then DASHA-PP-SYNC-MVR needs*

$$T := \widetilde{\mathcal{O}}\left(\frac{\omega+1}{p_{\mathrm{a}}} + \frac{d}{p_{\mathrm{a}}\zeta_{\mathcal{C}}} + \frac{\sigma^2}{p_{\mathrm{a}}\mu n\varepsilon B} + \frac{L}{\mu} + \frac{\omega}{p_{\mathrm{a}}\mu\sqrt{n}}\left(\frac{L_\sigma}{\sqrt{B}} + \widehat{L}\right) + \left(\frac{\sqrt{d}}{p_{\mathrm{a}}\mu\sqrt{\zeta_{\mathcal{C}}n}} + \frac{\sigma}{p_{\mathrm{a}}n\mu^{3/2}\sqrt{\varepsilon B}}\right)\left(\frac{L_\sigma}{\sqrt{B}} + \widehat{L}\right)\right).$$

*communication rounds to get an $\varepsilon$-solution, the expected communication complexity is equal to $\widetilde{\mathcal{O}}\left(\zeta_{\mathcal{C}} T\right),$ and the expected number of stochastic gradient calculations per node equals $\widetilde{\mathcal{O}}(BT),$ where $\zeta_{\mathcal{C}}$ is the expected density from Definition 12.*

The proof of this corollary almost repeats the proof of Corollary 6. Note that we can skip the initialization procedure and initialize $h_i^0$ and $g_i^0$, for instance, with zeros because the initialization error is under a logarithm.

Let us assume that $\frac{d}{\zeta_{\mathcal{C}}} = \Theta\left(\omega\right)$ (holds for the RandK compressor), then the convergence rate of DASHA-PP-SYNC-MVR is

$$\widetilde{\mathcal{O}}\left(\frac{\omega+1}{p_{\mathrm{a}}} + \frac{\sigma^2}{p_{\mathrm{a}}\mu n \varepsilon B} + \frac{L}{\mu} + \frac{\omega}{p_{\mathrm{a}}\mu\sqrt{n}}\left(\frac{L_\sigma}{\sqrt{B}} + \widehat{L}\right) + \frac{\sigma}{p_{\mathrm{a}}n\mu^{3/2}\sqrt{\varepsilon B}}\left(\frac{L_\sigma}{\sqrt{B}} + \widehat{L}\right)\right). \tag{34}$$

Comparing (34) with the rate of DASHA-PP-MVR (31), one can see that DASHA-PP-SYNC-MVR improves the suboptimal term $\mathcal{P}_2$ from (31). However, Corollary 5 reveals that we can escape these suboptimal regimes by choosing the parameter $K$ of RandK compressors in a particular way.

### G.1 Proof for DASHA-PP-SYNC-MVR

In this section, we provide the proof of the convergence rate for DASHA-PP-SYNC-MVR. There are four different sources of randomness in Algorithm 8: the first one from random samples $\xi^{t+1}$, the second one from compressors $\{\mathcal{C}_i\}_{i=1}^n$, the third one from availability of nodes, and the fourth one from $c^{t+1}$. We define $\mathrm{E}_k\left[\cdot\right]$, $\mathrm{E}_{\mathcal{C}}\left[\cdot\right]$, $\mathrm{E}_{p_{\mathrm{a}}}\left[\cdot\right]$ and $\mathrm{E}_{p_{\mathrm{mega}}}\left[\cdot\right]$ to be conditional expectations w.r.t. $\xi^{t+1}$, $\{\mathcal{C}_i\}_{i=1}^n$, availability, and $c^{t+1}$, accordingly, conditioned on all previous randomness. Moreover, we define $\mathrm{E}_{t+1}\left[\cdot\right]$ to be a conditional expectation w.r.t. all randomness in iteration $t+1$ conditioned on all previous randomness.

Let us denote

$$k_{i,1}^{t+1} := \frac{1}{B'}\sum_{k=1}^{B'}\nabla f_i(x^{t+1};\xi_{ik}^{t+1}) - \frac{1}{B'}\sum_{k=1}^{B'}\nabla f_i(x^t;\xi_{ik}^{t+1}) - \frac{b}{p_{\mathrm{mega}}}\left(h_i^t - \frac{1}{B'}\sum_{k=1}^{B'}\nabla f_i(x^t;\xi_{ik}^{t+1})\right),$$

$$k_{i,2}^{t+1} := \frac{1}{B}\sum_{j=1}^{B}\nabla f_i(x^{t+1};\xi_{ij}^{t+1}) - \frac{1}{B}\sum_{j=1}^{B}\nabla f_i(x^t;\xi_{ij}^{t+1}),$$

$$h_{i,1}^{t+1} := \begin{cases} h_i^t + \frac{1}{p_{\mathrm{a}}}k_{i,1}^{t+1}, & i^{\mathrm{th}} \text{ node is } \textit{participating}, \\ h_i^t, & \text{otherwise}, \end{cases}$$

$$h_{i,2}^{t+1} := \begin{cases} h_i^t + \frac{1}{p_{\mathrm{a}}}k_{i,2}^{t+1}, & i^{\mathrm{th}} \text{ node is } \textit{participating}, \\ h_i^t, & \text{otherwise}, \end{cases}$$

$$g_{i,1}^{t+1} := \begin{cases} g_i^t + \frac{1}{p_{\mathrm{a}}}k_{i,1}^{t+1} - \frac{a}{p_{\mathrm{a}}}\left(g_i^t - h_i^t\right), & i^{\mathrm{th}} \text{ node is } \textit{participating}, \\ g_i^t, & \text{otherwise}, \end{cases}$$

$$g_{i,2}^{t+1} := \begin{cases} g_i^t + \mathcal{C}_i\left(\frac{1}{p_{\mathrm{a}}}k_{i,2}^{t+1} - \frac{a}{p_{\mathrm{a}}}\left(g_i^t - h_i^t\right)\right), & i^{\mathrm{th}} \text{ node is } \textit{participating}, \\ g_i^t, & \text{otherwise}, \end{cases}$$

$h_1^{t+1} := \frac{1}{n}\sum_{i=1}^n h_{i,1}^{t+1}$, $h_2^{t+1} := \frac{1}{n}\sum_{i=1}^n h_{i,2}^{t+1}$, $g_1^{t+1} := \frac{1}{n}\sum_{i=1}^n g_{i,1}^{t+1}$, and $g_2^{t+1} := \frac{1}{n}\sum_{i=1}^n g_{i,2}^{t+1}$.
Note, that

$$h^{t+1} = \begin{cases} h_1^{t+1}, & c^{t+1} = 1, \\ h_2^{t+1}, & c^{t+1} = 0, \end{cases}$$

and

$$g^{t+1} = \begin{cases} g_1^{t+1}, & c^{t+1} = 1, \\ g_2^{t+1}, & c^{t+1} = 0 \end{cases}$$

First, we will prove two lemmas.

**Lemma 13.** *Suppose that Assumptions 3, 5, 7 and 8 hold and let us consider sequences $\{g_i^{t+1}\}_{i=1}^n$ and $\{h_i^{t+1}\}_{i=1}^n$ from Algorithm 8, then*

$$
\mathrm{E}_{\mathcal{C}}\left[\mathrm{E}_{p_{\mathrm{a}}}\left[\mathrm{E}_{p_{mega}}\left[\left\|g^{t+1}-h^{t+1}\right\|^2\right]\right]\right]
$$

$$
\leq \frac{2\left(1-p_{mega}\right)\omega}{n^2 p_{\mathrm{a}}}\sum_{i=1}^n\left\|k_{i,2}^{t+1}\right\|^2 + \left(\frac{\left(p_{\mathrm{a}}-p_{\mathrm{aa}}\right)a^2}{n^2 p_{\mathrm{a}}^2} + \frac{2\left(1-p_{mega}\right)a^2\omega}{n^2 p_{\mathrm{a}}}\right)\sum_{i=1}^n\left\|g_i^t-h_i^t\right\|^2
$$

$$
+ (1-a)^2\left\|g^t-h^t\right\|^2,
$$

*and*

$$
\mathrm{E}_{\mathcal{C}}\left[\mathrm{E}_{p_{\mathrm{a}}}\left[\mathrm{E}_{p_{mega}}\left[\left\|g_i^{t+1}-h_i^{t+1}\right\|^2\right]\right]\right]
$$

$$
\leq \frac{2\left(1-p_{mega}\right)\omega}{p_{\mathrm{a}}}\left\|k_{i,2}^{t+1}\right\|^2 + \left(\frac{\left(1-p_{\mathrm{a}}\right)a^2}{p_{\mathrm{a}}} + \frac{2\left(1-p_{mega}\right)a^2\omega}{p_{\mathrm{a}}}\right)\left\|g_i^t-h_i^t\right\|^2
$$

$$
+ (1-a)^2\left\|g_i^t-h_i^t\right\|^2, \quad \forall i \in [n].
$$

*Proof.* First, we get the bound for $\mathrm{E}_{t+1}\left[\left\|g^{t+1}-h^{t+1}\right\|^2\right]$:

$$
\mathrm{E}_{\mathcal{C}}\left[\mathrm{E}_{p_{\mathrm{a}}}\left[\mathrm{E}_{p_{\mathrm{mega}}}\left[\left\|g^{t+1}-h^{t+1}\right\|^2\right]\right]\right]
$$

$$
= p_{\mathrm{mega}}\mathrm{E}_{p_{\mathrm{a}}}\left[\left\|g_1^{t+1}-h_1^{t+1}\right\|^2\right] + \left(1-p_{\mathrm{mega}}\right)\mathrm{E}_{\mathcal{C}}\left[\mathrm{E}_{p_{\mathrm{a}}}\left[\left\|g_2^{t+1}-h_2^{t+1}\right\|^2\right]\right].
$$

Using

$$
\mathrm{E}_{p_{\mathrm{a}}}\left[g_{i,1}^{t+1}-h_{i,1}^{t+1}\right] = g_i^t + k_{i,1}^{t+1} - a\left(g_i^t-h_i^t\right) - h_i^t - k_{i,1}^{t+1} = (1-a)\left(g_i^t-h_i^t\right)
$$

and

$$
\mathrm{E}_{\mathcal{C}}\left[\mathrm{E}_{p_{\mathrm{a}}}\left[g_{i,2}^{t+1}-h_{i,2}^{t+1}\right]\right] = g_i^t + k_{i,2}^{t+1} - a\left(g_i^t-h_i^t\right) - h_i^t - k_{i,2}^{t+1} = (1-a)\left(g_i^t-h_i^t\right),
$$

we have

$$
\mathrm{E}_{\mathcal{C}}\left[\mathrm{E}_{p_{\mathrm{a}}}\left[\mathrm{E}_{p_{\mathrm{mega}}}\left[\left\|g^{t+1}-h^{t+1}\right\|^2\right]\right]\right]
$$

$$
\overset{(16)}{=} p_{\mathrm{mega}}\mathrm{E}_{p_{\mathrm{a}}}\left[\left\|g_1^{t+1}-h_1^{t+1}-\mathrm{E}_{p_{\mathrm{a}}}\left[g_1^{t+1}-h_1^{t+1}\right]\right\|^2\right]
$$

$$
+ \left(1-p_{\mathrm{mega}}\right)\mathrm{E}_{\mathcal{C}}\left[\mathrm{E}_{p_{\mathrm{a}}}\left[\left\|g_2^{t+1}-h_2^{t+1}-\mathrm{E}_{p_{\mathrm{a}}}\left[g_2^{t+1}-h_2^{t+1}\right]\right\|^2\right]\right]
$$

$$
+ (1-a)^2\left\|g^t-h^t\right\|^2.
$$

We can use Lemma 1 two times with i) $r_i = g_i^t - h_i^t$ and $s_i = -a\left(g_i^t-h_i^t\right)$ and ii) $r_i = g_i^t - h_i^t$ and $s_i = p_{\mathrm{a}}\mathcal{C}_i\left(\frac{1}{p_{\mathrm{a}}}k_{i,2}^{t+1} - \frac{a}{p_{\mathrm{a}}}\left(g_i^t-h_i^t\right)\right) - k_{i,2}^{t+1}$, to obtain

$$
\mathrm{E}_{\mathcal{C}}\left[\mathrm{E}_{p_{\mathrm{a}}}\left[\mathrm{E}_{p_{\mathrm{mega}}}\left[\left\|g^{t+1}-h^{t+1}\right\|^2\right]\right]\right]
$$

$$
\leq \frac{p_{\mathrm{mega}}a^2\left(p_{\mathrm{a}}-p_{\mathrm{aa}}\right)}{n^2 p_{\mathrm{a}}^2}\sum_{i=1}^n\left\|g_i^t-h_i^t\right\|^2
$$

$$
+ \left(1-p_{\mathrm{mega}}\right)\left(\frac{1}{n^2 p_{\mathrm{a}}}\sum_{i=1}^n\mathrm{E}_{\mathcal{C}}\left[\left\|p_{\mathrm{a}}\mathcal{C}_i\left(\frac{1}{p_{\mathrm{a}}}k_{i,2}^{t+1} - \frac{a}{p_{\mathrm{a}}}\left(g_i^t-h_i^t\right)\right) - \left(k_{i,2}^{t+1} - a\left(g_i^t-h_i^t\right)\right)\right\|^2\right]\right)
$$

$$
+ \left(1-p_{\mathrm{mega}}\right)\left(\frac{a^2\left(p_{\mathrm{a}}-p_{\mathrm{aa}}\right)}{n^2 p_{\mathrm{a}}^2}\sum_{i=1}^n\left\|g_i^t-h_i^t\right\|^2\right)
$$

$$
+ (1-a)^2\left\|g^t-h^t\right\|^2
$$

$$
= \frac{a^2\left(p_{\mathrm{a}}-p_{\mathrm{aa}}\right)}{n^2 p_{\mathrm{a}}^2}\sum_{i=1}^n\left\|g_i^t-h_i^t\right\|^2
$$

$$+ (1 - p_{\text{mega}}) \left( \frac{p_{\text{a}}}{n^2} \sum_{i=1}^{n} \mathrm{E}_{\mathcal{C}} \left[ \left\| \mathcal{C}_i \left( \frac{1}{p_{\text{a}}} k_{i,2}^{t+1} - \frac{a}{p_{\text{a}}} \left( g_i^t - h_i^t \right) \right) - \left( \frac{1}{p_{\text{a}}} k_{i,2}^{t+1} - \frac{a}{p_{\text{a}}} \left( g_i^t - h_i^t \right) \right) \right\|^2 \right] \right)$$

$$+ (1 - a)^2 \left\| g^t - h^t \right\|^2$$

$$\leq \frac{a^2 (p_{\text{a}} - p_{\text{aa}})}{n^2 p_{\text{a}}^2} \sum_{i=1}^{n} \left\| g_i^t - h_i^t \right\|^2$$

$$+ \frac{(1 - p_{\text{mega}}) p_{\text{a}} \omega}{n^2} \sum_{i=1}^{n} \left\| \frac{1}{p_{\text{a}}} k_{i,2}^{t+1} - \frac{a}{p_{\text{a}}} \left( g_i^t - h_i^t \right) \right\|^2$$

$$+ (1 - a)^2 \left\| g^t - h^t \right\|^2$$

$$= \frac{a^2 (p_{\text{a}} - p_{\text{aa}})}{n^2 p_{\text{a}}^2} \sum_{i=1}^{n} \left\| g_i^t - h_i^t \right\|^2$$

$$+ \frac{(1 - p_{\text{mega}}) \omega}{n^2 p_{\text{a}}} \sum_{i=1}^{n} \left\| k_{i,2}^{t+1} - a \left( g_i^t - h_i^t \right) \right\|^2$$

$$+ (1 - a)^2 \left\| g^t - h^t \right\|^2 .$$

In the last inequality, we use Assumption 7. Next, using (15), we have

$$\mathrm{E}_{\mathcal{C}} \left[ \mathrm{E}_{p_{\text{a}}} \left[ \mathrm{E}_{p_{\text{mega}}} \left[ \left\| g^{t+1} - h^{t+1} \right\|^2 \right] \right] \right]$$

$$\leq \frac{2 (1 - p_{\text{mega}}) \omega}{n^2 p_{\text{a}}} \sum_{i=1}^{n} \left\| k_{i,2}^{t+1} \right\|^2 + \left( \frac{(p_{\text{a}} - p_{\text{aa}}) a^2}{n^2 p_{\text{a}}^2} + \frac{2 (1 - p_{\text{mega}}) \omega a^2}{n^2 p_{\text{a}}} \right) \sum_{i=1}^{n} \left\| g_i^t - h_i^t \right\|^2$$

$$+ (1 - a)^2 \left\| g^t - h^t \right\|^2 .$$

The second inequality can be proved almost in the same way:

$$\mathrm{E}_{\mathcal{C}} \left[ \mathrm{E}_{p_{\text{a}}} \left[ \mathrm{E}_{p_{\text{mega}}} \left[ \left\| g_i^{t+1} - h_i^{t+1} \right\|^2 \right] \right] \right]$$

$$= p_{\text{mega}} \mathrm{E}_{p_{\text{a}}} \left[ \left\| g_{i,1}^{t+1} - h_{i,1}^{t+1} \right\|^2 \right] + (1 - p_{\text{mega}}) \mathrm{E}_{\mathcal{C}} \left[ \mathrm{E}_{p_{\text{a}}} \left[ \left\| g_{i,2}^{t+1} - h_{i,2}^{t+1} \right\|^2 \right] \right]$$

$$\overset{(16)}{=} p_{\text{mega}} \mathrm{E}_{p_{\text{a}}} \left[ \left\| g_{i,1}^{t+1} - h_{i,1}^{t+1} - (1 - a) \left( g_i^t - h_i^t \right) \right\|^2 \right] + (1 - p_{\text{mega}}) \mathrm{E}_{\mathcal{C}} \left[ \mathrm{E}_{p_{\text{a}}} \left[ \left\| g_{i,2}^{t+1} - h_{i,2}^{t+1} \right\|^2 \right] \right]$$

$$+ p_{\text{mega}} (1 - a)^2 \left\| g_i^t - h_i^t \right\|^2$$

$$= \frac{p_{\text{mega}} (1 - p_{\text{a}}) a^2}{p_{\text{a}}} \left\| g_i^t - h_i^t \right\|^2 + (1 - p_{\text{mega}}) \mathrm{E}_{\mathcal{C}} \left[ \mathrm{E}_{p_{\text{a}}} \left[ \left\| g_{i,2}^{t+1} - h_{i,2}^{t+1} \right\|^2 \right] \right]$$

$$+ p_{\text{mega}} (1 - a)^2 \left\| g_i^t - h_i^t \right\|^2$$

$$\overset{(16)}{=} \frac{p_{\text{mega}} (1 - p_{\text{a}}) a^2}{p_{\text{a}}} \left\| g_i^t - h_i^t \right\|^2 + (1 - p_{\text{mega}}) \mathrm{E}_{\mathcal{C}} \left[ \mathrm{E}_{p_{\text{a}}} \left[ \left\| g_{i,2}^{t+1} - h_{i,2}^{t+1} - (1 - a) \left( g_i^t - h_i^t \right) \right\|^2 \right] \right]$$

$$+ (1 - a)^2 \left\| g_i^t - h_i^t \right\|^2$$

$$= \frac{p_{\text{mega}} (1 - p_{\text{a}}) a^2}{p_{\text{a}}} \left\| g_i^t - h_i^t \right\|^2$$

$$+ (1 - p_{\text{mega}}) p_{\text{a}} \mathrm{E}_{\mathcal{C}} \left[ \left\| g_i^t + \mathcal{C}_i \left( \frac{1}{p_{\text{a}}} k_{i,2}^{t+1} - \frac{a}{p_{\text{a}}} \left( g_i^t - h_i^t \right) \right) - \left( h_i^t + \frac{1}{p_{\text{a}}} k_{i,2}^{t+1} \right) - (1 - a) \left( g_i^t - h_i^t \right) \right\|^2 \right]$$

$$+ (1 - p_{\text{mega}}) (1 - p_{\text{a}}) \left\| g_i^t - h_i^t - (1 - a) \left( g_i^t - h_i^t \right) \right\|^2$$

$$+ (1 - a)^2 \left\| g_i^t - h_i^t \right\|^2$$

$$= \frac{p_{\text{mega}} (1 - p_{\text{a}}) a^2}{p_{\text{a}}} \left\| g_i^t - h_i^t \right\|^2$$

$$+ (1 - p_{\text{mega}}) p_{\text{a}} \mathrm{E}_{\mathcal{C}} \left[ \left\| \mathcal{C}_i \left( \frac{1}{p_{\text{a}}} k_{i,2}^{t+1} - \frac{a}{p_{\text{a}}} \left( g_i^t - h_i^t \right) \right) - \left( \frac{1}{p_{\text{a}}} k_{i,2}^{t+1} - a \left( g_i^t - h_i^t \right) \right) \right\|^2 \right]$$

$$+ \left(1 - p_{\mathrm{mega}}\right)\left(1 - p_{\mathrm{a}}\right) a^2 \left\| g_i^t - h_i^t \right\|^2$$

$$+ \left(1 - a\right)^2 \left\| g_i^t - h_i^t \right\|^2$$

$$\overset{(16)}{=} \left( \frac{p_{\mathrm{mega}}(1 - p_{\mathrm{a}})a^2}{p_{\mathrm{a}}} + \frac{\left(1 - p_{\mathrm{mega}}\right)\left(1 - p_{\mathrm{a}}\right)a^2}{p_{\mathrm{a}}} \right) \left\| g_i^t - h_i^t \right\|^2$$

$$+ \left(1 - p_{\mathrm{mega}}\right) p_{\mathrm{a}} \mathrm{E}_{\mathcal{C}} \left[ \left\| \mathcal{C}_i \left( \frac{1}{p_{\mathrm{a}}} k_{i,2}^{t+1} - \frac{a}{p_{\mathrm{a}}} \left(g_i^t - h_i^t\right) \right) - \left( \frac{1}{p_{\mathrm{a}}} k_{i,2}^{t+1} - \frac{a}{p_{\mathrm{a}}} \left(g_i^t - h_i^t\right) \right) \right\|^2 \right]$$

$$+ \left(1 - a\right)^2 \left\| g_i^t - h_i^t \right\|^2$$

$$= \frac{(1 - p_{\mathrm{a}})a^2}{p_{\mathrm{a}}} \left\| g_i^t - h_i^t \right\|^2$$

$$+ \left(1 - p_{\mathrm{mega}}\right) p_{\mathrm{a}} \mathrm{E}_{\mathcal{C}} \left[ \left\| \mathcal{C}_i \left( \frac{1}{p_{\mathrm{a}}} k_{i,2}^{t+1} - \frac{a}{p_{\mathrm{a}}} \left(g_i^t - h_i^t\right) \right) - \left( \frac{1}{p_{\mathrm{a}}} k_{i,2}^{t+1} - \frac{a}{p_{\mathrm{a}}} \left(g_i^t - h_i^t\right) \right) \right\|^2 \right]$$

$$+ \left(1 - a\right)^2 \left\| g_i^t - h_i^t \right\|^2$$

$$\leq \frac{(1 - p_{\mathrm{a}})a^2}{p_{\mathrm{a}}} \left\| g_i^t - h_i^t \right\|^2$$

$$+ \frac{\left(1 - p_{\mathrm{mega}}\right)\omega}{p_{\mathrm{a}}} \left\| k_{i,2}^{t+1} - a\left(g_i^t - h_i^t\right) \right\|^2$$

$$+ \left(1 - a\right)^2 \left\| g_i^t - h_i^t \right\|^2$$

$$\overset{(15)}{\leq} \frac{2\left(1 - p_{\mathrm{mega}}\right)\omega}{p_{\mathrm{a}}} \left\| k_{i,2}^{t+1} \right\|^2 + \left( \frac{(1 - p_{\mathrm{a}})a^2}{p_{\mathrm{a}}} + \frac{2\left(1 - p_{\mathrm{mega}}\right)a^2\omega}{p_{\mathrm{a}}} \right) \left\| g_i^t - h_i^t \right\|^2$$

$$+ \left(1 - a\right)^2 \left\| g_i^t - h_i^t \right\|^2 .$$

$\square$

**Lemma 14.** *Suppose that Assumptions 3, 5, 6 and 8 hold and let us consider sequence* $\{h_i^{t+1}\}_{i=1}^n$ *from Algorithm 8, then*

$$\mathrm{E}_k \left[ \mathrm{E}_{p_{\mathrm{a}}} \left[ \mathrm{E}_{p_{mega}} \left[ \left\| h^{t+1} - \nabla f(x^{t+1}) \right\|^2 \right] \right] \right]$$

$$\leq \frac{2b^2\sigma^2}{np_{mega}p_{\mathrm{a}}B'} + \left( \frac{2p_{mega}L_\sigma^2}{np_{\mathrm{a}}B'} \left(1 - \frac{b}{p_{mega}}\right)^2 + \frac{\left(1 - p_{mega}\right)L_\sigma^2}{np_{\mathrm{a}}B} + \frac{2\left(p_{\mathrm{a}} - p_{\mathrm{aa}}\right)\widehat{L}^2}{np_{\mathrm{a}}^2} \right) \left\| x^{t+1} - x^t \right\|^2$$

$$+ \frac{2\left(p_{\mathrm{a}} - p_{\mathrm{aa}}\right)b^2}{n^2 p_{\mathrm{a}}^2 p_{mega}} \sum_{i=1}^n \left\| h_i^t - \nabla f_i(x^t) \right\|^2 + \left( p_{mega} \left(1 - \frac{b}{p_{mega}}\right)^2 + \left(1 - p_{mega}\right) \right) \left\| h^t - \nabla f(x^t) \right\|^2 ,$$

$$\mathrm{E}_k \left[ \mathrm{E}_{p_{\mathrm{a}}} \left[ \mathrm{E}_{p_{mega}} \left[ \left\| h_i^{t+1} - \nabla f_i(x^{t+1}) \right\|^2 \right] \right] \right]$$

$$\leq \frac{2b^2\sigma^2}{p_{\mathrm{a}}p_{mega}B'} + \left( \frac{2p_{mega}L_\sigma^2}{p_{\mathrm{a}}B'} \left(1 - \frac{b}{p_{mega}}\right)^2 + \frac{\left(1 - p_{mega}\right)L_\sigma^2}{p_{\mathrm{a}}B} + \frac{2\left(1 - p_{\mathrm{a}}\right)L_i^2}{p_{\mathrm{a}}} \right) \left\| x^{t+1} - x^t \right\|^2$$

$$+ \frac{2\left(1 - p_{\mathrm{a}}\right)b^2}{p_{mega}p_{\mathrm{a}}} \left\| h_i^t - \nabla f_i(x^t) \right\|^2 + \left( p_{mega} \left(1 - \frac{b}{p_{mega}}\right)^2 + \left(1 - p_{mega}\right) \right) \left\| h_i^t - \nabla f_i(x^t) \right\|^2 , \quad \forall i \in [n],$$

*and*

$$\mathrm{E}_k \left[ \left\| k_{i,2}^{t+1} \right\|^2 \right] \leq \left( \frac{L_\sigma^2}{B} + L_i^2 \right) \left\| x^{t+1} - x^t \right\|^2 , \quad \forall i \in [n],$$

*Proof.* First, we prove the bound for $\mathrm{E}_k \left[ \mathrm{E}_{p_{\mathrm{a}}} \left[ \mathrm{E}_{p_{\mathrm{mega}}} \left[ \left\| h^{t+1} - \nabla f(x^{t+1}) \right\|^2 \right] \right] \right]$. Using

$$\mathrm{E}_k \left[ \mathrm{E}_{p_{\mathrm{a}}} \left[ h_{i,1}^{t+1} \right] \right]$$

$$= h_i^t + \mathrm{E}_k \left[ \frac{1}{B'} \sum_{k=1}^{B'} \nabla f_i(x^{t+1}; \xi_{ik}^{t+1}) - \frac{1}{B'} \sum_{k=1}^{B'} \nabla f_i(x^t; \xi_{ik}^{t+1}) - \frac{b}{p_{\mathrm{mega}}} \left( h_i^t - \frac{1}{B'} \sum_{k=1}^{B'} \nabla f_i(x^t; \xi_{ik}^{t+1}) \right) \right]$$

$$= h_i^t + \nabla f_i(x^{t+1}) - \nabla f_i(x^t) - \frac{b}{p_{\mathrm{mega}}} \left( h_i^t - \nabla f_i(x^t) \right)$$

and

$$\mathrm{E}_k \left[ \mathrm{E}_{p_{\mathrm{a}}} \left[ h_{i,2}^{t+1} \right] \right]$$

$$= h_i^t + \mathrm{E}_k \left[ \frac{1}{B} \sum_{j=1}^{B} \nabla f_i(x^{t+1}; \xi_{ij}^{t+1}) - \frac{1}{B} \sum_{j=1}^{B} \nabla f_i(x^t; \xi_{ij}^{t+1}) \right]$$

$$= h_i^t + \nabla f_i(x^{t+1}) - \nabla f_i(x^t),$$

we have

$$\mathrm{E}_k \left[ \mathrm{E}_{p_{\mathrm{a}}} \left[ \mathrm{E}_{p_{\mathrm{mega}}} \left[ \left\| h^{t+1} - \nabla f(x^{t+1}) \right\|^2 \right] \right] \right]$$

$$= p_{\mathrm{mega}} \mathrm{E}_k \left[ \mathrm{E}_{p_{\mathrm{a}}} \left[ \left\| h_1^{t+1} - \nabla f(x^{t+1}) \right\|^2 \right] \right] + (1 - p_{\mathrm{mega}}) \mathrm{E}_k \left[ \mathrm{E}_{p_{\mathrm{a}}} \left[ \left\| h_2^{t+1} - \nabla f(x^{t+1}) \right\|^2 \right] \right]$$

$$\overset{(16)}{=} p_{\mathrm{mega}} \mathrm{E}_k \left[ \mathrm{E}_{p_{\mathrm{a}}} \left[ \left\| h_1^{t+1} - \mathrm{E}_k \left[ \mathrm{E}_{p_{\mathrm{a}}} \left[ h_1^{t+1} \right] \right] \right\|^2 \right] \right] + (1 - p_{\mathrm{mega}}) \mathrm{E}_k \left[ \mathrm{E}_{p_{\mathrm{a}}} \left[ \left\| h_2^{t+1} - \mathrm{E}_k \left[ \mathrm{E}_{p_{\mathrm{a}}} \left[ h_2^{t+1} \right] \right] \right\|^2 \right] \right]$$

$$+ \left( p_{\mathrm{mega}} \left( 1 - \frac{b}{p_{\mathrm{mega}}} \right)^2 + (1 - p_{\mathrm{mega}}) \right) \left\| h^t - \nabla f(x^t) \right\|^2.$$

We can use Lemma 1 two times with i) $r_i = h_i^t$ and $s_i = k_{i,1}^{t+1}$ and ii) $r_i = h_i^t$ and $s_i = k_{i,2}^{t+1}$, to obtain

$$\mathrm{E}_k \left[ \mathrm{E}_{p_{\mathrm{a}}} \left[ \mathrm{E}_{p_{\mathrm{mega}}} \left[ \left\| h^{t+1} - \nabla f(x^{t+1}) \right\|^2 \right] \right] \right]$$

$$\leq p_{\mathrm{mega}} \left( \frac{1}{n^2 p_{\mathrm{a}}} \sum_{i=1}^{n} \mathrm{E}_k \left[ \left\| k_{i,1}^{t+1} - \mathrm{E}_k \left[ k_{i,1}^{t+1} \right] \right\|^2 \right] + \frac{p_{\mathrm{a}} - p_{\mathrm{aa}}}{n^2 p_{\mathrm{a}}^2} \sum_{i=1}^{n} \left\| \nabla f_i(x^{t+1}) - \nabla f_i(x^t) - \frac{b}{p_{\mathrm{mega}}} \left( h_i^t - \nabla f_i(x^t) \right) \right\|^2 \right)$$

$$+ (1 - p_{\mathrm{mega}}) \left( \frac{1}{n^2 p_{\mathrm{a}}} \sum_{i=1}^{n} \mathrm{E}_k \left[ \left\| k_{i,2}^{t+1} - \mathrm{E}_k \left[ k_{i,2}^{t+1} \right] \right\|^2 \right] + \frac{p_{\mathrm{a}} - p_{\mathrm{aa}}}{n^2 p_{\mathrm{a}}^2} \sum_{i=1}^{n} \left\| \nabla f_i(x^{t+1}) - \nabla f_i(x^t) \right\|^2 \right)$$

$$+ \left( p_{\mathrm{mega}} \left( 1 - \frac{b}{p_{\mathrm{mega}}} \right)^2 + (1 - p_{\mathrm{mega}}) \right) \left\| h^t - \nabla f(x^t) \right\|^2$$

$$\overset{(15)}{\leq} \frac{p_{\mathrm{mega}}}{n^2 p_{\mathrm{a}}} \sum_{i=1}^{n} \mathrm{E}_k \left[ \left\| k_{i,1}^{t+1} - \mathrm{E}_k \left[ k_{i,1}^{t+1} \right] \right\|^2 \right]$$

$$+ \frac{1 - p_{\mathrm{mega}}}{n^2 p_{\mathrm{a}}} \sum_{i=1}^{n} \mathrm{E}_k \left[ \left\| k_{i,2}^{t+1} - \mathrm{E}_k \left[ k_{i,2}^{t+1} \right] \right\|^2 \right]$$

$$+ \frac{2 \left( p_{\mathrm{a}} - p_{\mathrm{aa}} \right)}{n^2 p_{\mathrm{a}}^2} \sum_{i=1}^{n} \left\| \nabla f_i(x^{t+1}) - \nabla f_i(x^t) \right\|^2$$

$$+ \frac{2 \left( p_{\mathrm{a}} - p_{\mathrm{aa}} \right) b^2}{n^2 p_{\mathrm{a}}^2 p_{\mathrm{mega}}} \sum_{i=1}^{n} \left\| h_i^t - \nabla f_i(x^t) \right\|^2 + \left( p_{\mathrm{mega}} \left( 1 - \frac{b}{p_{\mathrm{mega}}} \right)^2 + (1 - p_{\mathrm{mega}}) \right) \left\| h^t - \nabla f(x^t) \right\|^2.$$

$$\tag{35}$$

Let us consider $\mathrm{E}_k \left[ \left\| k_{i,1}^{t+1} - \mathrm{E}_k \left[ k_{i,1}^{t+1} \right] \right\|^2 \right].$

$$\mathrm{E}_k \left[ \left\| k_{i,1}^{t+1} - \mathrm{E}_k \left[ k_{i,1}^{t+1} \right] \right\|^2 \right]$$

$$= \mathrm{E}_k \left[ \left\| \frac{1}{B'} \sum_{k=1}^{B'} \nabla f_i(x^{t+1}; \xi_{ik}^{t+1}) - \frac{1}{B'} \sum_{k=1}^{B'} \nabla f_i(x^t; \xi_{ik}^{t+1}) - \frac{b}{p_{\mathrm{mega}}} \left( h_i^t - \frac{1}{B'} \sum_{k=1}^{B'} \nabla f_i(x^t; \xi_{ik}^{t+1}) \right) \right. \right.$$

$$- \left(\nabla f_i(x^{t+1}) - \nabla f_i(x^t) - \frac{b}{p_{\text{mega}}}\left(h_i^t - \nabla f_i(x^t)\right)\right)\bigg\|^2\Bigg]$$

$$= \mathrm{E}_k\Bigg[\bigg\|\frac{1}{B'}\sum_{k=1}^{B'}\nabla f_i(x^{t+1};\xi_{ik}^{t+1}) - \frac{1}{B'}\sum_{k=1}^{B'}\nabla f_i(x^t;\xi_{ik}^{t+1}) + \frac{b}{p_{\text{mega}}}\left(\frac{1}{B'}\sum_{k=1}^{B'}\nabla f_i(x^t;\xi_{ik}^{t+1})\right)$$

$$- \left(\nabla f_i(x^{t+1}) - \nabla f_i(x^t) + \frac{b}{p_{\text{mega}}}\left(\nabla f_i(x^t)\right)\right)\bigg\|^2\Bigg]$$

$$= \frac{1}{B'^2}\sum_{k=1}^{B'}\mathrm{E}_k\Bigg[\bigg\|\frac{b}{p_{\text{mega}}}\left(\nabla f_i(x^{t+1};\xi_{ik}^{t+1}) - \nabla f_i(x^{t+1})\right)$$

$$+ \left(1 - \frac{b}{p_{\text{mega}}}\right)\left(\nabla f_i(x^{t+1};\xi_{ik}^{t+1}) - \nabla f_i(x^t;\xi_{ik}^{t+1}) - \left(\nabla f_i(x^{t+1}) - \nabla f_i(x^t)\right)\right)\bigg\|^2\Bigg],$$

where we used independence of the mini-batch samples. Using (15), we get

$$\mathrm{E}_k\left[\left\|k_{i,1}^{t+1} - \mathrm{E}_k\left[k_{i,1}^{t+1}\right]\right\|^2\right]$$

$$\leq \frac{2b^2}{B'^2 p_{\text{mega}}^2}\sum_{k=1}^{B'}\mathrm{E}_k\left[\left\|\nabla f_i(x^{t+1};\xi_{ik}^{t+1}) - \nabla f_i(x^{t+1})\right\|^2\right]$$

$$+ \frac{2}{B'^2}\left(1 - \frac{b}{p_{\text{mega}}}\right)^2\sum_{k=1}^{B'}\mathrm{E}_k\left[\left\|\nabla f_i(x^{t+1};\xi_{ik}^{t+1}) - \nabla f_i(x^t;\xi_{ik}^{t+1}) - \left(\nabla f_i(x^{t+1}) - \nabla f_i(x^t)\right)\right\|^2\right].$$

Due to Assumptions 5 and 6, we have

$$\mathrm{E}_k\left[\left\|k_{i,1}^{t+1} - \mathrm{E}_k\left[k_{i,1}^{t+1}\right]\right\|^2\right] \leq \frac{2b^2\sigma^2}{B' p_{\text{mega}}^2} + \frac{2L_\sigma^2}{B'}\left(1 - \frac{b}{p_{\text{mega}}}\right)^2\left\|x^{t+1} - x^t\right\|^2. \tag{36}$$

Next, we estimate the bound for $\mathrm{E}_k\left[\left\|k_{i,2}^{t+1} - \mathrm{E}_k\left[k_{i,2}^{t+1}\right]\right\|^2\right]$.

$$\mathrm{E}_k\left[\left\|k_{i,2}^{t+1} - \mathrm{E}_k\left[k_{i,2}^{t+1}\right]\right\|^2\right]$$

$$= \mathrm{E}_k\Bigg[\bigg\|\frac{1}{B}\sum_{j=1}^{B}\nabla f_i(x^{t+1};\xi_{ij}^{t+1}) - \frac{1}{B}\sum_{j=1}^{B}\nabla f_i(x^t;\xi_{ij}^{t+1}) - \left(\nabla f_i(x^{t+1}) - \nabla f_i(x^t)\right)\bigg\|^2\Bigg]$$

$$= \frac{1}{B^2}\sum_{j=1}^{B}\mathrm{E}_k\left[\left\|\nabla f_i(x^{t+1};\xi_{ij}^{t+1}) - \nabla f_i(x^t;\xi_{ij}^{t+1}) - \left(\nabla f_i(x^{t+1}) - \nabla f_i(x^t)\right)\right\|^2\right].$$

Due to Assumptions 6, we have

$$\mathrm{E}_k\left[\left\|k_{i,2}^{t+1} - \mathrm{E}_k\left[k_{i,2}^{t+1}\right]\right\|^2\right] \leq \frac{L_\sigma^2}{B}\left\|x^{t+1} - x^t\right\|^2. \tag{37}$$

Plugging (36) and (37) into (35), we obtain

$$\mathrm{E}_k\left[\mathrm{E}_{p_a}\left[\mathrm{E}_{p_{\text{mega}}}\left[\left\|h^{t+1} - \nabla f(x^{t+1})\right\|^2\right]\right]\right]$$

$$\leq \frac{p_{\text{mega}}}{n p_a}\left(\frac{2b^2\sigma^2}{B' p_{\text{mega}}^2} + \frac{2L_\sigma^2}{B'}\left(1 - \frac{b}{p_{\text{mega}}}\right)^2\left\|x^{t+1} - x^t\right\|^2\right)$$

$$+ \frac{(1 - p_{\text{mega}})L_\sigma^2}{n p_a B}\left\|x^{t+1} - x^t\right\|^2$$

$$+ \frac{2(p_a - p_{aa})}{n^2 p_a^2}\sum_{i=1}^{n}\left\|\nabla f_i(x^{t+1}) - \nabla f_i(x^t)\right\|^2$$

$$+ \frac{2\left(p_{\mathrm{a}} - p_{\mathrm{aa}}\right)b^2}{n^2 p_{\mathrm{a}}^2 p_{\mathrm{mega}}} \sum_{i=1}^{n} \left\| h_i^t - \nabla f_i(x^t) \right\|^2 + \left( p_{\mathrm{mega}} \left( 1 - \frac{b}{p_{\mathrm{mega}}} \right)^2 + (1 - p_{\mathrm{mega}}) \right) \left\| h^t - \nabla f(x^t) \right\|^2.$$

Using Assumption 3, we get

$$\mathrm{E}_k \left[ \mathrm{E}_{p_{\mathrm{a}}} \left[ \mathrm{E}_{p_{\mathrm{mega}}} \left[ \left\| h^{t+1} - \nabla f(x^{t+1}) \right\|^2 \right] \right] \right]$$

$$\leq \frac{2b^2 \sigma^2}{n p_{\mathrm{mega}} p_{\mathrm{a}} B'} + \left( \frac{2 p_{\mathrm{mega}} L_\sigma^2}{n p_{\mathrm{a}} B'} \left( 1 - \frac{b}{p_{\mathrm{mega}}} \right)^2 + \frac{(1 - p_{\mathrm{mega}}) L_\sigma^2}{n p_{\mathrm{a}} B} + \frac{2\left(p_{\mathrm{a}} - p_{\mathrm{aa}}\right)\widehat{L}^2}{n p_{\mathrm{a}}^2} \right) \left\| x^{t+1} - x^t \right\|^2$$

$$+ \frac{2\left(p_{\mathrm{a}} - p_{\mathrm{aa}}\right)b^2}{n^2 p_{\mathrm{a}}^2 p_{\mathrm{mega}}} \sum_{i=1}^{n} \left\| h_i^t - \nabla f_i(x^t) \right\|^2 + \left( p_{\mathrm{mega}} \left( 1 - \frac{b}{p_{\mathrm{mega}}} \right)^2 + (1 - p_{\mathrm{mega}}) \right) \left\| h^t - \nabla f(x^t) \right\|^2.$$

Using almost the same derivations, we can prove the second inequality:

$$\mathrm{E}_k \left[ \mathrm{E}_{p_{\mathrm{a}}} \left[ \mathrm{E}_{p_{\mathrm{mega}}} \left[ \left\| h_i^{t+1} - \nabla f_i(x^{t+1}) \right\|^2 \right] \right] \right]$$

$$= p_{\mathrm{mega}} \mathrm{E}_k \left[ \mathrm{E}_{p_{\mathrm{a}}} \left[ \left\| h_{i,1}^{t+1} - \nabla f_i(x^{t+1}) \right\|^2 \right] \right] + (1 - p_{\mathrm{mega}}) \mathrm{E}_k \left[ \mathrm{E}_{p_{\mathrm{a}}} \left[ \left\| h_{i,2}^{t+1} - \nabla f_i(x^{t+1}) \right\|^2 \right] \right]$$

$$\overset{(16)}{=} p_{\mathrm{mega}} \mathrm{E}_k \left[ \mathrm{E}_{p_{\mathrm{a}}} \left[ \left\| h_{i,1}^{t+1} - \mathrm{E}_k \left[ \mathrm{E}_{p_{\mathrm{a}}} \left[ h_{i,1}^{t+1} \right] \right] \right\|^2 \right] \right] + (1 - p_{\mathrm{mega}}) \mathrm{E}_k \left[ \mathrm{E}_{p_{\mathrm{a}}} \left[ \left\| h_{i,2}^{t+1} - \mathrm{E}_k \left[ \mathrm{E}_{p_{\mathrm{a}}} \left[ h_{i,2}^{t+1} \right] \right] \right\|^2 \right] \right]$$

$$+ \left( p_{\mathrm{mega}} \left( 1 - \frac{b}{p_{\mathrm{mega}}} \right)^2 + (1 - p_{\mathrm{mega}}) \right) \left\| h_i^t - \nabla f_i(x^t) \right\|^2$$

$$= p_{\mathrm{mega}} p_{\mathrm{a}} \mathrm{E}_k \left[ \left\| h_i^t + \frac{1}{p_{\mathrm{a}}} k_{i,1}^{t+1} - \left( h_i^t + \mathrm{E}_k \left[ k_{i,1}^{t+1} \right] \right) \right\|^2 \right]$$

$$+ p_{\mathrm{mega}} \left( 1 - p_{\mathrm{a}} \right) \left\| h_i^t - \left( h_i^t + \mathrm{E}_k \left[ k_{i,1}^{t+1} \right] \right) \right\|^2$$

$$+ (1 - p_{\mathrm{mega}}) p_{\mathrm{a}} \mathrm{E}_k \left[ \left\| h_i^t + \frac{1}{p_{\mathrm{a}}} k_{i,2}^{t+1} - \left( h_i^t + \mathrm{E}_k \left[ k_{i,2}^{t+1} \right] \right) \right\|^2 \right]$$

$$+ (1 - p_{\mathrm{mega}})(1 - p_{\mathrm{a}}) \left\| h_i^t - \left( h_i^t + \mathrm{E}_k \left[ k_{i,2}^{t+1} \right] \right) \right\|^2$$

$$+ \left( p_{\mathrm{mega}} \left( 1 - \frac{b}{p_{\mathrm{mega}}} \right)^2 + (1 - p_{\mathrm{mega}}) \right) \left\| h_i^t - \nabla f_i(x^t) \right\|^2$$

$$= p_{\mathrm{mega}} p_{\mathrm{a}} \mathrm{E}_k \left[ \left\| \frac{1}{p_{\mathrm{a}}} k_{i,1}^{t+1} - \mathrm{E}_k \left[ k_{i,1}^{t+1} \right] \right\|^2 \right]$$

$$+ p_{\mathrm{mega}} \left( 1 - p_{\mathrm{a}} \right) \left\| \nabla f_i(x^{t+1}) - \nabla f_i(x^t) - \frac{b}{p_{\mathrm{mega}}} \left( h_i^t - \nabla f_i(x^t) \right) \right\|^2$$

$$+ (1 - p_{\mathrm{mega}}) p_{\mathrm{a}} \mathrm{E}_k \left[ \left\| \frac{1}{p_{\mathrm{a}}} k_{i,2}^{t+1} - \mathrm{E}_k \left[ k_{i,2}^{t+1} \right] \right\|^2 \right]$$

$$+ (1 - p_{\mathrm{mega}})(1 - p_{\mathrm{a}}) \left\| \nabla f_i(x^{t+1}) - \nabla f_i(x^t) \right\|^2$$

$$+ \left( p_{\mathrm{mega}} \left( 1 - \frac{b}{p_{\mathrm{mega}}} \right)^2 + (1 - p_{\mathrm{mega}}) \right) \left\| h_i^t - \nabla f_i(x^t) \right\|^2$$

$$\overset{(16)}{=} \frac{p_{\mathrm{mega}}}{p_{\mathrm{a}}} \mathrm{E}_k \left[ \left\| k_{i,1}^{t+1} - \mathrm{E}_k \left[ k_{i,1}^{t+1} \right] \right\|^2 \right]$$

$$+ \frac{(1 - p_{\mathrm{mega}})}{p_{\mathrm{a}}} \mathrm{E}_k \left[ \left\| k_{i,2}^{t+1} - \mathrm{E}_k \left[ k_{i,2}^{t+1} \right] \right\|^2 \right]$$

$$+ \frac{p_{\mathrm{mega}} \left( 1 - p_{\mathrm{a}} \right)}{p_{\mathrm{a}}} \left\| \nabla f_i(x^{t+1}) - \nabla f_i(x^t) - \frac{b}{p_{\mathrm{mega}}} \left( h_i^t - \nabla f_i(x^t) \right) \right\|^2$$

$$+ \frac{(1 - p_{\mathrm{mega}})(1 - p_{\mathrm{a}})}{p_{\mathrm{a}}} \left\| \nabla f_i(x^{t+1}) - \nabla f_i(x^t) \right\|^2$$

$$+ \left( p_{\text{mega}} \left( 1 - \frac{b}{p_{\text{mega}}} \right)^2 + (1 - p_{\text{mega}}) \right) \left\| h_i^t - \nabla f_i(x^t) \right\|^2$$

$$\overset{(15)}{\leq} \frac{p_{\text{mega}}}{p_{\text{a}}} \mathrm{E}_k \left[ \left\| k_{i,1}^{t+1} - \mathrm{E}_k \left[ k_{i,1}^{t+1} \right] \right\|^2 \right]$$

$$+ \frac{(1 - p_{\text{mega}})}{p_{\text{a}}} \mathrm{E}_k \left[ \left\| k_{i,2}^{t+1} - \mathrm{E}_k \left[ k_{i,2}^{t+1} \right] \right\|^2 \right]$$

$$+ \frac{2(1 - p_{\text{a}})}{p_{\text{a}}} \left\| \nabla f_i(x^{t+1}) - \nabla f_i(x^t) \right\|^2$$

$$+ \frac{2(1 - p_{\text{a}}) b^2}{p_{\text{mega}} p_{\text{a}}} \left\| h_i^t - \nabla f_i(x^t) \right\|^2 + \left( p_{\text{mega}} \left( 1 - \frac{b}{p_{\text{mega}}} \right)^2 + (1 - p_{\text{mega}}) \right) \left\| h_i^t - \nabla f_i(x^t) \right\|^2 .$$

Using (36) and (37), we get

$$\mathrm{E}_k \left[ \mathrm{E}_{p_{\text{a}}} \left[ \mathrm{E}_{p_{\text{mega}}} \left[ \left\| h_i^{t+1} - \nabla f_i(x^{t+1}) \right\|^2 \right] \right] \right]$$

$$\leq \frac{2b^2 \sigma^2}{p_{\text{a}} p_{\text{mega}} B'} + \frac{2 p_{\text{mega}} L_\sigma^2}{p_{\text{a}} B'} \left( 1 - \frac{b}{p_{\text{mega}}} \right)^2 \left\| x^{t+1} - x^t \right\|^2$$

$$+ \frac{(1 - p_{\text{mega}}) L_\sigma^2}{p_{\text{a}} B} \left\| x^{t+1} - x^t \right\|^2$$

$$+ \frac{2(1 - p_{\text{a}})}{p_{\text{a}}} \left\| \nabla f_i(x^{t+1}) - \nabla f_i(x^t) \right\|^2$$

$$+ \frac{2(1 - p_{\text{a}}) b^2}{p_{\text{mega}} p_{\text{a}}} \left\| h_i^t - \nabla f_i(x^t) \right\|^2 + \left( p_{\text{mega}} \left( 1 - \frac{b}{p_{\text{mega}}} \right)^2 + (1 - p_{\text{mega}}) \right) \left\| h_i^t - \nabla f_i(x^t) \right\|^2 .$$

Next, due to Assumption 3, we obtain

$$\mathrm{E}_k \left[ \mathrm{E}_{p_{\text{a}}} \left[ \mathrm{E}_{p_{\text{mega}}} \left[ \left\| h_i^{t+1} - \nabla f_i(x^{t+1}) \right\|^2 \right] \right] \right]$$

$$\leq \frac{2b^2 \sigma^2}{p_{\text{a}} p_{\text{mega}} B'} + \left( \frac{2 p_{\text{mega}} L_\sigma^2}{p_{\text{a}} B'} \left( 1 - \frac{b}{p_{\text{mega}}} \right)^2 + \frac{(1 - p_{\text{mega}}) L_\sigma^2}{p_{\text{a}} B} + \frac{2(1 - p_{\text{a}}) L_i^2}{p_{\text{a}}} \right) \left\| x^{t+1} - x^t \right\|^2$$

$$+ \frac{2(1 - p_{\text{a}}) b^2}{p_{\text{mega}} p_{\text{a}}} \left\| h_i^t - \nabla f_i(x^t) \right\|^2 + \left( p_{\text{mega}} \left( 1 - \frac{b}{p_{\text{mega}}} \right)^2 + (1 - p_{\text{mega}}) \right) \left\| h_i^t - \nabla f_i(x^t) \right\|^2 .$$

The third inequality can be proved with the help of (37) and Assumption 3.

$$\mathrm{E}_k \left[ \left\| k_{i,2}^{t+1} \right\|^2 \right]$$

$$\overset{(16)}{=} \mathrm{E}_k \left[ \left\| k_{i,2}^{t+1} - \mathrm{E}_k \left[ k_{i,2}^{t+1} \right] \right\|^2 \right] + \left\| \nabla f_i(x^{t+1}) - \nabla f_i(x^t) \right\|^2$$

$$\leq \frac{L_\sigma^2}{B} \left\| x^{t+1} - x^t \right\|^2 + \left\| \nabla f_i(x^{t+1}) - \nabla f_i(x^t) \right\|^2$$

$$\leq \left( \frac{L_\sigma^2}{B} + L_i^2 \right) \left\| x^{t+1} - x^t \right\|^2 .$$

$\square$

**Theorem 11.** *Suppose that Assumptions 1, 2, 3, 5, 6, 7 and 8 hold. Let us take* $a = \frac{p_{\text{a}}}{2\omega + 1}$, $b = \frac{p_{\text{mega}} p_{\text{a}}}{2 - p_{\text{a}}}$, *probability* $p_{\text{mega}} \in (0,1]$, *batch size* $B' \geq B \geq 1$

$$\gamma \leq \left( L + \sqrt{ \frac{8(2\omega + 1)\omega}{n p_{\text{a}}^2} \left( \widehat{L}^2 + \frac{L_\sigma^2}{B} \right) + \frac{16}{n p_{\text{mega}} p_{\text{a}}^2} \left( \left( 1 - \frac{p_{\text{aa}}}{p_{\text{a}}} \right) \widehat{L}^2 + \frac{L_\sigma^2}{B} \right) } \right)^{-1},$$

*and* $h_i^0 = g_i^0$ *for all* $i \in [n]$ *in Algorithm 8. Then*

$$\mathrm{E} \left[ \left\| \nabla f(\widehat{x}^T) \right\|^2 \right] \leq \frac{1}{T} \left[ \frac{2\Delta_0}{\gamma} + \frac{4}{p_{\text{mega}} p_{\text{a}}} \left\| h^0 - \nabla f(x^0) \right\|^2 + \frac{4 \left( 1 - \frac{p_{\text{aa}}}{p_{\text{a}}} \right)}{n p_{\text{mega}} p_{\text{a}}} \frac{1}{n} \sum_{i=1}^n \left\| h_i^0 - \nabla f_i(x^0) \right\|^2 \right]$$

$$+ \frac{12\sigma^2}{nB'}.$$

*Proof.* Due to Lemma 2 and the update step from Line 5 in Algorithm 8, we have

$\mathrm{E}_{t+1}\left[f(x^{t+1})\right]$

$$\leq \mathrm{E}_{t+1}\left[f(x^t) - \frac{\gamma}{2}\left\|\nabla f(x^t)\right\|^2 - \left(\frac{1}{2\gamma} - \frac{L}{2}\right)\left\|x^{t+1} - x^t\right\|^2 + \frac{\gamma}{2}\left\|g^t - \nabla f(x^t)\right\|^2\right]$$

$$= \mathrm{E}_{t+1}\left[f(x^t) - \frac{\gamma}{2}\left\|\nabla f(x^t)\right\|^2 - \left(\frac{1}{2\gamma} - \frac{L}{2}\right)\left\|x^{t+1} - x^t\right\|^2 + \frac{\gamma}{2}\left\|g^t - h^t + h^t - \nabla f(x^t)\right\|^2\right]$$

$$\overset{(16)}{\leq} \mathrm{E}_{t+1}\left[f(x^t) - \frac{\gamma}{2}\left\|\nabla f(x^t)\right\|^2 - \left(\frac{1}{2\gamma} - \frac{L}{2}\right)\left\|x^{t+1} - x^t\right\|^2 + \gamma\left(\left\|g^t - h^t\right\|^2 + \left\|h^t - \nabla f(x^t)\right\|^2\right)\right].$$

Let us fix constants $\kappa, \eta, \nu, \rho \in [0, \infty)$ that we will define later. Considering Lemma 13, Lemma 14, and the law of total expectation, we obtain

$$\mathrm{E}\left[f(x^{t+1})\right] + \kappa\mathrm{E}\left[\left\|g^{t+1} - h^{t+1}\right\|^2\right] + \eta\mathrm{E}\left[\frac{1}{n}\sum_{i=1}^{n}\left\|g_i^{t+1} - h_i^{t+1}\right\|^2\right]$$

$$+ \nu\mathrm{E}\left[\left\|h^{t+1} - \nabla f(x^{t+1})\right\|^2\right] + \rho\mathrm{E}\left[\frac{1}{n}\sum_{i=1}^{n}\left\|h_i^{t+1} - \nabla f_i(x^{t+1})\right\|^2\right]$$

$$\leq \mathrm{E}\left[f(x^t) - \frac{\gamma}{2}\left\|\nabla f(x^t)\right\|^2 - \left(\frac{1}{2\gamma} - \frac{L}{2}\right)\left\|x^{t+1} - x^t\right\|^2 + \gamma\left(\left\|g^t - h^t\right\|^2 + \left\|h^t - \nabla f(x^t)\right\|^2\right)\right]$$

$$+ \kappa\mathrm{E}\left[\mathrm{E}_k\left[\mathrm{E}_{\mathcal{C}}\left[\mathrm{E}_{p_a}\left[\mathrm{E}_{p_{\mathrm{mega}}}\left[\left\|g^{t+1} - h^{t+1}\right\|^2\right]\right]\right]\right]\right]$$

$$+ \eta\mathrm{E}\left[\mathrm{E}_k\left[\mathrm{E}_{\mathcal{C}}\left[\mathrm{E}_{p_a}\left[\mathrm{E}_{p_{\mathrm{mega}}}\left[\frac{1}{n}\sum_{i=1}^{n}\left\|g_i^{t+1} - h_i^{t+1}\right\|^2\right]\right]\right]\right]\right]$$

$$+ \nu\mathrm{E}\left[\mathrm{E}_k\left[\mathrm{E}_{p_a}\left[\mathrm{E}_{p_{\mathrm{mega}}}\left[\left\|h^{t+1} - \nabla f(x^{t+1})\right\|^2\right]\right]\right]\right]$$

$$+ \rho\mathrm{E}\left[\mathrm{E}_k\left[\mathrm{E}_{p_a}\left[\mathrm{E}_{p_{\mathrm{mega}}}\left[\frac{1}{n}\sum_{i=1}^{n}\left\|h_i^{t+1} - \nabla f_i(x^{t+1})\right\|^2\right]\right]\right]\right]$$

$$\leq \mathrm{E}\left[f(x^t) - \frac{\gamma}{2}\left\|\nabla f(x^t)\right\|^2 - \left(\frac{1}{2\gamma} - \frac{L}{2}\right)\left\|x^{t+1} - x^t\right\|^2 + \gamma\left(\left\|g^t - h^t\right\|^2 + \left\|h^t - \nabla f(x^t)\right\|^2\right)\right]$$

$$+ \kappa\mathrm{E}\left(\frac{2(1 - p_{\mathrm{mega}})\omega}{np_a}\left(\frac{L_\sigma^2}{B} + \widehat{L}^2\right)\left\|x^{t+1} - x^t\right\|^2\right.$$

$$+ \left(\frac{(p_a - p_{aa})a^2}{n^2 p_a^2} + \frac{2(1 - p_{\mathrm{mega}})a^2\omega}{n^2 p_a}\right)\sum_{i=1}^{n}\left\|g_i^t - h_i^t\right\|^2 + (1 - a)^2\left\|g^t - h^t\right\|^2\right)$$

$$+ \eta\mathrm{E}\left(\frac{2(1 - p_{\mathrm{mega}})\omega}{p_a}\left(\frac{L_\sigma^2}{B} + \widehat{L}^2\right)\left\|x^{t+1} - x^t\right\|^2\right.$$

$$+ \left(\frac{(1 - p_a)a^2}{p_a} + \frac{2(1 - p_{\mathrm{mega}})a^2\omega}{p_a}\right)\frac{1}{n}\sum_{i=1}^{n}\left\|g_i^t - h_i^t\right\|^2 + (1 - a)^2\left\|g_i^t - h_i^t\right\|^2\right)$$

$$+ \nu\mathrm{E}\left(\frac{2b^2\sigma^2}{np_{\mathrm{mega}}p_a B'} + \left(\frac{2p_{\mathrm{mega}}L_\sigma^2}{np_a B'}\left(1 - \frac{b}{p_{\mathrm{mega}}}\right)^2 + \frac{(1 - p_{\mathrm{mega}})L_\sigma^2}{np_a B} + \frac{2(p_a - p_{aa})\widehat{L}^2}{np_a^2}\right)\left\|x^{t+1} - x^t\right\|^2\right.$$

$$+ \frac{2(p_a - p_{aa})b^2}{n^2 p_a^2 p_{\mathrm{mega}}}\sum_{i=1}^{n}\left\|h_i^t - \nabla f_i(x^t)\right\|^2 + \left(p_{\mathrm{mega}}\left(1 - \frac{b}{p_{\mathrm{mega}}}\right)^2 + (1 - p_{\mathrm{mega}})\right)\left\|h^t - \nabla f(x^t)\right\|^2\right)$$

$$+ \rho\mathrm{E}\left(\frac{2b^2\sigma^2}{p_a p_{\mathrm{mega}} B'} + \left(\frac{2p_{\mathrm{mega}}L_\sigma^2}{p_a B'}\left(1 - \frac{b}{p_{\mathrm{mega}}}\right)^2 + \frac{(1 - p_{\mathrm{mega}})L_\sigma^2}{p_a B} + \frac{2(1 - p_a)\widehat{L}^2}{p_a}\right)\left\|x^{t+1} - x^t\right\|^2\right.$$

$$+ \frac{2(1-p_{\mathrm{a}})b^2}{np_{\mathrm{mega}}p_{\mathrm{a}}} \sum_{i=1}^{n} \left\| h_i^t - \nabla f_i(x^t) \right\|^2 + \left( p_{\mathrm{mega}} \left(1 - \frac{b}{p_{\mathrm{mega}}}\right)^2 + (1 - p_{\mathrm{mega}}) \right) \frac{1}{n} \sum_{i=1}^{n} \left\| h_i^t - \nabla f_i(x^t) \right\|^2 \right).$$

Let us simplify the last inequality. Since $B' \geq B$ and $b = \frac{p_{\mathrm{mega}}p_{\mathrm{a}}}{2-p_{\mathrm{a}}} \leq p_{\mathrm{mega}}$, we have $1 - p_{\mathrm{mega}} \leq 1$,

$$\frac{2p_{\mathrm{mega}}L_\sigma^2}{p_{\mathrm{a}}B'} \left(1 - \frac{b}{p_{\mathrm{mega}}}\right)^2 \leq \frac{2p_{\mathrm{mega}}L_\sigma^2}{p_{\mathrm{a}}B},$$

$$\left( p_{\mathrm{mega}} \left(1 - \frac{b}{p_{\mathrm{mega}}}\right)^2 + (1 - p_{\mathrm{mega}}) \right) \leq 1 - b,$$

and

$$\left( \frac{2(1-p_{\mathrm{a}})b^2}{p_{\mathrm{mega}}p_{\mathrm{a}}} + p_{\mathrm{mega}} \left(1 - \frac{b}{p_{\mathrm{mega}}}\right)^2 + (1 - p_{\mathrm{mega}}) \right) \leq 1 - b.$$

Thus

$$\mathrm{E}\left[ f(x^{t+1}) \right] + \kappa \mathrm{E}\left[ \left\| g^{t+1} - h^{t+1} \right\|^2 \right] + \eta \mathrm{E}\left[ \frac{1}{n} \sum_{i=1}^{n} \left\| g_i^{t+1} - h_i^{t+1} \right\|^2 \right]$$

$$+ \nu \mathrm{E}\left[ \left\| h^{t+1} - \nabla f(x^{t+1}) \right\|^2 \right] + \rho \mathrm{E}\left[ \frac{1}{n} \sum_{i=1}^{n} \left\| h_i^{t+1} - \nabla f_i(x^{t+1}) \right\|^2 \right]$$

$$\leq \mathrm{E}\left[ f(x^t) - \frac{\gamma}{2} \left\| \nabla f(x^t) \right\|^2 - \left( \frac{1}{2\gamma} - \frac{L}{2} \right) \left\| x^{t+1} - x^t \right\|^2 + \gamma \left( \left\| g^t - h^t \right\|^2 + \left\| h^t - \nabla f(x^t) \right\|^2 \right) \right]$$

$$+ \kappa \mathrm{E}\left( \frac{2\omega}{np_{\mathrm{a}}} \left( \frac{L_\sigma^2}{B} + \widehat{L}^2 \right) \left\| x^{t+1} - x^t \right\|^2 \right.$$

$$+ \frac{((2\omega + 1)p_{\mathrm{a}} - p_{\mathrm{aa}})a^2}{n^2 p_{\mathrm{a}}^2} \sum_{i=1}^{n} \left\| g_i^t - h_i^t \right\|^2 + (1 - a)^2 \left\| g^t - h^t \right\|^2 \right)$$

$$+ \eta \mathrm{E}\left( \frac{2\omega}{p_{\mathrm{a}}} \left( \frac{L_\sigma^2}{B} + \widehat{L}^2 \right) \left\| x^{t+1} - x^t \right\|^2 \right.$$

$$+ \frac{(2\omega + 1 - p_{\mathrm{a}})a^2}{p_{\mathrm{a}}} \frac{1}{n} \sum_{i=1}^{n} \left\| g_i^t - h_i^t \right\|^2 + (1 - a)^2 \left\| g_i^t - h_i^t \right\|^2 \right)$$

$$+ \nu \mathrm{E}\left( \frac{2b^2\sigma^2}{np_{\mathrm{mega}}p_{\mathrm{a}}B'} + \left( \frac{2L_\sigma^2}{np_{\mathrm{a}}B} + \frac{2(p_{\mathrm{a}} - p_{\mathrm{aa}})\widehat{L}^2}{np_{\mathrm{a}}^2} \right) \left\| x^{t+1} - x^t \right\|^2 \right.$$

$$+ \frac{2(p_{\mathrm{a}} - p_{\mathrm{aa}})b^2}{n^2 p_{\mathrm{a}}^2 p_{\mathrm{mega}}} \sum_{i=1}^{n} \left\| h_i^t - \nabla f_i(x^t) \right\|^2 + (1 - b) \left\| h^t - \nabla f(x^t) \right\|^2 \right)$$

$$+ \rho \mathrm{E}\left( \frac{2b^2\sigma^2}{p_{\mathrm{a}}p_{\mathrm{mega}}B'} + \left( \frac{2L_\sigma^2}{p_{\mathrm{a}}B} + \frac{2(1-p_{\mathrm{a}})\widehat{L}^2}{p_{\mathrm{a}}} \right) \left\| x^{t+1} - x^t \right\|^2 \right.$$

$$+ (1 - b) \frac{1}{n} \sum_{i=1}^{n} \left\| h_i^t - \nabla f_i(x^t) \right\|^2 \right).$$

After rearranging the terms, we get

$$\mathrm{E}\left[ f(x^{t+1}) \right] + \kappa \mathrm{E}\left[ \left\| g^{t+1} - h^{t+1} \right\|^2 \right] + \eta \mathrm{E}\left[ \frac{1}{n} \sum_{i=1}^{n} \left\| g_i^{t+1} - h_i^{t+1} \right\|^2 \right]$$

$$+ \nu \mathrm{E}\left[ \left\| h^{t+1} - \nabla f(x^{t+1}) \right\|^2 \right] + \rho \mathrm{E}\left[ \frac{1}{n} \sum_{i=1}^{n} \left\| h_i^{t+1} - \nabla f_i(x^{t+1}) \right\|^2 \right]$$

$$\leq \mathrm{E}\left[f(x^t)\right] - \frac{\gamma}{2}\mathrm{E}\left[\left\|\nabla f(x^t)\right\|^2\right]$$

$$- \left(\frac{1}{2\gamma} - \frac{L}{2} - \frac{2\kappa\omega}{np_{\mathrm{a}}}\left(\frac{L_\sigma^2}{B} + \widehat{L}^2\right) - \frac{2\eta\omega}{p_{\mathrm{a}}}\left(\frac{L_\sigma^2}{B} + \widehat{L}^2\right)\right.$$

$$\left. - \nu\left(\frac{2L_\sigma^2}{np_{\mathrm{a}}B} + \frac{2\left(p_{\mathrm{a}} - p_{\mathrm{aa}}\right)\widehat{L}^2}{np_{\mathrm{a}}^2}\right) - \rho\left(\frac{2L_\sigma^2}{p_{\mathrm{a}}B} + \frac{2(1 - p_{\mathrm{a}})\widehat{L}^2}{p_{\mathrm{a}}}\right)\right)\mathrm{E}\left[\left\|x^{t+1} - x^t\right\|^2\right]$$

$$+ \left(\gamma + \kappa\left(1 - a\right)^2\right)\mathrm{E}\left[\left\|g^t - h^t\right\|^2\right]$$

$$+ \left(\kappa\frac{\left(\left(2\omega + 1\right)p_{\mathrm{a}} - p_{\mathrm{aa}}\right)a^2}{np_{\mathrm{a}}^2} + \eta\left(\frac{(2\omega + 1 - p_{\mathrm{a}})a^2}{p_{\mathrm{a}}} + (1 - a)^2\right)\right)\mathrm{E}\left[\frac{1}{n}\sum_{i=1}^n\left\|g_i^t - h_i^t\right\|^2\right]$$

$$+ \left(\gamma + \nu\left(1 - b\right)\right)\mathrm{E}\left[\left\|h^t - \nabla f(x^t)\right\|^2\right]$$

$$+ \left(\nu\frac{2\left(p_{\mathrm{a}} - p_{\mathrm{aa}}\right)b^2}{np_{\mathrm{a}}^2 p_{\mathrm{mega}}} + \rho(1 - b)\right)\mathrm{E}\left[\frac{1}{n}\sum_{i=1}^n\left\|h_i^t - \nabla f_i(x^t)\right\|^2\right]$$

$$+ \left(\frac{2\nu b^2}{np_{\mathrm{mega}}p_{\mathrm{a}}} + \frac{2\rho b^2}{p_{\mathrm{a}}p_{\mathrm{mega}}}\right)\frac{\sigma^2}{B'}.$$

Let us take $\kappa = \frac{\gamma}{a}$, thus $\gamma + \kappa\left(1 - a\right)^2 \leq \kappa$ and

$$\mathrm{E}\left[f(x^{t+1})\right] + \frac{\gamma}{a}\mathrm{E}\left[\left\|g^{t+1} - h^{t+1}\right\|^2\right] + \eta\mathrm{E}\left[\frac{1}{n}\sum_{i=1}^n\left\|g_i^{t+1} - h_i^{t+1}\right\|^2\right]$$

$$+ \nu\mathrm{E}\left[\left\|h^{t+1} - \nabla f(x^{t+1})\right\|^2\right] + \rho\mathrm{E}\left[\frac{1}{n}\sum_{i=1}^n\left\|h_i^{t+1} - \nabla f_i(x^{t+1})\right\|^2\right]$$

$$\leq \mathrm{E}\left[f(x^t)\right] - \frac{\gamma}{2}\mathrm{E}\left[\left\|\nabla f(x^t)\right\|^2\right]$$

$$- \left(\frac{1}{2\gamma} - \frac{L}{2} - \frac{2\gamma\omega}{anp_{\mathrm{a}}}\left(\frac{L_\sigma^2}{B} + \widehat{L}^2\right) - \frac{2\eta\omega}{p_{\mathrm{a}}}\left(\frac{L_\sigma^2}{B} + \widehat{L}^2\right)\right.$$

$$\left. - \nu\left(\frac{2L_\sigma^2}{np_{\mathrm{a}}B} + \frac{2\left(p_{\mathrm{a}} - p_{\mathrm{aa}}\right)\widehat{L}^2}{np_{\mathrm{a}}^2}\right) - \rho\left(\frac{2L_\sigma^2}{p_{\mathrm{a}}B} + \frac{2(1 - p_{\mathrm{a}})\widehat{L}^2}{p_{\mathrm{a}}}\right)\right)\mathrm{E}\left[\left\|x^{t+1} - x^t\right\|^2\right]$$

$$+ \frac{\gamma}{a}\mathrm{E}\left[\left\|g^t - h^t\right\|^2\right]$$

$$+ \left(\frac{\gamma\left(\left(2\omega + 1\right)p_{\mathrm{a}} - p_{\mathrm{aa}}\right)a}{np_{\mathrm{a}}^2} + \eta\left(\frac{(2\omega + 1 - p_{\mathrm{a}})a^2}{p_{\mathrm{a}}} + (1 - a)^2\right)\right)\mathrm{E}\left[\frac{1}{n}\sum_{i=1}^n\left\|g_i^t - h_i^t\right\|^2\right]$$

$$+ \left(\gamma + \nu\left(1 - b\right)\right)\mathrm{E}\left[\left\|h^t - \nabla f(x^t)\right\|^2\right]$$

$$+ \left(\nu\frac{2\left(p_{\mathrm{a}} - p_{\mathrm{aa}}\right)b^2}{np_{\mathrm{a}}^2 p_{\mathrm{mega}}} + \rho(1 - b)\right)\mathrm{E}\left[\frac{1}{n}\sum_{i=1}^n\left\|h_i^t - \nabla f_i(x^t)\right\|^2\right]$$

$$+ \left(\frac{2\nu b^2}{np_{\mathrm{mega}}p_{\mathrm{a}}} + \frac{2\rho b^2}{p_{\mathrm{a}}p_{\mathrm{mega}}}\right)\frac{\sigma^2}{B'}.$$

Next, since $a = \frac{p_{\mathrm{a}}}{2\omega + 1}$, we have $\left(\frac{(2\omega + 1 - p_{\mathrm{a}})a^2}{p_{\mathrm{a}}} + (1 - a)^2\right) \leq 1 - a$. We the choice $\eta = \frac{\gamma\left(\left(2\omega + 1\right)p_{\mathrm{a}} - p_{\mathrm{aa}}\right)}{np_{\mathrm{a}}^2}$, we guarantee $\frac{\gamma\left(\left(2\omega + 1\right)p_{\mathrm{a}} - p_{\mathrm{aa}}\right)a}{np_{\mathrm{a}}^2} + \eta\left(\frac{(2\omega + 1 - p_{\mathrm{a}})a^2}{p_{\mathrm{a}}} + (1 - a)^2\right) \leq \eta$ and

$$\mathrm{E}\left[f(x^{t+1})\right] + \frac{\gamma\left(2\omega + 1\right)}{p_{\mathrm{a}}}\mathrm{E}\left[\left\|g^{t+1} - h^{t+1}\right\|^2\right] + \frac{\gamma\left(\left(2\omega + 1\right)p_{\mathrm{a}} - p_{\mathrm{aa}}\right)}{np_{\mathrm{a}}^2}\mathrm{E}\left[\frac{1}{n}\sum_{i=1}^n\left\|g_i^{t+1} - h_i^{t+1}\right\|^2\right]$$

$$+ \nu\mathrm{E}\left[\left\|h^{t+1} - \nabla f(x^{t+1})\right\|^2\right] + \rho\mathrm{E}\left[\frac{1}{n}\sum_{i=1}^n\left\|h_i^{t+1} - \nabla f_i(x^{t+1})\right\|^2\right]$$

$$
\leq \mathrm{E}\left[f(x^t)\right] - \frac{\gamma}{2}\mathrm{E}\left[\left\|\nabla f(x^t)\right\|^2\right]
$$

$$
- \left(\frac{1}{2\gamma} - \frac{L}{2} - \frac{2\gamma\left(2\omega+1\right)\omega}{np_{\mathrm{a}}^2}\left(\frac{L_\sigma^2}{B} + \widehat{L}^2\right) - \frac{2\gamma\left(\left(2\omega+1\right)p_{\mathrm{a}} - p_{\mathrm{aa}}\right)\omega}{np_{\mathrm{a}}^3}\left(\frac{L_\sigma^2}{B} + \widehat{L}^2\right)\right.
$$

$$
\left. - \nu\left(\frac{2L_\sigma^2}{np_{\mathrm{a}}B} + \frac{2\left(p_{\mathrm{a}} - p_{\mathrm{aa}}\right)\widehat{L}^2}{np_{\mathrm{a}}^2}\right) - \rho\left(\frac{2L_\sigma^2}{p_{\mathrm{a}}B} + \frac{2(1-p_{\mathrm{a}})\widehat{L}^2}{p_{\mathrm{a}}}\right)\right)\mathrm{E}\left[\left\|x^{t+1} - x^t\right\|^2\right]
$$

$$
+ \frac{\gamma\left(2\omega+1\right)}{p_{\mathrm{a}}}\mathrm{E}\left[\left\|g^t - h^t\right\|^2\right] + \frac{\gamma\left(\left(2\omega+1\right)p_{\mathrm{a}} - p_{\mathrm{aa}}\right)}{np_{\mathrm{a}}^2}\mathrm{E}\left[\frac{1}{n}\sum_{i=1}^n\left\|g_i^t - h_i^t\right\|^2\right]
$$

$$
+ \left(\gamma + \nu\left(1-b\right)\right)\mathrm{E}\left[\left\|h^t - \nabla f(x^t)\right\|^2\right]
$$

$$
+ \left(\nu\frac{2\left(p_{\mathrm{a}} - p_{\mathrm{aa}}\right)b^2}{np_{\mathrm{a}}^2 p_{\mathrm{mega}}} + \rho(1-b)\right)\mathrm{E}\left[\frac{1}{n}\sum_{i=1}^n\left\|h_i^t - \nabla f_i(x^t)\right\|^2\right]
$$

$$
+ \left(\frac{2\nu b^2}{np_{\mathrm{mega}}p_{\mathrm{a}}} + \frac{2\rho b^2}{p_{\mathrm{a}}p_{\mathrm{mega}}}\right)\frac{\sigma^2}{B'}
$$

$$
\leq \mathrm{E}\left[f(x^t)\right] - \frac{\gamma}{2}\mathrm{E}\left[\left\|\nabla f(x^t)\right\|^2\right]
$$

$$
- \left(\frac{1}{2\gamma} - \frac{L}{2} - \frac{4\gamma\left(2\omega+1\right)\omega}{np_{\mathrm{a}}^2}\left(\frac{L_\sigma^2}{B} + \widehat{L}^2\right)\right.
$$

$$
\left. - \nu\left(\frac{2L_\sigma^2}{np_{\mathrm{a}}B} + \frac{2\left(p_{\mathrm{a}} - p_{\mathrm{aa}}\right)\widehat{L}^2}{np_{\mathrm{a}}^2}\right) - \rho\left(\frac{2L_\sigma^2}{p_{\mathrm{a}}B} + \frac{2(1-p_{\mathrm{a}})\widehat{L}^2}{p_{\mathrm{a}}}\right)\right)\mathrm{E}\left[\left\|x^{t+1} - x^t\right\|^2\right]
$$

$$
+ \frac{\gamma\left(2\omega+1\right)}{p_{\mathrm{a}}}\mathrm{E}\left[\left\|g^t - h^t\right\|^2\right] + \frac{\gamma\left(\left(2\omega+1\right)p_{\mathrm{a}} - p_{\mathrm{aa}}\right)}{np_{\mathrm{a}}^2}\mathrm{E}\left[\frac{1}{n}\sum_{i=1}^n\left\|g_i^t - h_i^t\right\|^2\right]
$$

$$
+ \left(\gamma + \nu\left(1-b\right)\right)\mathrm{E}\left[\left\|h^t - \nabla f(x^t)\right\|^2\right]
$$

$$
+ \left(\nu\frac{2\left(p_{\mathrm{a}} - p_{\mathrm{aa}}\right)b^2}{np_{\mathrm{a}}^2 p_{\mathrm{mega}}} + \rho(1-b)\right)\mathrm{E}\left[\frac{1}{n}\sum_{i=1}^n\left\|h_i^t - \nabla f_i(x^t)\right\|^2\right]
$$

$$
+ \left(\frac{2\nu b^2}{np_{\mathrm{mega}}p_{\mathrm{a}}} + \frac{2\rho b^2}{p_{\mathrm{a}}p_{\mathrm{mega}}}\right)\frac{\sigma^2}{B'},
$$

where simplified the term using $p_{\mathrm{aa}} \geq 0$. Let us take $\nu = \frac{\gamma}{b}$ to obtain

$$
\mathrm{E}\left[f(x^{t+1})\right] + \frac{\gamma\left(2\omega+1\right)}{p_{\mathrm{a}}}\mathrm{E}\left[\left\|g^{t+1} - h^{t+1}\right\|^2\right] + \frac{\gamma\left(\left(2\omega+1\right)p_{\mathrm{a}} - p_{\mathrm{aa}}\right)}{np_{\mathrm{a}}^2}\mathrm{E}\left[\frac{1}{n}\sum_{i=1}^n\left\|g_i^{t+1} - h_i^{t+1}\right\|^2\right]
$$

$$
+ \frac{\gamma}{b}\mathrm{E}\left[\left\|h^{t+1} - \nabla f(x^{t+1})\right\|^2\right] + \rho\mathrm{E}\left[\frac{1}{n}\sum_{i=1}^n\left\|h_i^{t+1} - \nabla f_i(x^{t+1})\right\|^2\right]
$$

$$
\leq \mathrm{E}\left[f(x^t)\right] - \frac{\gamma}{2}\mathrm{E}\left[\left\|\nabla f(x^t)\right\|^2\right]
$$

$$
- \left(\frac{1}{2\gamma} - \frac{L}{2} - \frac{4\gamma\left(2\omega+1\right)\omega}{np_{\mathrm{a}}^2}\left(\frac{L_\sigma^2}{B} + \widehat{L}^2\right)\right.
$$

$$
\left. - \left(\frac{2\gamma L_\sigma^2}{bnp_{\mathrm{a}}B} + \frac{2\gamma\left(p_{\mathrm{a}} - p_{\mathrm{aa}}\right)\widehat{L}^2}{bnp_{\mathrm{a}}^2}\right) - \rho\left(\frac{2L_\sigma^2}{p_{\mathrm{a}}B} + \frac{2(1-p_{\mathrm{a}})\widehat{L}^2}{p_{\mathrm{a}}}\right)\right)\mathrm{E}\left[\left\|x^{t+1} - x^t\right\|^2\right]
$$

$$
+ \frac{\gamma\left(2\omega+1\right)}{p_{\mathrm{a}}}\mathrm{E}\left[\left\|g^t - h^t\right\|^2\right] + \frac{\gamma\left(\left(2\omega+1\right)p_{\mathrm{a}} - p_{\mathrm{aa}}\right)}{np_{\mathrm{a}}^2}\mathrm{E}\left[\frac{1}{n}\sum_{i=1}^n\left\|g_i^t - h_i^t\right\|^2\right]
$$

$$
+ \frac{\gamma}{b}\mathrm{E}\left[\left\|h^t - \nabla f(x^t)\right\|^2\right]
$$

$$+ \left( \frac{2\gamma \left(p_{\mathrm{a}} - p_{\mathrm{aa}}\right) b}{n p_{\mathrm{a}}^2 p_{\mathrm{mega}}} + \rho(1 - b) \right) \mathrm{E} \left[ \frac{1}{n} \sum_{i=1}^n \left\| h_i^t - \nabla f_i(x^t) \right\|^2 \right]$$

$$+ \left( \frac{2\gamma b}{n p_{\mathrm{mega}} p_{\mathrm{a}}} + \frac{2\rho b^2}{p_{\mathrm{a}} p_{\mathrm{mega}}} \right) \frac{\sigma^2}{B'}.$$

Next, we take $\rho = \frac{2\gamma(p_{\mathrm{a}} - p_{\mathrm{aa}})}{n p_{\mathrm{a}}^2 p_{\mathrm{mega}}}$, thus

$$\mathrm{E}\left[ f(x^{t+1}) \right] + \frac{\gamma \left(2\omega + 1\right)}{p_{\mathrm{a}}} \mathrm{E}\left[ \left\| g^{t+1} - h^{t+1} \right\|^2 \right] + \frac{\gamma \left(\left(2\omega + 1\right) p_{\mathrm{a}} - p_{\mathrm{aa}}\right)}{n p_{\mathrm{a}}^2} \mathrm{E}\left[ \frac{1}{n} \sum_{i=1}^n \left\| g_i^{t+1} - h_i^{t+1} \right\|^2 \right]$$

$$+ \frac{\gamma}{b} \mathrm{E}\left[ \left\| h^{t+1} - \nabla f(x^{t+1}) \right\|^2 \right] + \frac{2\gamma \left(p_{\mathrm{a}} - p_{\mathrm{aa}}\right)}{n p_{\mathrm{a}}^2 p_{\mathrm{mega}}} \mathrm{E}\left[ \frac{1}{n} \sum_{i=1}^n \left\| h_i^{t+1} - \nabla f_i(x^{t+1}) \right\|^2 \right]$$

$$\leq \mathrm{E}\left[ f(x^t) \right] - \frac{\gamma}{2} \mathrm{E}\left[ \left\| \nabla f(x^t) \right\|^2 \right]$$

$$- \left( \frac{1}{2\gamma} - \frac{L}{2} - \frac{4\gamma \left(2\omega + 1\right) \omega}{n p_{\mathrm{a}}^2} \left( \frac{L_\sigma^2}{B} + \widehat{L}^2 \right) \right.$$

$$- \left( \frac{2\gamma L_\sigma^2}{b n p_{\mathrm{a}} B} + \frac{2\gamma \left(p_{\mathrm{a}} - p_{\mathrm{aa}}\right) \widehat{L}^2}{b n p_{\mathrm{a}}^2} \right) - \left( \frac{2\gamma \left(p_{\mathrm{a}} - p_{\mathrm{aa}}\right)}{n p_{\mathrm{a}}^2 p_{\mathrm{mega}}} \right) \left( \frac{2 L_\sigma^2}{p_{\mathrm{a}} B} + \frac{2(1 - p_{\mathrm{a}}) \widehat{L}^2}{p_{\mathrm{a}}} \right) \right) \mathrm{E}\left[ \left\| x^{t+1} - x^t \right\|^2 \right]$$

$$+ \frac{\gamma \left(2\omega + 1\right)}{p_{\mathrm{a}}} \mathrm{E}\left[ \left\| g^t - h^t \right\|^2 \right] + \frac{\gamma \left(\left(2\omega + 1\right) p_{\mathrm{a}} - p_{\mathrm{aa}}\right)}{n p_{\mathrm{a}}^2} \mathrm{E}\left[ \frac{1}{n} \sum_{i=1}^n \left\| g_i^t - h_i^t \right\|^2 \right]$$

$$+ \frac{\gamma}{b} \mathrm{E}\left[ \left\| h^t - \nabla f(x^t) \right\|^2 \right] + \frac{2\gamma \left(p_{\mathrm{a}} - p_{\mathrm{aa}}\right)}{n p_{\mathrm{a}}^2 p_{\mathrm{mega}}} \mathrm{E}\left[ \frac{1}{n} \sum_{i=1}^n \left\| h_i^t - \nabla f_i(x^t) \right\|^2 \right]$$

$$+ \left( \frac{2\gamma b}{n p_{\mathrm{mega}} p_{\mathrm{a}}} + \frac{4\gamma \left(p_{\mathrm{a}} - p_{\mathrm{aa}}\right) b^2}{n p_{\mathrm{a}}^3 p_{\mathrm{mega}}^2} \right) \frac{\sigma^2}{B'}.$$

Since $\frac{p_{\mathrm{mega}} p_{\mathrm{a}}}{2} \leq b \leq p_{\mathrm{mega}} p_{\mathrm{a}}$ and $1 - p_{\mathrm{a}} \leq 1 - \frac{p_{\mathrm{aa}}}{p_{\mathrm{a}}} \leq 1$, we get

$$\mathrm{E}\left[ f(x^{t+1}) \right] + \frac{\gamma \left(2\omega + 1\right)}{p_{\mathrm{a}}} \mathrm{E}\left[ \left\| g^{t+1} - h^{t+1} \right\|^2 \right] + \frac{\gamma \left(\left(2\omega + 1\right) p_{\mathrm{a}} - p_{\mathrm{aa}}\right)}{n p_{\mathrm{a}}^2} \mathrm{E}\left[ \frac{1}{n} \sum_{i=1}^n \left\| g_i^{t+1} - h_i^{t+1} \right\|^2 \right]$$

$$+ \frac{\gamma}{b} \mathrm{E}\left[ \left\| h^{t+1} - \nabla f(x^{t+1}) \right\|^2 \right] + \frac{2\gamma \left(p_{\mathrm{a}} - p_{\mathrm{aa}}\right)}{n p_{\mathrm{a}}^2 p_{\mathrm{mega}}} \mathrm{E}\left[ \frac{1}{n} \sum_{i=1}^n \left\| h_i^{t+1} - \nabla f_i(x^{t+1}) \right\|^2 \right]$$

$$\leq \mathrm{E}\left[ f(x^t) \right] - \frac{\gamma}{2} \mathrm{E}\left[ \left\| \nabla f(x^t) \right\|^2 \right]$$

$$- \left( \frac{1}{2\gamma} - \frac{L}{2} - \frac{4\gamma \left(2\omega + 1\right) \omega}{n p_{\mathrm{a}}^2} \left( \frac{L_\sigma^2}{B} + \widehat{L}^2 \right) \right.$$

$$- \left( \frac{4\gamma L_\sigma^2}{n p_{\mathrm{mega}} p_{\mathrm{a}}^2 B} + \frac{4\gamma \left(p_{\mathrm{a}} - p_{\mathrm{aa}}\right) \widehat{L}^2}{n p_{\mathrm{mega}} p_{\mathrm{a}}^3} \right) - \left( \frac{4\gamma L_\sigma^2}{n p_{\mathrm{mega}} p_{\mathrm{a}}^2 B} + \frac{4\gamma(1 - p_{\mathrm{a}}) \widehat{L}^2}{n p_{\mathrm{mega}} p_{\mathrm{a}}^2} \right) \right) \mathrm{E}\left[ \left\| x^{t+1} - x^t \right\|^2 \right]$$

$$+ \frac{\gamma \left(2\omega + 1\right)}{p_{\mathrm{a}}} \mathrm{E}\left[ \left\| g^t - h^t \right\|^2 \right] + \frac{\gamma \left(\left(2\omega + 1\right) p_{\mathrm{a}} - p_{\mathrm{aa}}\right)}{n p_{\mathrm{a}}^2} \mathrm{E}\left[ \frac{1}{n} \sum_{i=1}^n \left\| g_i^t - h_i^t \right\|^2 \right]$$

$$+ \frac{\gamma}{b} \mathrm{E}\left[ \left\| h^t - \nabla f(x^t) \right\|^2 \right] + \frac{2\gamma \left(p_{\mathrm{a}} - p_{\mathrm{aa}}\right)}{n p_{\mathrm{a}}^2 p_{\mathrm{mega}}} \mathrm{E}\left[ \frac{1}{n} \sum_{i=1}^n \left\| h_i^t - \nabla f_i(x^t) \right\|^2 \right]$$

$$+ \frac{6\gamma \sigma^2}{n B'}$$

$$\leq \mathrm{E}\left[ f(x^t) \right] - \frac{\gamma}{2} \mathrm{E}\left[ \left\| \nabla f(x^t) \right\|^2 \right]$$

$$-\left(\frac{1}{2\gamma} - \frac{L}{2} - \frac{4\gamma(2\omega+1)\omega}{np_{\mathrm{a}}^2}\left(\frac{L_\sigma^2}{B} + \widehat{L}^2\right) - \left(\frac{8\gamma L_\sigma^2}{np_{\mathrm{mega}}p_{\mathrm{a}}^2 B} + \frac{8\gamma\left(1 - \frac{p_{\mathrm{aa}}}{p_{\mathrm{a}}}\right)\widehat{L}^2}{np_{\mathrm{mega}}p_{\mathrm{a}}^2}\right)\right)\mathrm{E}\left[\|x^{t+1} - x^t\|^2\right]$$

$$+ \frac{\gamma(2\omega+1)}{p_{\mathrm{a}}}\mathrm{E}\left[\|g^t - h^t\|^2\right] + \frac{\gamma((2\omega+1)p_{\mathrm{a}} - p_{\mathrm{aa}})}{np_{\mathrm{a}}^2}\mathrm{E}\left[\frac{1}{n}\sum_{i=1}^{n}\|g_i^t - h_i^t\|^2\right]$$

$$+ \frac{\gamma}{b}\mathrm{E}\left[\|h^t - \nabla f(x^t)\|^2\right] + \frac{2\gamma(p_{\mathrm{a}} - p_{\mathrm{aa}})}{np_{\mathrm{a}}^2 p_{\mathrm{mega}}}\mathrm{E}\left[\frac{1}{n}\sum_{i=1}^{n}\|h_i^t - \nabla f_i(x^t)\|^2\right]$$

$$+ \frac{6\gamma\sigma^2}{nB'}.$$

Using Lemma 4 and the assumption about $\gamma$, we get

$$\mathrm{E}\left[f(x^{t+1})\right] + \frac{\gamma(2\omega+1)}{p_{\mathrm{a}}}\mathrm{E}\left[\|g^{t+1} - h^{t+1}\|^2\right] + \frac{\gamma((2\omega+1)p_{\mathrm{a}} - p_{\mathrm{aa}})}{np_{\mathrm{a}}^2}\mathrm{E}\left[\frac{1}{n}\sum_{i=1}^{n}\|g_i^{t+1} - h_i^{t+1}\|^2\right]$$

$$+ \frac{\gamma}{b}\mathrm{E}\left[\|h^{t+1} - \nabla f(x^{t+1})\|^2\right] + \frac{2\gamma(p_{\mathrm{a}} - p_{\mathrm{aa}})}{np_{\mathrm{a}}^2 p_{\mathrm{mega}}}\mathrm{E}\left[\frac{1}{n}\sum_{i=1}^{n}\|h_i^{t+1} - \nabla f_i(x^{t+1})\|^2\right]$$

$$\leq \mathrm{E}\left[f(x^t)\right] - \frac{\gamma}{2}\mathrm{E}\left[\|\nabla f(x^t)\|^2\right]$$

$$+ \frac{\gamma(2\omega+1)}{p_{\mathrm{a}}}\mathrm{E}\left[\|g^t - h^t\|^2\right] + \frac{\gamma((2\omega+1)p_{\mathrm{a}} - p_{\mathrm{aa}})}{np_{\mathrm{a}}^2}\mathrm{E}\left[\frac{1}{n}\sum_{i=1}^{n}\|g_i^t - h_i^t\|^2\right]$$

$$+ \frac{\gamma}{b}\mathrm{E}\left[\|h^t - \nabla f(x^t)\|^2\right] + \frac{2\gamma(p_{\mathrm{a}} - p_{\mathrm{aa}})}{np_{\mathrm{a}}^2 p_{\mathrm{mega}}}\mathrm{E}\left[\frac{1}{n}\sum_{i=1}^{n}\|h_i^t - \nabla f_i(x^t)\|^2\right]$$

$$+ \frac{6\gamma\sigma^2}{nB'}.$$

It is left to apply Lemma 3 with

$$\Psi^t = \frac{(2\omega+1)}{p_{\mathrm{a}}}\mathrm{E}\left[\|g^t - h^t\|^2\right] + \frac{((2\omega+1)p_{\mathrm{a}} - p_{\mathrm{aa}})}{np_{\mathrm{a}}^2}\mathrm{E}\left[\frac{1}{n}\sum_{i=1}^{n}\|g_i^t - h_i^t\|^2\right]$$

$$+ \frac{1}{b}\mathrm{E}\left[\|h^t - \nabla f(x^t)\|^2\right] + \frac{2\left(1 - \frac{p_{\mathrm{aa}}}{p_{\mathrm{a}}}\right)}{np_{\mathrm{a}}p_{\mathrm{mega}}}\mathrm{E}\left[\frac{1}{n}\sum_{i=1}^{n}\|h_i^t - \nabla f_i(x^t)\|^2\right]$$

and $C = \frac{6\sigma^2}{nB'}$ to conclude the proof. $\qquad\square$

**Corollary 6.** *Suppose that assumptions from Theorem 11 hold, probability* $p_{mega} = \min\left\{\frac{\zeta_C}{d}, \frac{n\varepsilon B}{\sigma^2}\right\}$, *batch size* $B' = \Theta\left(\frac{\sigma^2}{n\varepsilon}\right)$, *and* $h_i^0 = g_i^0 = \frac{1}{B_{\mathrm{init}}}\sum_{k=1}^{B_{\mathrm{init}}}\nabla f_i(x^0; \xi_{ik}^0)$ *for all* $i \in [n]$, *initial batch size* $B_{\mathrm{init}} = \Theta\left(\frac{B}{p_{mega}\sqrt{p_{\mathrm{a}}}}\right) = \Theta\left(\max\left\{\frac{Bd}{\sqrt{p_{\mathrm{a}}}\zeta_C}, \frac{\sigma^2}{\sqrt{p_{\mathrm{a}}}n\varepsilon}\right\}\right)$, *then* DASHA-PP-SYNC-MVR *needs*

$$T := \mathcal{O}\left(\frac{\Delta_0}{\varepsilon}\left[L + \left(\frac{\omega}{p_{\mathrm{a}}\sqrt{n}} + \sqrt{\frac{d}{p_{\mathrm{a}}^2\zeta_C n}}\right)\left(\widehat{L} + \frac{L_\sigma}{\sqrt{B}}\right) + \frac{\sigma}{p_{\mathrm{a}}\sqrt{\varepsilon}n}\left(\frac{\widehat{L}}{\sqrt{B}} + \frac{L_\sigma}{B}\right)\right] + \frac{\sigma^2}{\sqrt{p_{\mathrm{a}}}n\varepsilon B}\right).$$

*communication rounds to get an* $\varepsilon$*-solution, the expected communication complexity is equal to* $\mathcal{O}\left(d + \zeta_C T\right)$, *and the expected number of stochastic gradient calculations per node equals* $\mathcal{O}(B_{\mathrm{init}} + BT)$, *where* $\zeta_C$ *is the expected density from Definition 12.*

*Proof.* Due to the choice of $B'$, we have

$$\mathrm{E}\left[\left\|\nabla f(\widehat{x}^T)\right\|^2\right] \leq \frac{1}{T}\left[2\Delta_0\left(L + \sqrt{\frac{8\left(2\omega+1\right)\omega}{np_\mathrm{a}^2}\left(\widehat{L}^2 + \frac{L_\sigma^2}{B}\right) + \frac{16}{np_\mathrm{mega}p_\mathrm{a}^2}\left(\left(1 - \frac{p_\mathrm{aa}}{p_\mathrm{a}}\right)\widehat{L}^2 + \frac{L_\sigma^2}{B}\right)}\right)\right.$$

$$\left. + \frac{4}{p_\mathrm{mega}p_\mathrm{a}}\left\|h^0 - \nabla f(x^0)\right\|^2 + \frac{4\left(1 - \frac{p_\mathrm{aa}}{p_\mathrm{a}}\right)}{np_\mathrm{mega}p_\mathrm{a}}\frac{1}{n}\sum_{i=1}^n\left\|h_i^0 - \nabla f_i(x^0)\right\|^2\right]$$

$$+ \frac{2\varepsilon}{3}.$$

Using

$$\mathrm{E}\left[\left\|h^0 - \nabla f(x^0)\right\|^2\right] = \mathrm{E}\left[\left\|\frac{1}{n}\sum_{i=1}^n\frac{1}{B_\mathrm{init}}\sum_{k=1}^{B_\mathrm{init}}\nabla f_i(x^0;\xi_{ik}^0) - \nabla f(x^0)\right\|^2\right] \leq \frac{\sigma^2}{nB_\mathrm{init}}$$

and

$$\frac{1}{n^2}\sum_{i=1}^n\mathrm{E}\left[\left\|h_i^0 - \nabla f_i(x^0)\right\|^2\right] = \frac{1}{n^2}\sum_{i=1}^n\mathrm{E}\left[\left\|\frac{1}{B_\mathrm{init}}\sum_{k=1}^{B_\mathrm{init}}\nabla f_i(x^0;\xi_{ik}^0) - \nabla f_i(x^0)\right\|^2\right] \leq \frac{\sigma^2}{nB_\mathrm{init}},$$

we have

$$\mathrm{E}\left[\left\|\nabla f(\widehat{x}^T)\right\|^2\right] \leq \frac{1}{T}\left[2\Delta_0\left(L + \sqrt{\frac{8\left(2\omega+1\right)\omega}{np_\mathrm{a}^2}\left(\widehat{L}^2 + \frac{L_\sigma^2}{B}\right) + \frac{16}{np_\mathrm{mega}p_\mathrm{a}^2}\left(\left(1 - \frac{p_\mathrm{aa}}{p_\mathrm{a}}\right)\widehat{L}^2 + \frac{L_\sigma^2}{B}\right)}\right)\right.$$

$$\left. + \frac{8\sigma^2}{np_\mathrm{mega}p_\mathrm{a}B_\mathrm{init}}\right]$$

$$+ \frac{2\varepsilon}{3}.$$

Therefore, we can take the following $T$ to get $\varepsilon$–solution.

$$T = \mathcal{O}\left(\frac{1}{\varepsilon}\left[\Delta_0\left(L + \sqrt{\frac{\omega^2}{np_\mathrm{a}^2}\left(\widehat{L}^2 + \frac{L_\sigma^2}{B}\right) + \frac{1}{np_\mathrm{mega}p_\mathrm{a}^2}\left(\widehat{L}^2 + \frac{L_\sigma^2}{B}\right)}\right) + \frac{\sigma^2}{np_\mathrm{mega}p_\mathrm{a}B_\mathrm{init}}\right]\right)$$

Considering the choice of $p_\mathrm{mega}$ and $B_\mathrm{init}$, we obtain

$$T = \mathcal{O}\left(\frac{1}{\varepsilon}\left[\Delta_0\left(L + \left(\frac{\omega}{p_\mathrm{a}\sqrt{n}} + \sqrt{\frac{d}{p_\mathrm{a}^2\zeta_\mathcal{C}n}}\right)\left(\widehat{L} + \frac{L_\sigma}{\sqrt{B}}\right) + \frac{\sigma}{p_\mathrm{a}\sqrt{\varepsilon n}}\left(\frac{\widehat{L}}{\sqrt{B}} + \frac{L_\sigma}{B}\right)\right) + \frac{\sigma^2}{np_\mathrm{mega}p_\mathrm{a}B_\mathrm{init}}\right]\right)$$

$$= \mathcal{O}\left(\frac{\Delta_0}{\varepsilon}\left[L + \left(\frac{\omega}{p_\mathrm{a}\sqrt{n}} + \sqrt{\frac{d}{p_\mathrm{a}^2\zeta_\mathcal{C}n}}\right)\left(\widehat{L} + \frac{L_\sigma}{\sqrt{B}}\right) + \frac{\sigma}{p_\mathrm{a}\sqrt{\varepsilon n}}\left(\frac{\widehat{L}}{\sqrt{B}} + \frac{L_\sigma}{B}\right)\right] + \frac{\sigma^2}{\sqrt{p_\mathrm{a}}n\varepsilon B}\right).$$

The expected communication complexity equals $\mathcal{O}\left(d + p_\mathrm{mega}d + (1 - p_\mathrm{mega})\zeta_\mathcal{C}\right) = \mathcal{O}\left(d + \zeta_\mathcal{C}\right)$ and the expected number of stochastic gradient calculations per node equals $\mathcal{O}\left(B_\mathrm{init} + p_\mathrm{mega}B' + (1 - p_\mathrm{mega})B\right) = \mathcal{O}\left(B_\mathrm{init} + B\right).$ $\qquad\square$

**Theorem 13.** *Suppose that Assumptions 1, 2, 3, 5, 6, 7, 8 and 9 hold. Let us take* $a = \frac{p_{\mathrm{a}}}{2\omega+1}$, $b = \frac{p_{mega}p_{\mathrm{a}}}{2-p_{\mathrm{a}}}$, *probability* $p_{mega} \in (0,1]$, *batch size* $B' \geq B \geq 1$,

$$
\gamma \leq \min\left\{ \left( L + \sqrt{\frac{16\left(2\omega+1\right)\omega}{np_{\mathrm{a}}^2}\left(\frac{L_\sigma^2}{B} + \widehat{L}^2\right) + \left(\frac{48L_\sigma^2}{np_{mega}p_{\mathrm{a}}^2 B} + \frac{24\left(1 - \frac{p_{\mathrm{aa}}}{p_{\mathrm{a}}}\right)\widehat{L}^2}{np_{mega}p_{\mathrm{a}}^2}\right)} \right)^{-1}, \frac{a}{2\mu}, \frac{b}{2\mu} \right\},
$$

*and* $h_i^0 = g_i^0$ *for all* $i \in [n]$ *in Algorithm 8. Then*

$$
\mathrm{E}\left[f(x^T) - f^*\right]
$$

$$
\leq (1 - \gamma\mu)^T \left( \Delta_0 + \frac{2\gamma}{b}\left\| h^0 - \nabla f(x^0) \right\|^2 + \frac{8\gamma\left(p_{\mathrm{a}} - p_{\mathrm{aa}}\right)}{np_{\mathrm{a}}^2 p_{mega}} \frac{1}{n}\sum_{i=1}^n \left\| h_i^0 - \nabla f_i(x^0) \right\|^2 \right) + \frac{20\sigma^2}{\mu n B'}.
$$

*Proof.* Let us fix constants $\kappa, \eta, \nu, \rho \in [0, \infty)$ that we will define later. As in the proof of Theorem 11, we can get

$$
\mathrm{E}\left[f(x^{t+1})\right] + \kappa\mathrm{E}\left[\left\| g^{t+1} - h^{t+1} \right\|^2\right] + \eta\mathrm{E}\left[\frac{1}{n}\sum_{i=1}^n \left\| g_i^{t+1} - h_i^{t+1} \right\|^2\right]
$$

$$
+ \nu\mathrm{E}\left[\left\| h^{t+1} - \nabla f(x^{t+1}) \right\|^2\right] + \rho\mathrm{E}\left[\frac{1}{n}\sum_{i=1}^n \left\| h_i^{t+1} - \nabla f_i(x^{t+1}) \right\|^2\right]
$$

$$
\leq \mathrm{E}\left[f(x^t)\right] - \frac{\gamma}{2}\mathrm{E}\left[\left\| \nabla f(x^t) \right\|^2\right]
$$

$$
- \left( \frac{1}{2\gamma} - \frac{L}{2} - \frac{2\kappa\omega}{np_{\mathrm{a}}}\left(\frac{L_\sigma^2}{B} + \widehat{L}^2\right) - \frac{2\eta\omega}{p_{\mathrm{a}}}\left(\frac{L_\sigma^2}{B} + \widehat{L}^2\right) \right.
$$

$$
\left. - \nu\left(\frac{2L_\sigma^2}{np_{\mathrm{a}}B} + \frac{2\left(p_{\mathrm{a}} - p_{\mathrm{aa}}\right)\widehat{L}^2}{np_{\mathrm{a}}^2}\right) - \rho\left(\frac{2L_\sigma^2}{p_{\mathrm{a}}B} + \frac{2(1 - p_{\mathrm{a}})\widehat{L}^2}{p_{\mathrm{a}}}\right) \right)\mathrm{E}\left[\left\| x^{t+1} - x^t \right\|^2\right]
$$

$$
+ \left( \gamma + \kappa\left(1 - a\right)^2 \right)\mathrm{E}\left[\left\| g^t - h^t \right\|^2\right]
$$

$$
+ \left( \kappa\frac{\left(\left(2\omega+1\right)p_{\mathrm{a}} - p_{\mathrm{aa}}\right)a^2}{np_{\mathrm{a}}^2} + \eta\left(\frac{(2\omega+1 - p_{\mathrm{a}})a^2}{p_{\mathrm{a}}} + \left(1 - a\right)^2\right) \right)\mathrm{E}\left[\frac{1}{n}\sum_{i=1}^n \left\| g_i^t - h_i^t \right\|^2\right]
$$

$$
+ \left( \gamma + \nu\left(1 - b\right) \right)\mathrm{E}\left[\left\| h^t - \nabla f(x^t) \right\|^2\right]
$$

$$
+ \left( \nu\frac{2\left(p_{\mathrm{a}} - p_{\mathrm{aa}}\right)b^2}{np_{\mathrm{a}}^2 p_{mega}} + \rho(1 - b) \right)\mathrm{E}\left[\frac{1}{n}\sum_{i=1}^n \left\| h_i^t - \nabla f_i(x^t) \right\|^2\right]
$$

$$
+ \left( \frac{2\nu b^2}{np_{mega}p_{\mathrm{a}}} + \frac{2\rho b^2}{p_{\mathrm{a}}p_{mega}} \right)\frac{\sigma^2}{B'}.
$$

Let us take $\kappa = \frac{2\gamma}{a}$, thus $\gamma + \kappa\left(1 - a\right)^2 \leq \left(1 - \frac{a}{2}\right)\kappa$ and

$$
\mathrm{E}\left[f(x^{t+1})\right] + \frac{2\gamma}{a}\mathrm{E}\left[\left\| g^{t+1} - h^{t+1} \right\|^2\right] + \eta\mathrm{E}\left[\frac{1}{n}\sum_{i=1}^n \left\| g_i^{t+1} - h_i^{t+1} \right\|^2\right]
$$

$$
+ \nu\mathrm{E}\left[\left\| h^{t+1} - \nabla f(x^{t+1}) \right\|^2\right] + \rho\mathrm{E}\left[\frac{1}{n}\sum_{i=1}^n \left\| h_i^{t+1} - \nabla f_i(x^{t+1}) \right\|^2\right]
$$

$$
\leq \mathrm{E}\left[f(x^t)\right] - \frac{\gamma}{2}\mathrm{E}\left[\left\| \nabla f(x^t) \right\|^2\right]
$$

$$
- \left( \frac{1}{2\gamma} - \frac{L}{2} - \frac{4\gamma\omega}{anp_{\mathrm{a}}}\left(\frac{L_\sigma^2}{B} + \widehat{L}^2\right) - \frac{2\eta\omega}{p_{\mathrm{a}}}\left(\frac{L_\sigma^2}{B} + \widehat{L}^2\right) \right)
$$

$$-\nu\left(\frac{2L_\sigma^2}{np_aB}+\frac{2\left(p_a-p_{aa}\right)\widehat{L}^2}{np_a^2}\right)-\rho\left(\frac{2L_\sigma^2}{p_aB}+\frac{2(1-p_a)\widehat{L}^2}{p_a}\right)\right)\mathrm{E}\left[\left\|x^{t+1}-x^t\right\|^2\right]$$

$$+\left(1-\frac{a}{2}\right)\frac{2\gamma}{a}\mathrm{E}\left[\left\|g^t-h^t\right\|^2\right]$$

$$+\left(\frac{2\gamma\left(\left(2\omega+1\right)p_a-p_{aa}\right)a}{np_a^2}+\eta\left(\frac{(2\omega+1-p_a)a^2}{p_a}+(1-a)^2\right)\right)\mathrm{E}\left[\frac{1}{n}\sum_{i=1}^n\left\|g_i^t-h_i^t\right\|^2\right]$$

$$+\left(\gamma+\nu\left(1-b\right)\right)\mathrm{E}\left[\left\|h^t-\nabla f(x^t)\right\|^2\right]$$

$$+\left(\nu\frac{2\left(p_a-p_{aa}\right)b^2}{np_a^2p_{\mathrm{mega}}}+\rho(1-b)\right)\mathrm{E}\left[\frac{1}{n}\sum_{i=1}^n\left\|h_i^t-\nabla f_i(x^t)\right\|^2\right]$$

$$+\left(\frac{2\nu b^2}{np_{\mathrm{mega}}p_a}+\frac{2\rho b^2}{p_ap_{\mathrm{mega}}}\right)\frac{\sigma^2}{B'}.$$

Next, since $a=\frac{p_a}{2\omega+1}$, we have $\left(\frac{(2\omega+1-p_a)a^2}{p_a}+(1-a)^2\right)\leq1-a$. We the choice $\eta=\frac{2\gamma((2\omega+1)p_a-p_{aa})}{np_a^2}$, we guarantee $\frac{\gamma((2\omega+1)p_a-p_{aa})a}{np_a^2}+\eta\left(\frac{(2\omega+1-p_a)a^2}{p_a}+(1-a)^2\right)\leq\left(1-\frac{a}{2}\right)\eta$ and

$$\mathrm{E}\left[f(x^{t+1})\right]+\frac{2\gamma(2\omega+1)}{p_a}\mathrm{E}\left[\left\|g^{t+1}-h^{t+1}\right\|^2\right]+\frac{2\gamma\left(\left(2\omega+1\right)p_a-p_{aa}\right)}{np_a^2}\mathrm{E}\left[\frac{1}{n}\sum_{i=1}^n\left\|g_i^{t+1}-h_i^{t+1}\right\|^2\right]$$

$$+\nu\mathrm{E}\left[\left\|h^{t+1}-\nabla f(x^{t+1})\right\|^2\right]+\rho\mathrm{E}\left[\frac{1}{n}\sum_{i=1}^n\left\|h_i^{t+1}-\nabla f_i(x^{t+1})\right\|^2\right]$$

$$\leq\mathrm{E}\left[f(x^t)\right]-\frac{\gamma}{2}\mathrm{E}\left[\left\|\nabla f(x^t)\right\|^2\right]$$

$$-\left(\frac{1}{2\gamma}-\frac{L}{2}-\frac{8\gamma\left(2\omega+1\right)\omega}{np_a^2}\left(\frac{L_\sigma^2}{B}+\widehat{L}^2\right)\right.$$

$$-\nu\left(\frac{2L_\sigma^2}{np_aB}+\frac{2\left(p_a-p_{aa}\right)\widehat{L}^2}{np_a^2}\right)-\rho\left(\frac{2L_\sigma^2}{p_aB}+\frac{2(1-p_a)\widehat{L}^2}{p_a}\right)\right)\mathrm{E}\left[\left\|x^{t+1}-x^t\right\|^2\right]$$

$$+\left(1-\frac{a}{2}\right)\frac{2\gamma(2\omega+1)}{p_a}\mathrm{E}\left[\left\|g^t-h^t\right\|^2\right]$$

$$+\left(1-\frac{a}{2}\right)\frac{2\gamma\left(\left(2\omega+1\right)p_a-p_{aa}\right)}{np_a^2}\mathrm{E}\left[\frac{1}{n}\sum_{i=1}^n\left\|g_i^t-h_i^t\right\|^2\right]$$

$$+\left(\gamma+\nu\left(1-b\right)\right)\mathrm{E}\left[\left\|h^t-\nabla f(x^t)\right\|^2\right]$$

$$+\left(\nu\frac{2\left(p_a-p_{aa}\right)b^2}{np_a^2p_{\mathrm{mega}}}+\rho(1-b)\right)\mathrm{E}\left[\frac{1}{n}\sum_{i=1}^n\left\|h_i^t-\nabla f_i(x^t)\right\|^2\right]$$

$$+\left(\frac{2\nu b^2}{np_{\mathrm{mega}}p_a}+\frac{2\rho b^2}{p_ap_{\mathrm{mega}}}\right)\frac{\sigma^2}{B'},$$

where simplified the term using $p_{aa}\geq0$. Let us take $\nu=\frac{2\gamma}{b}$ to obtain

$$\mathrm{E}\left[f(x^{t+1})\right]+\frac{2\gamma(2\omega+1)}{p_a}\mathrm{E}\left[\left\|g^{t+1}-h^{t+1}\right\|^2\right]+\frac{2\gamma\left(\left(2\omega+1\right)p_a-p_{aa}\right)}{np_a^2}\mathrm{E}\left[\frac{1}{n}\sum_{i=1}^n\left\|g_i^{t+1}-h_i^{t+1}\right\|^2\right]$$

$$+\frac{2\gamma}{b}\mathrm{E}\left[\left\|h^{t+1}-\nabla f(x^{t+1})\right\|^2\right]+\rho\mathrm{E}\left[\frac{1}{n}\sum_{i=1}^n\left\|h_i^{t+1}-\nabla f_i(x^{t+1})\right\|^2\right]$$

$$\leq\mathrm{E}\left[f(x^t)\right]-\frac{\gamma}{2}\mathrm{E}\left[\left\|\nabla f(x^t)\right\|^2\right]$$

$$
-\left(\frac{1}{2\gamma} - \frac{L}{2} - \frac{8\gamma\,(2\omega+1)\,\omega}{np_{\mathrm a}^2}\left(\frac{L_\sigma^2}{B} + \widehat{L}^2\right)\right.
$$

$$
\left.- \left(\frac{4\gamma L_\sigma^2}{bnp_{\mathrm a}B} + \frac{4\gamma\,(p_{\mathrm a}-p_{\mathrm{aa}})\,\widehat{L}^2}{bnp_{\mathrm a}^2}\right) - \rho\left(\frac{2L_\sigma^2}{p_{\mathrm a}B} + \frac{2(1-p_{\mathrm a})\widehat{L}^2}{p_{\mathrm a}}\right)\right)\mathrm{E}\left[\left\|x^{t+1}-x^t\right\|^2\right]
$$

$$
+ \left(1-\frac{a}{2}\right)\frac{2\gamma(2\omega+1)}{p_{\mathrm a}}\mathrm{E}\left[\left\|g^t-h^t\right\|^2\right] + \left(1-\frac{a}{2}\right)\frac{2\gamma\,((2\omega+1)\,p_{\mathrm a}-p_{\mathrm{aa}})}{np_{\mathrm a}^2}\mathrm{E}\left[\frac{1}{n}\sum_{i=1}^{n}\left\|g_i^t-h_i^t\right\|^2\right]
$$

$$
+ \left(1-\frac{b}{2}\right)\frac{2\gamma}{b}\mathrm{E}\left[\left\|h^t-\nabla f(x^t)\right\|^2\right]
$$

$$
+ \left(\frac{4\gamma\,(p_{\mathrm a}-p_{\mathrm{aa}})\,b}{np_{\mathrm a}^2 p_{\mathrm{mega}}} + \rho(1-b)\right)\mathrm{E}\left[\frac{1}{n}\sum_{i=1}^{n}\left\|h_i^t-\nabla f_i(x^t)\right\|^2\right]
$$

$$
+ \left(\frac{4\gamma b}{np_{\mathrm{mega}}p_{\mathrm a}} + \frac{2\rho b^2}{p_{\mathrm a}p_{\mathrm{mega}}}\right)\frac{\sigma^2}{B'},
$$

Next, we take $\rho = \frac{8\gamma(p_{\mathrm a}-p_{\mathrm{aa}})}{np_{\mathrm a}^2 p_{\mathrm{mega}}}$, thus

$$
\mathrm{E}\left[f(x^{t+1})\right] + \frac{2\gamma(2\omega+1)}{p_{\mathrm a}}\mathrm{E}\left[\left\|g^{t+1}-h^{t+1}\right\|^2\right] + \frac{2\gamma\,((2\omega+1)\,p_{\mathrm a}-p_{\mathrm{aa}})}{np_{\mathrm a}^2}\mathrm{E}\left[\frac{1}{n}\sum_{i=1}^{n}\left\|g_i^{t+1}-h_i^{t+1}\right\|^2\right]
$$

$$
+ \frac{2\gamma}{b}\mathrm{E}\left[\left\|h^{t+1}-\nabla f(x^{t+1})\right\|^2\right] + \frac{8\gamma\,(p_{\mathrm a}-p_{\mathrm{aa}})}{np_{\mathrm a}^2 p_{\mathrm{mega}}}\mathrm{E}\left[\frac{1}{n}\sum_{i=1}^{n}\left\|h_i^{t+1}-\nabla f_i(x^{t+1})\right\|^2\right]
$$

$$
\le \mathrm{E}\left[f(x^t)\right] - \frac{\gamma}{2}\mathrm{E}\left[\left\|\nabla f(x^t)\right\|^2\right]
$$

$$
-\left(\frac{1}{2\gamma} - \frac{L}{2} - \frac{8\gamma\,(2\omega+1)\,\omega}{np_{\mathrm a}^2}\left(\frac{L_\sigma^2}{B} + \widehat{L}^2\right)\right.
$$

$$
\left.- \left(\frac{4\gamma L_\sigma^2}{bnp_{\mathrm a}B} + \frac{4\gamma\,(p_{\mathrm a}-p_{\mathrm{aa}})\,\widehat{L}^2}{bnp_{\mathrm a}^2}\right) - \left(\frac{8\gamma\,(p_{\mathrm a}-p_{\mathrm{aa}})}{np_{\mathrm a}^2 p_{\mathrm{mega}}}\right)\left(\frac{2L_\sigma^2}{p_{\mathrm a}B} + \frac{2(1-p_{\mathrm a})\widehat{L}^2}{p_{\mathrm a}}\right)\right)\mathrm{E}\left[\left\|x^{t+1}-x^t\right\|^2\right]
$$

$$
+ \left(1-\frac{a}{2}\right)\frac{2\gamma(2\omega+1)}{p_{\mathrm a}}\mathrm{E}\left[\left\|g^t-h^t\right\|^2\right] + \left(1-\frac{a}{2}\right)\frac{2\gamma\,((2\omega+1)\,p_{\mathrm a}-p_{\mathrm{aa}})}{np_{\mathrm a}^2}\mathrm{E}\left[\frac{1}{n}\sum_{i=1}^{n}\left\|g_i^t-h_i^t\right\|^2\right]
$$

$$
+ \left(1-\frac{b}{2}\right)\frac{2\gamma}{b}\mathrm{E}\left[\left\|h^t-\nabla f(x^t)\right\|^2\right] + \left(1-\frac{b}{2}\right)\frac{8\gamma\,(p_{\mathrm a}-p_{\mathrm{aa}})}{np_{\mathrm a}^2 p_{\mathrm{mega}}}\mathrm{E}\left[\frac{1}{n}\sum_{i=1}^{n}\left\|h_i^t-\nabla f_i(x^t)\right\|^2\right]
$$

$$
+ \left(\frac{4\gamma b}{np_{\mathrm{mega}}p_{\mathrm a}} + \frac{16\gamma\,(p_{\mathrm a}-p_{\mathrm{aa}})\,b^2}{np_{\mathrm a}^3 p_{\mathrm{mega}}^2}\right)\frac{\sigma^2}{B'},
$$

Since $\frac{p_{\mathrm{mega}}p_{\mathrm a}}{2} \le b \le p_{\mathrm{mega}}p_{\mathrm a}$ and $1-p_{\mathrm a} \le 1-\frac{p_{\mathrm{aa}}}{p_{\mathrm a}} \le 1$, we get

$$
\mathrm{E}\left[f(x^{t+1})\right] + \frac{2\gamma(2\omega+1)}{p_{\mathrm a}}\mathrm{E}\left[\left\|g^{t+1}-h^{t+1}\right\|^2\right] + \frac{2\gamma\,((2\omega+1)\,p_{\mathrm a}-p_{\mathrm{aa}})}{np_{\mathrm a}^2}\mathrm{E}\left[\frac{1}{n}\sum_{i=1}^{n}\left\|g_i^{t+1}-h_i^{t+1}\right\|^2\right]
$$

$$
+ \frac{2\gamma}{b}\mathrm{E}\left[\left\|h^{t+1}-\nabla f(x^{t+1})\right\|^2\right] + \frac{8\gamma\,(p_{\mathrm a}-p_{\mathrm{aa}})}{np_{\mathrm a}^2 p_{\mathrm{mega}}}\mathrm{E}\left[\frac{1}{n}\sum_{i=1}^{n}\left\|h_i^{t+1}-\nabla f_i(x^{t+1})\right\|^2\right]
$$

$$
\le \mathrm{E}\left[f(x^t)\right] - \frac{\gamma}{2}\mathrm{E}\left[\left\|\nabla f(x^t)\right\|^2\right]
$$

$$
-\left(\frac{1}{2\gamma} - \frac{L}{2} - \frac{8\gamma\,(2\omega+1)\,\omega}{np_{\mathrm a}^2}\left(\frac{L_\sigma^2}{B} + \widehat{L}^2\right)\right.
$$

$$
\left.- \left(\frac{8\gamma L_\sigma^2}{np_{\mathrm{mega}}p_{\mathrm a}^2 B} + \frac{8\gamma\,(p_{\mathrm a}-p_{\mathrm{aa}})\,\widehat{L}^2}{np_{\mathrm{mega}}p_{\mathrm a}^3}\right) - \left(\frac{16\gamma L_\sigma^2}{np_{\mathrm{mega}}p_{\mathrm a}^2 B} + \frac{16\gamma(1-p_{\mathrm a})\widehat{L}^2}{np_{\mathrm{mega}}p_{\mathrm a}^2}\right)\right)\mathrm{E}\left[\left\|x^{t+1}-x^t\right\|^2\right]
$$

$$
\begin{aligned}
&+ \left(1 - \frac{a}{2}\right) \frac{2\gamma(2\omega + 1)}{p_{\mathrm{a}}} \mathrm{E}\left[\left\|g^t - h^t\right\|^2\right] + \left(1 - \frac{a}{2}\right) \frac{2\gamma\left((2\omega + 1)p_{\mathrm{a}} - p_{\mathrm{aa}}\right)}{np_{\mathrm{a}}^2} \mathrm{E}\left[\frac{1}{n}\sum_{i=1}^n \left\|g_i^t - h_i^t\right\|^2\right] \\
&+ \left(1 - \frac{b}{2}\right) \frac{2\gamma}{b} \mathrm{E}\left[\left\|h^t - \nabla f(x^t)\right\|^2\right] + \left(1 - \frac{b}{2}\right) \frac{8\gamma\left(p_{\mathrm{a}} - p_{\mathrm{aa}}\right)}{np_{\mathrm{a}}^2 p_{\mathrm{mega}}} \mathrm{E}\left[\frac{1}{n}\sum_{i=1}^n \left\|h_i^t - \nabla f_i(x^t)\right\|^2\right] \\
&+ \frac{20\gamma\sigma^2}{nB'} \\
\leq{}& \mathrm{E}\left[f(x^t)\right] - \frac{\gamma}{2}\mathrm{E}\left[\left\|\nabla f(x^t)\right\|^2\right] \\
&- \left(\frac{1}{2\gamma} - \frac{L}{2} - \frac{8\gamma(2\omega + 1)\omega}{np_{\mathrm{a}}^2}\left(\frac{L_\sigma^2}{B} + \widehat{L}^2\right) - \left(\frac{24\gamma L_\sigma^2}{np_{\mathrm{mega}}p_{\mathrm{a}}^2 B} + \frac{24\gamma\left(1 - \frac{p_{\mathrm{aa}}}{p_{\mathrm{a}}}\right)\widehat{L}^2}{np_{\mathrm{mega}}p_{\mathrm{a}}^2}\right)\right)\mathrm{E}\left[\left\|x^{t+1} - x^t\right\|^2\right] \\
&+ \left(1 - \frac{a}{2}\right) \frac{2\gamma(2\omega + 1)}{p_{\mathrm{a}}} \mathrm{E}\left[\left\|g^t - h^t\right\|^2\right] + \left(1 - \frac{a}{2}\right) \frac{2\gamma\left((2\omega + 1)p_{\mathrm{a}} - p_{\mathrm{aa}}\right)}{np_{\mathrm{a}}^2} \mathrm{E}\left[\frac{1}{n}\sum_{i=1}^n \left\|g_i^t - h_i^t\right\|^2\right] \\
&+ \left(1 - \frac{b}{2}\right) \frac{2\gamma}{b} \mathrm{E}\left[\left\|h^t - \nabla f(x^t)\right\|^2\right] + \left(1 - \frac{b}{2}\right) \frac{8\gamma\left(p_{\mathrm{a}} - p_{\mathrm{aa}}\right)}{np_{\mathrm{a}}^2 p_{\mathrm{mega}}} \mathrm{E}\left[\frac{1}{n}\sum_{i=1}^n \left\|h_i^t - \nabla f_i(x^t)\right\|^2\right] \\
&+ \frac{20\gamma\sigma^2}{nB'}.
\end{aligned}
$$

Using Lemma 4 and the assumption about $\gamma$, we get

$$
\begin{aligned}
&\mathrm{E}\left[f(x^{t+1})\right] + \frac{2\gamma(2\omega + 1)}{p_{\mathrm{a}}} \mathrm{E}\left[\left\|g^{t+1} - h^{t+1}\right\|^2\right] + \frac{2\gamma\left((2\omega + 1)p_{\mathrm{a}} - p_{\mathrm{aa}}\right)}{np_{\mathrm{a}}^2} \mathrm{E}\left[\frac{1}{n}\sum_{i=1}^n \left\|g_i^{t+1} - h_i^{t+1}\right\|^2\right] \\
&+ \frac{2\gamma}{b} \mathrm{E}\left[\left\|h^{t+1} - \nabla f(x^{t+1})\right\|^2\right] + \frac{8\gamma\left(p_{\mathrm{a}} - p_{\mathrm{aa}}\right)}{np_{\mathrm{a}}^2 p_{\mathrm{mega}}} \mathrm{E}\left[\frac{1}{n}\sum_{i=1}^n \left\|h_i^{t+1} - \nabla f_i(x^{t+1})\right\|^2\right] \\
\leq{}& \mathrm{E}\left[f(x^t)\right] - \frac{\gamma}{2}\mathrm{E}\left[\left\|\nabla f(x^t)\right\|^2\right] \\
&+ \left(1 - \frac{a}{2}\right) \frac{2\gamma(2\omega + 1)}{p_{\mathrm{a}}} \mathrm{E}\left[\left\|g^t - h^t\right\|^2\right] + \left(1 - \frac{a}{2}\right) \frac{2\gamma\left((2\omega + 1)p_{\mathrm{a}} - p_{\mathrm{aa}}\right)}{np_{\mathrm{a}}^2} \mathrm{E}\left[\frac{1}{n}\sum_{i=1}^n \left\|g_i^t - h_i^t\right\|^2\right] \\
&+ \left(1 - \frac{b}{2}\right) \frac{2\gamma}{b} \mathrm{E}\left[\left\|h^t - \nabla f(x^t)\right\|^2\right] + \left(1 - \frac{b}{2}\right) \frac{8\gamma\left(p_{\mathrm{a}} - p_{\mathrm{aa}}\right)}{np_{\mathrm{a}}^2 p_{\mathrm{mega}}} \mathrm{E}\left[\frac{1}{n}\sum_{i=1}^n \left\|h_i^t - \nabla f_i(x^t)\right\|^2\right] \\
&+ \frac{20\gamma\sigma^2}{nB'}.
\end{aligned}
$$

Due to $\gamma \leq \frac{a}{2\mu}$ and $\gamma \leq \frac{b}{2\mu}$, we have

$$
\begin{aligned}
&\mathrm{E}\left[f(x^{t+1})\right] + \frac{2\gamma(2\omega + 1)}{p_{\mathrm{a}}} \mathrm{E}\left[\left\|g^{t+1} - h^{t+1}\right\|^2\right] + \frac{2\gamma\left((2\omega + 1)p_{\mathrm{a}} - p_{\mathrm{aa}}\right)}{np_{\mathrm{a}}^2} \mathrm{E}\left[\frac{1}{n}\sum_{i=1}^n \left\|g_i^{t+1} - h_i^{t+1}\right\|^2\right] \\
&+ \frac{2\gamma}{b} \mathrm{E}\left[\left\|h^{t+1} - \nabla f(x^{t+1})\right\|^2\right] + \frac{8\gamma\left(p_{\mathrm{a}} - p_{\mathrm{aa}}\right)}{np_{\mathrm{a}}^2 p_{\mathrm{mega}}} \mathrm{E}\left[\frac{1}{n}\sum_{i=1}^n \left\|h_i^{t+1} - \nabla f_i(x^{t+1})\right\|^2\right] \\
\leq{}& \mathrm{E}\left[f(x^t)\right] - \frac{\gamma}{2}\mathrm{E}\left[\left\|\nabla f(x^t)\right\|^2\right] \\
&+ \left(1 - \gamma\mu\right) \frac{2\gamma(2\omega + 1)}{p_{\mathrm{a}}} \mathrm{E}\left[\left\|g^t - h^t\right\|^2\right] + \left(1 - \gamma\mu\right) \frac{2\gamma\left((2\omega + 1)p_{\mathrm{a}} - p_{\mathrm{aa}}\right)}{np_{\mathrm{a}}^2} \mathrm{E}\left[\frac{1}{n}\sum_{i=1}^n \left\|g_i^t - h_i^t\right\|^2\right] \\
&+ \left(1 - \gamma\mu\right) \frac{2\gamma}{b} \mathrm{E}\left[\left\|h^t - \nabla f(x^t)\right\|^2\right] + \left(1 - \gamma\mu\right) \frac{8\gamma\left(p_{\mathrm{a}} - p_{\mathrm{aa}}\right)}{np_{\mathrm{a}}^2 p_{\mathrm{mega}}} \mathrm{E}\left[\frac{1}{n}\sum_{i=1}^n \left\|h_i^t - \nabla f_i(x^t)\right\|^2\right] \\
&+ \frac{20\gamma\sigma^2}{nB'}.
\end{aligned}
$$

It is left to apply Lemma 11 with

$$
\begin{aligned}
\Psi^t \;=\;\; & \frac{2(2\omega+1)}{p_{\mathrm{a}}}\mathrm{E}\left[\left\|g^t-h^t\right\|^2\right]+\frac{2\left((2\omega+1)\,p_{\mathrm{a}}-p_{\mathrm{aa}}\right)}{np_{\mathrm{a}}^2}\mathrm{E}\left[\frac{1}{n}\sum_{i=1}^{n}\left\|g_i^t-h_i^t\right\|^2\right] \\[2mm]
+\;\; & \frac{2}{b}\mathrm{E}\left[\left\|h^t-\nabla f(x^t)\right\|^2\right]+\frac{8\left(p_{\mathrm{a}}-p_{\mathrm{aa}}\right)}{np_{\mathrm{a}}^2 p_{\mathrm{mega}}}\mathrm{E}\left[\frac{1}{n}\sum_{i=1}^{n}\left\|h_i^t-\nabla f_i(x^t)\right\|^2\right]
\end{aligned}
$$

and $C=\frac{20\sigma^2}{nB'}$ to conclude the proof. $\qquad\square$

