*3.*
$$p_{\mathrm{aa}} \le p_{\mathrm{a}}^2,$$

*where $p_{\mathrm{a}} \in (0, 1]$ and $p_{\mathrm{aa}} \in [0, 1]$. Let us take random independent vectors $s_i \in \mathbb{R}^d$ for all $i \in [n]$, nonrandom vector $r_i \in \mathbb{R}^d$ for all $i \in [n]$, and random vectors*

$$v_i = \begin{cases} r_i + \frac{1}{p_{\mathrm{a}}} s_i, i \in S, \\ r_i, i \notin S, \end{cases}$$

*then*

$$\mathrm{E}\left[\left\|\frac{1}{n}\sum_{i=1}^n v_i - \mathrm{E}\left[\frac{1}{n}\sum_{i=1}^n v_i\right]\right\|^2\right]$$

$$= \frac{1}{n^2 p_{\mathrm{a}}} \sum_{i=1}^n \mathrm{E}\left[\|s_i - \mathrm{E}\left[s_i\right]\|^2\right] + \frac{p_{\mathrm{a}} - p_{\mathrm{aa}}}{n^2 p_{\mathrm{a}}^2} \sum_{i=1}^n \|\mathrm{E}\left[s_i\right]\|^2 + \frac{p_{\mathrm{aa}} - p_{\mathrm{a}}^2}{p_{\mathrm{a}}^2} \left\|\frac{1}{n}\sum_{i=1}^n \mathrm{E}\left[s_i\right]\right\|$$

$$\le \frac{1}{n^2 p_{\mathrm{a}}} \sum_{i=1}^n \mathrm{E}\left[\|s_i - \mathrm{E}\left[s_i\right]\|^2\right] + \frac{p_{\mathrm{a}} - p_{\mathrm{aa}}}{n^2 p_{\mathrm{a}}^2} \sum_{i=1}^n \|\mathrm{E}\left[s_i\right]\|^2.$$

*Proof.* Let us define additional constants $p_{\mathrm{an}}$ and $p_{\mathrm{nn}}$, such that

1.
$$\mathbf{Prob}\left(i \in S, j \notin S\right) = p_{\mathrm{an}}, \quad \forall i \ne j \in [n],$$

2.
$$\mathbf{Prob}\left(i \notin S, j \notin S\right) = p_{\mathrm{nn}}, \quad \forall i \ne j \in [n].$$

Note, that

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

$$

$$
+ \left( \frac{24 b \gamma \omega (2\omega + 1)}{n p_{\mathrm{a}}^2 B} + \frac{6\gamma}{n p_{\mathrm{a}} B} \right) \frac{2 m L_{\max}^2}{p_{\mathrm{a}} B}
$$

Considering that $b \leq \frac{p_{\mathrm{a}} B}{m}$ and $b \geq \frac{p_{\mathrm{a}} B}{2m}$, we obtain

$$
\frac{4\gamma \omega (2\omega + 1)}{n p_{\mathrm{a}}^2} \left( \frac{2 L_{\max}^2}{B} + 2 \widehat{L}^2 \right)
$$

$$
+ \left( \frac{2\gamma L_{\max}^2}{bn p_{\mathrm{a}} B} + \frac{2\gamma \left( p_{\mathrm{a}} - p_{\mathrm{aa}} \right) \widehat{L}^2}{bn p_{\mathrm{a}}^2} \right)
$$

$$
+ \left( \frac{8 b \gamma \omega (2\omega + 1)}{n p_{\mathrm{a}}^2} + \frac{2\gamma \left( p_{\mathrm{a}} - p_{\mathrm{aa}} \right)}{n p_{\mathrm{a}}^2} \right) \left( \frac{2 L_{\max}^2}{p_{\mathrm{a}} B} + \frac{2 (1 - p_{\mathrm{a}}) \widehat{L}^2}{p_{\mathrm{a}}} \right)
$$

$$
+ \left( \frac{24 b \gamma \omega (2\omega + 1)}{n p_{\mathrm{a}}^2 B} + \frac{6\gamma}{n p_{\mathrm{a}} B} \right) \frac{2 m L_{\max}^2}{p_{\mathrm{a}} B}
$$

$$
\leq \frac{36\gamma \omega (2\omega + 1)}{n p_{\mathrm{a}}^2} \left( \frac{2 L_{\max}^2}{B} + 2 \widehat{L}^2 \right) + \left( \frac{18\gamma L_{\max}^2}{bn p_{\mathrm{a}} B} + \frac{6\gamma \left( p_{\mathrm{a}} - p_{\mathrm{aa}} \right) \widehat{L}^2}{bn p_{\mathrm{a}}^2} \right)
$$

$$
\leq \frac{36\gamma \omega (2\omega + 1)}{n p_{\mathrm{a}}^2} \left( \frac{2 L_{\max}^2}{B} + 2 \widehat{L}^2 \right) + \left( \frac{36 m \gamma L_{\max}^2}{n p_{\mathrm{a}}^2 B^2} + \frac{12 m \gamma \left( p_{\mathrm{a}} - p_{\mathrm{aa}} \right) \widehat{L}^2}{B n p_{\mathrm{a}}^3} \right).
$$

All in all, we have

$$
\mathrm{E}\left[ f(x^{t+1}) \right] + \frac{\gamma (2\omega + 1)}{p_{\mathrm{a}}} \mathrm{E}\left[ \left\| g^{t+1} - h^{t+1} \right\|^2 \right] + \frac{\gamma ((2\omega + 1) p_{\mathrm{a}} - p_{\mathrm{aa}})}{n p_{\mathrm{a}}^2} \mathrm{E}\left[ \frac{1}{n} \sum_{i=1}^{n} \left\| g_i^{t+1} - h_i^{t+1} \right\|^2 \right]
$$

$$
+ \frac{\gamma}{b} \mathrm{E}\left[ \left\| h^{t+1} - \nabla f(x^{t+1}) \right\|^2 \right] + \left( \frac{8 b \gamma \omega (2\omega + 1)}{n p_{\mathrm{a}}^2} + \frac{2\gamma \left( p_{\mathrm{a}} - p_{\mathrm{aa}} \right)}{n p_{\mathrm{a}}^2} \right) \mathrm{E}\left[ \frac{1}{n} \sum_{i=1}^{n} \left\| h_i^{t+1} - \nabla f_i(x^{t+1}) \right\|^2 \right]
$$

$$
+ \left( \frac{24 b \gamma \omega (2\omega + 1)}{n p_{\mathrm{a}}^2 B} + \frac{6\gamma}{n p_{\mathrm{a}} B} \right) \mathrm{E}\left[ \frac{1}{nm} \sum_{i=1}^{n} \sum_{j=1}^{m} \left\| h_{ij}^{t+1} - \nabla f_{ij}(x^{t+1}) \right\|^2 \right]
$$

$$
\leq \mathrm{E}\left[ f(x^t) \right] - \frac{\gamma}{2} \mathrm{E}\left[ \left\| \nabla f(x^t) \right\|^2 \right]
$$

$$
+ \frac{\gamma (2\omega + 1)}{p_{\mathrm{a}}} \mathrm{E}\left[ \left\| g^t - h^t \right\|^2 \right] + \frac{\gamma ((2\omega + 1) p_{\mathrm{a}} - p_{\mathrm{aa}})}{n p_{\mathrm{a}}^2} \mathrm{E}\left[ \frac{1}{n} \sum_{i=1}^{n} \left\| g_i^t - h_i^t \right\|^2 \right]
$$

$$
- \left( \frac{1}{2\gamma} - \frac{L}{2} - \frac{36\gamma \omega (2\omega + 1)}{n p_{\mathrm{a}}^2} \left( \frac{2 L_{\max}^2}{B} + 2 \widehat{L}^2 \right) - \left( \frac{36 m \gamma L_{\max}^2}{n p_{\mathrm{a}}^2 B^2} + \frac{12 m \

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

_{\mathrm{init}}} + \frac{b\omega^2\sigma^2}{np_{\mathrm{a}}^2 B_{\mathrm{init}}} + \frac{\sigma^2}{np_{\mathrm{a}}B_{\mathrm{init}}}\right]\right).$$

Using the choice of $B_{\mathrm{init}}$ and $b$, we obtain

$$T = \mathcal{O}\left(\frac{1}{\varepsilon}\left[\Delta_0\left(L + \frac{\omega}{p_{\mathrm{a}}\sqrt{n}}\left(\widehat{L} + \frac{L_\sigma}{\sqrt{B}}\right) + \sqrt{\frac{\sigma^2}{p_{\mathrm{a}}^2\varepsilon n^2 B}}\left(\mathbb{1}_{p_{\mathrm{a}}}\widehat{L} + \frac{L_\sigma}{\sqrt{B}}\right)\right)\right.\right.$$
$$\left.\left.+ \frac{\sigma^2}{\sqrt{p_{\mathrm{a}}}nB} + \frac{b^2\omega^2\sigma^2}{np_{\mathrm{a}}^{5/2}B} + \frac{b\sigma^2}{p_{\mathrm{

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

^2} + \frac{(1-p_{\text{page}})L_{\max}^2}{np_aB}\right) - \rho\left(\frac{2(1-p_a)\widehat{L}^2}{p_a} + \frac{(1-p_{\text{page}})L_{\max}^2}{p_aB}\right)\right)\mathrm{E}\left[\left\|x^{t+1} - x^t\right\|^2\right]$$

$$+\left(\gamma + \nu\left(p_{\text{page}}\left(1 - \frac{b}{p_{\text{page}}}\right)^2 + (1-p_{\text{page}})\right)\right)\mathrm{E}\left[\left\|h^t - \nabla f(x^t)\right\|^2\right]$$

$$+\left(\frac{20b^2\gamma\omega(2\omega+1)}{np_a^2 p_{\text{page}}} + \frac{2\nu(p_a - p_{aa})b^2}{np_a^2 p_{\text{page}}}\right.$$

$$\left.+\rho\left(\frac{2(1-p_a)b^2}{p_a p_{\text{page}}} + p_{\text{page}}\left(1 - \frac{b}{p_{\text{page}}}\right)^2 + (1-p_{\text{page}})\right)\right)\mathrm{E}\left[\frac{1}{n}\sum_{i=1}^n\left\|h_i^t - \nabla f_i(x^t)\right\|^2\right].$$

Due to $b = \frac{p_{\text{page}}p_a}{2 - p_a} \le p_{\text{page}}$, one can show that $\left(p_{\text{page}}\left(1 - \frac{b}{p_{\text{page}}}\right)^2 + (1-p_{\text{page}})\right) \le 1 - b$. Thus, if we take $\nu = \frac{2\gamma}{b}$, then

$$\left(\gamma + \nu\left(p_{\text{page}}\left(1 - \frac{b}{p_{\text{page}}}\right)^2 + (1-p_{\text{page}})\right)\right) \le \gamma + \nu(1-b) = \left(1 - \frac{b}{2}\right)\nu,$$

therefore

$$\mathrm{E}\left[f(x^{t+1})\right] + \frac{2\gamma(2\omega+1)}{p_a}\mathrm{E}\left[\left\|g^{t+1} - h^{t+1}\right\|^2\right] + \frac{4\gamma((2\omega+1)p_a - p_{aa})}{np_a^2}\mathrm{E}\left[\frac{1}{n}\sum_{i=1}^n\left\|g_i^{t+1} - h_i^{t+1}\right\|^2\right]$$

$$+\frac{2\gamma}{b}\mathrm{E}\left[\left\|h^{t+1} - \nabla f(x^{t+1})\right\|^2\right] + \rho\mathrm{E}\left[\frac{1}{n}\sum_{i=1}^n\left\|h_i^{t+1} - \nabla f_i(x^{t+1})\right\|^2\right]$$

$$\le \mathrm{E}\left[f(x^t)\right] - \frac{\gamma}{2}\mathrm{E}\left[\left\|\nabla f(x^t)\right\|^2\right]$$

$$+(1-\gamma\mu)\frac{2\gamma(2\omega+1)}{p_a}\mathrm{E}\left[\left\|g^t - h^t\right\|^2\right] + (1-\gamma\mu)\frac{4\gamma((2\omega+1)p_a - p_{aa})}{np_a^2}\mathrm{E}\left[\frac{1}{n}\sum_{i=1}^n\left\|g_i^t - h_i^t\right\|^2\right]$$

$$-\left(\frac{1}{2\gamma} - \frac{L}{2} - \frac{10\gamma\omega(2\omega+1)}{np_a^2}\left(2\widehat{L}^2 + \frac{(1-p_{\text{page}})L_{\max}^2}{B}\right)\right.$$

$$\left.-\frac{2\gamma}{bnp_a}\left(2\left(1 - \frac{p_{aa}}{p_a}\right)\widehat{L}^2 + \frac{(1-p_{\text{page}})L_{\max}^2}{B}\right) - \rho\left(\frac{2(1-p_a)\widehat{L}^2}{p_a} + \frac{(1-p_{\text{page}})L_{\max}^2}{p_aB}\right)\right)\mathrm{E}\left[\left\|x^{t+1} - x^t\right\|^2\right]$$

$$+\left(1 - \frac{b}{2}\right)\frac{2\gamma}{b}\mathrm{E}\left[\left\|h^t - \nabla f(x^t)\right\|^2\right]$$

$$+\left(\frac{20b^2\gamma\omega(2\omega+1)}{np_a^2 p_{\text{page}}} + \frac{4\gamma(p_a - p_{aa})b}{np_a^2 p_{\text{page}}}\right.$$

$$\left.+\rho\left(\frac{2(1-p_a)b^2}{p_a p_{\text{page}}} + p_{\text{page}}\left(1 - \frac{b}{p_{\text{page}}}\right)^2 + (1-p_{\text{page}})\right)\right)\mathrm{E}\left[\frac{1}{n}\sum_{i=1}^n\left\|h_i^t - \nabla f_i(x^t)\right\|^2\right].$$

Next, with the choice of $b = \frac{p_{\text{page}}p_a}{2 - p_a}$, we ensure that

$$\left(\frac{2(1-p_a)b^2}{p_a p_{\text{page}}} + p_{\text{page}}\left(1 - \frac{b}{p_{\text{page}}}\right)^2 + (1-p_{\text{page}})\right) \le 1 - b.$$

If we take $\rho = \frac{40b\gamma\omega(2\omega+1)}{np_a^2 p_{\text{page}}} + \frac{8\gamma(p_a - p_{aa})}{np_a^2 p_{\text{page}}}$, then

$$\left(\frac{20b^2\gamma\omega(2\omega+1)}{np_a^2 p_{\text{page}}} + \frac{4\gamma(p_a - p_{aa})b}{np_a^2 p_{\text{page}}} + \rho\left(\frac{2(1-p_a)b^2}{p_

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

_{\rm a}B}+\frac{2\left(p_{\rm a}-p_{\rm aa}\right)\widehat{L}^2}{np_{\rm a}^2}\right)-\rho\left(\frac{2L_\sigma^2}{p_{\rm a}B}+\frac{2(1-p_{\rm a})\widehat{L}^2}{p_{\rm a}}\right)\right)\mathrm{E}\left[\left\|x^{t+1}-x^t\right\|^2\right]$$

$$+\left(1-\frac{a}{2}\right)\frac{2\gamma}{a}\mathrm{E}\left[\left\|g^t-h^t\right\|^2\right]$$

$$+\left(\frac{2\gamma\left(\left(2\omega+1\right)p_{\rm a}-p_{\rm aa}\right)a}{np_{\rm a}^2}+\eta\left(\frac{(2\omega+1-p_{\rm a})a^2}{p_{\rm a}}+(1-a)^2\right)\right)\mathrm{E}\left[\frac{1}{n}\sum_{i=1}^n\left\|g_i^t-h_i^t\right\|^2\right]$$

$$+\left(\gamma+\nu\left(1-b\right)\right)\mathrm{E}\left[\left\|h^t-\nabla f(x^t)\right\|^2\right]$$

$$+\left(\nu\frac{2\left(p_{\rm a}-p_{\rm aa}\right)b^2}{np_{\rm a}^2p_{\rm mega}}+\rho(1-b)\right)\mathrm{E}\left[\frac{1}{n}\sum_{i=1}^n\left\|h_i^t-\nabla f_i(x^t)\right\|^2\right]$$

$$+\left(\frac{2\nu b^2}{np_{\rm mega}p_{\rm a}}+\frac{2\rho b^2}{p_{\rm a}p_{\rm mega}}\right)\frac{\sigma^2}{B'}.$$

Next, since $a=\frac{p_{\rm a}}{2\omega+1}$, we have $\left(\frac{(2\omega+1-p_{\rm a})a^2}{p_{\rm a}}+(1-a)^2\right)\leq 1-a$. We the choice $\eta=\frac{2\gamma((2\omega+1)p_{\rm a}-p_{\rm aa})}{np_{\rm a}^2}$, we guarantee $\frac{\gamma((2\omega+1)p_{\rm a}-p_{\rm aa})a}{np_{\rm a}^2}+\eta\left(\frac{(2\omega+1-p_{\rm a})a^2}{p_{\rm a}}+(1-a)^2\right)\leq\left(1-\frac{a}{2}\right)\eta$ and

$$\mathrm{E}\left[f(x^{t+1})\right]+\frac{2\gamma(2\omega+1)}{p_{\rm a}}\mathrm{E}\left[\left\|g^{t+1}-h^{t+1}\right\|^2\right]+\frac{2\gamma\left(\left(2\omega+1\right)p_{\rm a}-p_{\rm aa}\right)}{np_{\rm a}^2}\mathrm{E}\left[\frac{1}{n}\sum_{i=1}^n\left\|g_i^{t+1}-h_i^{t+1}\right\|^2\right]$$

$$+\nu\mathrm{E}\left[\left\|h^{t+1}-\nabla f(x^{t+1})\right\|^2\right]+\rho\mathrm{E}\left[\frac{1}{n}\sum_{i=1}^n\left\|h_i^{t+1}-\nabla f_i(x^{t+1})\right\|^2\right]$$

$$\leq\mathrm{E}\left[f(x^t)\right]-\frac{\gamma}{2}\mathrm{E}\left[\left\|\nabla f(x^t)\right\|^2\right]$$

$$-\left(\frac{1}{2\gamma}-\frac{L}{2}-\frac{8\gamma\left(2\omega+1\right)\omega}{np_{\rm a}^2}\left(\frac{L_\sigma^2}{B}+\widehat{L}^2\right)\right.$$

$$-\nu\left(\frac{2L_\sigma^2}{np_{\rm a}B}+\frac{2\left(p_{\rm a}-p_{\rm aa}\right)\widehat{L}^2}{np_{\rm