# OpenReview forum: "A Computation and Communication Efficient Method for Distributed Nonconvex Problems in the Partial Participation Setting"
_NeurIPS.cc/2023/Conference — NeurIPS 2023 poster_

### Official Review · Reviewer_MQ7w · 2023-06-26

**Soundness:** 3 good
**Presentation:** 2 fair
**Contribution:** 3 good
**Rating:** 6
**Confidence:** 4

**Summary:**

This research focuses on distributed learning problems and introduces a novel approach that incorporates three key elements: variance reduction, compression, and partial participation. It extends the DASHA method to address the partial participation scenario in both finite-sum and stochastic scenarios. In the context of partial participation, the proposed method achieves state-of-the-art communication complexity. Furthermore, in the scenario involving variance reduction and partial participation (without compression), it achieves optimal oracle complexity without relying on strict bounded gradient assumptions.

**Strengths:**

The extension of DASHA to accommodate the partial participation setting is nontrivial addition. The analysis involving partial participation introduces additional complexity, but this work successfully establishes state-of-the-art bounds (although not that surprising).

**Weaknesses:**

The presentation can be further improved in several ways to enhance clarity and readability. Firstly, instead of listing the assumptions and results for all three settings (full gradient, finite sum, stochastic) consecutively, it would be beneficial to present each setting individually with a particular focus on the stochastic setting, which is the main focus of this work. This approach would make it easier for the reader to understand and follow the discussion. Additionally, by adopting this approach, it would be possible to cover the results for both the nonconvex and PL condition cases in the main text.

Furthermore, there are a few minor typos that need to be addressed. The set $\mathbb{R}^d$ is written as $\mathbb{R}$ in various places, specifically lines 35, 44, 51, and 59. Additionally, in line 164, "what" should be replaced with "which." Finally, in line 405, "dependeces" should be corrected to "dependencies."

Another aspect that requires attention is the presentation of the algorithm. Currently, it lacks motivation and clarity. For instance, it is unclear how step (9) is derived to address the issue in steps (7) to (8), where it is mentioned that these updates are adapted to ensure the validity of the proof. Providing a clearer explanation or justification for step (9) would greatly improve the understanding of the algorithm and its purpose.


Although the specific setup presented in this study has not been previously explored, and its analysis poses challenges, it can be viewed as a minor blend of previous setups, and the results can be considered as a slight extension of the work done in DASHA.

Finally, the scope of the experiments conducted in this research is somewhat limited, as they do not encompass the more intriguing nonconvex neural network scenarios. Furthermore, there is a notable absence of numerical comparisons with other methods, such as FRECON or any other method within the full gradient setup. While I understand that space constraints may be a factor, with careful organization, it may be possible to allocate some room for such comparisons. One possible approach could involve dedicating the main body solely to the stochastic setting, while relegating the finite-sum case to the appendix. This would create additional space for the inclusion of numerical comparisons and provide a more comprehensive evaluation of the proposed approach.


**Questions:**

After examining equations (7)-(8), it is mentioned that step (7) requires further updating if the same idea as in the full gradient step is utilized. The question arises as to why this is necessary. Would the method still function if only the participating nodes updated $h$ while the others retained the previous values? Essentially, is this a limitation of the analysis or would the method diverge under such circumstances?

Another query pertains to practical applications that satisfy Assumption 6. Is there a specific real-world scenario that meets this criterion? Additionally, it is worth exploring whether achieving state-of-the-art bounds in the stochastic setting is possible without relying on Assumption 6.



**Limitations:**

Yes.

---

> ### Author Rebuttal · Authors · 2023-08-06
>
> Thank you for the review!
>
> > The presentation can be further improved in several ways to enhance clarity and readability. ...
>
> We agree. One can see that we separate assumptions in the parts "1. Gradient Setting.," "2. Finite-Sum Setting," and "3. Stochastic Setting," which begin at Lines 33, 38, and 43, accordingly. We tried to separate assumptions for each particular setting.
>
> > Although the specific setup presented in this study has not been previously explored, and its analysis poses challenges, it can be viewed as a minor blend of previous setups, and the results can be considered as a slight extension of the work done in DASHA.
>
> 1. The **partial participation setting** with **stochastic gradients** was explored in many previous papers, including FedAvg, FedPAGE, MIME, and CE-LSGD. However, these methods have weaknesses (see Tables 1 and 2).
> 2. The **partial participation setting** with **stochastic gradients** and **compression** was explored in FRECON, MARINA and DASHA (gradient setting only).
>
> Our paper also considers **partial participation setting** with **stochastic gradients** and **compression,** so we consider the setup that was explored in the papers from 2., and partially explored the papers from 1.
>
> > Finally, the scope of the experiments conducted in this research is somewhat limited ... Furthermore, there is a notable absence of numerical comparisons with other methods, such as FRECON or any other method within the full gradient setup. While I understand that space constraints may be a factor, with careful organization, it may be possible to allocate some room for such comparisons. ...
>
> **Note that we added new experiments that are presented in a separate file in the "global" comment to all reviewers called "Author Rebuttal by Authors".** Click "Revisions" if you can not find the pdf.
>
> There we compare our new method with other baselines, MARINA and FRECON. Also, in Section A, we compare with DASHA. If our paper gets accepted, we will be allowed an additional content page, where we will be happy to add our numerical experiments.
>
> > Another aspect that requires attention is the presentation of the algorithm. Currently, it lacks motivation and clarity. For instance, it is unclear how step (9) is derived to address the issue in steps (7) to (8), where it is mentioned that these updates are adapted to ensure the validity of the proof. ...
>
> > After examining equations (7)-(8), it is mentioned that step (7) requires further updating if the same idea as in the full gradient step is utilized. The question arises as to why this is necessary. ...
>
> In Lines 162-167, we discuss that DASHA-MVR *can not* work in the partial participation scenario because $h_{i}^{t+1}$ requires the points $x^{t+1}$ and $x^{t}$ to make the update (7). We can not evaluate (7) if the $i^{th}$ does not participate. The step (9) resolves this problem because $h_{i}^{t+1} = h_{i}^{t}$  when the $i^{th}$ does not participate. However, we can not use (7) anymore if the $i^{th}$ does not participate because **the proofs do not work** in such a case. So we redesigned (7) and obtained (9) to make new proofs work.
>
> Let us give an intuition of how we come up with (9). Imagine that we do not have partial participation. Then, using the results from DASHA, we know that (7) works:
>
> $h_i^{t+1} = \nabla f_i(x^{t+1};\xi^{t+1}_{i}) + (1 - b) (h_i^t - \nabla f_i(x^{t};\xi_i^{t+1})).$ ( * )
>
> Consider that we the partial participation. If we understood the question correctly, the reviewer asks if the following strategy will work:
>
> $h_i^{t+1} = \begin{cases}\nabla f_i(x^{t+1};\xi^{t+1}_{i}) + (1 - b) (h_i^t - \nabla f_i(x^{t};\xi_i^{t+1})),& p \\\\ h_i^{t}, & 1 - p\end{cases}$ ( ** )
>
> Let us take the expectation of $h_i^{t+1}$ w.r.t. the partial participation randomness. We get
>
> $$E_p [h_i^{t+1}] = p (\nabla f_i(x^{t+1};\xi^{t+1}_{i}) + (1 - b) (h_i^t - \nabla f_i(x^{t};\xi_i^{t+1}))) + (1 - p) h_i^{t}$$
> $$=p \nabla f_i(x^{t+1};\xi^{t+1}_i) +p (1 - b) (h_i^t - \nabla f_i(x^{t};\xi_i^{t+1}) + (1 - p) h_i^{t}$$
>
> Note that this $E_p [h_i^{t+1}]$ is not equal to r.h.s of ( * ). At the same time, if we take $h_i^{t+1}$ from (9), one can show that $E_p [h_i^{t+1}]$ equals to ( * )!  This indicates that we can not use ( ** ) in the partial participation setting, and the right choice is (9).
>
> > Another query pertains to practical applications that satisfy Assumption 6. Is there a specific real-world scenario that meets this criterion? Additionally, it is worth exploring whether achieving state-of-the-art bounds in the stoch. setting is possible without relying on Assumption 6.
>
> Assumption 6 is crucial to use the variance reduction technique and obtain the $1 / \varepsilon^{3/2}$ rate. This assumption is the mean-squared smoothness property. It was shown \[1\] that this is crucial to get the $1 / \varepsilon^{3/2}$ rate. Without this assumption, one can only get the $1 / \varepsilon^{2}$ rate (the rate of the vanilla SGD method). In the context of the ML and FL problems, this assumption is not strong. For instance, for the logistic regression problems, this assumption holds. In other words, this assumption requires the smoothness of $f_i(x, \xi)$ w.r.t $x,$ which holds for most ML problems.
>
> > Regarding the Main theorems, there is uncertainty surrounding the meaning of $\omega$. Does it represent any positive scalar, or is there a specific definition or interpretation attributed to it?
>
> $\omega \geq 0$ is the variance of the compressor from Definition 1. For instance, consider the Rand$K$ compressor that takes $K$ random values of a vector scaled by $d / K.$ Using the definition, one can show that this compressor is unbiased and $\omega = \frac{d}{K} - 1.$ So the more we compress ($K$ smaller), the larger is $
> \omega.$ If let say $K = d$ (we don't compress), then $\omega = 0.$ Since we use compressors in our method, $\omega$ affects our convergence rates.
>
> \[1\]: Arjevani et. al Lower Bounds for Non-Convex Stochastic Optimization

---

> > ### Comment · Reviewer_MQ7w · 2023-08-11
> >
> > I'd like to confirm that I've reviewed the rebuttal. The authors have appropriately taken into account my comments. I suggest including the reasoning behind the derivation of equation (9) for better understanding. Lastly, I maintain the same evaluation because, in my view, while the contributions are valuable, the paper holds a moderately significant impact.

---

> > > ### Author Response · Authors · 2023-08-11
> > > **Respond**
> > >
> > > Thank you for your comments and time!

---

### Official Review · Reviewer_wGcA · 2023-07-07

**Soundness:** 3 good
**Presentation:** 3 good
**Contribution:** 3 good
**Rating:** 6
**Confidence:** 3

**Summary:**

This paper introduces a novel method for distributed optimization and federated learning that integrates three critical components: variance reduction of stochastic gradients, partial participation, and compressed communication. The proposed method is shown to have optimal oracle complexity and leading-edge communication complexity in a setting of partial participation. A significant advantage of this method is that it effectively blends variance reduction and partial participation, thus providing optimal oracle complexity without the necessity of all nodes' participation or the bounded gradients assumption.

**Strengths:**

One of the primary strengths of this paper is the integration of three crucial elements in distributed optimization and federated learning. By encompassing variance reduction of stochastic gradients, partial participation, and compressed communication into a single method, the authors have created a holistic approach that takes into account the multifaceted nature of distributed optimization.

The authors demonstrate that their method achieves optimal oracle complexity, which is an important measure of the efficiency of an algorithm. This is a significant contribution, suggesting that the method has a high degree of efficiency in processing the data.

**Weaknesses:**

Despite the significant theoretical contributions, this paper falls short in one crucial aspect, which restricts its overall impact and practical applicability.

Lack of Empirical Validation: The major limitation of this paper is the absence of empirical experiments to substantiate the theoretical findings. While the theoretical results are compelling, they need to be complemented with practical validation to establish the practicality and effectiveness of the proposed method. The lack of experimental results makes it difficult to ascertain the method's performance in real-world scenarios, to gauge its scalability with varying data sizes and distributions, and to compare it with existing methods.

**Questions:**

See weakness

---

> ### Author Rebuttal · Authors · 2023-08-06
>
> > Despite the significant theoretical contributions, this paper falls short in one crucial aspect, which restricts its overall impact and practical applicability.
>
> > Lack of Empirical Validation: The major limitation of this paper is the absence of empirical experiments to substantiate the theoretical findings. While the theoretical results are compelling, they need to be complemented with practical validation to establish the practicality and effectiveness of the proposed method. The lack of experimental results makes it difficult to ascertain the method's performance in real-world scenarios, to gauge its scalability with varying data sizes and distributions, and to compare it with existing methods.
>
> We agree experiments are important, and that is why we have experiments in Section A. **Note that we added new experiments that are presented in a separate file in the "global" comment to all reviewers called "Author Rebuttal by Authors".** Click "Revisions" if you can not find the pdf.  We compare our new method with other baselines, MARINA and FRECON, that support compressed communication in the partial participation setting.
>
> We show that our new method requires significantly fewer communication rounds to find an $\varepsilon$-stationary point. We believe that these experiments, together with the experiments from Section A in the main paper, give us strong evidence that our method has an excellent practical performance.
>
> Moreover, please see Section A. The DASHA \[1\] method is considered to be the current SOTA method in *the full participation setting.* So, it is reasonable to compare our new method with DASHA. However, our method supports partial participation, and from the discussions in Section 6, we know that partial participation should degenerate the convergence rate up to $1 / p.$ When $p = 1$ ($s = 100$, full participation regime) we observe that our method and DASHA have almost the same convergence rates. Then we take $p < 1$ by taking the number of clients $s$ smaller and observe that the degeneration of the convergence rates is up to $1/p$ factor. This is the expected dependence because some clients do not participate.
>
> In total, we compare our new method with three previous baselines that support compressed communication with different datasets and settings. If our paper gets accepted, we will be allowed an additional content page, where we will be happy to add our numerical experiments.
>
> *In view of the extra experiments, we kindly ask the reviewer to reconsider the score. If the reviewer thinks that there should be some other experiments that would enhance the paper, then let us know.*
>
> \[1\]: Tyurin A, et al. DASHA: Distributed nonconvex optimization with communication compression and optimal oracle complexity (ICLR 2023)

---

> > ### Comment · Reviewer_wGcA · 2023-08-10
> >
> > Thanks for the extra experiments. I have updated rating accordingly.

---

> > > ### Author Response · Authors · 2023-08-10
> > > **thanks!**
> > >
> > > Thanks, much appreciated!

---

### Official Review · Reviewer_ZASs · 2023-07-19

**Soundness:** 4 excellent
**Presentation:** 4 excellent
**Contribution:** 2 fair
**Rating:** 5
**Confidence:** 4

**Summary:**

This manuscript considers distributed non-convex optimization in the federated learning setting. It proposes an algorithm DASHA-PP that brings three important features of federated learning together: i) variance reduction, ii) compressed communication, and iii) partial participation. The authors derive the oracle and communication complexities of finding an $\epsilon$-stationary point of the smooth non-convex objective. It is the first method that includes those three important features.

The algorithm presented in the paper is built upon DASHA, presented in [1]. In particular, in its form in [1], DASHA could work in the partial participation case only when the exact gradient oracle was available. In this manuscript, the authors extended the partial participation property of DASHA to the stochastic and finite-sum settings.

[1] Tyurin, Alexander and Peter Richt'arik. “DASHA: Distributed Nonconvex Optimization with Communication Compression, Optimal Oracle Complexity, and No Client Synchronization.” ArXiv abs/2202.01268 (2022): n. pag.

**Strengths:**

* I think the paper is very well-written.

* The problem the manuscript considers is an important problem for the large-scale federated learning setting, where including all three features, i.e., partial participation/compressed communication/stochastic gradient, is inevitable.

**Weaknesses:**

* The main technical contribution of the manuscript over [1] is the update rule given in Eq. (9). Similar update rule adjustments are frequently used in bandit optimization literature, cf [Section 2, 2] and [Section 6, 2].

* Besides the new update rule, it seems to me that the mathematical derivations in the manuscript are very similar to those in [1]. In that sense, I find the contribution of the manuscript is rather incremental.

[1] Tyurin, Alexander and Peter Richt'arik. “DASHA: Distributed Nonconvex Optimization with Communication Compression, Optimal Oracle Complexity, and No Client Synchronization.” ArXiv abs/2202.01268 (2022): n. pag.

[2] Cesa-Bianchi, Nicolò and Gábor Lugosi. “Prediction, learning, and games.” (2006).

**Questions:**

NA

---

> ### Author Rebuttal · Authors · 2023-08-06
>
> Thank you for the comments!
>
> > Besides the new update rule, it seems to me that the mathematical derivations in the manuscript are very similar to those in [1]. In that sense, I find the contribution of the manuscript is rather incremental.
>
> In Lines 168-182, we describe the difference between the proof techniques. The partial participation setting required us to rethink the proofs from the previous methods. Note that we had two challenges: *develop method* and *prove convergence.* When you have a developed method and you know that it is a correct method, it is much easier to prove the convergence rate (still may be very challenging though). However, at the beginning of our research journey, we didn't have either the method, nor the proof technique.
>
>
> > The main technical contribution of the manuscript over [1] is the update rule given in Eq. (9). Similar update rule adjustments are frequently used in bandit optimization literature, cf [Section 2, 2] and [Section 6, 2].
>
> * We agree that there are many connections in mathematics. One can also argue that there is a connection to the Markov chain with infinitely many states, where we move to the next state with probability $p,$ or stay at the same state with probability $1 - p.$
> * But we respectfully disagree that this is a weakness of our paper because our paper and \[2\] solve completely different problems. Even if we knew about \[2\], how could it help us to design the first step (the update rules of $h^{t+1}_i$ and $g^{t+1}_i$) of (9) from our paper? The connections between (9) and LABEL EFFICIENT FORECASTER from [2, p. 130] are not straightforward except for the fact that they both choose a step based on a random variable.
>
> Please let us know if you have any questions.
>
> \[1\] Tyurin, Alexander and Peter Richt'arik. “DASHA: Distributed Nonconvex Optimization with Communication Compression, Optimal Oracle Complexity, and No Client Synchronization.” ArXiv abs/2202.01268 (2022): n. pag.
>
> \[2\] Cesa-Bianchi, Nicolò and Gábor Lugosi. “Prediction, learning, and games.” (2006)

---

> > ### Comment · Reviewer_ZASs · 2023-08-12
> >
> > I thank the authors for their response. I keep my overall acceptance score.

---

### Official Review · Reviewer_QePY · 2023-07-23

**Soundness:** 4 excellent
**Presentation:** 3 good
**Contribution:** 2 fair
**Rating:** 7
**Confidence:** 2

**Summary:**

The paper injects existing algorithm for distributed optimisation in the compressed and variance reduced setting with a mechanism for partial participation of nodes. Theoretical guarantees are given in the general non-convex setting, but also under PL-condition. The theoretical results are verified empirically in a classification problem.

**Strengths:**

The paper considers the setting of partial participation, which is of importance in distributed optimisation. The setting of partial participation  considered, is general enough to include at least two specific partial participation strategies. The theoretical results match the state of the art guarantees for variance reduced algorithms with limited communication up to scaling of $1/p_a$, where $p_a$ governs the probability of some node to participate in the gradient estimation during some iteration of the algorithm. I find this result elegant and I like that this scaling is pretty much observed in the experiments. The authors give some intuition on why passing from the full to the partial participation setting is non-trivial.

**Weaknesses:**

The paper is very technical (if one wishes to follow proofs of the claims) and builds to an extensive amount of previous work, thus it is difficult to be appreciated by readers that do not belong to this specific research community. The appendix is about 90 (!) pages and a significant portion of them is essential to the proper understanding of this work, thus it's quite hard to review it in the context of neurips (the results are believable though). Since the topic is distributed computing with limited communication, I would expect a wider experimental analysis of the proposed algorithms in various tasks (also would be good some of the experiments to be in the main text).



**Questions:**

I struggle a bit to understand the intuition behind Algorithms 3 and 4. For algorithm 3, I don't see immediately the connection with the PAGE algorithm and, for Algorithm 4, I don't see which algorithm tries to generalise. Could you elaborate?

Is it possible to show that the chosen batch sizes of $B=O((L_{max}/\hat L)^2)$ and $O((L_{\sigma}/\hat L)^2)$ are at most $n$? Otherwise, there is a chance that such a batch size is not available.

Minor: there are some issues with the writing, e.g.:
"it is not trivial in each order one should apply the expectations"
"We are not fighting for the full generality"
" The main goal of our paper is to develop a method for the nonconvex distributed optimization"
"From the beginning of federated learning era, the partial participation..."

I would recommend to the authors to go through the text once more and improve the quality of writing.

**Limitations:**

The authors do not discuss limitations of their work, but I cannot see any profound ones. Perhaps the issue of batch sizes specified in my questions deserves a bit more attention.

---

> ### Author Rebuttal · Authors · 2023-08-06
>
> Thank you for the review!
>
> > The paper is very technical (if one wishes to follow proofs of the claims) and builds to an extensive amount of previous work, thus it is difficult to be appreciated by readers that do not belong to this specific research community. The appendix is about 90 (!) pages and a significant portion of them is essential to the proper understanding of this work, thus it's quite hard to review it in the context of neurips (the results are believable though). Since the topic is distributed computing with limited communication, I would expect a wider experimental analysis of the proposed algorithms in various tasks (also would be good some of the experiments to be in the main text).
>
> * The flow of our proofs is quite standard for the literature on optimization methods. One can look at the proofs of celebrated optimization methods (e.g. \[1\]). We agree that the appendix is large and admit it in Lines 183-187. However, the size of the appendix is justified by the number of new methods that we developed. And, in Lines 183-187, we suggest how to approach the proofs.
> * Our main goal was to develop methods with strong *theoretical* guarantees. Once we've done that, we expect that the developed methods will have the best performance in the analyzed setting with any loss functions, datasets, and parameters. **Note that we added new experiments that are presented in a separate file in the "global" comment to all reviewers called "Author Rebuttal by Authors".** Click "Revisions" if you can not find the pdf.
>
>
> > I struggle a bit to understand the intuition behind Algorithms 3 and 4. For algorithm 3, I don't see immediately the connection with the PAGE algorithm and, for Algorithm 4, I don't see which algorithm tries to generalise. Could you elaborate?
>
> Regarding Algorithm 4, it does not try to generalize any algorithm because it is somewhat new. However, it has connections to ZeroSARAH \[2\] which has stronger assumptions on the functions $f_i$ (see Table 2). For Algorithm 3, one can compare it to Line 4 of Algorithm 1 in \[3\]: we also use the probability switching technique. However, Algorithm 3 is not exactly PAGE because, with probability $p$, Algorithm 3 calculates $\nabla f_i(x^{t+1}) - \nabla f_i(x^{t}) - \frac{b}{p} (h^t_i - \nabla f_i(x^{t}))$ instead of $\nabla f_i(x^{t+1}).$ This modification is necessary for the partial participation setting.
>
> > Is it possible to show that the chosen batch sizes ...
>
> In the nonconvex case, these quantities can be as large as possible (we can not bound them by $n.$ Note that they are lower bounded by $1$). However, we never say that $B = (L_{\max} / \widehat{L})^2.$ Instead of it, we only say that $B = O ((L_{\max} / \widehat{L})^2),$ which means that $B \leq C * (L_{\max} / \widehat{L})^2$ for some constant $C \geq 1$ (This the definition of the Big-O notation). So it is always possible to choose a batch size $B.$
>
> \[1\]: G. Lan First-order and stochastic optimization methods for machine learning
> \[2\]: Z. Li. et al. ZeroSARAH: Efficient nonconvex finite-sum optimization with zero full gradient computation
> \[3\]: Z. Li. et al. PAGE: A simple and optimal probabilistic gradient estimator for nonconvex optimization

---

> > ### Comment · Reviewer_QePY · 2023-08-12
> >
> > I thank the authors for their response. I am satisfied by it and keep my overall acceptance score.

---

### Author Rebuttal · Authors · 2023-08-06

Dear AC and Reviewers,

We are now presenting new **extra experiments**. See the attached pdf to this rebuttal. If you do not see the pdf here, click "Revisions," after "Author Rebuttal by Authors" and there you will find a link to the pdf

In these experiments, we compare our new method with the previous theoretical state-of-the-art baselines (FRECON and MARINA) that work with the **partial participation setting** and **compression**. We choose only those baselines that support compressed communication from Table 1. We compare the algorithms on two datasets (real-sim and MNIST) and in stochastic and finite-sum settings.

We show that our new method requires significantly fewer communication rounds to find an $\varepsilon$-stationary point. **We believe that these experiments, together with the experiments from Section A in the main paper, give us strong evidence that our method has an excellent practical performance.**

Even more important, we have a theory that provides new state-of-the-art convergence guarantees in the considered settings.

Thank you! Please let us know if you have any additional questions or suggestions.

---

### Decision · Program_Chairs · 2023-09-21

**Decision:**

Accept (poster)

**Comment:**

This paper considers the partial participation setting for federated learning. By bringing together variance reduction and compressed communication to the partial participation setting, the paper extends the DASHA algorithm to stochastic and finite-sum settings, and the new DASHA-PP method achieves the optimal oracle complexity. Overall, I agree with the reviewers that this work provides an important contribution, and so I would recommend acceptance.